# Adversarial testing of global neuronal workspace and integrated information theories of consciousness

Cogitate Consortium*, Oscar Ferrante[1,40], Urszula Gorska-Klimowska[2,40], Simon Henin[3,40], Rony Hirschhorn[4,40], Aya Khalaf[5,40], Alex Lepauvre[6,7,40], Ling Liu[8,9,10,40], David Richter[7,11,40], Yamil Vidal[7,40], Niccolò Bonacchi[12,13], Tanya Brown[6], Praveen Sripad[6], Marcelo Armendariz[14,15], Katarina Bendtz[14,15], Tara Ghafari[1,16], Dorottya Hetenyi[1,17], Jay Jeschke[3], Csaba Kozma[2,18], David R. Mazumder[14], Stephanie Montenegro[3], Alia Seedat[3], Abdelrahman Sharafeldin[19], Shujun Yang[20], Sylvain Baillet[21], David J. Chalmers[22], Radoslaw M. Cichy[23,24,25], Francis Fallon[26], Theofanis I. Panagiotaropoulos[27,28], Hal Blumenfeld[5], Floris P. de Lange[7], Sasha Devore[3], Ole Jensen[16,29], Gabriel Kreiman[14,15], Huan Luo[8,30,31], Melanie Boly[2,32], Stanislas Dehaene[33,34], Christof Koch[35,36], Giulio Tononi[2], Michael Pitts[37,41], Liad Mudrik[4,38,41] & Lucia Melloni[3,6,39,41 ✉]

Different theories explain how subjective experience arises from brain activity[1,2]. These theories have independently accrued evidence, but have not been directly compared[3]. Here we present an open science adversarial collaboration directly juxtaposing integrated information theory (IIT)[4,5] and global neuronal workspace theory (GNWT)[6–10] via a theory-neutral consortium[11–13]. The theory proponents and the consortium developed and preregistered the experimental design, divergent predictions, expected outcomes and interpretation thereof[12]. Human participants (*n* = 256) viewed suprathreshold stimuli for variable durations while neural activity was measured with functional magnetic resonance imaging, magnetoencephalography and intracranial electroencephalography. We found information about conscious content in visual, ventrotemporal and inferior frontal cortex, with sustained responses in occipital and lateral temporal cortex reflecting stimulus duration, and content-specific synchronization between frontal and early visual areas. These results align with some predictions of IIT and GNWT, while substantially challenging key tenets of both theories. For IIT, a lack of sustained synchronization within the posterior cortex contradicts the claim that network connectivity specifies consciousness. GNWT is challenged by the general lack of ignition at stimulus offset and limited representation of certain conscious dimensions in the prefrontal cortex. These challenges extend to other theories of consciousness that share some of the predictions tested here[14–17]. Beyond challenging the theories, we present an alternative approach to advance cognitive neuroscience through principled, theory-driven, collaborative research and highlight the need for a quantitative framework for systematic theory testing and building.

Philosophers and scientists have sought to explain the subjective nature of consciousness (for example, the feeling of pain or of seeing a colourful rainbow) and how it relates to physical processes in the brain[18,19]. This quest has led to various theories of consciousness evolving in parallel[1–3] and often providing incompatible accounts of the neural basis of consciousness[1,2]. Furthermore, empirical support for a given theory is often highly dependent on methodological choices, pointing towards a confirmation bias in theory testing[3]. Convergence on a broadly accepted neuroscientific theory of consciousness will have profound medical, societal and ethical implications.

To advance this goal, we tested two theories of consciousness, through a large-scale, open-science adversarial collaboration[11,12,20–22] aimed at accelerating progress in consciousness research by building on constructive disagreement. We brought together proponents of IIT[4,5] and GNWT[6,23], in addition to theory-neutral researchers. The group identified differential existing and novel predictions of the two theories and developed an experimental design to test them (Fig. 1a). We preregistered these predictions, including pass or fail criteria, expected outcomes and their interpretation ex-ante[11,12]. We focus on IIT and GNWT, among other widely discussed theories (for example,

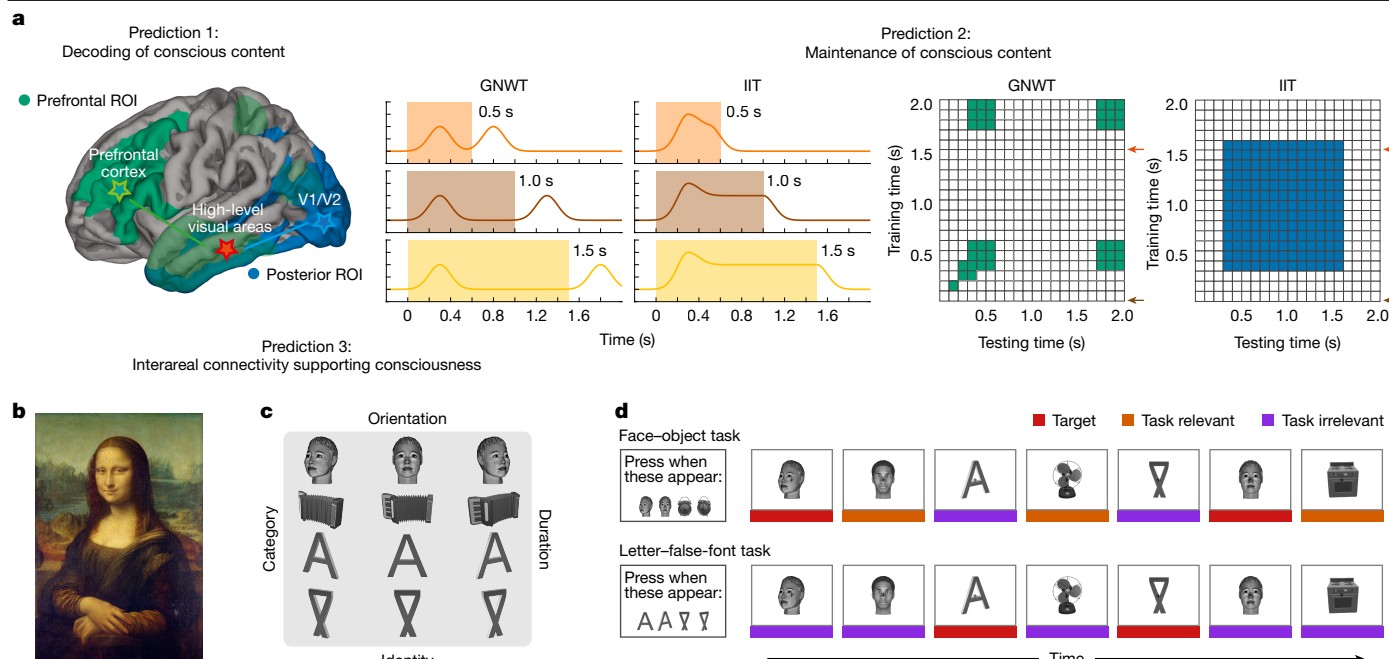

**Fig. 1 | Predictions and experimental design. a**, Predictions of IIT and GNWT. For prediction 1 (decoding of conscious content), IIT predicts maximal decoding of conscious content in posterior brain areas, whereas GNWT emphasizes a necessary role for the PFC. For prediction 2 (maintenance of conscious content), IIT posits that conscious content is actively maintained in the posterior cortex, whereas GNWT predicts brief content-specific ignition (approximately 0.3–0.5 s) in the PFC at stimulus onset and offset, with content stored in a non-conscious silent state between these events. Waveforms (left) and temporal generalization matrices (right) illustrate the predicted amplitude-based and information-based temporal profiles: coloured rectangles indicate the three stimulus durations for PFC (GNWT) and posterior cortex (IIT; left); the arrows mark stimulus onset (brown) and offset (red), whereas predicted temporal generalization is depicted in green (GNWT) and blue (IIT; right). For prediction 3 (interareal connectivity supporting consciousness), the stars and arrows on the brain diagram illustrate predicted synchrony patterns, with green representing GNWT and blue representing IIT. Brain surface is from Freesurfer. **b**, Conscious experience is multifaceted. For instance, viewing the Mona Lisa involves experiencing it as occupying a specific spatial location, categorizing it as a face, recognizing an identity and noting its leftward orientation, with this complex experience maintained over time. **c**, To manipulate conscious content, stimuli varied across four dimensions: category (faces, objects, letters and false fonts), identity (different exemplars within each category), orientation (left, right and front views) and duration (0.5 s, 1.0 s and 1.5 s). Example stimuli are shown. **d**, Experimental paradigm. Participants detected predefined targets (for example, a specific face and object or a letter and false font) in sequences of single, high-contrast stimuli. Each trial contained three stimulus types: targets (red; coloured frames for illustration only), task-relevant stimuli (orange-red; same categories as targets) and task-irrelevant stimuli (purple; other categories). Blank intervals between stimuli are not depicted. Object stimulus images in panels **c**,**d** are courtesy of Michael J. Tarr, Carnegie Mellon University, http://www.tarrlab.org/; face stimuli were created using FaceGen Modeler 3.1.

recurrent processing theory and higher-order theories[1,2]), because they feature prominently in consciousness science, as demonstrated by a recent systematic review of the literature[3].

IIT and GNWT explain consciousness differently: IIT proposes that consciousness is the intrinsic ability of a neuronal network to influence itself, as determined by the amount of maximally irreducible integrated information (phi) supported by a network in a state. On the basis of theoretical and neuroanatomical considerations, IIT suggests that a complex of maximum phi probably resides primarily in the posterior cerebral cortex, in a temporo–parietal–occipital 'hot zone'[4,5,24,25]. GNWT instead posits that consciousness arises from global broadcasting and late amplification (or 'ignition') of information across interconnected networks of higher-order sensory, parietal and especially prefrontal cortex (PFC)[6,9,23].

Both theories have a mathematical or computational core (integrated information for IIT and global workspace for GNWT) and proposed biological implementations (posterior cortex versus PFC and associated areas, respectively). Although it is difficult to test the mathematical or computational core of either theory directly, their competing biological implementations are empirically testable with current methodologies. Thus, our study focuses on brain regions where the predictions diverge most notably—posterior cortex for IIT and PFC for GNWT, rather than the associated areas in higher-order sensory or parietal cortex—to facilitate maximally diagnostic experiments.

One consequence of this biological focus is that theorists could respond to challenging data by modifying the proposed biological implementation while retaining the mathematical or computational core of a theory. Another consequence is that some predictions (and their associated consequences) may overlap with other theories of consciousness that share similar biological bases, such as higher-order theories[16,17] in the PFC and local recurrency theories[14,26] in the visual cortex. Although these are inherent aspects of studying theoretical proposals about neural mechanisms of consciousness, the results are expected to help the community make more informed judgements about the tested theories (for rationale, see the preregistration document[27]).

## Preregistered predictions and analyses

We tested three preregistered, peer-reviewed predictions of IIT and GNWT[12] for how the brain enables conscious experience (Fig. 1a). Prediction 1 addresses the cortical areas holding information about different aspects of conscious content. IIT predicts that conscious content is maximal in posterior brain areas, whereas GNWT predicts a necessary role for PFC. Prediction 2 pertains to the maintenance of conscious percepts over time[28–30]. IIT predicts that conscious content is actively maintained by neural activity in the posterior 'hot zone' throughout the duration of a conscious experience, whereas GNWT

predicts ignition events in PFC at stimulus onset and offset, updating the global workspace, with activity-silent information maintenance in between[31]. Prediction 3 examines interareal connectivity during conscious perception. IIT predicts sustained short-range connectivity within the posterior cortex, linking low-level sensory (V1/V2) with high-level category-selective areas (for example, fusiform face area and lateral occipital cortex), whereas GNWT predicts long-range connectivity between high-level category-selective areas and PFC. The combination of predictions, tested through highly powered, multimodal studies, places a high bar for either theory to pass, rendering failures more informative. Predictions were differentially weighted on the basis of their centrality to the theory and methodological considerations (Extended Data Table 1; for an additional preregistered non-critical analysis, see section 8 in Supplementary Information).

To empirically test these predictions, we investigated the content and temporal extent of conscious visual experiences, focusing on their phenomenological richness and multifaceted nature, even for a single stimulus. For instance, when viewing the Mona Lisa (Fig. 1b), one experiences it as having a specific identity, orientation and location in visual space for as long as one looks at the painting. To approximate such multifaceted experiences, we manipulated several attributes of conscious content by presenting suprathreshold visual stimuli across four different categories (faces, objects, letters and false fonts), each containing 20 unique identities shown in three orientations (front, left and right view) and for three durations (0.5, 1.0 and 1.5 s). In each block, participants were instructed to detect two infrequent target stimuli from either the pictorial (face–object) or symbolic (letter–false fonts) stimulus categories (for example, a specific face or object), making these categories task relevant for that block (Fig. 1c,d).

This paradigm offers several advantages. First, it provides robust conditions to test the predictions of the theories by focusing on clearly experienced conscious content, studied through a high signal-to-noise, suprathreshold, fully attended single stimulus at fixation. This amplifies the significance of any challenges to the theories, as they cannot be explained by weak signals. Second, it minimizes task and report confounds, isolating neural activity specifically related to consciousness. Third, it allows testing of novel predictions to address previously unexplored questions, that is, how experience is maintained over time, refining theories and yielding new insights.

All research was conducted by theory-neutral teams to minimize confirmatory bias. We evaluated the predictions of theories in 256 participants performing the same behavioural task in three neuroimaging modalities: functional magnetic resonance imaging (fMRI; $n = 120$), magnetoencephalography (MEG; $n = 102$) and intracranial electroencephalography (iEEG; $n = 34$). To overcome the spatial and temporal limitations of each modality, we combined whole-brain, non-invasive fMRI and MEG with invasive iEEG, ensuring methodological rigour. Combined with large sample sizes, this minimizes the likelihood that negative results are due to methodological or sensitivity issues. Data collection occurred in two (or three) independent laboratories for each modality to guarantee generalization across groups of participants, instruments and experimenters. To foster informativeness, reproducibility and robustness, we (1) separated theory proponents from data acquisition and analysis to minimize bias and post-hoc interpretation, (2) used a multimodal approach that maximizes spatiotemporal resolution and coverage for a stringent and comprehensive tests of the theories in humans, (3) predefined large samples to increase statistical power, (4) followed standardized[32] and preregistered protocols[12] to reduce setup differences and confirmatory bias[22] (see sections 1 and 2 in Supplementary Information), and (5) implemented an analysis optimization phase (one-third of the sample) followed by a final testing phase (two-thirds of the sample) on independent data for result validation[33]. Consequently, this large-scale international effort aimed at implementing a rigorous adversarial collaboration framework, thereby establishing a precedent for an alternative scientific approach.

## Decoding of conscious content

According to IIT, PFC is not necessary for consciousness. Consequently, decoding conscious content should be most effective from the posterior cortex, and adding PFC activity as additional information should not improve decoding accuracy. This prediction was considered non-critical for testing IIT, as the theory focuses on the intrinsic, causal perspective of information within a neural substrate rather than the amount of information decodable from the perspective of an extrinsic observer[5]. By contrast, GNWT posits that conscious content can be decoded from PFC activity. Both theories predict that conscious content should be evident in theory-relevant areas independently of other cognitive processes (for example, report and task); thus, conscious content should be present irrespective of task manipulations[34,35]. This prediction was tested by evaluating the decoding accuracy of stimulus category (faces–objects (pictorial) and letters–false fonts (symbolic)) and orientation (left, right and front facing) in all theory-relevant areas. All stimulus categories alternated between being task-relevant and task-irrelevant across blocks (Fig. 1d). Stimulus orientation, being orthogonal to the task, remained task-irrelevant in all blocks.

On the basis of our preregistered predictions and pre-approved interpretations[27] (Extended Data Table 1), the theories would pass the test if decoding is possible for both category (in at least one category pairing) and orientation (in at least one category), but would fail otherwise. Testing both category and orientation decoding constitutes a stringent test, as it requires two conditions to be satisfied, increasing the likelihood of failures[36], while capturing the critical multidimensionality of conscious content, that is, phenomenological richness (Fig. 1b).

For decoding of category, we tested whether information is present in the relevant regions irrespective of the task, using a cross-task generalization approach (see Methods).

Here we report the most robust results for decoding of category (faces–objects) and orientation (left, right and front views of faces). Qualitatively similar results were observed for decoding of letters–false fonts (Extended Data Fig. 1a–d). Results for orientation decoding were consistent across stimulus categories and data modalities in posterior cortex, but mostly absent in PFC (see section 5.1.2 in Supplementary Information).

In the iEEG data, pattern classifiers were trained on high gamma frequency band (70–150 Hz), which correlates with spiking activity[37,38], at each time point in the task-irrelevant condition, and tested in the task-relevant condition, for each stimulus duration and category, and across all electrodes within the theory-relevant region of interests (ROIs; see Fig. 2a for a visualization of ROIs and Methods for anatomical ROI definitions). In the posterior ROIs, face–object decoding showed significant cross-task generalization (more than 95% accuracy) for the approximate duration of the stimulus (Fig. 2b, top row). In the PFC ROIs, significant cross-task face–object decoding accuracy (approximately 70%) was also evident, but the temporal generalization of this decoding was restricted to approximately 0.2–0.4 s (Fig. 2b, bottom row). Training on task-relevant and testing on task-irrelevant trials showed similar results (Extended Data Fig. 1e; within-task decoding is presented in Extended Data Fig. 2).

Although electrode coverage across our sample of iEEG patients ($n = 29$ for decoding analyses) was exceptional in the relevant brain regions (Fig. 2a; PFC ROIs $n_{electrodes} = 576$, posterior ROIs $n_{electrodes} = 583$), we further analysed a larger population of healthy participants ($n = 65$) using MEG, focusing on theory-relevant ROIs (see Methods). Here too, cross-task generalization of face–object decoding was significant in both posterior and prefrontal ROIs (Fig. 2c) within the theory-predicted time windows. Cross-temporal generalization of decoding in MEG was sustained in posterior ROIs and brief in PFC ROIs for all three stimulus durations (see section 5.1.1.2 in Supplementary Information).

We leveraged the higher spatial resolution of fMRI ($n = 73$) to complement the analysis. A searchlight approach (see Methods) revealed

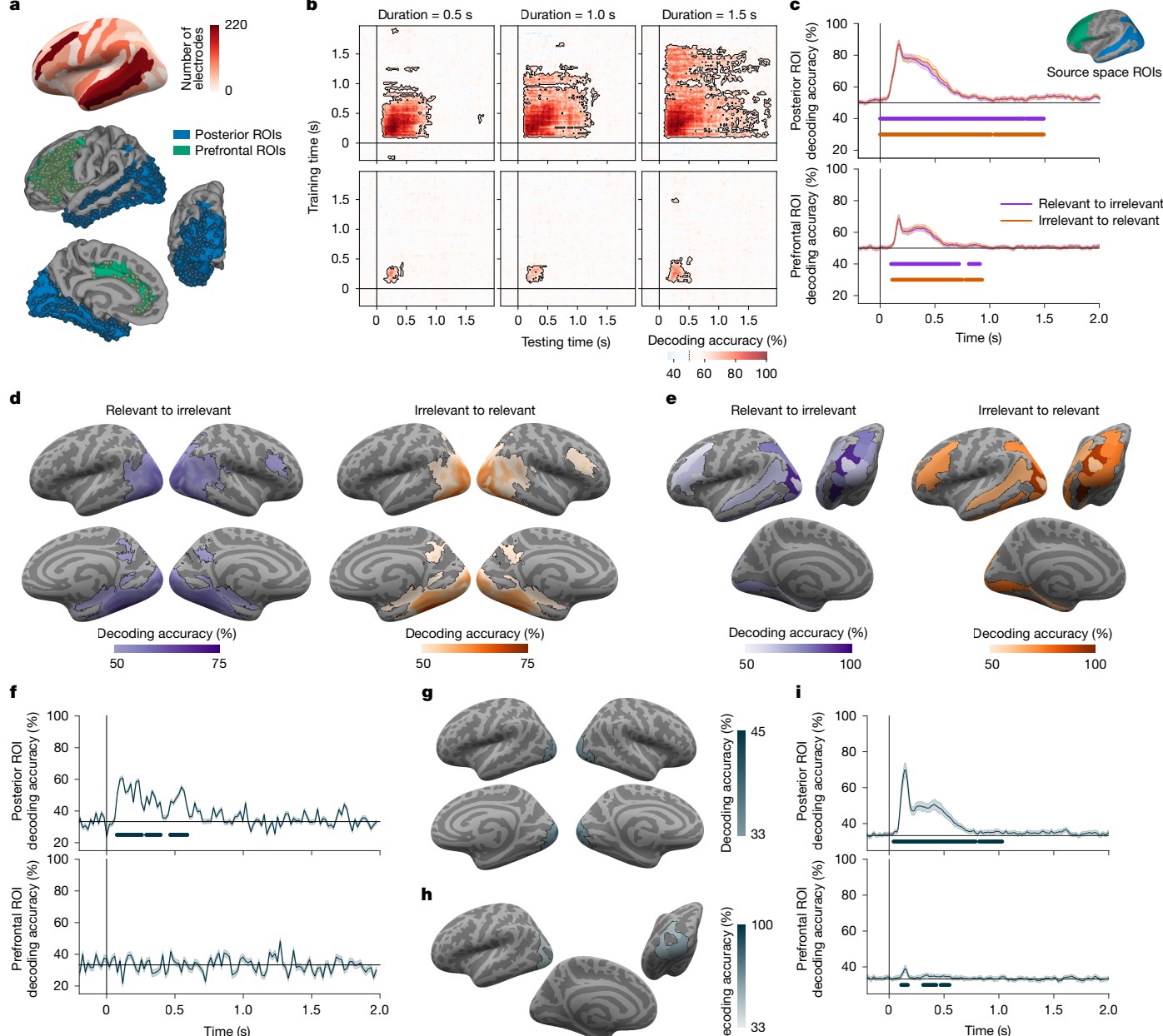

**Fig. 2 | Prediction 1: decoding of conscious content. a**, Spatial coverage of intracranial electrodes ($n_{patients}$ = 29) on a standard inflated cortical surface map (top), and within theory-defined ROIs (bottom): posterior (blue; $n_{electrodes}$ = 583) and prefrontal (green; $n_{electrodes}$ = 576). **b**, iEEG cross-task temporal generalization of decoding of high-gamma signal. Pattern classifiers were trained to discriminate stimulus category (faces–objects) in the task-irrelevant condition at each time point and tested in the task-relevant condition across all time-points. Columns denote stimulus durations (0.5 s (left), 1.0 s (centre) and 1.5 s (right)), and rows indicate theory ROIs (posterior (top) and prefrontal (bottom)). Contoured red-shaded regions depict significant above-chance (50%) decoding. Here and below, significance was evaluated through a non-parametric cluster-based permutation test ($P < 0.05$; two-sided). **c**, MEG average cross-task decoding of stimulus category ($n$ = 65) from task-relevant to task-irrelevant stimuli (purple) and vice versa (orange), separately for the posterior (top) and prefrontal (bottom) ROIs, depicted on inflated cortical surfaces (posterior in blue and prefrontal in green), across durations, using pseudotrial aggregation. Underlying lines indicate significance. The shading depicts 95% CI across participants. **d**, fMRI searchlight cross-task decoding of stimulus category ($n$ = 73), collapsed across durations, from task-relevant stimuli to task-irrelevant stimuli (left; purple) or vice versa (right; orange). The outlined coloured regions on the inflated cortical surfaces (left–right lateral views; right–left medial views (bottom)) indicate significant above-chance decoding. **e**, iEEG ROIs significant cross-task decoding of stimulus category, collapsed across durations. Conventions are as in panel **d**, displayed from a left lateral (top left), posterior (top right) and left medial (bottom) views. **f**, iEEG average decoding of stimulus orientation (left, right and front) within posterior (top) and prefrontal (bottom) ROIs, collapsed across durations. Underlying lines indicate above-chance (33%) decoding. The shading depicts 95% CI estimated across cross-validation folds. **g**, fMRI searchlight decoding of face orientation (left, right and front). Regions with significantly above-chance (33%) decoding accuracies are outlined in blue. **h**, iEEG ROIs decoding of face orientation (left, right and front). Conventions are as in panel **g**. **i**, MEG ROIs average decoding of face orientation (left, right and front). Conventions are as in panel **f**. Brain surfaces in panels **a**,**c**–**e**,**g**,**h** are from Freesurfer.

distributed, robust cross-task generalization (approximately 75%) in the striate and extrastriate, ventral temporal and intraparietal cortex (Fig. 2d and Extended Data Table 2). Generalization in PFC had lower accuracy (approximately 60%) and was spatially restricted to middle and inferior frontal cortex regions (Fig. 2d). Theory-relevant ROIs defined in the Destrieux atlas yielded comparable results (see section

5.1.1.3 in Supplementary Information). These results closely matched those from iEEG-restricted to theory-specified ROIs and time windows (Fig. 2e). Hence, across modalities, face–object decoding occurred in both posterior and prefrontal ROIs, consistent with IIT and GNWT predictions.

Given the rich and multidimensional nature of conscious content, we assessed the decoding of stimulus orientation, which was always task-irrelevant. We obtained divergent results for IIT and GNWT: decoding of face orientation (left, right or front views) was achieved in posterior but not in prefrontal ROIs, both with iEEG (Fig. 2f,h, approximately 95% with pseudotrial aggregation; Extended Data Fig. 3a) and the fMRI searchlight approach (Fig. 2g, approximately 45%). Decoding of face orientation was robust from MEG cortical time series in posterior ROIs (approximately 75% with pseudotrial aggregation), but was weaker, yet above chance (35%), in prefrontal ROIs (Fig. 2i), with a possibility of signal leakage from posterior regions (Extended Data Fig. 3b). Bayesian testing further validated these findings. For iEEG, Bayes factor (BF)$_{01}$ values (5.11–8.65) supported the null hypothesis of no face orientation decoding in prefrontal regions. fMRI Bayesian analysis revealed substantial-to-very-strong support for the null hypothesis in 34–55% of prefrontal voxels (BF$_{01}$: 3–71.5), with support for the alternative hypothesis in only 1–9% of voxels, whereas the rest remained inconclusive. Across all modalities, orientation decoding was observed for letters and false fonts—but not objects—in posterior, but not prefrontal, ROIs (see section 5.1.2 in Supplementary Information).

Finally, we tested IIT's prediction that prefrontal regions do not contribute further information beyond that specified by posterior areas (or may even degrade performance by introducing noise)[39]. If PFC activity increased decoding accuracy, IIT would be challenged, whereas no improvement in decoding accuracy would align with both IIT and GNWT, as GNWT posits that PFC workspace neurons broadcast but do not add information. We compared the performance of decoders trained exclusively on posterior ROIs with those trained on posterior and prefrontal ROIs together (Extended Data Fig. 3c; see Methods). Across critical time-resolved methods (iEEG and MEG) and various PFC ROI definitions, adding prefrontal ROIs did not improve—and in some cases reduced—category and orientation decoding (Extended Data Fig. 3d,e and see section 5.1.3 in Supplementary Information for non-critical fMRI results). Bayesian testing confirmed these findings: we found strong evidence against increased decoding accuracy when including PFC ROIs for category decoding (face–object: iEEG BF$_{01}$ = 1.94 × 10$^4$ and MEG BF$_{01}$ = 3.05; letter–false font: iEEG BF$_{01}$ = 1.91 × 10$^5$ and MEG BF$_{01}$ = 4.70) and face orientation (iEEG BF$_{01}$ = 1,205 and MEG BF$_{01}$ = 3.26).

## Maintenance of conscious content

According to IIT, the network that specifies the content of consciousness in posterior cortex is actively maintained over the duration of the conscious experience (manipulated here via different stimulus durations). By contrast, GNWT predicts brief, content-specific ignition in PFC within 0.3–0.5 s after stimulus onset, as the workspace is updated[12]. Activity then decays back to baseline, with information maintained in a latent state, until another ignition marks the offset of the current percept and the onset of a new percept (for example, the fixation screen following stimulus offset). Thus, although the underlying brain responses (the workspace update) are temporally discrete (that is, an onset and an offset response), the conscious experience can be temporally continuous, spanning from one update to the next.

Following the preregistration[27] (Extended Data Table 1), IIT would be challenged if sustained content-specific information and activation tracking stimulus duration were absent in the posterior cortex. GNWT would be challenged if transient prefrontal activation (at stimulus onset and offset) was not observed. These patterns were expected for at least one conscious feature (category, identity or orientation). We assessed activation strength as a function of stimulus duration and the informational content of this activation within theory-relevant ROIs. For IIT, both activation and information content were critical predictions, jointly determining result interpretation. For GNWT, activation was the primary measure owing to the difficulty of reliably detecting content-specific reinstatement at stimulus offset. The temporal predictions were tested in time-resolved iEEG and MEG data. We focused on the task-irrelevant condition as it best isolates neural activity related to consciousness while minimizing confounds (see sections 6.1 and 6.2.9 in Supplementary Information for task-relevant results).

First, we tested the predictions of the theories by investigating iEEG neural activation as a function of stimulus duration using linear mixed models (LMMs; see Methods) to model the time course of neural activity in the high gamma. Among the 31 patients included, 29.5% (194 of 657) of posterior ROI electrodes and 18.7% (123 of 655) of PFC ROI electrodes exhibited high gamma responses to stimuli (see section 6.1.2 in Supplementary Information).

In posterior cortex ROIs, 25 electrodes (out of 657) measured sustained activity tracking stimulus duration (Extended Data Table 3 for electrode localization and section 6.1.1 in Supplementary Information for results of the full model), consistent with IIT's (Fig. 3a). Of these, 12 electrodes tracked duration independent of stimulus category, primarily in early visual areas (for example, occipital pole; Fig. 3b), whereas 13 showed category-specific duration tracking (mostly for faces) in the ventral temporal cortex (for example, lateral fusiform gyrus; Fig. 3b). Overall, only a small proportion of electrodes exhibited both category selectivity and duration tracking—for example, just 15% (8 of 53) of face-selective electrodes showed sustained activity as predicted by IIT, suggesting a sparse neural substrate. These responses were mostly localized to the lateral fusiform gyrus, whereas the majority of face-selective electrodes displayed transient activations at stimulus onset across the striate, extrastriate and ventral areas (see section 6.1.2 in Supplementary Information).

In PFC ROIs, 99 and 24 electrodes showed non-selective and category-selective onset responses, respectively (Fig. 3c). However, none of the 655 electrodes measured the temporal profile predicted by GNWT (that is, onset and offset). Bayesian analysis confirmed this result (BF$_{01}$ > 3 for all electrodes in PFC ROIs), providing stronger evidence for either an intercept-only or a time-varying amplitude model over the GNWT model, with or without category interaction. This null result was not due to analysis limitations, as the LMM successfully detected the GNWT-predicted pattern in 10 electrodes in other ROIs (in the striate or extrastriate cortex; Fig. 3b). An exploratory decoding analysis of stimulus duration with unrestricted temporal profiles identified only one electrode, in the inferior frontal sulcus, showing the GNWT-predicted pattern, although with transient responses occurring earlier than expected (0.15 s post-onset and post-offset; Fig. 3c). Additional control analyses confirmed the IIT-predicted pattern in posterior ROIs and the absence of the GNWT-predicted pattern in PFC ROIs (see sections 6.2.1–6.2.3 in Supplementary Information).

We used LMMs to track gamma frequency band (60–90 Hz) power changes from the MEG source time series across posterior (15 parcels) and PFC (11 parcels) ROIs. Although signals were strong in posterior areas, none of the theory-based models adequately fit the data (BF$_{01}$ > 3 for all parcels; see section 6.1.3.1 in Supplementary Information). We also examined alpha band activity (8–13 Hz), which negatively correlates with neural spiking activity[40,41]. Validation of theoretical predictions from iEEG and MEG data was inconclusive: no prefrontal iEEG electrodes showed the GNWT-predicted combination of an onset and offset response (BF$_{01}$ > 3 for all prefrontal electrodes); instead, this pattern appeared in posterior sites and in MEG data, including the anterior cingulate cortex. However, the MEG results were sensitive to parameter choices, and signal leakage from posterior sites could not be ruled out (see sections 6.1.1 and 6.1.3.2 in Supplementary Information).

Next, we used cross-temporal representational similarity analysis (RSA) on both iEEG and MEG source data, within each theory-relevant

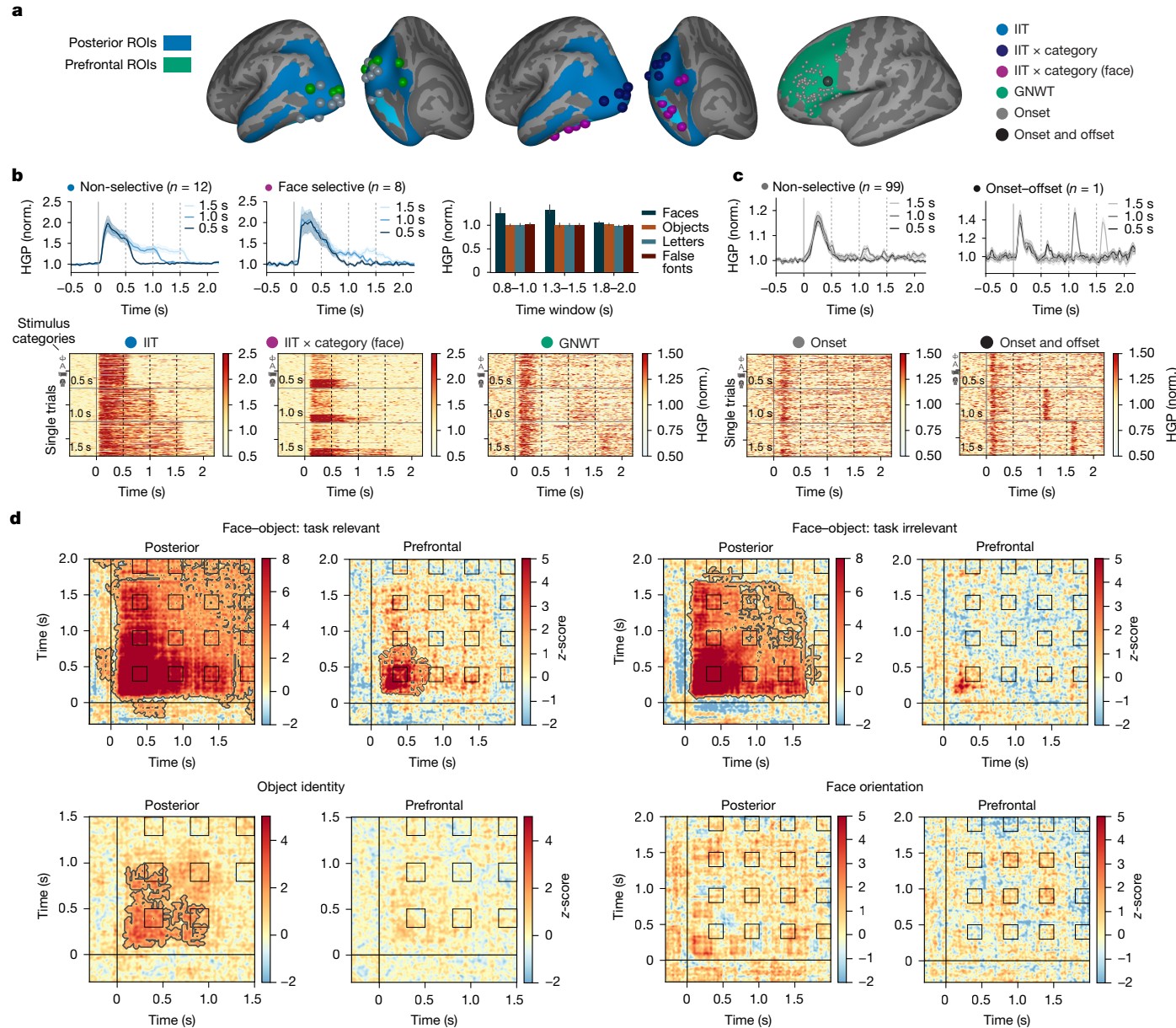

**Fig. 3 | Prediction 2: maintenance of conscious content. a**, Intracranial electrode localization on the MNI template, for posterior (left; blue; $n_{patients} = 31$ and $n_{electrodes} = 657$) and prefrontal (right; green; $n_{patients} = 31$ and $n_{electrodes} = 655$) ROIs. Electrodes are colour coded by response type based on model comparison (see Methods): sustained non-category-selective activation (light blue; $n = 12$), sustained category-selective activation (dark blue; $n = 5$), sustained face-selective activation (purple; $n = 8$), biphasic onset–offset activation in posterior areas (green; $n = 11$) and in PFC (black; $n = 1$), and onset-responsive activation in PFC (grey; $n = 99$). Brain surfaces are from Freesurfer. **b**, Posterior ROI activation. Time-series plots depict average high gamma (HG), separated by stimulus duration (0.5 s (dark), 1 s (medium) and 1.5 s (light)) for non-selective (left) and face-selective (middle) electrodes. The shading here and in panel **c** depicts standard error of the mean across electrodes and trials. The barplots (right) depict the average HG signal across sustained face-selective electrodes ($n = 8$) in 1.5-s trials, separated by category (faces in dark blue, objects in orange, letters in turquoise and false fonts in dark red) and theory-defined time windows (x axis). Raster plots show single-trial ($n = 320$) HG of individual electrodes

during task-irrelevant trials: a sustained non-selective (left), sustained face-selective (middle) and onset–offset (right) electrode. The rows depict single trials, sorted per duration (from top: 0.5, 1.0 and 1.5 s), and then category (from top: false fonts, letters, objects and faces). **c**, Prefrontal ROI activation. Time-series plots (top left) depict the average HG response per stimulus duration (shades of grey) for onset-responsive electrodes ($n = 99$) in task-irrelevant trials ($n = 320$). Average HG response per stimulus duration for a single electrode exhibiting onset–offset responses, with an earlier-than-predicted offset (top right). Raster plots for example onset (bottom left) and onset–offset (bottom right) responses are also shown. Conventions are as in panel **b**. **d**, Cross-temporal RSA matrices in posterior ($n_{patients} = 28$ and $n_{electrodes} = 583$) and prefrontal ($n_{patients} = 28$ and $n_{electrodes} = 576$) ROIs. Titles indicate the compared contrasts, and subtitles denote the ROIs. Matrix values represent $z$-scored, within-class-corrected correlation distances derived from a label shuffle null distribution. Contours denote significance (cluster-based permutation tests, $P < 0.05$, upper tail).

ROI, to test the predictions of IIT and GNWT about the temporal profile of the maintenance of conscious content (Fig. 1a, middle panel): sustained versus phasic (onset and offset) representation for IIT and GNWT, respectively. This test was critical for IIT only. Results for faces

and objects are presented below (see Extended Data Fig. 4 for similar results for letters–false fonts).

In iEEG, we calculated the correlation distance between high gamma activity patterns across 583 electrodes in posterior ($n_{patients} = 28$) and

576 electrodes in PFC ($n_{patients}$ = 28) ROIs separately (see Methods). We analysed the 1.5-s duration trials because this condition provided the strongest contrast between the temporal profiles predicted by the theories.

In posterior cortex ROIs, cross-temporal RSA revealed sustained face–object categorical representation, with larger correlation distances between categories (face–objects) than within category (face, object; compare Fig. 3d, left, with Fig. 1a). The RSA matrix matched the IIT model better than the GNWT model (see section 6.3 in Supplementary Information for results of all contrasts).

In PFC ROIs, cross-temporal RSA revealed transient face–object categorical representation at stimulus onset, but not at stimulus offset. Consequently, no significant correlation was found with the GNWT onset and offset model (compare Fig. 3d, right, with the predicted pattern in Fig. 1a). This pattern held even for the task-relevant condition, in which face–object information was stronger, more stable and longer lasting. Additional evidence for the absence of GNWT-predicted patterns in PFC ROIs emerged from three control analyses using (1) feature selection, which improved RSA in PFC; (2) modified time-windows to account for a potential earlier ignition at stimulus offset; and (3) a decoding analysis time-locked to stimulus offset to enhance sensitivity (see section 6.4 in Supplementary Information). These results align with two independent studies using comparable methods[42,43].

It has been argued that because conscious experiences are specific, the representation of identity and orientation are more stringent tests of the neural substrate of conscious experience[44] than of category. We thus also evaluated the predictions of the theories on these dimensions.

In posterior ROIs, iEEG revealed sustained object identity information, with smaller distances for same-identity objects than for different identities (Fig. 3d). The IIT model significantly correlated with the observed RSA matrix, providing a better fit than the GNWT model. Similar results were found for letter and false-font identity, but not for faces (Extended Data Fig. 4). In PFC ROIs, identity information was absent for all categories across analysed time windows (Fig. 3d, objects). Face orientation information appeared weakly in posterior ROIs at stimulus onset but was not sustained, decaying after 0.5 s, contrary to the predictions of IIT. No face orientation information was detected in PFC ROIs (Fig. 3d). Finally, the predictions of neither theory were supported for category, identity or orientation by the MEG data (see section 6.5 for Supplementary Information).

## Interareal connectivity

IIT predicts sustained gamma-band connectivity within the posterior cortex, that is, between high-level and low-level sensory areas (V1/V2), throughout any conscious visual experience. By contrast, GNWT predicts brief, late-phase metastable connectivity (more than 0.25 s) with information sharing between the PFC and category-specific areas, manifested in long-range gamma-band or beta-band synchronization[45].

On the basis of our preregistration (see Extended Data Table 1), IIT would be challenged in the absence of sustained content-specific synchronization between face–object selective areas and V1/V2; whereas a challenge for GNWT would be a lack of phasic connectivity (0.3–0.5 s) between category-selective areas and PFC. Given the dynamical nature of these predictions, iEEG and MEG provided the most informative empirical test. These predictions were tested by computing pairwise phase consistency (PPC)[46] between each category-selective time series (face-selective and object-selective nodes) and either the V1/V2 or the PFC time series in the intermediate (1.0 s) and long-stimulus-duration (1.5 s), task-irrelevant trials (see section 7.1.2 in Supplementary Information for task-relevant trials). Gamma activity was analysed because of its close link to neuronal spiking[47], which IIT considers a constituent property of the physical substrate of consciousness[5].

For iEEG, we analysed connectivity between electrodes showing face and object selectivity, using a different subset of electrodes to test connectivity with V1/V2 and PFC (see Methods; Extended Data Fig. 5a for ROIs and examples of face-selective and object-selective electrodes). Given the sparse electrode coverage, the requirement to focus on 'activated' electrodes (see Methods) was relaxed, although restricting it to only activated electrodes yielded similar results. We found increased category-selective synchrony between category-selective and V1/V2 electrodes (Extended Data Fig. 5b). These effects were early and brief (for example, less than 0.75 s), and restricted to low frequencies (2–25 Hz). This synchrony was mostly explained by the stimulus-evoked response (Extended Data Fig. 6a). These results fail to align with IIT's predictions: the activity was neither sustained nor observed in the gamma frequency band. Bayesian analysis further supported the null hypothesis ($BF_{01}$ = 1.15–4.9). No content-selective PPC was found between face-selective and object-selective electrodes and PFC electrodes in the relevant time window, contradicting the prediction of GNWT (Extended Data Fig. 6a; $BF_{01}$ = 2.62–5.32).

For MEG, we found selective synchronization between face-selective areas and both V1/V2 and PFC. These effects were again early, restricted to low frequencies (2–25 Hz), and mostly explained by stimulus-evoked responses (Extended Data Figs. 5d and 6b). Bayesian analysis of the gamma-band synchronization further supported the null hypothesis (all $BF_{01}$ > 3).

The results of the preregistered PPC metric for prediction 3, critical for both IIT and the GNWT, supported neither theory. PPC was chosen based on the mechanistic considerations of the theories because it assesses oscillatory phase. However, phase estimation from macroscopic recordings is susceptible to noise.

We thus used dynamic functional connectivity (DFC; see Methods), a metric sensitive to co-modulations of signal amplitude, after removing stimulus-evoked responses (Extended Data Fig. 6c,d includes the evoked response).

In iEEG, we observed significant connectivity between object-selective electrodes and V1/V2 (Fig. 4a), spanning multiple frequency bands, with the gamma band being the most predominant. In contrast to the predictions of IIT, the observed connectivity was brief. Connectivity between face-selective electrodes and V1/V2 was scarce, further supported by Bayesian analysis ($BF_{01}$ = 1.3 in favour of the null hypothesis). Significant connectivity was observed between PFC and both the face-selective and the object-selective areas in the gamma frequency band within the GNWT-predicted time window. For MEG, brief DFC in the alpha–beta frequency bands was found between face-selective nodes and both PFC and V1/V2 (Fig. 4b).

The exploratory DFC results in iEEG were consistent with the predictions of GNWT while challenging the predictions of IIT, as connectivity with V1/V2 was not sustained. However, V1/V2 were sparsely sampled in our population, with only 12 electrodes localized to V1/V2 compared with 472 in PFC.

Using fMRI, we evaluated connectivity across the entire cortex with homogeneous sampling. We computed the generalized psychophysiological interaction, defining the fusiform face area (FFA) and lateral occipital complex as seed regions (see Methods). Task-relevant and task-irrelevant trials were pooled to increase statistical power (see sections 7.1.1 and 12 in Supplementary Information for separate preregistered analyses). FFA showed content selective (face > object stimuli) connectivity with V1/V2, inferior frontal gyrus and intraparietal sulcus, consistent with predictions from both theories (Fig. 4c). No selective increase in interareal connectivity between object-selective nodes and PFC or V1/V2 was found in fMRI, even when separating task conditions (Extended Data Fig. 6f). Bayesian testing across prefrontal ROIs confirmed our findings with 62–94% of voxels across ROIs showing substantial evidence for the null hypothesis of no interareal connectivity ($BF_{01}$ = 3–7.75). Support for the alternative hypothesis was observed only in 0–4% of voxels. The remaining voxels showed inconclusive evidence.

To determine whether connectivity to PFC and V1/V2 was task driven in the generalized psychophysiological interaction, we explored the

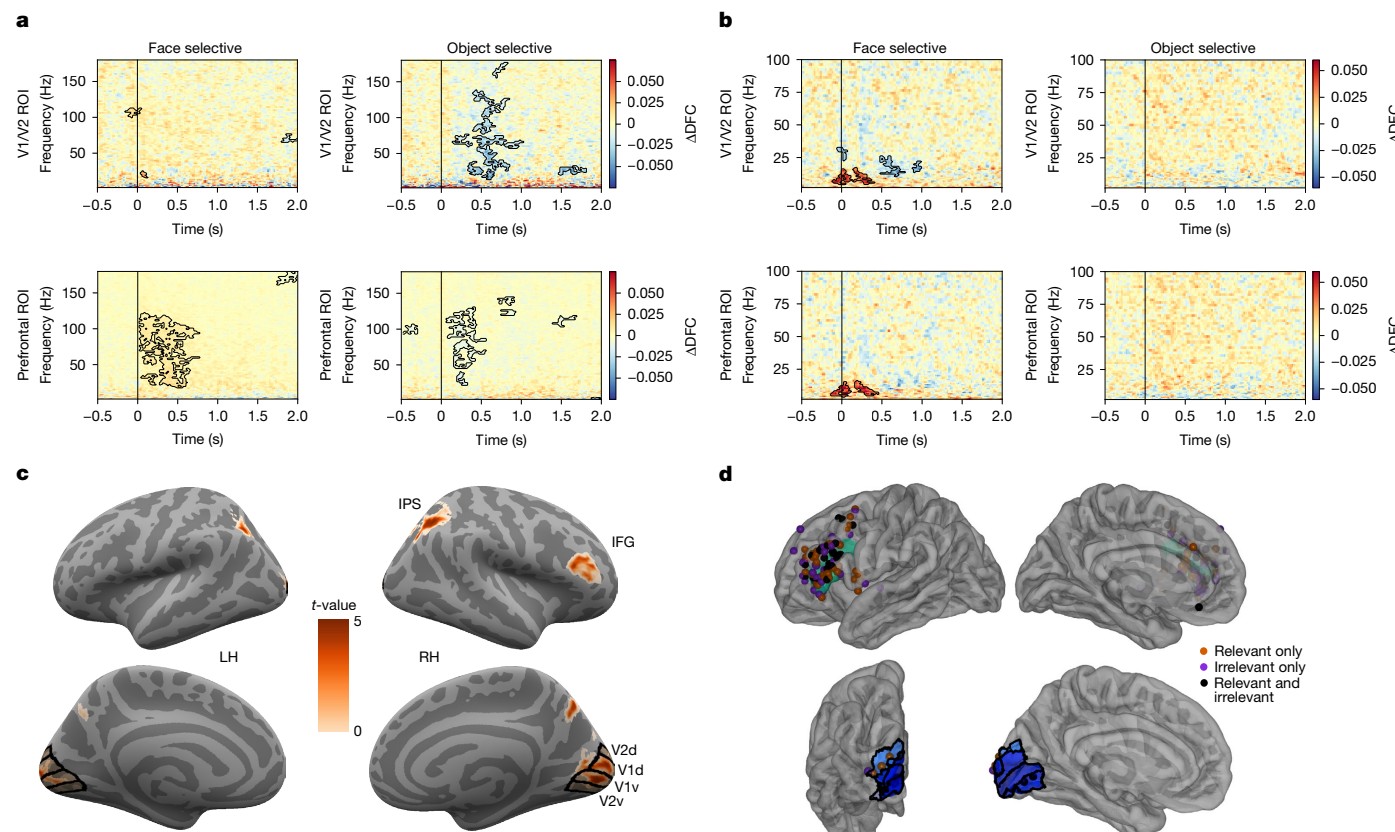

**Fig. 4 | Prediction 3: interareal connectivity. a**, iEEG DFC analysis of task-irrelevant trials revealed significant content-selective synchrony only for object-selective electrodes in V1/V2 (for example, top row; face-selective: $n_{patients}$ = 4 and $n_{electrodes}$ = 30; object-selective: $n_{patients}$= 4 and $n_{electrodes}$ = 21), while showing significant content-selective synchrony for both categories in the PFC ROI (bottom row; face selective: $n_{patients}$ = 19 and $n_{electrodes}$ = 81; object selective: $n_{patients}$ = 14 and $n_{electrodes}$ = 57). Here and in panel **b**, significance was assessed using a cluster-based permutation analysis ($P$ < 0.05, two-sided) and the colour bars represent the average change in the DFC between conditions. **b**, MEG DFC analysis of task-irrelevant trials ($n$ = 65) revealed significant content-selective synchrony below 25 Hz for the face-selective GED filter in both V1/V2 (top left) and PFC (bottom left), but not for the object-selective GED filter (right panels). **c**, fMRI generalized psychophysiological interaction (gPPI; $n$ = 70) on

task-relevant and task-irrelevant trials combined revealed significant content-selective connectivity when FFA is used as the analysis seed. Various significant regions showing task-related connectivity with the FFA seed were observed including V1/V2, right intraparietal sulcus (IPS) and right inferior frontal gyrus (IFG). LH, left hemisphere; RH, right hemisphere. **d**, Analysis of iEEG face-selective DFC synchrony across tasks is shown at the single-electrode level in PFC (top) and V1/V2 (bottom) ROIs. Electrodes showing significant synchrony (tested using a permutation test, FDR-corrected, $P$ < 0.05) in relevant (orange-red), irrelevant (purple) or combined relevant and irrelevant (black) trials are shown (averaged over 70–120 Hz and 0–0.5-s time window). DFC synchrony was observed in both tasks, but restricted to IFG for the GNWT analysis and V2 regions for the IIT analysis, consistent with the fMRI gPPI analysis shown in panel **c**. Brain surfaces in panels **c**,**d** are from Freesurfer.

iEEG data, separating task-relevant and irrelevant trials. We found task-independent, selective DFC connectivity (face > objects) for face-selective electrodes in both inferior frontal gyrus and V1/V2 (Fig. 4d).

## Discussion

This adversarial collaboration aimed to address confirmation biases by researchers, breaking theoretical echo chambers[3] and identifying the strengths and weaknesses of theories[2,48] by forcing them to be explicit and committal about their empirical predictions, rigorously testing them on common methodological grounds[22,49], and providing the means for theorists to change their minds given conflicting results[49]. Doing so catalyses our ability to evaluate and arbitrate between theories of consciousness. Embracing this spirit, and adhering to guidelines for structuring adversarial collaborations[21], we opted for a three-voice discussion format, acknowledging that despite stringent testing of incompatible theoretical views, different interpretations of the same evidence may persist. Below, the theory-neutral consortium presents the main challenges our findings pose to the theories, based on the preregistered predictions, methods and analysis agreed on in advance with the adversaries. Then, adversaries offer their own interpretation

of the findings and future directions (see sections 12 and 13 in Supplementary Information).

Extended Data Fig. 7 summarizes the key results, including the criteria used to assess whether findings support or contradict the theories. This summary covers both central and peripheral findings related to theory evaluation. The consortium adopted Lakatos' sophisticated falsificationist approach to philosophy of science[13,50], emphasizing that challenged predictions provide more valuable insights than those confirmed by the data (see section 11 in Supplementary Information). Outcomes are weighted differentially across predictions and with respect to the different brain imaging modalities (Extended Data Table 1). This approach ensures a nuanced evaluation of the theories, highlighting areas of strength and those requiring further refinement.

For IIT, the lack of sustained synchronization within posterior cortex represents the most direct challenge, based on our preregistration. This is incompatible with IIT's claim that the state of the neural network, including its activity and connectivity, specifies the degree and content of consciousness[5]. Although this null result could stem from methodological limitations (for example, limited iEEG sampling of V1/V2 areas), our multimodal and highly powered study provided the best conditions so far for evaluating the prediction. We urge IIT

proponents to direct future efforts to evaluate this prediction and determine the implications of this failure.

More broadly, although IIT passed the predefined criteria for the duration prediction (number 2), there was no evidence for a sustained representation of orientation, despite being a property of the consciously perceived stimuli[25]. This is an informative challenge for IIT, as orientation decoding was robust across all three data modalities, leaving open the question of whether and how information about orientation is maintained over time.

For GNWT, the most substantial challenge based on our preregistered criteria pertains to its account for the maintenance of a conscious percept over time and, in particular, the lack of ignition at stimulus offset. This result is unlikely to stem from sensitivity limitations, as offset responses were robustly found elsewhere (for example, visual areas); and in PFC, strong onset responses were found to the very same stimuli. The lack of ignition at stimulus offset is especially surprising given the change in conscious experience at the onset of the blank fixation screen. This clear update to the content of consciousness should have been represented somehow by the global workspace[12]. Thus, that aspect of consciousness remains unexplained within the GNWT framework.

Another key challenge for GNWT pertains to representing the contents of experience: although we found representation of category in PFC irrespective of the task, thereby demonstrating the sensitivity of our methods, no representation of identity was found, and representation of orientation was evident only in MEG (signal leakage notwithstanding); although these dimensions were clearly a part of the conscious experience of participants of the stimuli. This raises the question of whether PFC is involved in broadcasting all conscious content, as predicted by GNWT[23], or only a subset (for example, abstract concepts and categories, rather than low-level details), in which case the role of PFC in consciousness might need to be redefined.

Before this study, predictions from IIT and GNWT were typically tested with one data modality at a time[23,24], leaving room for negative results to be easily attributed to the limitations of the chosen modality[51]. We combined multiple techniques (iEEG, MEG and fMRI) to mitigate these limitations, cross-compensating for their weaknesses. This methodological approach was mutually agreed upon by the theory leaders before data collection and results disclosure as the most powerful and conclusive approach, making both positive and negative findings more meaningful.

Although this study was designed around IIT and GNWT, the results may have implications for other theories of consciousness. For example, the prediction of GNWT about PFC is shared by those higher-order theories of consciousness that hypothesize that PFC actually supplies the content of visual consciousness (for example, ref. 17), rather than those that take it to merely enable the consciousness of content that is located in posterior visual areas (for example, refs. 52,53). As a result, the failures to confirm this prediction challenge not only GNWT but also those higher-order theories[54]. Predictions 2 and 3, about timing and connectivity, are more distinctive to GNWT but could also be shared by other theories. Likewise, the non-core prediction 1 about the posterior cortex by IIT is also shared by many theories (for example, recurrent processing theory[14]), and its prediction 2 about timing may be shared by some posterior theories of consciousness, such as the local recurrency theory[15]. Its prediction 3 about interareal connectivity is more distinctive to IIT (for example, it is not shared by synchrony theory[55]), so the challenge here is more specific as well.

Our study focused on the contents of consciousness (for example, category, identity, orientation and duration), linking brain activity to subjective phenomenology. This departs from the traditional contrastive method, which compares the presence and absence of consciousness but conflates it with other cognitive processes (for example, decision-making or memory formation)[56–58]. Some might argue that our approach tracks stimulus processing rather than consciousness. Yet, our aim is to challenge and potentially falsify[50,59] IIT and GNWT, by

examining where their predictions differ, rather than to discover the neural correlates of consciousness. In this context, what might seem like a weakness—focusing on the presence of fully attended, consciously experienced stimuli—is actually beneficial for testing the primary positive predictions of the theories and their failures. This is because such failures are harder to dismiss owing to weak signals. Thus, our approach assesses whether the proposed neural mechanisms are truly necessary for consciousness.

Our study, although comprehensive, is not without limitations. First, we cannot entirely rule out residual task engagement with respect to category, although our design ensured that orientation and duration remained task-irrelevant, so the results on these dimensions cannot be explained by task-related effects. Second, although we aimed to capture multiple aspects of consciousness, our approach still falls short of encompassing its full phenomenal richness. Third, despite the high spatial and temporal resolution of our data, it lacks single-unit recordings, which are typically restricted to patients with epilepsy and selected brain regions. Ongoing studies in animal models, as part of a separate adversarial collaboration, can accordingly complement our findings.

Beyond directly challenging the theories, our study raises broader questions about theory testing and development across disciplines. A key challenge is how to weigh predictions and integrate evidence across different analyses and measurement techniques (for example, fMRI, MEG and iEEG). We adopted a lenient falsificationist approach, in which evidence for any tested feature (for example, decoding of category or orientation) was sufficient to rule out failure, rather than requiring consistency across all features. However, a formal framework is urgently needed to quantitatively integrate evidence, accounting for prediction centrality, measurement error and cross-sample consistency. Such a framework would enhance systematic theory building in an era of accumulating results[60].

After reviewing the results and the discussions by adversaries, readers might expect a definitive verdict on the two theories under evaluation. Instead, we invite readers to weigh the evidence themselves—considering the support for each preregistered prediction, the breadth of the data, the sophistication of the methods and analyses, and the cognitive biases that shape interpretation. Scientific progress is rarely a matter of simple verdicts; evidence is filtered through previous beliefs and motivations[61], making theory evaluation a dynamic process. By presenting results and adversarial responses transparently, we embrace the openness needed for science to converge on robust explanations of complex phenomena such as consciousness.

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

¹Centre for Human Brain Health, School of Psychology, University of Birmingham, Birmingham, UK. ²Department of Psychiatry, University of Wisconsin-Madison, Madison, WI, USA. ³Department of Neurology, New York University Grossman School of Medicine, New York, NY, USA. ⁴Sagol School of Neuroscience, Tel Aviv University, Tel Aviv, Israel. ⁵Department of Neurology, Yale School of Medicine, New Haven, CT, USA. ⁶Neural Circuits, Consciousness and Cognition Research Group, Max Planck Institute for Empirical Aesthetics, Frankfurt am Main, Germany. ⁷Donders Institute for Brain, Cognition and Behaviour, Radboud University Nijmegen, Nijmegen, the Netherlands. ⁸School of Psychological and Cognitive Sciences, Peking University, Beijing, China. ⁹Cognitive Science and Allied Health School, Beijing Language and Culture University, Beijing, China. ¹⁰Speech and Hearing Impairment and Brain Computer Interface LAB, Beijing Language and Culture University, Beijing, China. ¹¹Mind, Brain and Behavior Research Center (CIMCYC), University of Granada, Granada, Spain. ¹²William James Center for Research, ISPA - Instituto Universitário, Lisbon, Portugal. ¹³Champalimaud Research, Lisbon, Portugal. ¹⁴Boston Children's Hospital, Harvard Medical School, Boston, MA, USA¹⁵Center for Brains, Minds and Machines, Cambridge, MA, USA. ¹⁶Wellcome Centre for Integrative Neuroscience, Oxford Centre for Human Brain Activity, Department of Psychiatry, University of Oxford, Oxford, UK. ¹⁷Department of Imaging Neuroscience, UCL Queen Square Institute of Neurology, University College London, London, UK. ¹⁸CNNP Lab, School of Computing, Newcastle University, Newcastle upon Tyne, UK. ¹⁹Georgia Institute of Technology, Atlanta, GA, USA. ²⁰Department of Psychology, University of Amsterdam, Amsterdam, the Netherlands²¹Montreal Neurological Institute, McGill University, Montreal, Québec, Canada. ²²Department of Philosophy, New York University, New York, NY, USA. ²³Department of Education and Psychology, Freie Universität Berlin, Berlin, Germany. ²⁴Berlin School of Mind and Brain, Faculty of Philosophy, Humboldt-Universität zu Berlin, Berlin, Germany. ²⁵Bernstein Center for Computational Neuroscience Berlin, Berlin, Germany. ²⁶Philosophy Department, Psychology Department, St John's University, Queens, NY, USA. ²⁷Department of Psychology, National and Kapodistrian University of Athens, Athens, Greece. ²⁸Centre for Basic Research, Biomedical Research Foundation of the Academy of Athens (BRFAA), Athens, Greece. ²⁹Department of Experimental Psychology, University of Oxford, Oxford, UK. ³⁰IDG/McGovern Institute for Brain Research, Peking University, Beijing, China. ³¹Key Laboratory of Machine Perception (Ministry of Education), Peking University, Beijing, China. ³²Department of Neurology, University of Wisconsin-Madison, Madison, WI, USA. ³³Cognitive Neuroimaging Unit, Commissariat à l'Energie Atomique (CEA), Institut National de la Santé et de la Recherche Médicale (INSERM), Université Paris-Saclay, NeuroSpin Center, Gif-sur-Yvette, France. ³⁴Collège de France, Université Paris-Sciences-Lettres (PSL), Paris, France. ³⁵Allen Institute, Seattle, WA, USA. ³⁶Tiny Blue Dot Foundation, Santa Monica, CA, USA. ³⁷Psychology Department, Reed College, Portland, OR, USA. ³⁸School of Psychological Sciences, Tel Aviv University, Tel Aviv, Israel. ³⁹Predictive Brain Department, Research Center One Health Ruhr, University Alliance Ruhr, Faculty of Psychology, Ruhr University Bochum, Bochum, Germany. ⁴⁰These authors contributed equally: Oscar Ferrante, Urszula Gorska-Klimowska, Simon Henin, Rony Hirschhorn, Aya Khalaf, Alex Lepauvre, Ling Liu, David Richter, Yamil Vidal. ⁴¹These authors jointly supervised this work: Michael Pitts, Liad Mudrik, Lucia Melloni. e-mail: lucia.mellonibuljevic@rub.de

## Methods

### Ethics statement

The experiment was approved by the institutional ethics committees of each participating data-collecting laboratory, including the Science, Technology, Engineering and Mathematics Ethical Review Committee at the Centre for Human Brain Research, University of Birmingham (ERN_18-0226AP20); the Committee for Protecting Human and Animal Subjects at the School of Psychological and Cognitive Sciences, Peking University (2020-05-07e); the Commissie Mensgebonden Onderzoek Regio Arnhem-Nijmegen at the Centre for Cognitive Neuroimaging at Donders Institute (NL45659.091.14); the Human Research Protection Program Institutional Review Board at Yale School of Medicine (2000027591); the Office of Science and Research Institutional Review Board at New York University Langone Health (i14-02101_CR6); the Boston Children's Hospital Institutional Review Board at Children's Hospital Corporation d/b/a Boston Children's Hospital (04-05-065R); the Institutional Review Board at the University of Wisconsin-Madison (ID: 2017-1299); and the Ethics Council of the Max Planck Society at Max Planck Institute for Empirical Aesthetics (Nr. 2017_12). All participants and patients provided oral and written informed consent before participating in the study. All study procedures were carried out in accordance with the Declaration of Helsinki. Patients were also informed that clinical care was not affected by participation in the study.

### Participants

Healthy participants and patients with pharmaco-resistant focal epilepsy participated in this study. The datasets reported here consist of: (1) behaviour, eye tracking and iEEG data collected at the Comprehensive Epilepsy Center at New York University (NYU) Langone Health, the Brigham and Women's Hospital, the Boston Children's Hospital (Harvard), and the University of Wisconsin School of Medicine and Public Health (WU). (2) Behaviour, eye tracking, MEG and EEG data collected at the Centre for Human Brain Health (CHBH) of the University of Birmingham (UB), and at the Center for MRI Research of Peking University (PKU). (3) Behaviour, eye tracking and fMRI data collected at the Yale Magnetic Resonance Research Center (MRRC) and at the Donders Centre for Cognitive Neuroimaging (DCCN), of Radboud University Nijmegen. For both the MEG and fMRI datasets, one-third of the data that passed quality tests (henceforth, the optimization dataset; see the section 'Preregistration' for details about quality test criteria[27]) were used to optimize the analysis methods, which were subsequently added to the preregistration as an additional amendment. These preregistered analyses were then run on the remaining two-thirds of the data (henceforth, the replication dataset) and constitute the data reported in the main study. This procedure was not used for the iEEG data due to the serendipitous nature of the recording and electrode placement, the rarity of this type of data and the increased difficulty of data collection due to the COVID-19 pandemic.

A total of 97 healthy participants were included in the MEG sample (mean age of 22.79 ± 3.59 years, 54 females, all right handed), 32 of those datasets were included in the optimization phase (mean age of 22.50 ± 3.43 years, 19 females, all right handed), and 65 in the replication sample (mean age of 22.93 ± 3.66, 35 females, all right handed). Five additional participants were excluded from the MEG dataset: two because of failure to meet predefined behavioural criteria (that is, Hits of less than 80% and/or False Alarms > 20%), two because of excessive noise from sensors, and one because of incorrect sensor reconstruction. A total of 108 healthy participants were included in the fMRI sample (mean age of 23.28 ± 3.46 years, 70 females, 105 right handed); 35 of those datasets were included in the optimization sample (mean age of 23.26 ± 3.64 years, 21 females, 34 right handed) and 73 in the replication sample (mean age of 23.29 ± 3.37, 49 females, 71 right handed). Twelve additional participants were excluded from the fMRI dataset: eight because of motion artefacts, two because of insufficient coverage and two because of incomplete data (with respect to these last two participants, see section 14 of the Supplementary Information for deviations from the preregistration document). For the iEEG arm of the project, a total of 34 patients were recruited. Two patients were excluded owing to incomplete data. Demographic, medical and neuropsychological scores for each patient, when available, are reported in Supplementary Table 25. Three iEEG patients whose behaviour fell slightly short of the predefined behavioural criteria (that is Hits of less than 70%, FA > 30%) were nonetheless included given the difficulty of obtaining additional iEEG data (see section 14 in Supplementary Information for deviation from the preregistration).

### Experimental procedure

**Experimental design.** To test critical predictions of the theories, five experimental manipulations were included in the experimental design: (1) four stimulus categories (faces, objects, letters and false fonts), (2) 20 stimulus identities (20 different exemplars per stimulus category), (3) three stimulus orientations (front, left and right view), (4) three stimulus durations (0.5 s, 1.0 s and 1.5 s), and (5) task relevance (relevant targets, relevant non-targets and irrelevant).

Stimulus category, stimulus identity and stimulus orientation served to test predictions about the representation of the content of consciousness in different brain areas by the theories. In addition, stimulus duration served to test predictions about the temporal dynamics of sustained conscious percepts and interareal synchronization between areas. Task relevance served to rule out the effect of task demands, as opposed to conscious perception per se, on the observed effects[62]. This aspect of the experimental design was inspired by ref. 63.

**Stimuli.** Four stimulus categories were used: faces, objects, letters and false fonts. These stimuli naturally fell into two clearly distinct groups: pictures (faces and objects) and symbols (letters and false fonts). These natural couplings were aimed at creating a clear difference between task-relevant and task-irrelevant stimuli in each trial block (see the section 'Procedure'). All stimuli covered a squared aperture at an average visual angle of 6° by 6°. Face stimuli were created with FaceGen Modeler 3.1; letter and false font stimuli were generated with MAXON CINEMA 4D Studio (RC - R20) 20.059; object stimuli were taken from the Object Databank[64]. Stimuli were grey scaled and equated for luminance and size. To facilitate face individuation, faces had different hairstyles and belonged to different ethnicities and genders. Equal proportions of male and female faces were presented. The orientation of the stimuli was manipulated, such that half of the stimuli from each category had a side view (30° and −30° horizontal viewing angle, left and right orientation) and the other half had a front view (0°).

**Procedure.** Participants performed a non-speeded target detection task (see Supplementary Video 1). The experiment was divided into runs, with four blocks in each run (see the section 'Trial counts'). On a given block, participants viewed a sequence of single, supra-threshold, foveally presented stimuli belonging to one of four stimulus categories and presented for one of three stimulus durations onto a fixation cross that was present throughout the experiment. Within each block, half of the stimuli were task-relevant and half were task-irrelevant. To manipulate task relevance, at the beginning of each block participants were instructed to detect the rare occurrences of two target stimulus identities, one from each relevant category (for pictures, face–object; for symbols, letter–false font), irrespective of their orientation. This was specified by presenting the instruction 'detect face A and object B' or 'detect letter C and false font D', accompanied by images for each target (see Fig. 1d). Targets did not repeat across blocks. Each run contained two blocks of the face–object task and two blocks of the letter–false font task, with block order counterbalanced across runs.

Accordingly, each block contained three different trial types: (1) targets: the two stimuli being detected (for example, the specific face

and object identities); (2) task-relevant stimuli: all other stimuli from the task-relevant categories (for example, the non-target faces–objects); and (3) task-irrelevant stimuli: all stimuli from the two other categories (for example, letters–false fonts). An advantage of this design is that the three trial types enabled a differentiation of neural responses related to task goal, task relevance and simply consciously seeing a stimulus. We confirmed that participants were conscious of the stimuli in both the task-relevant and task-irrelevant trials in a separate experiment, which included a surprise memory test (see section 3 in Supplementary Information).

Stimuli were presented for one of three durations (0.5 s, 1.0 s or 1.5 s), followed by a blank period of a variable duration to complete an overall trial length fixed at 2.0 s. For the MEG and iEEG version, random jitter was added at the end of each trial (mean inter-trial interval of 0.4 s, jittered 0.2–2.0 s, truncated exponential distribution) to avoid periodic presentation of the stimuli. The mean trial length was 2.4 s. For the fMRI protocol, timing was adjusted as follows: the random jitter between trials was increased (mean inter-trial interval of 3 s, jittered 2.5–10 s, with truncated exponential distribution), with each trial lasting approximately 5.5 s. This modification helped with avoiding non-linearities in BOLD signal, which may affect fMRI decoding[65]. Second, to increase detection efficacy for amplitude-based analyses, three additional baseline periods (blank screen) of 12 s each were included per run (total of 24). The identity of the stimuli was randomized with the constraint that they appeared equally across durations and tasks conditions. Participants were further instructed to maintain central fixation on a black circle with a white cross and another black circle in the middle throughout each trial (see Supplementary Fig. 1d and Supplementary Video 1 for a demonstration of the experimental paradigm).

**Trial counts.** The MEG study consisted of 10 runs containing 4 blocks each with 34–38 trials per block, 32 non-targets (8 per category) and 2–6 targets, for a total of 1,440 trials. The same design was used for iEEG, but with half the runs (5 runs total), resulting in a total of 720 trials. For fMRI, there were 8 runs containing 4 blocks each with 17–19 trials per block, 16 non-targets (4 per category) and 1–3 targets, for a total of 576 trials. Rest breaks between runs and blocks were included.

### Data acquisition
**Behavioural data acquisition.** The task was run on Matlab (PKU: R2018b; DCCN, UB and Yale: R2019b; Harvard: R2020b; NYU: R2020a, and WU: 2021a) using Psychtoolbox (v3)[66]. The iEEG version of the task was run on a Dell Precision 5540 laptop, with a 15.6″ Ultrasharp screen at NYU and Harvard and on a Dell D29M PC with an Acer 19.1″ screen in WU. Participants responded using an eight-button response box (Millikey LH-8; response hand (or hands) varied based on the setting in the patient's room). The MEG version was run on a custom PC at UB and a Dell XPS desktop PC on PKU. Stimuli were displayed on a screen placed in front of the participants with a PROPixx DLP LED projector (VPixx Technologies). Participants responded with both hands using two 5-button response boxes (NAtA or SINORAD). The fMRI version was run on an MSI laptop at Yale and a Dell Desktop PC at DCCN. In DCCN, stimuli were presented on an MRI compatible Cambridge Research Systems BOLD screen 32″ IPS LCD monitor, and in Yale they were presented on a Psychology Software Tools Hyperion projection system to project stimuli on the mirror fixed to the head coil. Participants responded with their right hand using a 2 × 2 current designs response box at Yale and a 1 × 4 current designs response box at DCCN.

**Eye tracking data acquisition.** For the iEEG setup, eye tracking and pupillometry data were collected using a EyeLink 1000 Plus in remote mode, sampled monocularly at 500 Hz (from the left eye at WU, and depending on the setup at Harvard), or on a Tobii-4C eye tracker, sampled binocularly at 90 Hz (NYU). The MEG and fMRI laboratories used the MEG-compatible and fMRI-compatible EyeLink 1000 Plus Eye-tracker

system (SR Research) to collect data at 1,000 Hz. For MEG, eye tracking data were acquired binocularly. For fMRI, data were acquired monocularly from either the left or the right eye, in DCCN and Yale, respectively. For all recordings, a 9-point calibration was performed (besides Harvard, where a 13-point calibration was used) at the beginning of the experiment, and recalibration was carried out as needed at the beginning of each block or run.

**iEEG data acquisition.** Brain activity was recorded with a combination of intracranial subdural platinum-iridium electrodes embedded in SILASTIC sheets (2.3-mm diameter contacts, Ad-Tech Medical Instrument and PMT Corporation) and/or depth stereo-electroencephalographic platinum-iridium electrodes (PMT Corporation; 0.8 mm in diameter, 2.0-mm length cylinders; separated from adjacent contacts by 1.5–2.43 mm), or depth stereo-electroencephalographic platinum-iridium electrodes (BF08R-SP21X-0C2, Ad-Tech Medical; 1.28 mm in diameter, 1.57 mm in length, 3–5.5-mm spacing). Electrodes were arranged as grid arrays (either 8 × 8 with 10-mm centre-to-centre spacing, 8 × 16 contacts with 3-mm spacing, or hybrid macro–micro 8 × 8 contacts with 10-mm spacing and 64 integrated microcontacts with 5-mm spacing), linear strips (1 × 8/12 contacts), depth electrodes (1 × 8/12 contacts) or a combination thereof. Recordings from grid, strip and depth electrode arrays were done using a Natus Quantum amplifier or a Neuralynx Atlas amplifier. A total of 4,057 electrodes (892 grids, 346 strips and 2,819 depths) were implanted across 32 patients with drug-resistant focal epilepsy undergoing clinically motivated invasive monitoring. A total of 3,512 electrodes (780 grids, 307 strips and 2,425 depths) that were unaffected by epileptic activity, artefacts or electrical noise were used in subsequent analyses. To determine the electrode localization for each patient, a post-operative computed tomography scan and a pre-operative T1 MRI were acquired and co-registered.

**MEG data acquisition.** MEG was acquired using a 306-sensor TRIUX MEGIN system, comprising 204 planar gradiometres and 102 magnetometres in a helmet-shaped array. The MEG gantry was positioned at 68° for optimal coverage of frontal and posterior brain areas. Simultaneous EEG was recorded using an integrated EEG system and a 64-channel electrode cap (EEG data are not reported here, but are included in the shared dataset). During acquisition, MEG and EEG data were bandpass filtered (0.01 and 330 Hz) and sampled at 1,000 Hz. The location of the head fiducials, the shape of the head, the positions of the 64 EEG electrodes and the head position indicator (HPI) coil locations relative to anatomical landmarks were collected with a 3D digitizer system (Polhemus Isotrack). ECG was recorded with a set of bipolar electrodes placed on the chest of the participant. Two sets of bipolar electrodes were placed around the eyes (two at the outer canthi of the right and left eyes and two above and below the centre of the right eye) to record eye movements and blinks (EOG). Ground and reference electrodes were placed on the back of the neck and on the right cheek, respectively. The head position of participants on the MEG system was measured at the beginning and end of each run, and also before and after each resting period, using four HPI coils placed on the EEG cap, next to the left and right mastoids and over left and right frontal areas.

**Anatomical MRI data acquisition.** For source localization of the MEG data with individual realistic head modelling, a high-resolution T1-weighted MRI volume (3 T Siemens MRI Prisma scanner) was acquired per participant. Anatomical scans were acquired either with a 32-channel coil (repetition time (TR)/echo time (TE) = 2,000/2.03 ms; inversion time (TI) = 880 ms; 8° flip angle; field of view = 256 × 256 × 208 mm; 208 slices; 1-mm isotropic voxels, UB) or a 64-channel coil (TR/TE = 2,530/2.98 ms; TI = 1,100 ms; 7° flip angle; field of view = 224 × 256 × 192 mm, 192 slice, 0.5 × 0.5 × 1 mm voxels, PKU). The FreeSurfer standard template was used (fsaverage) for participants lacking an anatomical scan (n = 5).

**fMRI data acquisition.** MRI data were acquired using a 32-channel head coil on a 3 T Prisma scanner. A session included high-resolution anatomical T1-weighted MPRAGE images (GRAPPA acceleration factor = 2, TR/TE = 2,300/3.03 ms, 8° flip angle, 192 slices, 1-mm isotropic voxels), and a whole-brain T2*-weighted multiband-4 sequence (TR/TE = 1,500/39.6 ms, 75° flip angle, 68 slices, voxel size of 2 mm isotropic, anterior/posterior (A/P) phase-encoding direction, field of view = 210 mm, bandwith (BW) = 2,090 Hz px$^{-1}$). A single-band reference image was acquired before each run. To correct for susceptibility distortions, additional scans using the same T2*-weighted sequence, but with inverted phase-encoding direction (inverted readout/phase-encoding (RO/PE) polarity), were collected while the participant was resting at multiple points throughout the experiment.

### Preprocessing and analysis details
For readability, we first detail the preprocessing protocols for each of the modalities (iEEG, MEG and fMRI) separately. Then, we describe the different analyses, combining information across the modalities, while noting any differences between them.

### Analysis strategy
As part of our testing framework, after excluding a limited number of participants due to data quality checks, we conducted an initial optimization phase on one-third of the MEG ($n$ = 32) and fMRI ($n$ = 35) datasets to evaluate data quality across sites and to optimize analysis pipelines. Following the optimization phase, pipelines were preregistered[27] and applied to the novel datasets containing twice as much data (MEG $n$ = 65 and fMRI $n$ = 73).

In the main paper, we report results obtained on the novel, previously unexamined datasets. For iEEG, given the smaller sample, a different analysis strategy was implemented. We refer the reader to the iEEG methods section and text in the main paper for numbers of participants that were entered in each analysis. Results from the optimization phase are reported in section 4 of Supplementary Information. The results of the optimization phase and the preregistered replication phase were compared and deemed to be largely compatible, with some minor exceptions (section 4 of Supplementary Information).

**iEEG preprocessing.** Data were converted to BIDS[67] and preprocessed using MNE-Python (v0.24)[68], and custom-written functions in Python and Matlab. Preprocessing steps included downsampling to 512 Hz, detrending, bad channel rejection, line noise and harmonic removal, and re-referencing. Electrodes were re-referenced to a Laplacian scheme[69], whereas bipolar referencing was used for electrodes at the edge of a strip, grid or stereo EEG, and the signal was localized at the midpoint (Euclidean distance) between the two electrodes. Electrodes with no direct neighbours were discarded. Seizure-onset zone electrodes, those localized outside the brain and/or containing no signal or high amplitude noise level were discarded. Line noise and harmonics were removed using a one-pass, zero-phase non-causal band-stop FIR filter.

The high-gamma power (70–150 Hz) was obtained by bandpass filtering the raw signal in eight successive 10-Hz-wide frequency bands, computing the envelope using a standard Hilbert transform, and normalizing it (dividing) by the mean power per frequency band across the entire recording. To produce a single high-gamma envelope time series, all frequency bands were averaged together[70]. Most analyses focused on the high-gamma power as it closely correlated with neural spiking activity[71] and with the BOLD signal[37]. To obtain the event-related potentials (ERPs), the raw signal was low-pass-filtered at 30 Hz with a one-pass, zero-phase, non-causal low-pass FIR filter. Epochs were segmented between 1-s pre-stimulus until 2.5-s post-stimulus of interest.

**Surface reconstruction and electrode localization.** Electrode positions were determined based on a computed tomography scan coregistered with a pre-implant T1-weighted MRI. A 3D reconstruction of the brain of each patient was computed using FreeSurfer (http://surfer.nmr.mgh.harvard.edu). For visualization, the electrode positions for individual participants were converted to the Montreal Neurological Institute (MNI)152 space. As each theory specified a set of anatomical ROIs, after electrode localization, electrodes were labelled according to the Freesurfer-based Destrieux atlas segmentation[72,73] and/or Wang atlas segmentation[74].

**Identification of task-responsive channels.** To identify task-responsive electrodes, we computed the area under the curve (AUC) for the baseline (−0.3 to 0 s) and the stimulus-evoked period (0.05–0.35 s) separately for the task-relevant and task-irrelevant conditions, and compared them per electrode using a Wilcoxon sign-rank test, corrected for false discovery rate (FDR)[75]. A Bayesian $t$-test[76] was used to quantify evidence for non-responsiveness.

**Identification of category-selective channels.** To determine category selectivity for faces, objects, letters and false fonts in the high gamma, we followed the method of Kadipasaoglu and colleagues[77]. Per category, we computed a $d'$ (AUC of 0.05–0.4 s) comparing the activation between the category of interest ($u_j$) and each of the other categories ($u_i$), normalized by the standard deviation of each category:

$$d' = \frac{u_j - \frac{1}{N}\sum_i^N u_i}{\sqrt{\frac{1}{2}\left(\sigma_j^2 + \frac{1}{N}\sum_i^N \sigma_i^2\right)}} \; ; i \neq j$$

A permutation test (10,000 permutations) was used to evaluate significance. $d'$ was computed for the task-relevant and task-irrelevant conditions separately. An electrode was considered selective if it showed selectivity on both tasks.

**Multivariate analysis electrodes combination.** Owing to the sparse and highly variable coverage of iEEG data, all collected electrodes were combined into a 'super participant' multivariate analyses (RSA and decoding). To create a single-trial matrix for the super participant, we equated the trial matrices of all our participants by subsampling to the lowest number of trials in the relevant conditions. Participants that did not complete the full experiment were discarded ($n$ = 3), resulting in a total of 29 participants with 583 electrodes in posterior ROIs and 576 electrodes in prefrontal ROIs. For analyses on stimuli identities, stimuli that were presented less than three times to any of the participants across intermediate and long trials in the task-relevant and task-irrelevant trials were discarded. We then subsampled the trials for each identity to three trials per participant. The subsampling procedure was repeated 100 times to avoid random fluctuations induced by the subsampling. The analysis was computed for each repetition and average across repetitions.

**MEG preprocessing.** The MEG data were converted to BIDS[78] using MNE-BIDS[79], and preprocessed following the FLUX Pipeline[80] in MNE-Python (v0.24.0)[68]. Preprocessing steps included MEG sensor reconstruction using a semi-automatic detection algorithm and signal-space separation[81] to reduce environmental artefacts. FastICA[82] was used to detect and remove cardiac and ocular components from the data for each participant (mean = 2.90 components, s.d. = 0.92). Before ICA, data were segmented, and segments containing muscle artefacts were removed. After preprocessing, data were epoched into 3.5-s segments (1-s pre-stimulus to 2.5-s post-stimulus onset). Trials in which gradiometre values exceeded 5,000 fT cm$^{-1}$, magnetometres exceeded 5,000 fT and/or the trial contained muscle artefacts were rejected from the MEG dataset. Finally, to be included in the analyses, participants should have a minimum of 30 clean trials per condition. No participants were excluded because of not meeting this criterion.

**Source modelling.** MEG source modelling was performed using the dynamic statistical parametric mapping method[83], based on depth-weighted minimum-norm estimates (MNEs)[84,85], on epoched and baseline (−0.5 s to 0 s before stimulus onset) corrected data. To build a forward model, the MRI images were manually aligned to the digitized head shape. A single shell boundary elements model was constructed in MNE-Python based on the inner skull surface derived from FreeSurfer[72,73], to create a volumetric forwards model (5-mm grid) covering the full-brain volume. The lead field matrix was then calculated according to the head position with respect to the MEG sensor array. A noise covariance matrix for the baseline and a covariance matrix for the active time window were calculated and the combined (that is, sum) covariance matrix was used with the forwards model to create a common spatial filter. Data were spatially pre-whitened using the covariance matrix from the baseline interval to combine gradiometre and magnetometre data[86].

**fMRI preprocessing.** Source DICOM data were converted to BIDS using BIDScoin (v3.6.3)[87]. This includes converting DICOM data to NIfTI using dcm2niix[88] and creating event files using custom Python codes. BIDS compliance of the resulting dataset was controlled using BIDS-Validator. Subsequently, MRI data quality control was performed using MRIQC (0.16.1)[89] and custom scripts for data rejection. All (f)MRI data were preprocessed using fMRIPrep (20.2.3)[90], based on Nipype (1.6.1)[91]. For further details on the fMRIprep pipeline, see preregistration. Custom scripts used NumPy (1.19.2)[92] and Pandas (1.1.3)[93].

**Analysis-specific functional preprocessing.** Additional, analysis-specific, fMRI data preprocessing was performed using FSL 6.0.2 (FMRIB Software Library)[94], Statistical Parametric Mapping (SPM 12) software[95], and custom Python scripts (using NiBabel (3.2.2)[96] and SciPy (1.8.0)[97] after the above-outlined general preprocessing. Functional data for univariate data analyses were spatially smoothed (Gaussian kernel with full-width at half-maximum of 5 mm), grand mean scaled and temporal high-pass filtered (128 s). No spatial smoothing was applied for multivariate analyses.

**Contrast of parameter estimates.** We modelled BOLD signal responses to the experimental variables by fitting voxel-wise general linear model (GLM) to the data of each run using FSL FEAT. The following regressors were modelled in an event-related approach, with event duration corresponding to the stimulus duration (that is, 0.5 s, 1.0 s and 1.5 s), and convolved with a double gamma haemodynamic response function: 12 regressors of interest (targets, task-relevant and task-irrelevant stimuli per stimulus category, that is, faces, objects, letters and false fonts; and a regressors of no interest, that is, target screen display). We included the first-order temporal derivatives of the regressors of interest, and a set of nuisance regressors: 24 motion regressors (FMRIB Software Library (FSL)'s standard + extended set of motion parameters) plus a cerebrospinal fluid (CSF) and a white matter (WM) tissue regressor. Each of the 12 regressors of interest was contrasted against an implicit baseline (used in the putative Neural Correlates of Consciousness analysis; see below). In addition, we obtained contrast of parameter estimates for 'relevant faces versus relevant objects', 'relevant letters versus relevant false fonts', 'irrelevant faces versus irrelevant objects', 'irrelevant letters versus irrelevant false fonts' (used for the definition of decoding ROIs), 'relevant and irrelevant faces versus relevant and irrelevant objects' and 'all stimuli versus baseline' (used for the definition of seeds for the generalized psychophysiological interaction (gPPI) analysis). Data were averaged across runs per participant using FSL's fixed-effects analysis and subsequently averaged across participants using FSL's FLAME1 mixed-effect analysis. Gaussian random-field cluster thresholding was used to correct for multiple comparisons, using the default settings of FSL, with a cluster formation threshold of one-sided $P < 0.001$ ($z \geq 3.1$) and a cluster significance threshold of $P < 0.05$.

**Anatomical ROIs.** ROIs were defined a priori in consultation with the adversarial theories. They were determined per participant based on the Destrieux atlas[73] including both hemispheres, and then resampled to standard MNI space (see Supplementary Table 26). For the connectivity analysis, areas V1/V2 (combining dorsal and ventral) were defined based on the Wang cortical parcellation[74]. For details on the process of selecting the ROIs and the justification of the ROI selection in the context of this study, see section 10 in Supplementary Information. All anatomical segmentations were performed using Freesurfer (6.0.1)[72].

**Behavioural analyses.** Log-linear-corrected $d'$ (ref. 98), false alarms and reaction times were computed per category and stimulus duration, separately (false alarms were also calculated per task relevance, without duration) and per modality (iEEG, MEG and fMRI). These measures were compared with linear–logistic mixed models, where appropriate. For the former, we report analysis of variance omnibus $F$-tests, and for the latter, omnibus $\chi^2$ test from an analysis of deviance. We approximated degrees of freedom using the Satterthwaite method[99]. Pairwise $t$-tests following significant interactions were Bonferroni corrected. To estimate Bayesian information criterion (BIC) differences between the original and null logistic models, we used the $P$ values and sample size[100] (p_to_bf package in R).

**Eye-tracking analyses.** For Eyelink, gaze and pupil data were segmented, and trials with missing data were excluded. Blinks were detected using the Hershman algorithm[101], and removed with 200-ms padding[102]. The Eyelink standard parser algorithm was used for saccade and fixation detection. Saccades were further corroborated using the Engbert and Kliegl[103] algorithm. Fixations were baseline corrected (−0.25 s to 0 s). Mean fixation distance, mean blink rate, mean saccade amplitude and mean pupil size were compared in a LMM with category and task relevance as fixed effects, and participant and item as random effects. Separate analyses were carried out on the first 0.5 s after stimulus onset including all trials; and on the 1.5-s trials including time window (0–0.5 s, 0.5–1.0 s and 1.0–1.5 s) as fixed effects. BIC was used to test the models against the null hypothesis models. For Tobii, gaze coordinate data were segmented, missing data were excluded and coordinates were baseline corrected to depict heatmaps of patients' gaze. Of note, the coordinate data were not added to the LMMs due to its poorer quality with respect to the EyeLink data.

**Decoding analysis.** All decoding analyses were performed using a linear support vector machine (SVM; scikit learn (0.23.2), https://scikit-learn.org/) classifier. Below, we explain how this was done for each one of the predictions.

iEEG decoding was done on the high-gamma signal, averaged over non-overlapping windows of 0.02 s separately for electrodes located in the GNWT and IIT ROIs. The top 200 electrodes (selectKbest[104]), as determined by a $F$-test within a given set of electrodes from the theory ROIs, were used as features for the classifier. Two-hundred features were selected to provide a balance between model optimization (for example, feature selection) and participant representation (for example, electrodes or features coming from multiple participants). Statistical significance of decoding performance was assessed via permutation test, randomly permuting the sample labels and repeating the decoding analysis 1,000 times, corrected for multiple comparisons using a cluster-based correction (cluster mass inference with cluster forming threshold at $P < 0.05$)[105,106]. Also, to assess the decoding accuracy within unique ROIs (for example, S_temporal_sup of the Destrieux atlas), separate classifiers were trained using all electrodes in a given parcel. Each classifier was fitted using all electrodes in a parcel and time window (GNWT: 0.3–0.5 s, IIT: 0.3–1.5 s) as features, resulting in a single accuracy value per parcel. SelectKbest (200 features for iEEG) feature selection and fivefold cross-validation with three

repetitions was used. To assess the statistical significance of the decoding accuracy within unique ROIs (so only one accuracy score is obtained per ROI), *P* values obtained via permutation tests were corrected for multiple comparisons across all ROIs using FDR correction ($q \leq 0.05$[75]). To compute Bayes factors on the decoding accuracy values, we used a β-binomial approach that compares the marginal likelihood under a point-null hypothesis against a flat $B(\alpha = 1, \beta = 1)$ alternative prior, yielding an analytic Bayes factor. We then derived the null hypothesis parameters from the empirical null distribution by updating a tight prior centred at chance level ($B(\alpha = 1,000, \beta = 1,000)$) with the shuffle-based accuracies, thereby incorporating any bias present in the null distribution.

MEG decoding was done on bandpass-filtered (1–40 Hz) and downsampled (100 Hz) data. The reconstructed source-level MEG data within a subset of the predefined anatomical ROIs (GNWT: 'G_and_S_cingul-Ant', 'G_and_S_cingul-Mid-Ant', 'G_and_S_cingul-Mid-Post', 'G_front_middle', 'S_front_inf', 'S_front_sup'; IIT: 'G_cuneus', 'G_oc-temp_lat-fusifor', 'G_oc-temp_med-Lingual', 'Pole_occipital', 'S_calcarine', 'S_oc_sup_and_transversal', as they showed high response to the stimulus on the optimization dataset) were extracted for further analysis (500 vertices and 800 vertices per hemisphere for each of the anatomical ROI defined by the theories). We applied temporal smoothing (0.05-s window, 0.01-s sliding window), computed pseudotrials[107], normalized the data and selected the top 30 features within a given ROI as features for the different classifiers. A group-level one-sample *t*-test per time point was performed on the decoding accuracy results, corrected for multiple comparisons using a cluster-based correction[106].

The overall decoding strategy for fMRI was similar to that used on the iEEG and MEG data, yet with some differences. A multivariate pattern analysis approach was used on the pattern of BOLD activity over voxels. A non-spatially smoothed parameter estimate map was obtained by fitting a GLM per event with that event as the regressor of interest and all the other remaining events as one regressor of no interest[108] as implemented in NiBetaSeries (0.6.0) package. The model also included the 24 nuisance regressors described in the 'fMRI preprocessing' section.

Decoding was performed using whole-brain and ROI-based approaches. The whole-brain analysis was performed using a searchlight approach with 4-mm radius. For ROI-based decoding, decoding ROIs were defined based on functional fMRI contrasts (see the 'fMRI preprocessing' section) and constrained with pre-defined anatomical ROIs (see Extended Data Table 2 on anatomical ROIs). A one-sample permutation test was used to determine whether decoding significantly exceeded chance level within each ROI. FDR was used to correct for multiple comparisons across ROIs. For whole-brain decoding, a cluster-based permutation test was used to evaluate the decoding statistical significance across participants ($P < 0.05$), complemented by Bayesian analysis. In addition, stimulus versus baseline searchlight decoding was performed using leave-one-run out cross-validation, and the resultant decoding accuracy maps were used as input for the multivariate putative NCC analysis (see below). To perform stimulus versus baseline decoding, we subsampled the stimuli trials to a 2:1 ratio with respect to baseline. The SVM cost function was weighted by the number of trials from each class. Plots were generated using Matplotlib (3.3.2)[109].

**Decoding schemes for the different predictions.** To test GNWT and IIT decoding predictions, stimulus category (faces versus objects and letters versus false fonts) was decoded separately for the task-relevant and task-irrelevant conditions (within-task category decoding), whereas orientation (front view versus left view versus right view) was decoded on the combined data from the two task conditions. In addition, cross-task category decoding from the task-relevant to task-irrelevant condition and vice versa was performed to test generalization by training classifiers on one condition and testing on the other condition. Both within-task category and orientation decoding were performed in a leave-one-run-out cross-validation scheme for fMRI and in an *k*-fold cross-validation scheme for MEG and iEEG.

For category decoding, trials from each task condition (that is, task relevant or task irrelevant) were extracted for each category comparison of interest: 160 face/160 objects classification, 160 letters/160 false-fonts classification within each task-relevant condition for MEG, and half the trials for iEEG. For fMRI, there were 64 trials for each category in each task-relevant condition. For orientation decoding, task-relevant and task-irrelevant trials were collapsed within category to increase the signal-to-noise ratio, resulting in 160 front, 80 left and 80 right trials per category for MEG, and half these numbers for iEEG. For fMRI, there were 64 front and 32 left and right trials per category. Decoding was evaluated using accuracy measures, tested against 50% chance level for category decoding (binary classification) and against 33% chance level for orientation decoding (three-class classification). For orientation decoding, balanced accuracy was used due to the unbalanced number of trials for the different orientations. The SVM cost function was weighted by the number of trials per class to reduce bias to the class with the highest number.

$$\text{Balanced accuracy} = \frac{1}{3}(\text{Sensitivity}_{\text{front}} + \text{Sensitivity}_{\text{right}} + \text{Sensitivity}_{\text{left}})$$

For within-task decoding (for example, classification of categories across time), a classifier at each time point was trained and tested separately using a fivefold cross-validation (with three separate repeats of cross-validation). For cross-task decoding (task relevant → irrelevant and task irrelevant → relevant), each SVM model was trained on one task (for example, faces–objects in the task-relevant condition) and tested on the second task (for example, faces–objects in the task-irrelevant condition). As cross-decoding in iEEG data is performed across all pooled electrodes, an additional cross-validation step was performed on this modality data to provide a confidence metric (for example, confidence intervals) using a fivefold cross-validation with three repetitions (for example, train on 80% of task 1, and test on held-out 20% of task 2).

Within-task temporal generalization was performed by training a classifier at each time point (using selectKbest feature selection) and testing its performance across all time points using the same set of selected features and three repetitions of fivefold cross-validation. To generalize from one task to another across all time points, cross-temporal generalization was used: a classifier was trained at each time point in task 1 (for example, task relevant) using selectKbest feature selection, and tested across all time points in task 2 (for example, task irrelevant) using the same set of selected features. Cross-validation was performed in the same manner as in cross-decoding.

Additional decoding analyses were performed on all trials aligned to the stimulus onset (for example, −0.2 to 2 s relative to stimulus onset) and stimulus offset (−0.5 to 0.5 s around stimulus offset). For the latter analysis, all trials from different durations were aligned to the stimulus offset.

To assess the prediction of IIT that included prefrontal regions along with posterior regions to the decoding of categories will not significantly affect decoding accuracy, we performed an additional decoding analysis in which the decoding performance of electrodes from the IIT region were compared with the decoding performance when electrodes from both the posterior + PFC ROIs are included. The PFC ROIs included all PFC ROIs except for inferior frontal sulcus, as it belongs to the IIT extended ROIs. Posterior ROI included all IIT ROIs shown in Supplementary Table 26. The analysis compared the decoding accuracy for a model including all electrodes from posterior regions to a separate model in which electrodes (features) from posterior and PFC regions were combined (for example, feature combination). Training and testing of the individual models followed all previously described cross-validation procedures, and model comparison was performed

using a variance-corrected paired $t$-test[110] and complemented with Bayesian analysis.

We also tested this prediction on the fMRI data. To select features to be used for both analyses, the face versus object contrast for each participant was masked by a predefined anatomical posterior ROIs as well as PFC anatomical ROIs, defined the same way as described above. Within each of the two ROIs, the 150 voxels that are most selective to each of the to-be-decoded stimuli were defined as the decoding ROIs (300 voxels total) for each participant. The first analysis compared the decoding accuracies for a model that included 300 voxels from the posterior ROIs as features to another model that included 600 voxels (300 features from each ROI). In the second analysis, two separate models were constructed, calibrated and combined as described above. For the two analyses, model comparison was performed using a group-level one-sample permutation test to determine if accuracies obtained by combining posterior and PFC ROIs were significantly higher than the accuracies obtained based on posterior ROIs only. FDR was used to correct for multiple comparisons. Bayesian analysis was performed to quantify evidence for the null hypothesis that adding prefrontal ROIs will not improve decoding accuracy.

**Duration analysis.** Neural responses were extracted from three windows of interest (0.8–1.0 s, 1.3–1.5 s and 1.8–2.0 s) and compared using LMMs. Four theory agnostic models were fitted: a null model, a duration model (three durations), a windows of interest model, and a duration and windows of interest model. Two theory models were fitted: the GNWT model predicts activation (ignition) following stimulus offset (0.3–0.5 s) independent of duration, with virtually no response in between. The IIT model predicts sustained activation for the duration of the stimulus returning to baseline after stimulus offset. Both theoretical models were complemented with an interaction term between category (faces, objects, letters and false fonts) and the theories' predictors, to account for regions showing selective responses to categories. BIC was used to define the winning model and we computed Bayes factors based on the difference in BIC values, comparing the GNWT model (with or without interaction) against either the null model (intercept only) or the time-window model (capturing amplitude changes over time)[111].

Models for iEEG were fitted per electrode on the predefined ROIs, using the high-gamma (AUC), alpha (8–13 Hz, obtained through Morlet wavelets, $f = 8$–13 Hz, in 1-Hz steps; $f/2$ cycles, AUC), and ERPs (peak to peak) as signal, separately for task-relevant and task-irrelevant condition.

MEG models were fitted to source data on the predefined ROIs, using the gamma (60–90 Hz) and alpha (8–13 Hz) bands as signal, separately for task-relevant and task-irrelevant conditions. Time-frequency analyses were performed on source-data using Morlet wavelets ($f = 8$–13 Hz, in 1-Hz steps; $f/2$ cycles; $f = 60$–90 Hz, in 2-Hz steps; $f/4$ cycles) and were baseline corrected. Spectral activity was computed for each vertex, baseline corrected and then averaged across trials within each parcel included in the ROIs, yielding a unique time course per ROI parcel. In addition, a single-source time course capturing the entire prefrontal ROI and the posterior ROI was computed by averaging the spectral activity within an ROI. Models were fitted on each parcel and ROI, as defined by the theories.

**Representational similarity analysis.** To examine how the neural representations evolved over time in response to the different stimulus properties (that is, category, orientation and identity representation), we performed cross-temporal RSA on source-level MEG data and iEEG high-gamma power within each of the theory-defined ROIs, using all trials. Specifically, at each set of data points, we computed a representational dissimilarity matrix (RDM) by calculating the correlation distance (1 − Pearson's $r$, Fisher corrected) between all pairs of stimuli (the pre-registration document described a different method that was however updated to optimize trial numbers; see section 14 in Supplementary Information for justification). Next, to quantify the representational space occupied by one class versus another, we computed the average within-class distances versus the average between-class distances. This analysis was performed in a cross-temporal manner, in which RDMs were computed between all stimuli at time point $t_1$ and the corresponding set of stimuli at time points $t_1, t_2, …t_n$.

Long trials (1.5 s) were used to investigate category and orientation representation. As specific identities were repeated a limited number of times per duration, both intermediate (1.0 s) and long (1.5 s) trials were combined and equated in duration by cropping the 1–1.5-s time interval for long trials. This was done to allow for the analysis of at least three (3) presentations of the same identity.

To evaluate the theoretical predictions about when significant content representation should occur, we subsampled the observed cross-temporal representational matrices in four time windows (0.3–0.5 s, 0.8–1.0 s, 1.3–1.5 s and 1.8–2.0 s). The subsampled matrices were correlated to the model matrices predicted by GNWT and IIT (see Fig. 1a, right panel) using Kendall's tau correlation. If the correlation was significant (see below) for at least one of the predicted matrices, we computed the difference between the transformed correlation $((r + 1)/2)$ to each theory, and compared this difference against a random distribution to obtain a $P$ value. If the correlation with the theory-predicted pattern in the theory ROI was significantly higher than the other model, we considered the theory prediction to be fulfilled.

To generate a null distribution of cross-temporal RSA surrogate matrices, we repeated the procedure outlined above 1,024 times, randomly shuffling the labels. Next, the observed RSA matrix was $z$-scored using the null distribution as:

$$z_{i,j} = \frac{\text{obs}_{i,j} - \mu_{\text{surr}_{i,j}}}{\sigma_{\text{surr}_{i,j}}}$$

Where $\text{obs}_{i,j}$ is the observed within-versus-between class difference at time points $i$ and $j$, and $\mu_{\text{surr}_{i,j}}$ and $\sigma_{\text{surr}_{i,j}}$ are the mean and standard deviation of the surrogate representational similarity matrix at time points $i$ and $j$, respectively. Cluster-based permutation tests[112], $z$-score threshold of $z = 1.5$ for clustering, were used to evaluate significance. RSA surrogates were also used to assess the significance of the correlation between the observed matrices and the predicted matrices of the theories. First, a null distribution of possible correlations was generated for each of the theories by correlating each of the surrogate matrices to each of the theory-predicted matrices. Next, a $P$ value was obtained for each theory-predicted matrix, by locating its observed correlation within the null correlation distribution. The same procedure was used to assess the significance of the difference in correlation to IIT and GNWT matrices (for example, each of the surrogate matrices was correlated to each of the theory-predicted matrices and the difference between the two was computed). $P$ values were FDR corrected ($q \le 0.05$)[75].

For iEEG, the high-gamma power per electrode within the predefined anatomical ROI was averaged in 0.02-s non-overlapping windows. Electrodes were used as features for the RDM. The data were vectorized across all electrodes within a ROI (for example, samples × significant electrodes) to compute the RDMs. A total of 576 and 583 electrodes entered this analysis for the prefrontal and posterior ROI, respectively. The resultant RDM was subjected to a PCA, and the first two dimensions were plotted against each other to produce a 2D projection of dissimilarity scores across all pairs for each of the 100 subsampling repetitions. The PCA components were aligned across repetitions using Procrustes alignment and averaged together for visualization purposes[113,114].

For MEG, the same analysis was run on the source reconstructed data within the predefined anatomical ROIs used for the decoding analysis, bandpass filtered (1–40 Hz) and downsampled (100 Hz). For the category and orientation analysis, pseudotrials and temporal moving-average methods were used to optimize the RSA analysis and improve the signal-to-noise ratio. For identity, single trials were used.

Vertices within the ROIs were used as features. The statistical testing differed from that conducted on the iEEG data, as it was performed at the participant level. Similarly to the iEEG analysis, we first tested whether the correlation between the data and the model predicted by each theory was greater than zero using the Kendall's tau measure, and then compared between the theories using the Mann–Whitney $U$ rank test on two independent samples.

**Functional connectivity analysis.** For both iEEG and MEG, PPC[46] was computed between each category-selective time series (face selective and object selective) and either the V1/V2 or the PFC time series.

For iEEG, the PPC analysis included electrodes in V1/V2 visual areas, in PFC ROIs (see Supplementary Table 26), and face-selective and object-selective electrodes (see 'Identification of task-responsive channels'), as long as they were 'active' during the task. As both theories predict different types of activation (for example, ignition versus sustained activation), channels were categorized as active if they showed an increase in high-gamma power relative to baseline (−0.5 to −0.3 s, $P < 0.05$, signed-rank test) evaluated across all trials (task relevant + irrelevant, intermediate + long trials, combined across both categories), for the 0.3–0.5-s window (GNWT), or in all time windows: 0.3–0.5 s, 0.5–0.8 s and 1.3–1.5 s (IIT).

For MEG, the category-selective single-trial time courses used to define the ROIs for PPC analysis were extracted using the generalized eigenvalue decomposition (GED) method[115]. Two GED spatial filters were built by contrasting either faces or objects against all other categories during the first 0.5 s after stimulus onset. Single-trial covariance matrices were computed separately for signal and reference for all vertices within the fusiform ROI identified from the FreeSurfer parcellation using the Desikan atlas[116], and the Euclidean distance between them was $z$-scored. Trials exceeding 3 $z$-scores were excluded. The reference covariance matrix was regularized to reduce overfitting and increase numerical stability. The GED was then performed on the two covariance matrices, resulting in $n$ (=rank of the data) pairs of eigenvectors and eigenvalues. The eigenvector associated with the highest eigenvalue was selected as a GED spatial filter, which in turn was applied to the data to compute the single-trial GED component time series. A GED spatial filter was extracted also for the PFC ROI, on parcels from the Destrieux atlas[73], to identify the distributed pattern of sources that are responsive to visually presented stimuli. Specifically, a spatial filter was built by contrasting source-level frontal slow-frequency activity (30-Hz low-pass filter) after stimulus onset (0–0.5 s) against baseline (−0.5 to 0 s). V1/V2 areas were identified using the Wang Atlas[74] and a singular values-decomposition approach. For the GED, the 1.0-s and 1.5-s duration trials were used to minimize overlap with the transient evoked at stimulus onset.

PPC was computed for each MEG time series–iEEG electrode pairing, for all face trials and object trials separately. Analyses were performed on 1.0-s and 1.5-s duration trials, separately on task-relevant and task-irrelevant trials and also combined to maximize statistical power. To compute synchrony, time-frequency analysis of the broadband MEG and LFP signal was performed using Morlet wavelets ($f = 2$–30 Hz, in 1-Hz steps; 4 cycles; $f = 30$–180 Hz for iEEG or $f = 30$–100 Hz for MEG, in 2-Hz steps, f/4 cycles), and PPC was then computed by taking the difference in phase angle between MEG time series–iEEG electrode at each time $t$ and frequency $f$ for a specific trial and computing PPC across all trials in a category (for example, faces) as:

$$PPC(f, t) = \frac{2}{N(N-1)} \sum_{j=1}^{N-1} \sum_{k=j+1}^{N} \cos(\theta_j(f, t) - \theta_k(f, t)), j = \{1, \dots, N \text{ trials}\}$$

$\theta_{j,k}(f, t) = \theta(f, t)_{e1 \text{ or GED filter}} - \theta(f, t)_{e2 \text{ or GED filter}}$, for all frequencies $f$ and at all times $t$.

For iEEG, PPC for each category-selective site was then averaged across all its pairings (for example, all PFC electrodes pairings or all

V1/V2 pairings within that patient). The variability in electrode coverage across patients precluded a within-participants analysis. Therefore, to achieve sufficient statistical power, we pooled all derived PPC values from one electrode pairing (for example, face selective to the PFC) across all patients into one ROI-specific analysis. A similar approach was used on the MEG parcels.

To quantify content-specific synchrony enhancement, the difference in PPC was computed between within-category and across-category trials (for example, for face-selective sites, the change in PPC was computed between faces versus objects trials) using a cluster-based permutation test[106]. This was done for both modalities.

As an exploratory analysis, we also investigated dynamic functional connectivity using the Gaussian copula mutual information[117] approach to evaluate the dependencies between time series. This power-based measure of connectivity was implemented using the conn_dfc method from the Frites Python package[118]. We used the same parameters as for the PPC analysis, with the following exceptions: for both MEG and iEEG, power was estimated through a multitaper-based method (using a frequency-dependent dynamic sliding window: 2–30 Hz, $T = 4$ cycles; 30–100 Hz, $T4/f$ using a 0.25-s sliding window). For iEEG, the high-frequency range was extended from 30 to 180 Hz, $T = 4/f$ cycles). DFC was performed per frequency band, 0.1-s sliding window and 0.02-s steps.

For fMRI, connectivity was assessed through gPPI implemented in SPM[119]. The FFA and lateral occipital cortex were defined as seed regions per participant based on an anatomically constrained functional contrast. Anatomically, FFA seeds were constrained to the 'inferior occipital gyrus (O3) and sulcus' and 'lateral occipito-temporal gyrus (fusiform gyrus; O4–T4)'. LOC seeds were constrained to the 'middle occipital gyrus (O2; lateral occipital gyrus)' and the 'middle occipital sulcus and lunatus sulcus' (Destrieux ROIs 2 and 21 for FFA, and ROIs 19 and 57 for LOC; see 'Anatomical ROIs').

Candidate seed voxels within the above-mentioned anatomical ROIs were defined as those with $z > 1$ in the contrast of parameter estimates of all stimuli versus baseline. Three participants with less than 300 candidate seed voxels were excluded from the analysis. This was done to ensure that the seed voxels were visually driven. Next, using an unthresholded contrast of parameter estimates between 'relevant and irrelevant faces' and 'relevant and irrelevant objects', the 300 voxels most responsive to faces within the FFA anatomical ROIs were selected for the FFA seed, and the 300 voxels most responsive to objects within the LOC anatomical ROIs were selected for the LOC seed.

gPPI analysis was performed per participant and seed region separately, including an interaction term between the seed time-series regressor (physiological term) and the task regressor (psychological term) at the participant-level GLM[119], separately for task-relevant and irrelevant conditions, and also combining across tasks to increase statistical power. For combined conditions, the model design matrix for each participant included regressors for task-relevant and task-irrelevant faces, objects, letters and false fonts collapsed across conditions (four regressors) as well as a regressor for targets (irrespective of their category), yielding five regressors in total. As for separated conditions, the model design matrix included regressors for task-relevant and task-irrelevant faces, objects, letters and false fonts (eight regressors) as well as a regressor for targets (irrespective of their category), yielding nine regressors in total. For each seed, group-level analysis was performed using a cluster-based permutation test (preferred over the preregistered FDR correction), complemented by Bayesian analysis. See section 14 in Supplementary Information for a justification of this change to evaluate the statistical significance of face > object contrast parameter estimates across participants ($P < 0.05$).

**Putative NCC analyses.** A series of conjunction analyses were performed on the fMRI data to identify (1) areas responsive to task goal, (2) areas responsive to task relevance, and (3) areas putatively involved

in the neural correlates of consciousness. We note that the contrasts proposed below might overestimate the neural correlates of consciousness and that the fast-event-related design adopted here might be suboptimal to detect activity changes in the salience network[120], that is, potentially underestimating some regions that might be involved in conscious processing. We therefore have adopted a conservative approach that distinguishes between areas that might participate in consciousness versus those that definitely do not.

The conjunction defining areas responsive to task goals was defined as [TaskRelTar > bsl] and [(TaskRelNonTar = bsl) and (TaskIrrel = bsl)]. This contrast captures areas that show an increase of BOLD signal for targets but not for other stimuli. The following conjunction identified areas responsive to task relevance: [(TaskRelTar > bsl) and (TaskRelNonTar ≠ bsl)] and [TaskIrrel = bsl]. This contrast identifies areas displaying differential activity for all task-relevant stimuli, but are insensitive to non-task-relevant stimuli. Finally, the following conjunction was used to identify the putative NCC areas: [(TaskRelNonTar (stim id) > bsl) and (TaskIrrel (stim id) > bsl)] or [(TaskRelNonTar (stim id) < bsl) and (TaskIrrel (stim id) < bsl)], critically detecting areas that are responsive to any stimulus category irrespective of task, with consistent activation or deactivation. Thus, this analysis casts a wide net to identify areas that can potentially be the neural correlates of consciousness, whereas excluding areas that do not respond to task-relevant or irrelevant stimuli (meaning that areas that respond both to the task and to the content of perception are still included).

To compute conjunctions, we first ran a GLM (see above) corrected for multiple comparisons (Gaussian random-field cluster-based inference). Equivalence to baseline was established using a JZS Bayes factor test, with a Cauchy prior ($r$ scale value of 0.707, as implemented in Pingouin (0.5.1)[121]. Evidence maps were thresholded at $BF_{01} > 3$. The thresholded $z$ maps and the Bayesian evidence maps on the group level were used for the conjunction analysis. For conjunctions including an 'unequal to', a 'logical and' operation was used between the directional $z$ maps, after thresholded maps were binarized. For the putative NCC contrast, conjunctions were performed separately for activations and deactivations, using a 'logical and' operator for the task-relevant and irrelevant $z$ maps. The resulting maps were combined using a 'logical or' operation to discard areas showing effects of opposite direction for task-relevant and task-irrelevant stimuli. This analysis was also done at the participant level, masked using the anatomical ROIs, to account for inter-participant variability. For each ROI, the proportion of participants with voxels included in the conjunction is reported. The multivariate version of the putative NCC analysis was done using the thresholded statistical maps obtained from the whole-brain searchlight decoding based on a participant-level stimulus versus baseline-decoding accuracy maps (for details regarding the decoding approach used, see 'Decoding analysis').

### Reporting summary

Further information on research design is available in the Nature Portfolio Reporting Summary linked to this article.

## Data availability

The full study protocol is available in the preregistration on the OSF webpage (https://osf.io/92tbg/)[27], including a detailed description of the experimental design, the predictions of the theories and agreed-on interpretations of the results, as well as iEEG, MEG and fMRI data acquisition details, preprocessing pipelines and data analysis procedures. Deviations from the preregistration are documented throughout the article and summarized in section 14 of the Supplementary Information. All data generated in this study are available under a CC BY 4.0 license. The M-EEG, fMRI and iEEG datasets are distributed through two methods: as downloadable data bundles and via an XNAT instance, which enables search functionality and single-participant downloads.

Data bundles can be accessed (https://www.arc-cogitate.com/data-bundles) in raw format (M-EEG raw[122], fMRI raw[123] and iEEG raw[124]) and BIDS format (M-EEG BIDS[125], fMRI BIDS[126] and iEEG BIDS[127]). Alternatively, the datasets are accessible via the Cogitate XNAT instance[128] (https://cogitate-data.ae.mpg.de). All distribution formats include robust metadata, and detailed documentation of experimental procedures and dataset structure is available (https://cogitate-consortium.github.io/cogitate-data/). For further inquiries, please contact the corresponding author.

## Code availability

Task and analysis codes have been shared under an MIT license. The task code[129] has been shared on GitHub (https://github.com/Cogitate-consortium/cogitate-experiment-code). The analysis code[130] has also been shared on GitHub (https://github.com/Cogitate-consortium/cogitate-msp1).

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

**Acknowledgements** We thank D. Potgieter for spearheading the ARC program; D. Kahneman for guidance to navigate adversarial collaborations; H. Berlin, W. Jaworski, H. Lau and C. Pennartz for insightful discussions during the 2-day meeting organized by the Templeton World Charity Foundation at the Allen Institute, Seattle in March 2018; H. Lau for helping conceptualize the proposed experiments; O. Devinsky, W. Doyle, P. Dugan and D. Friedman for supporting the recruitment and patient care at NYU; E. Yacoub, M. Kahana, P. Zeidman, K. Friston, J.-R. King, M. X. Cohen, M. A. Cohen, F. Al Roumi, S. Yuval-Greenberg, D. Draschkow and A. Brovelli for guidance on diverse data analyses; C. Schwiedrzik for insightful discussions and feedback throughout the different phases of this study (conceptualization, data analysis and initial draft); S. Brendecke and F. Bernouly for help with figures and stimulus materials; M. Smulders and S. Kusch for help with fMRI data acquisition; and the patients and their families for generously supporting this study. This research was supported by Templeton World Charity Foundation (TWCF0389 (https://www.templetonworldcharity.org/projects-resources/project-database/0389) and TWCF0486 (https://www.templetonworldcharity.org/projects-resources/project-database/0486)) and the Max Planck Society. The opinions expressed in this publication are those of the authors and do not necessarily reflect the views of the Templeton World Charity Foundation nor of the Max Planck Society.

**Author contributions** Initial conceptualization of the experiment and development of predictions were done by D.J.C., F.F., H.B., S. Dehaene, C. Koch, G.T., L. Mudrik, M.P. and L. Melloni. Development of the experimental framework, design of the adversarial collaboration and overall consortium structure was conducted by L. Mudrik, M.P. and L. Melloni. Refinement of experimental design for preregistration was done by S.B., D.J.C., R.M.C., F.F., T.I.P., H.B., O.J., F.P.d.L., H.L., M.B., S. Dehaene, C. Koch, G.T., L. Mudrik, M.P. and L. Melloni, with input from O.F., A.K., A.L., L.L., D.R., R.H. and U.G.-K. GNWT proponent was done by S. Dehaene. IIT proponents were done by G.T., M.B. and C. Koch. Funding was acquired by L. Mudrik, M.P. and L. Melloni. Pilot studies were performed by A.L., supervised by L. Melloni. Consortium management, team leadership and project management were performed by T.B., L. Mudrik, M.P. and L. Melloni. Project administration and leadership at each site were done by O.J. (Birmingham) and H.L. (Peking) for MEG site Principal Investigators; H.B. (Yale) and F.p.d.L. (DCCN) for fMRI site Principal Investigators; and S. Devore, S.H., L. Melloni (NYU), G.K. (Harvard) and M.B. (Wisconsin) for iEEG site Principal Investigators. Data release plan and metadata organization were done by N.B., P.S., T.B. and L. Melloni. Management and coordination of High Performance Computing resources were performed by A.L., T.B., P.S. and L. Melloni. Data curation was performed by L. Melloni, P.S., N.B. and T.B. Conceptualization of the experimental framework for standardization across sites was conducted by A.L., K.B., R.H., L. Mudrik and L. Melloni. The experimental framework was carried out by O.F., U.G.-K., A.K., L.L. D.R., K.B., S.M., A. Seedat, A. Sharafeldina and S.Y., along with site PIs. Stimuli were generated by A.L. and R.H. iEEG data were collected by J.J., S.M. and A. Seedat (NYU); M.A., K.B. and D.R.M. (Harvard); and U.G.-K. and C. Kozma (Wisconsin). MEG data were collected by O.F., T.G. and D.H. (Birmingham); and L.L. and S.Y. (PKU). fMRI data were collected by A.K. and A. Sharafeldin (Yale); and D.R. (DCCN). iEEG data quality were checked by U.G.-K., A.L., M.A., K.B., J.J., S.M. and A. Seedat, supervised by S.H., S. Devore, G.K. and L. Melloni. MEG data quality was checked by O.F., L.L., T.G. and D.H., supervised by O.J. and H.L. fMRI data quality was checked by A.K., D.R. and Y.V., supervised by H.B. Data quality checks for all modalities and eye-tracking data were done by U.G.-K., R.H. and C. Kozma, supervised by L. Mudrik and M.P. Dataset division for optimization and replication was performed by U.G.-K. Behavioural and eye-tracking data analyses were done by R.H., with support from C. Kozma and A.L., and supervised by L. Mudrik, M.P. and L. Melloni. iEEG decoding and synchrony analyses, unification across modalities and validation of methods were performed by S.H. iEEG preprocessing was done by A.L. and K.B. Responsiveness, RSA, activation and LMMs for theory testing were done by A.L. Supervision of the iEEG team was provided by G.K., L. Melloni and S. Devore. MEG preprocessing and source modelling were done by O.F. and L.L. Activation, connectivity and associated control analyses were performed by O.F. Decoding, RSA and associated control analyses were done by L.L. Supervision of the MEG team was provided by O.J. and H.L., with support L. Mudrik. fMRI searchlight and ROI decoding, gPPI, and stimulus versus baseline decoding were performed by A.K. Putative NCC, generation of seeds for gPPI and ROIs for decoding analyses, and definition and generation of anatomical ROIs were done by Y.V. Supervision for the fMRI team was provided by H.B. and F.P.d.L., with support by M.P. Analysis coordination across teams was done by T.B. Expert methodological advice was provided by S.B., R.M.C., T.I.P. and M.B. Visualizations were done by R.H. (behavioural and eye tracking); S.H. (iEEG decoding and synchrony); A.L. (iEEG RSA, activation, visual responsiveness, category selectivity, duration decoding and duration tracking), supervised by L. Melloni; O.F. (MEG activation and connectivity); L.L. (MEG decoding, RSA and leakage analysis), supervised by O.J., H.L. and L. Mudrik; A.K. (fMRI decoding and gPPI analyses); and Y.V. (fMRI pNCC analyses), supervised by H.B., F.P.d.L. and M.P. Overall figure integration was done by T.B., L. Mudrik, M.P. and L. Melloni. The original draft was written by L. Mudrik, M.P. and L. Melloni, supported by T.B. Discussions by adversaries of the draft were done by T.I.P., M.B., S. Dehaene, G.T. and C. Koch. The draft of the Supplementary Information was done by O.F., U.G.-K., S.H., R.H., A.K., A.L., L.L., D.R. and Y.V. All authors reviewed, edited, provided critical comments and approved the final manuscript before submission. A detailed authors contribution grid is available in Supplementary Fig. 55.

**Funding** Open access funding provided by Max Planck Society.

**Competing interests** C. Koch and G.T. are a board members and have a financial interest in Intrinsic Powers, a company developing a clinical device for assessing the presence and absence of consciousness in patients. C. Koch is the Chief Scientist of the Tiny Blue Dot Foundation in Santa Monica, CA. G.T. holds a patent for a method of assessing anaesthetization (patent no.: US 8,457,731 B2). S.Dehaene is a co-inventor on patent 2019 EP 2983586 ('Methods to monitor consciousness'); and is an associate at NeuroMeters, a company that applies these methods in clinical practice. None of these affiliations impose restrictions on publication or present conflicts of interest related to this study. The other authors declare no competing interests.

**Additional information**

**Correspondence and requests for materials** should be addressed to Lucia Melloni.

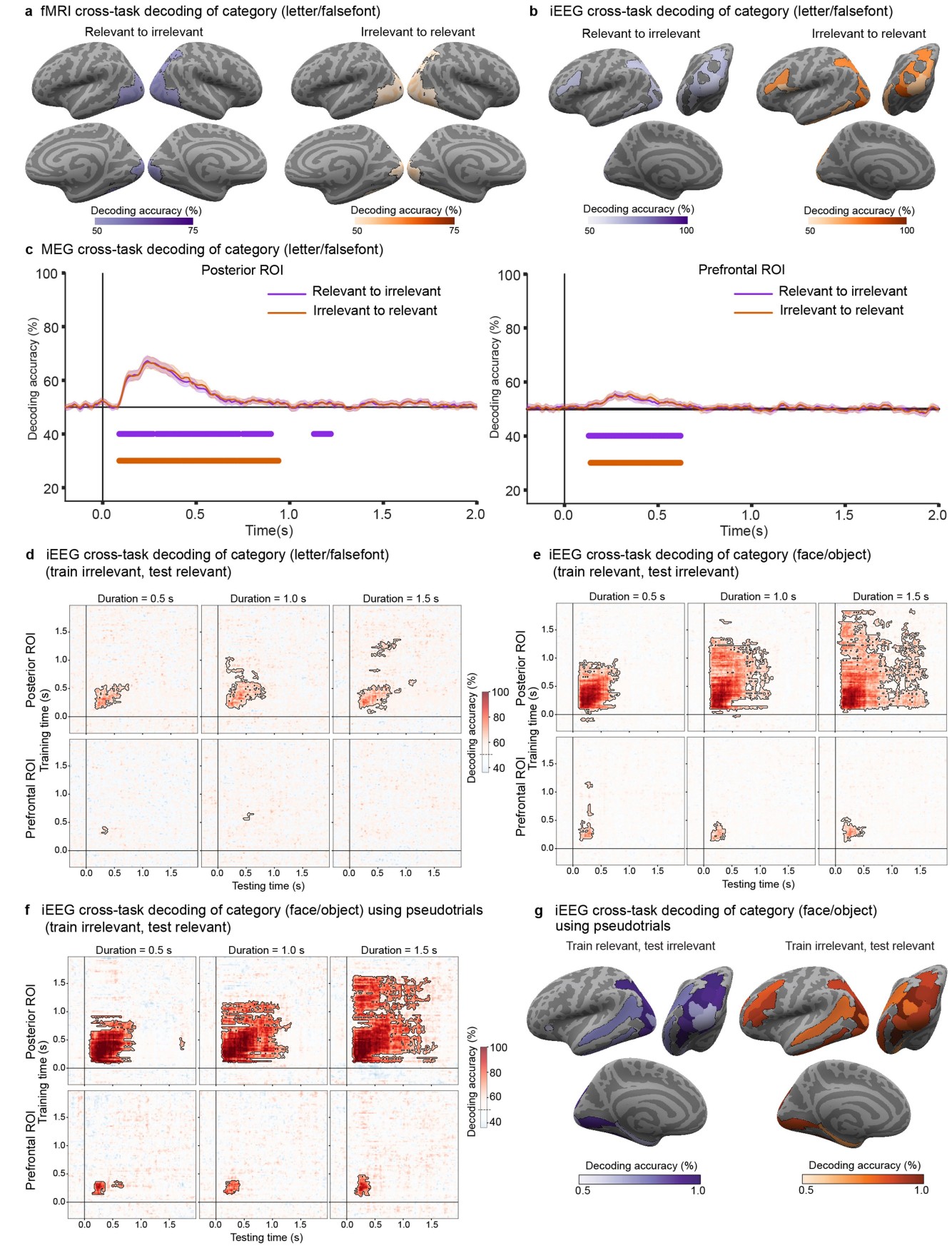

**a** fMRI cross-task decoding of category (letter/falsefont)

Relevant to irrelevant

Irrelevant to relevant

Decoding accuracy (%)

Decoding accuracy (%)

**b** iEEG cross-task decoding of category (letter/falsefont)

Relevant to irrelevant

Irrelevant to relevant

Decoding accuracy (%)

Decoding accuracy (%)

**c** MEG cross-task decoding of category (letter/falsefont)

Posterior ROI

Prefrontal ROI

Relevant to irrelevant
Irrelevant to relevant

**d** iEEG cross-task decoding of category (letter/falsefont)
(train irrelevant, test relevant)

**e** iEEG cross-task decoding of category (face/object)
(train relevant, test irrelevant)

**f** iEEG cross-task decoding of category (face/object) using pseudotrials
(train irrelevant, test relevant)

**g** iEEG cross-task decoding of category (face/object)
using pseudotrials

Train relevant, test irrelevant

Train irrelevant, test relevant

Decoding accuracy (%)

Decoding accuracy (%)

**Extended Data Fig. 1** | See next page for caption.

**Extended Data Fig. 1 | Prediction 1 Complementary results for decoding of conscious content. a**, fMRI searchlight decoding accuracies (letters-falsefonts), collapsed across durations. Pattern classifiers trained on relevant stimuli and tested on irrelevant stimuli (left, purple) or vice versa (right, orange): Outlined colored regions on the inflated cortical surfaces (top: lateral views; bottom: medial views) indicate significant above-chance (50%) decoding. Here and below, significance was evaluated through a cluster-based permutation test (p < 0.05; two-sided). Sample sizes as reported in Fig. 2. **b**, iEEG ROIs decoding accuracies (letters-falsefonts) collapsed across durations. Conventions as in **a**. The results are displayed on inflated surface maps from a left lateral (top left), posterior (top right) and left medial (bottom) views. **c**, MEG cross-task decoding of category (letters-falsefonts) when classifiers were trained on relevant stimuli and tested on irrelevant stimuli (purple); or vice versa (orange), separately for the whole posterior (left) and prefrontal (right) ROIs. Underlying lines indicate significantly above-chance (50%) decoding. Error bars depict 95% CI across participants. **d**, iEEG cross-task temporal generalization of category decoding (letters-falsefonts) classifiers trained on task-relevant stimuli and tested on task-irrelevant stimuli. Columns: stimulus durations (left: 0.5 s; center: 1.0 s; right: 1.5 s). Rows: theory ROIs (top: posterior; bottom: prefrontal). Contoured red-shaded regions depict significant above-chance (50%) decoding. **e**, iEEG cross-task temporal generalization of category decoding (faces-objects), classifiers were trained on task-relevant stimuli and tested on task-irrelevant ones. Conventions as in **d**. **f**, iEEG cross-task temporal generalization of category decoding (faces-objects) from task-irrelevant to task-relevant stimuli, yet using pseudotrial aggregation to boost decoding accuracy. Conventions as in **d**. **g**, iEEG ROI decoding accuracies (faces-objects) using pseudotrials. Conventions as in **b**. Brain surfaces in panels **a**, **b**, **g** are from Freesurfer.

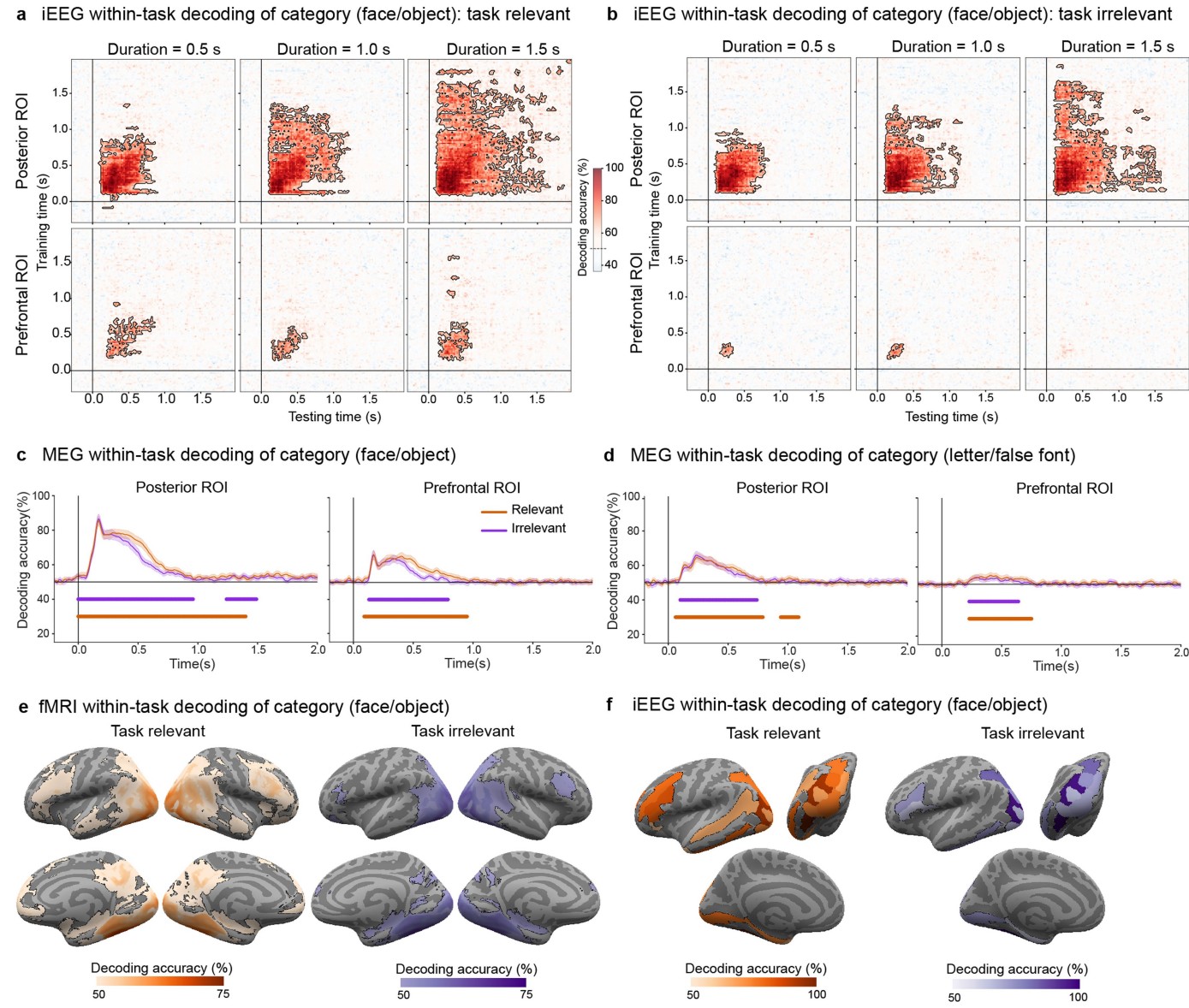

**a** iEEG within-task decoding of category (face/object): task relevant

**b** iEEG within-task decoding of category (face/object): task irrelevant

**c** MEG within-task decoding of category (face/object)

**d** MEG within-task decoding of category (letter/false font)

**e** fMRI within-task decoding of category (face/object)

**f** iEEG within-task decoding of category (face/object)

**Extended Data Fig. 2 | Within-task temporal generalization of decoding of stimulus category. a**, iEEG within-task temporal generalization decoding of category (faces-objects) for pattern classifiers trained and tested on task-relevant stimuli. As in Fig. 2b, columns represent stimulus durations (left: 0.5 s; center: 1.0 s; right: 1.5 s) and rows represent theory ROIs (top: posterior; bottom: prefrontal). Contoured red-shaded regions depict significant above-chance (50%) decoding. Here and below, significance was evaluated through a cluster-based permutation test (p < 0.05; two-sided). Sample size as in Fig. 2. **b**, iEEG within-task temporal generalization decoding of category for task-irrelevant stimuli. Conventions as in **a**. **c**, MEG within-task average decoding of category (faces-objects), for the task-relevant (orange) and the task-irrelevant (purple) conditions, in posterior (left) and prefrontal (right)

ROIs. Underlying lines depict significantly above-chance (50%) decoding assessed by cluster-based permutation test (p < 0.05). Error bars depict 95% CI estimated across participants. **d**, MEG within-task decoding of category (letters-falsefonts). Conventions as in **c**. **e**, fMRI searchlight decoding of category (faces-objects), collapsed across durations, for the task-relevant (left, orange) and task-irrelevant (right, purple) conditions. Outlined colors indicate regions on the inflated cortical surfaces showing significantly above-chance (50%) decoding (top: left/right lateral views; bottom: right/left medial views). **f**, iEEG ROIs decoding accuracies, collapsed across durations, within the task-relevant (left, orange) and the task-irrelevant (right, purple) stimuli. Same conventions as in **e**, with maps from a left lateral (top left), posterior (top right) and left medial (bottom) views. Brain surfaces in panels **e**,**f** are from Freesurfer.

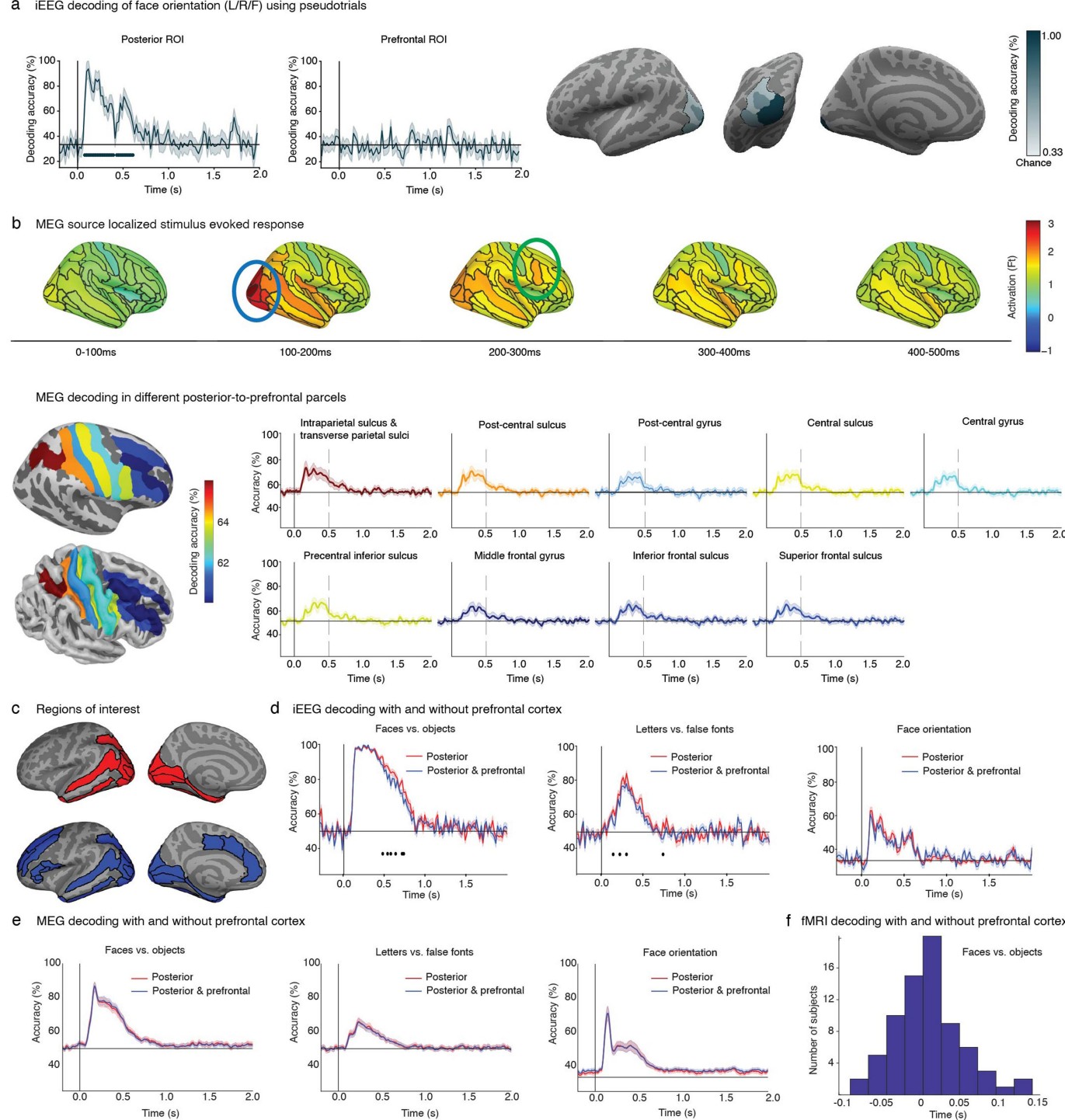

**Extended Data Fig. 3 |** See next page for caption.

**Extended Data Fig. 3 | Control analyses for the decoding prediction. a**, Left: iEEG ROIs decoding results of orientation (left/right/front view) over time as in Fig. 2, but using pseudotrials akin to the MEG analysis. Right: Regions with electrodes showing significant above-chance (33%) accuracies are indicated in outlined blue on the inflated surfaces (left: left lateral view; middle: posterior view; right: left medial view). Here and below, error bars depict 95% CI. Significance assessed using a cluster-based permutation test (p < 0.05, two-sided). Sample size as in Fig. 2. **b**, Two analyses were performed to evaluate potential leakage in MEG decoding, using independent data from the optimization phase (N = 32). Top: averaged stimulus-evoked response in face task-relevant trials, combined across durations, at different latencies, projected on the inflated surfaces. Activity in posterior areas (blue ellipse) showed the highest peak ~0.1-0.2 s, while prefrontal areas showed a later highest peak ~0.2-0.3 s. This challenges the leakage interpretation. Bottom: Analysis of face-object decoding in task-relevant trials across durations, separately within parcels in parietal and PFC. Left: Average decoding accuracy in an early time window (0.25-0.5 s) projected on two differently inflated surfaces to better depict gyri and sulci. Right: Time-resolved decoding of these parcels. Decoding is highest in posterior areas and lowest in anterior areas, with fairly similar time courses, suggesting a posterior-to-anterior gradient consistent with leakage. **c**, ROIs used in the decoding analysis including (blue) and excluding (red) PFC areas. **d**, iEEG decoding of faces-objects (left), letters-falsefonts (middle) and face orientation (right), with and without PFC (blue and red). Underlying lines indicate significantly worse decoding when including PFC. **e**, MEG decoding results, same conventions as in **d**. **f**, fMRI decoding of faces-objects. Histogram shows the differences in classification including and excluding frontal areas. fMRI accuracies including PFC show 1.2% increase compared to excluding PFC, observed in 56% of the participants. Notably, this slight increase was observed only in the combined features analysis and not the combined models' analysis (see Methods). Brain surfaces in panels **a-c** are from Freesurfer.

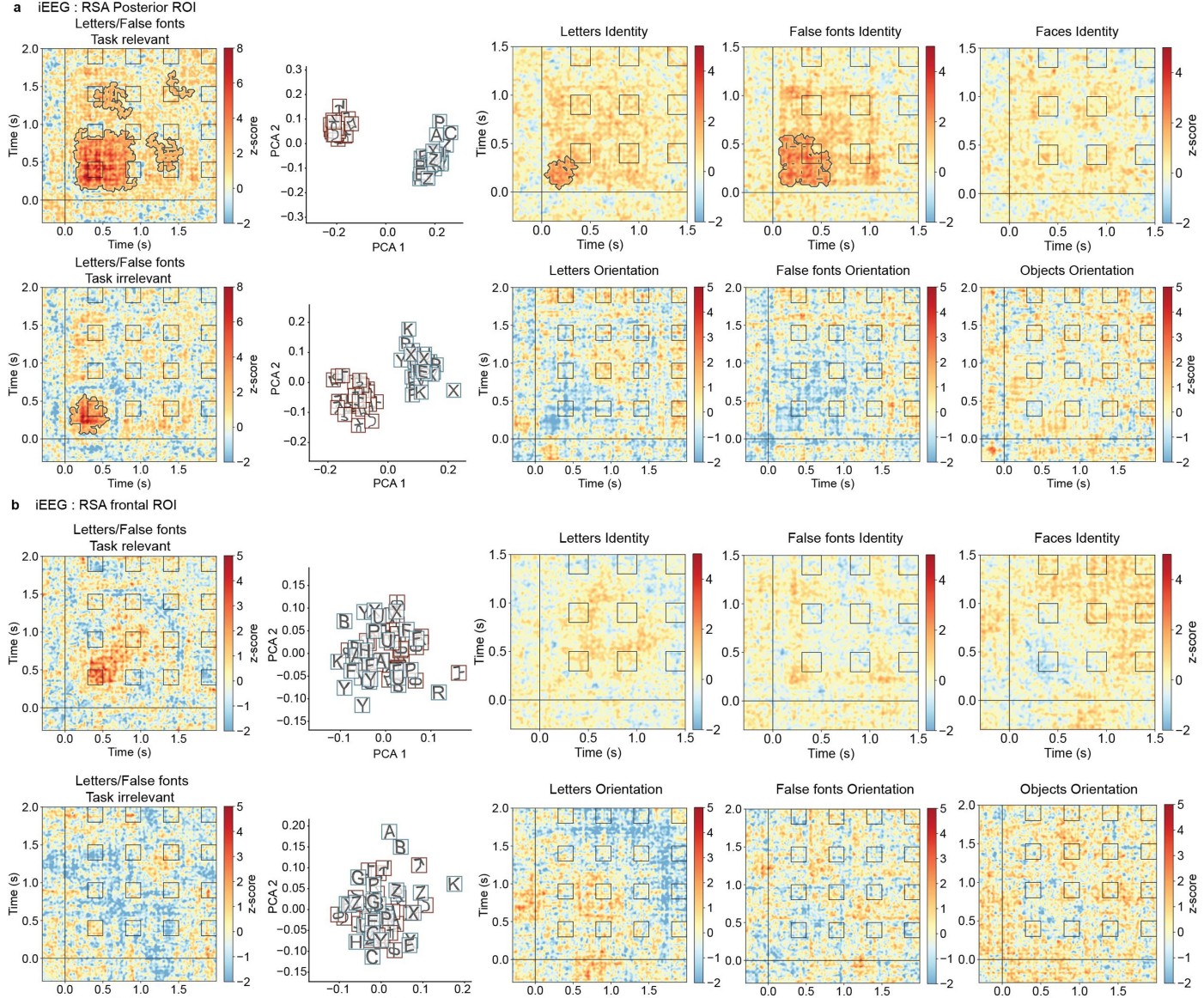

**Extended Data Fig. 4 | Maintenance of conscious content over time for stimulus categories, identity and orientation. a**, Cross-temporal representational similarity matrices in Posterior ROI ($N_{patients}$=28, $N_{electrodes}$ = 583). The leftmost column shows similarity for letters vs. false fonts, separately for task-relevant (left) and task-irrelevant (right) trials. Principal Component Analysis (PCA) plots at 0.3 s illustrate the separability between letters and false fonts. The top rightward column display similarity for identity, while bottom rows show similarity for orientation. Contours indicate statistical significance based on cluster-based permutation tests (upper tail test, α = 0.05). PCA illustrates clear separability between letters and false fonts in the posterior cortex at 0.3 s, regardless of task relevance (top – task-relevant, bottom – task-irrelevant). This separability was largely sustained in the task-relevant condition but diminished between ~0.95 and 1.4 s. In the task-irrelevant condition, separability was significant only for a brief period at the beginning. Identity information was significant for letters and false-fonts but not for faces. While identity information was not sustained throughout the entire stimulus duration, elevated z-scores up to 1 s suggest a potential limitation in statistical power. No statistically significant orientation information was observed for any category. Conventions as in Fig. 3. **b**, Cross-temporal representational similarity matrices in Prefrontal ROI ($N_{patients}$=28, $N_{electrodes}$ = 576) for the same contrasts as and following the same conventions as in **a**. No contrast yielded statistically significant results in the PFC ROI.

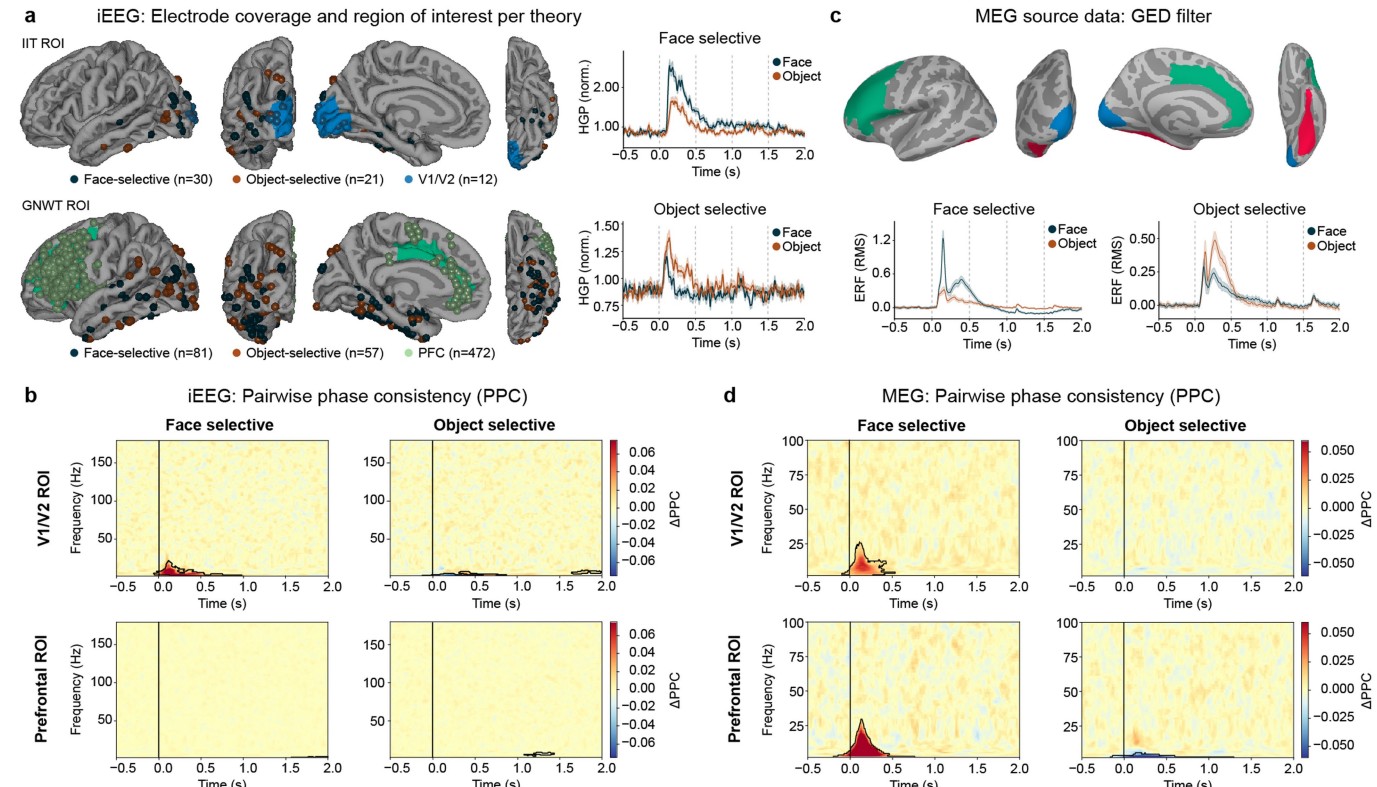

**Extended Data Fig. 5 | Prediction #3: Interareal connectivity preregistered analysis. a**, iEEG electrode coverage used to assess content-selective synchrony for IIT ROIs (top, $N_{patients}$ = 4) & GNWT ROIs (bottom, $N_{patients}$ = 21). Electrode coverage varied between ROIs as interareal connectivity was assessed between electrodes on a per-participant basis. In addition, two example category-selective electrodes are shown (right): one face-selective, and one object-selective. Error bars depict standard error of the mean. **b**, iEEG Pairwise phase consistency (PPC) analysis of task-irrelevant trials reveals significant content-selective synchrony (e.g. faces > objects for face-selective electrodes; left; objects > faces for object-selective electrodes; right) in V1/V2 ROIs (top row), but not in PFC ROIs (bottom row). Color bars represent the average change in PPC (face and object trials) for each node (face-selective, object-selective). Positive values reflect stronger connectivity for faces, while negative values reflect stronger connectivity for objects. **c**, MEG (N = 65)

cortical time-series were extracted per participant from cortical parcels in V1/V2 (blue), PFC (green) and in a fusiform (red) ROIs. Category-selective signals were obtained by creating a category-selective GED filter (i.e., contrasting face-object trials against any other stimulus category trials) on the activity extracted from the fusiform ROI. Face- (bottom left) and object-selective (bottom right) responses averaged across participants are shown at the bottom. Error bars depict 95% CI. Here and below, significance was assessed using cluster-based permutation tests, p < 0.05, two-sided. **d**, MEG PPC analysis of task-irrelevant trials (N = 65) reveals significant category-selective synchrony below 25 Hz for the face-selective GED filter (i.e., faces > objects for face-selective electrodes) in both V1/V2 (top row) and PFC ROIs (bottom row) and for the object-selective synchrony (objects > faces for object-selective electrodes) in the PFC ROI only. Brain surfaces in panels **a,c** are from Freesurfer.

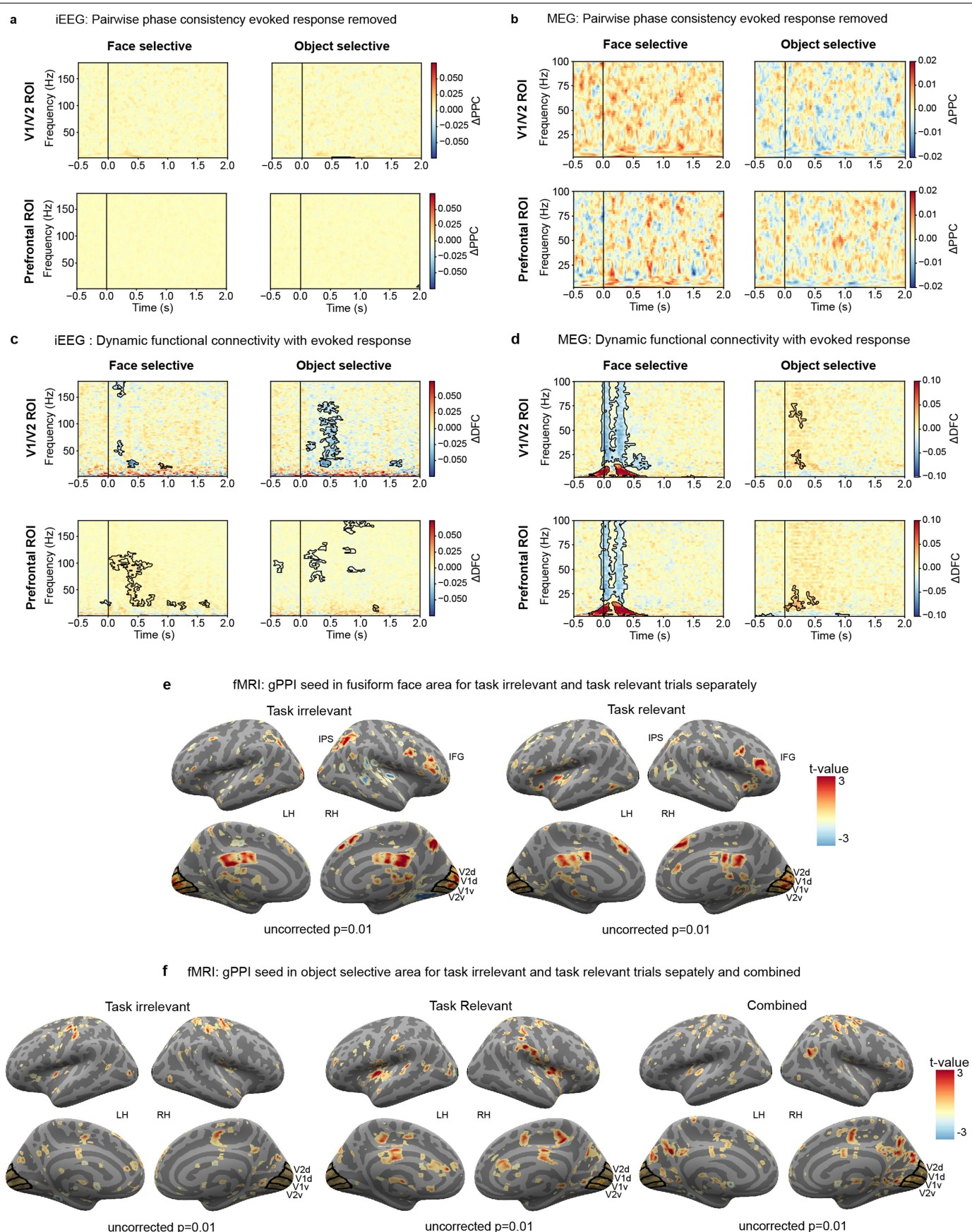

**a**  iEEG: Pairwise phase consistency evoked response removed

**b**  MEG: Pairwise phase consistency evoked response removed

**c**  iEEG : Dynamic functional connectivity with evoked response

**d**  MEG: Dynamic functional connectivity with evoked response

**e**  fMRI: gPPI seed in fusiform face area for task irrelevant and task relevant trials separately

**f**  fMRI: gPPI seed in object selective area for task irrelevant and task relevant trials sepately and combined

**Extended Data Fig. 6 |** See next page for caption.

**Extended Data Fig. 6 | Control analysis for the interareal connectivity prediction. a**, iEEG PPC analysis of task-irrelevant trials did not reveal any significant category-selective synchrony cluster in posterior (top) or PFC (bottom) ROIs after removing the evoked response. Same conventions, sample size and statistical tests as in Extended Data Fig. 5 are used here and below. **b**, MEG PPC analysis of task-irrelevant trials also did not reveal any synchrony cluster in any ROI after removing the evoked response. **c**, iEEG DFC analysis of task-irrelevant trials without removing the evoked response reveals significant content-selective connectivity between object-selective electrodes and V1/V2 electrodes (top-right), reflected as broadband (25–125 Hz) decrease in the change in DFC (e.g., faces < objects). Similar broadband changes in DFC (faces > objects) were observed for face-selective electrodes in PFC (bottom-left). Smaller significant effects were detected between face-selective and V1/V2 electrodes (top-left) and for object-selective and PFC electrodes (bottom-right). **d**, MEG DFC analysis of task-irrelevant trials without removing the evoked response reveal significant content-selective synchrony between the face-selective GED filter node and both V1/V2 (top-left) and PFC (bottom-left). This is reflected in an increase in low-frequency connectivity (< 25 Hz) combined with a decrease in high-frequency connectivity (25–100 Hz). Smaller yet significant effects were detected for the object-selective GED filter (right). **e**, Generalized psychophysiological interactions (gPPI) task-related connectivity analysis of task-irrelevant (left) and task-relevant (right) trials revealed weak clusters of content-selective connectivity with FFA as the analysis seed (p < 0.01, uncorrected). Common significant regions showing task-related connectivity in task-irrelevant, task-relevant, and combined conditions include V1/V2, right intraparietal sulcus (IPS), and right inferior frontal gyrus (IFG). **f**, gPPI task-related connectivity analysis of task-irrelevant (left), task-relevant (middle), and combined conditions revealed weak clusters of content-selective connectivity with lateral occipital complex (LOC) as the analysis seed (p < 0.01, uncorrected). Overall, no common significant regions showed task-related connectivity. Brain surfaces in panel **e**, **f** are from Freesurfer.

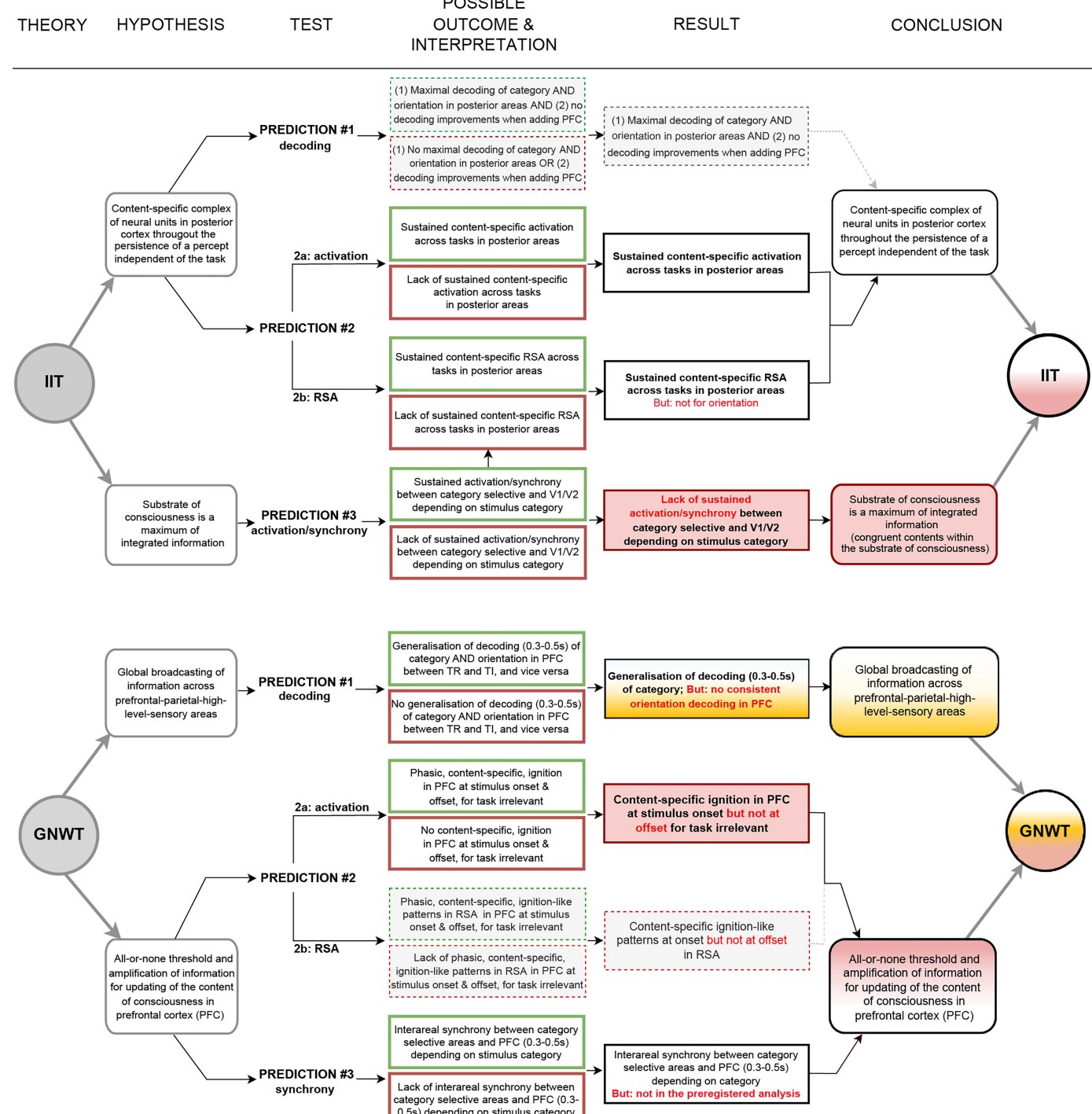

**Extended Data Fig. 7 | An overview of theoretical predictions, experimental outcomes and interpretations. Left:** Preregistered predictions of IIT (top) and GNWT (bottom) (see also ref. 12; Fig. 1). Key hypotheses (second column, Key hypotheses) are described alongside the three analyses used to test them (third column, Test): decoding (prediction #1; Fig. 2), activation & RSA (prediction #2; Fig. 3), and synchrony (prediction #3; Fig. 4). Potential outcomes and their interpretations are detailed in the fourth column (Possible outcome and interpretation), with outcomes aligning with predictions framed in green (pass) and contradictory outcomes framed in red (fail). Solid frames denote critical predictions, while dotted gray frames indicate non-critical predictions. This section reflects the theoretical expectations before the experiment. **Right:** summary of the actual findings, integrating results across modalities and analyses. Key findings for each prediction are described (fifth column; 'Result') with white denoting alignment with predictions, red indicating contradiction, white/red mixtures showing partial support or failure, and yellow indicating inconclusive results. Final conclusions synthesize these findings, using the same color coding. For IIT, the results mix a passed prediction (content-specific complex of neural units in posterior cortex, throughout the persistence of a percept, independent of the task) with a failure (maximum integrated information). For GNWT, the results consisted of a mixture of a partly challenged prediction (of an all-or-none threshold and amplification of information updating the content of consciousness in PFC) and a partly supported one, given the inconclusive result for orientation (of global broadcasting of information in the PFC). These results are discussed in the main text, including their implications for other consciousness theories.

**Extended Data Table 1 | Key Predictions and Integration of Evidence Across Planned Analyses**

| GNWT predictions | IIT predictions |
|---|---|
| **Prediction # 1: Decoding analyses** | |
| **(A1) Cross-task generalization of decoding of ANY CATEGORY that showed decoding in the TR[1] condition in ANY PREFRONTAL areas from Task irrelevant (TI) to Task relevant (TR) OR from TR to TI, *during 300-500ms post-stimulus onset*** | (A1) Cross-task generalization of decoding of ANY CATEGORY that showed decoding in the TR condition in ANY POSTERIOR from TI to TR *(data taken from any time window)* |
| **(A2) Decoding of ORIENTATION for ANY category in ANY PREFRONTAL area, *during 300-500ms post stimulus onset*** | (A2) Decoding of ORIENTATION, for ANY category in ANY POSTERIOR areas *(data taken from any time window)* |
| | (B1) NO increase in decoding accuracy[1] for ANY CATEGORY that showed decoding in the TR condition when adding non-specialized frontal areas (only for task irrelevant) *(data taken from any time window for posterior ROI and from 300-500ms post-stimulus onset for frontal areas)* |
| | (B2) NO increase in decoding accuracy[1] of ORIENTATION for ANY category that showed decoding when adding non-specialized frontal areas (only for task irrelevant) *(data taken from any time window for posterior ROI and from 300-500ms post-stimulus onset for frontal areas)* |
| A1 & A2 should be TRUE for MEG OR iEEG | B1 & B2 should be TRUE for MEG OR iEEG |
| **Prediction #2: Activation and representational similarity analyses** | |
| **(A) Phasic ignition in ANY PREFRONTAL area at stimulus ONSET (300-500ms post onset) AND OFFSET (300-500ms post offset) in TI for ALL stimulus durations for at least ONE cetgory in at least ONE measure of activation (ERP, High gamma, alpha)** | **(A) Content-specific sustained activation (from 300ms until the offset) in TI for ALL durations for at least ONE category in posterior cortical areas in at least ONE measure of activation (increased gamma, decreased alpha)** |
| (B) Phasic RSA during ONSET and OFFSET (300-500ms post stimulus onset/offset) in TI for 1.0 and 1.5 durations for at least ONE content (category OR orientation OR identity) in any PREFRONTAL area | **(B) Sustained RSA (from 300ms until the offset) in TI for 1.0 and 1.5 durations for at least ONE content (category OR orientation OR identity) in any POSTERIOR area (contingent on results of the blink control analysis)** |
| A should be TRUE for MEG OR iEEG | A & B should be TRUE for MEG OR iEEG |
| **Prediction #3: Synchronization analyses** | |
| **(A) Stronger synchronization between PFC and FFA for faces vs. objects during the 300-500ms time window in ANY technique[2], AND the STIMULUS difference should be larger than the TASK difference** | **(A) Stronger sustained (from 300ms until the offset) synchronization between (activated) V1/V2 and FFA for faces vs. objects for ALL durations in MEG/iEEG, AND the difference in the pattern of synchronization should be more consistent with the STIMULUS than with the TASK** |
| **(B) Stronger synchronization between PFC and LOC for objects vs. faces during 300-500ms time window in ANY technique, AND the STIMULUS difference should be larger than the TASK difference** | **(B) Stronger sustained (from 300ms until the offset) synchronization between (activated) V1/V2 and LOC for objects vs. faces for ALL durations in MEG/iEEG, AND the difference in the pattern of sumchronization should be more consistent with the STIMULUS than with the TASK** |
| A OR B | A OR B |
| Integration across predictions:<br><br>**Prediction #1 (Decoding)     AND**<br><br>**Prediction #2 (Activation)     AND**<br><br>**Prediction #3 (Synchronization)** | Integration across predictions:<br><br>**Prediction #2 (Activation)     AND**<br><br>**Prediction #2 (RSA)     AND**<br><br>**Prediction #3 (Synchronization)** |

[1] Assuming above chance (statistically significant) decoding.
[2] Although fMRI cannot be used to determine the temporal aspects of this prediction, we still consider a finding of synchronization between the expected areas as supporting evidence for GNWT.

Key predictions of each theory and plan for integrating outcomes across the different brain recording modalities and analyses. Each prediction (Bolded titles, light gray cells) is broken down to sub-predictions, which are then integrated together to provide the final conclusion per prediction (dark gray rows, appearing at the bottom for each prediction). Bolded predictions are the ones appearing on Extended Data Fig. 7 on the Preregistration, and are defined as the critical predictions for evaluating the theories. Numbered sub-predictions are the ones considered when integrating across sub-predictions to reach the final conclusion of each prediction (black rows). Finally, light red row denotes vertical integration across all predictions, to form the final conclusion for each theory based on its critical predictions.

**Extended Data Table 2 | Decoding of faces vs. objects in the theory-defined ROIs**

| Anatomical ROIs (Destrieux atlas) | Irrelevant-Relevant | | Relevant-irrelevant | | Irrelevant | | Relevant | |
|---|---|---|---|---|---|---|---|---|
| | n voxels | % voxels | n voxels | % voxels | n voxels | % voxels | n voxels | % voxels |
| **Posterior ROI** | | | | | | | | |
| G_and_S_occipital_inf | 1868 | 93 | 1866 | 93 | 1868 | 93 | 1876 | 93 |
| G_oc-temp_lat-fusifor | 2549 | 98 | 2550 | 98 | 2542 | 98 | 2561 | 99 |
| G_occipital_middle | 1979 | 80 | 1952 | 79 | 1909 | 76 | 2096 | 85 |
| S_oc_middle_and_Lunatus | 1009 | 100 | 1008 | 100 | 1000 | 100 | 1010 | 100 |
| G_cuneus | 600 | 24 | 542 | 22 | 587 | 23 | 1233 | 49 |
| G_occipital_sup | 1351 | 69 | 1295 | 66 | 1299 | 66 | 1302 | 66 |
| G_oc-temp_med-Lingual | 1403 | 47 | 1374 | 46 | 1375 | 46 | 1499 | 50 |
| G_oc-temp_med-Parahip | 430 | 30 | 408 | 29 | 432 | 31 | 521 | 37 |
| G_temporal_inf | 686 | 47 | 692 | 47 | 756 | 52 | 859 | 59 |
| Pole_occipital | 1952 | 80 | 1934 | 80 | 1870 | 77 | 1968 | 81 |
| Pole_temporal | 0 | 0 | 0 | 0 | 0 | 0 | 15 | 2 |
| S_calcarine | 448 | 18 | 427 | 18 | 395 | 16 | 657 | 27 |
| S_intrapariet_and_P_trans | 261 | 7 | 287 | 8 | 799 | 21 | 1670 | 44 |
| S_oc_sup_and_transversal | 1163 | 82 | 1166 | 82 | 1225 | 87 | 1230 | 87 |
| S_temporal_sup | 1100 | 22 | 944 | 19 | 820 | 17 | 2264 | 46 |
| | | | | | | | | |
| **PFC ROI** | | | | | | | | |
| G_and_S_cingul-Mid-Post | 0 | 0 | 0 | 0 | 0 | 0 | 0 | 0 |
| Lat_Fis-ant-Horizont | 0 | 0 | 0 | 0 | 0 | 0 | 1250 | 23 |
| Lat_Fis-ant-Vertical | 6 | 1 | 1 | 0 | 3 | 1 | 36 | 8 |
| G_and_S_cingul-Ant | 0 | 0 | 0 | 0 | 5 | 0 | 278 | 8 |
| G_and_S_cingul-Mid-Ant | 0 | 0 | 0 | 0 | 0 | 0 | 200 | 1 |
| G_front_inf-Opercular | 134 | 6 | 65 | 3 | 98 | 4 | 436 | 20 |
| G_front_inf-Orbital | 0 | 0 | 0 | 0 | 0 | 0 | 34 | 5 |
| G_front_inf-Triangul | 142 | 9 | 68 | 4 | 130 | 78 | 608 | 37 |
| G_front_middle | 50 | 1 | 15 | 0 | 154 | 3 | 1301 | 21 |
| S_front_middle | 0 | 0 | 4 | 0 | 29 | 1 | 86 | 4 |
| S_front_sup | 0 | 0 | 0 | 0 | 0 | 0 | 300 | 8 |
| S_front_inf | 164 | 8 | 89 | 4 | 184 | 9 | 1022 | 49 |

The table presents the number of voxels in each theory-defined ROI that were detected in the searchlight decoding of category (faces vs. objects; N=73), using a cluster-based permutation test (p<0.05). The results are presented separately for cross-task decoding (i.e., when classifiers are trained on the task-irrelevant trials and tested on task-relevant ones, or vice versa), as well as for within task decoding (irrelevant and relevant conditions).

**Extended Data Table 3 | Electrode locations found to be significant in the LMM analysis**

| Channel | x | y | z | Destrieux ROI | Wang ROI | Desikan ROI | Model |
|---|---|---|---|---|---|---|---|
| SE107-O2PH16 | -0.03618 | -0.08678 | 0.000733 | S_oc_middle_and_Lunatus | TO1 | ctx-lh-lateraloccipital | IIT x Cate |
| SE120-T3bOT10 | -0.05876 | -0.06964 | -0.02078 | G_oc-temp_lat-fusifor | Unknown | ctx-lh-fusiform | IIT x Cate |
| SE120-T3bOT9 | -0.05712 | -0.0689 | -0.02016 | G_oc-temp_lat-fusifor | Unknown | ctx-lh-fusiform | IIT x Cate |
| SF102-LO1 | -0.01976 | -0.10359 | 0.001174 | Pole_occipital | V2d | ctx-lh-lateraloccipital | IIT x Cate |
| SF102-LO2 | -0.02301 | -0.09792 | 0.005426 | Pole_occipital | V2d | ctx-lh-lateraloccipital | IIT x Cate |
| SF103-PIT1 | -0.04072 | -0.06213 | -0.02039 | G_oc-temp_lat-fusifor | Unknown | ctx-lh-fusiform | IIT x Cate |
| SF103-PIT2 | -0.04156 | -0.04393 | -0.02499 | G_oc-temp_lat-fusifor | Unknown | ctx-lh-fusiform | IIT x Cate |
| SF104-LO1 | -0.01396 | -0.10275 | 0.008659 | Pole_occipital | V2d | ctx-lh-lateraloccipital | IIT x Cate |
| SF104-LO2 | -0.01663 | -0.10338 | 0.005258 | Pole_occipital | V2d | ctx-lh-lateraloccipital | IIT x Cate |
| SF109-IO3 | 0.006178 | -0.07586 | -0.00279 | G_oc-temp_med-Lingual | V2v | ctx-rh-lingual | IIT x Cate |
| SF109-IO4 | 0.005093 | -0.07816 | -0.0047 | G_oc-temp_med-Lingual | V2v | ctx-rh-lingual | IIT x Cate |
| SF113-RIT1 | 0.038119 | -0.04974 | -0.02225 | G_oc-temp_lat-fusifor | Unknown | Cerebellum-Cortex | IIT x Cate |
| SF113-RIT2 | 0.040545 | -0.04845 | -0.02346 | G_oc-temp_lat-fusifor | Unknown | Cerebellum-Cortex | IIT x Cate |
| SE107-O1b3 | -0.01196 | -0.06305 | -0.00094 | G_oc-temp_med-Lingual | Unknown | ctx-lh-lingual | GNWT |
| SE107-O2PH14 | -0.03383 | -0.08203 | 6.93E-05 | S_oc_middle_and_Lunatus | LO2 | ctx-lh-lateraloccipital | GNWT |
| SE107-O2PH15 | -0.0354 | -0.08519 | 0.000512 | S_oc_middle_and_Lunatus | LO2 | ctx-lh-lateraloccipital | GNWT |
| SE108-O2b14 | -0.0294 | -0.09064 | -0.00472 | S_oc_middle_and_Lunatus | LO1 | ctx-lh-lateraloccipital | GNWT |
| SE120-O2*5 | -0.04225 | -0.09646 | -0.00451 | G_and_S_occipital_inf | Unknown | ctx-lh-lateraloccipital | GNWT |
| SE120-O2*6 | -0.04354 | -0.09769 | -0.00357 | G_and_S_occipital_inf | Unknown | ctx-lh-lateraloccipital | GNWT |
| SE120-T3c6 | -0.05264 | -0.08681 | 0.025426 | S_temporal_sup | Unknown | ctx-lh-inferiorparietal | GNWT |
| SF104-LO3 | -0.02255 | -0.10253 | 0.000551 | Pole_occipital | V2d | ctx-lh-lateraloccipital | GNWT |
| SF109-DL4 | 0.022039 | -0.07051 | 0.008421 | S_calcarine | Unknown | ctx-rh-pericalcarine | GNWT |
| SF109-DL5 | 0.02433 | -0.07204 | 0.008081 | S_calcarine | Unknown | ctx-rh-pericalcarine | GNWT |
| SF109-G45 | 0.04645 | -0.08224 | -0.00242 | G_occipital_middle | Unknown | ctx-rh-lateraloccipital | GNWT |
| SE108-O2b13 | -0.02856 | -0.08853 | -0.00505 | G_and_S_occipital_inf | Unknown | ctx-lh-lateraloccipital | IIT |
| SE110-O2*10 | 0.036288 | -0.1042 | -0.00079 | G_and_S_occipital_inf | Unknown | ctx-rh-lateraloccipital | IIT |
| SE110-O2*7 | 0.031792 | -0.09698 | -0.00721 | S_oc-temp_lat | Unknown | ctx-rh-lateraloccipital | IIT |
| SE110-O2*8 | 0.03359 | -0.09987 | -0.00464 | G_and_S_occipital_inf | Unknown | ctx-rh-lateraloccipital | IIT |
| SE110-O2*9 | 0.035389 | -0.10276 | -0.00207 | G_and_S_occipital_inf | Unknown | ctx-rh-lateraloccipital | IIT |
| SE120-O1b10 | -0.02828 | -0.11893 | 0.004408 | Pole_occipital | V2d | ctx-lh-lateraloccipital | IIT |
| SF102-LO3 | -0.0356 | -0.08904 | -0.00424 | G_occipital_middle | LO2 | ctx-lh-lateraloccipital | IIT |
| SF107-O1 | 0.024693 | -0.10108 | -0.00812 | Pole_occipital | Unknown | ctx-rh-lateraloccipital | IIT |
| SF107-O2 | 0.027381 | -0.09982 | -0.00773 | Pole_occipital | Unknown | ctx-rh-lateraloccipital | IIT |
| SF107-O3 | 0.042207 | -0.08618 | -0.00419 | G_occipital_middle | Unknown | ctx-rh-lateraloccipital | IIT |
| SF113-RO1 | 0.034984 | -0.08617 | 0.010333 | G_occipital_middle | V3B | ctx-rh-lateraloccipital | IIT |
| SF113-RO2 | 0.040244 | -0.08034 | 0.011692 | G_occipital_middle | LO2 | ctx-rh-inferiorparietal | IIT |

Electrodes location in MNI coordinates, as well as in the corresponding parcellations of the Destrieux Atlas, Wang Atlas and Desikan Atlas.

# Reporting Summary

## Statistics

For all statistical analyses, confirm that the following items are present in the figure legend, table legend, main text, or Methods section.

| n/a | Confirmed | |
|---|---|---|
| ☐ | ☒ | The exact sample size (*n*) for each experimental group/condition, given as a discrete number and unit of measurement |
| ☐ | ☒ | A statement on whether measurements were taken from distinct samples or whether the same sample was measured repeatedly |
| ☐ | ☒ | The statistical test(s) used AND whether they are one- or two-sided<br>*Only common tests should be described solely by name; describe more complex techniques in the Methods section.* |
| ☒ | ☐ | A description of all covariates tested |
| ☐ | ☒ | A description of any assumptions or corrections, such as tests of normality and adjustment for multiple comparisons |
| ☐ | ☒ | A full description of the statistical parameters including central tendency (e.g. means) or other basic estimates (e.g. regression coefficient) AND variation (e.g. standard deviation) or associated estimates of uncertainty (e.g. confidence intervals) |
| ☐ | ☒ | For null hypothesis testing, the test statistic (e.g. *F*, *t*, *r*) with confidence intervals, effect sizes, degrees of freedom and *P* value noted<br>*Give P values as exact values whenever suitable.* |
| ☐ | ☒ | For Bayesian analysis, information on the choice of priors and Markov chain Monte Carlo settings |
| ☒ | ☐ | For hierarchical and complex designs, identification of the appropriate level for tests and full reporting of outcomes |
| ☒ | ☐ | Estimates of effect sizes (e.g. Cohen's *d*, Pearson's *r*), indicating how they were calculated |

*Our web collection on statistics for biologists contains articles on many of the points above.*

## Software and code

Policy information about availability of computer code

| Data collection | iEEG - New York University (NYU), Harvard University (Harvard), University of Wisconsin (WI)<br>  Natus Quantum and Neuralynx amplifiers, Matlab versions: Harvard: R2020b; NYU: R2020a, WU: 2021a, Psychtoolbox v3, Eyetracking: Harvard and WU: Eyelink 1000+, NYU: Tobii4c<br><br>MEG- Birmingham University (BU)<br>  FaceGen Modeler 3.1, MAXON CINEMA 4D Studio (RC - R20) 20.059, Matlab R2019B, Psychtoolbox 3, MEGIN DALQ 6.0, Eyelink Host PC Software 5.15<br><br>MEG - Peking University (PKU)<br>  Matlab R2018B with Psychtoolbox 3, Eyelink Host PC Software 5.15, MEGIN DALQ 6.0<br><br>fMRI - Donders Institute, Centre for Cognitive Neuroimaging (DCCN) and Yale University (Yale)<br>  Matlab R2019b, Psychtoolbox v.3.<br><br>Pilot (MPIEA)<br>  Psychtoolbox 3 (Brainard, 1997; Pelli, 1997) on Matlab 2017a; Windows 10<br><br>*All experimental paradigm code, used for data collection, can be found at: https://doi.org/10.5281/zenodo.14362838 |
|---|---|
| Data analysis | iEEG<br>Python (v3.9) using open-source packages, MNE (0.24), NumPy (l.23.5), SciPy (l.9.3), Frites (0.4.3), nilearn (0.9.2), ibabel (4.0.2), JZS Bayes |

factor (Rouder, et al., 2009)

MEG (BU)
Python (3.9.7), MNE-python (0.24.0), Freesurfer (6.0.1), MNE-BIDS (0.8), Frites (0.4.3), FLUX (1.0)
Algorithms:
Signal-Space Separation (SSS), FastICA, dynamic statistical parametric mapping (dSPM), minimum-norm estimates (MNE), pairwise phase consistency (PPC) - All the algorithms above were performed as implemented in MNE-Python (see versions above).
MEG (PKU)
Python (3.9.7), MNE-python(0.24.0), Freesurfer (6.0.1), Spyder (5.3.3), MNE-bids (0.8), scikit-learn.
When conducting Bayesian analyses, we used the bayes_ttest function in the code "bayes_factor_fun.py" found in our shared github repository (see below).

fMRI (DCCN & Yale)
BIDScoin v3.6.3, BIDS-Validator, MRIQC 0.16.1, fMRIPrep 20.2.3, Nipype 1.6.1, FSL 6.0.2, SPM8, SPM12, Freesurfer 6.0.1, NiBetaSeries 0.6.0, Pingouin 0.5.1, NumPy 1.19.2, Pandas 1.1.3, NiBabel 3.2.2, SciPy 1.8.0, Matplotlib 3.3.2, Scikit-learn 0.23.2
Algorithms: JZS Bayes Factor (Rouder et al. 2009), Support Vector Machine classifier, and generalized Psycho-Physiological Interaction

Behavioural analysis
R 4.3.1, ordinal 2023.12.4, tidyR 1.3.0, dplyr 1.1.4, lmerTest 3.1.3, bayestestR 0.14.0, emmeans 1.10.4, Python 3.9, pandas 1.5.2, numpy 1.21.2, matplotlib 3.6.2, seaborn 0.12.1, scipy 1.7.1

Eye tracking analysis
R 4.3.1, ordinal 2023.12.4, tidyR 1.3.0, dplyr 1.1.4, lmerTest 3.1.3, bayestestR 0.14.0, emmeans 1.10.4, Python 3.9, numpy 1.21.2, pandas 1.5.2, scipy 1.7.1, pycircstat 0.0.2, astropy 4.3.post1, seaborn 0.12.1, matplotlib 3.6.2, statsmodels 0.14.0, matlab.engine 9.11.19 (must use this version), saccade analysis features are based on Engbert & Mergenthaler, 2006, but the vfac parameter is based on Engbert & Kliegl, 2003.

Pilot (MPIEA)
MATLAB 2019

**All analysis code (for all modalities) used for data analysis, can be found at: https://doi.org/10.5281/zenodo.13891328

For manuscripts utilizing custom algorithms or software that are central to the research but not yet described in published literature, software must be made available to editors and reviewers. We strongly encourage code deposition in a community repository (e.g. GitHub). See the Nature Portfolio guidelines for submitting code & software for further information.

# Data

Policy information about availability of data

All manuscripts must include a data availability statement. This statement should provide the following information, where applicable:
- Accession codes, unique identifiers, or web links for publicly available datasets
- A description of any restrictions on data availability
- For clinical datasets or third party data, please ensure that the statement adheres to our policy

The full study protocol is available in the preregistration on the OSF webpage (https://osf.io/92tbg/), including a detailed description of the experimental design, the theories' predictions and agreed-upon interpretations of the results, as well as iEEG, MEG, and fMRI data acquisition details, preprocessing pipelines, and data analysis procedures. Deviations from the preregistration are documented throughout the manuscript and summarized in Section 14 of the Supplementary Materials.
All data generated in this study are available under a CC BY 4.0 license. The M-EEG, fMRI, and iEEG datasets are distributed through two methods: as downloadable data bundles and via an XNAT instance, which enables search functionality and single-participant downloads. Data bundles can be accessed at https://www.arc-cogitate.com/data-bundles in raw format (M-EEG raw, fMRI raw, iEEG raw) and BIDS format (M-EEG BIDS, fMRI BIDS, iEEG BIDS). Alternatively, the datasets are accessible via the Cogitate XNAT instance at https://cogitate-data.ae.mpg.de. All distribution formats include robust metadata, and detailed documentation of experimental procedures and dataset structure is available at https://cogitate-consortium.github.io/cogitate-data/.

# Research involving human participants, their data, or biological material

Policy information about studies with human participants or human data. See also policy information about sex, gender (identity/presentation), and sexual orientation and race, ethnicity and racism.

| | |
|---|---|
| Reporting on sex and gender | Findings do not apply to only one sex or gender; gender was not considered in the study design and was determined based on self-reporting by participants/patients |
| Reporting on race, ethnicity, or other socially relevant groupings | No socially relevant categorization variables were collected during the study |
| Population characteristics | iEEG participants (mean age 30.88±13.94 years, 18 females, 26 right-handed), and all had a clinical diagnosis of epilepsy |
| | MEG participants: (mean age 22.79±3.59 years, 54 females, all right-handed), 32 of those datasets were included in the optimization phase (mean age 22.50±3.43 years, 19 females, all right-handed), and 65 in the replication sample (Age = 22.93±3.66, 35 of them females, all right-handed). |
| | fMRI participants (mean age 23.31±3.45 years, 72 female, 107 right handed), 35 of those datasets were included in the |

optimization sample (mean age 23.25±3.64 years, 21 females, 34 right handed), and 73 in the replication sample (mean age = 23.29±3.37, 49 females, 71 right-handed).

Pilot (MPI) Thirty-nine participants (26 females, aged between 18 and 59, mean=32.6, std=l2.82, all right-handed) took part in the study. All participants had normal or corrected-to-normal vision. They were recruited from the participant pool of the MPI and received monetary compensation for their participation

**Recruitment**

iEEG (NYU, Harvard, WI)
Participants for iEEG studies consisted of clinical patients admitted to the Epilepsy Monitoring Unit for the surgical management of epilepsy. Participants were approached to participate in research, and if agreeable, they were consented based on each site's IRB protocol for Research with Human Participants. Referral biases may exist, as not all epilepsy patients are referred for surgery, and those who are tend to share specific characteristics—such as disease severity, epilepsy risk factors, cognitive symptoms, comorbid conditions, and seizure onset zone. Since refractory epilepsy is often associated with cognitive dysfunction and neuropsychological disorders, this referral pattern is an important consideration. Patient enrollment is determined by the clinical teams at each site, not the iEEG research group, making this study akin to a cohort design. Our team only excludes patients who lack cognitive capacity to consent or are too young for participation.

MEG
Campus flyers advertisements and targeted mailing lists (BU); campus advertisements (PKU). Most participants were highly educated, middle/high-class university students for both sites.

fMRI (DCCN & Yale)
Both sites recruited neurotypical adult participants via flyers, online community listservs, and social media. In addition to this, the DCCN recruited participants from an existing pool (SONA subject management tool). There was no selection-bias or any other biased since we targeted enrollment of a representative sample across the different gender, racial, and ethnic groups. Due to safety requirements, anyone with a contraindication for MRI, would not be eligible to participate (e.g. metal or electronic implants, pregnant individuals or persons that experience claustrophobia).

Unlike most neuroscience studies that collect data from a single site and/ore single modality, our study minimizes site-specific biases by integrating data from seven sites across three continents. This diverse, multi-site approach enhances the generalizability and robustness of our findings, reducing the limitations associated with localized participant pools and single-laboratory methodologies.

Pilot (MPIEA)
All participants were recruited using an Max Planck Institute for Empirical Aesthetics (MPIEA) internal recruitment tool (MORLA).

**Ethics oversight**

Across our 7 data collection sites, ethics approvals metadata is as follows :
 1. Responsible institute
 2. Protocol number
 3. Approving committee

1. Centre for Human Brain Research, University of Birmingham
2. ERN_18-0226AP20
3. Science, Technology, Engineering and Mathematics Ethical Review Committee

1. School of Psychological and Cognitive Sciences, Peking University
2. 2020-05-07e
3. Committee for Protecting Human and Animal Subjects

1. Donders Institute (Centre for Cognitive Neuroimaging) - DCCN
2. File number : 2014-288NL number NL45659.091.14
3. Commissie Mensgebonden Onderzoek Regio Arnhem-Nijmegen

1. Yale School of Medicine
2. 2000027591
3. Human Research Protection Program Institutional Review Board

1. New York University Langone Health
2. i14-02101_CR6
3. Office of Science and Research Institutional Review Board

1. Children's Hospital Corporation d/b/a Boston Children's Hospital
2. 04-05-065R
3. Boston Children's Hospital Institutional Review Board (IRB)

1. University of Wisconsin-Madison
2. ID : 2017-1299
3. IRB UW-Madison

1. Max Planck Institute for Empirical Aesthetics (MPIEA)
2. Nr. 2017 12
3. Ethics Council of the Max Planck Society

Note that full information on the approval of the study protocol must also be provided in the manuscript.

# Field-specific reporting

Please select the one below that is the best fit for your research. If you are not sure, read the appropriate sections before making your selection.

☒ Life sciences ☐ Behavioural & social sciences ☐ Ecological, evolutionary & environmental sciences

For a reference copy of the document with all sections, see nature.com/documents/nr-reporting-summary-flat.pdf

# Life sciences study design

All studies must disclose on these points even when the disclosure is negative.

| Sample size | We collected data in 3 different neuroimaging modalities (iEEG, MEG, fMRI), in addition to behavioural and eye tracking data for all datasets, across 7 sites. Sample sizes for fMRI and MEG were determined as being 2.5 times larger than common sample sizes in the literature for that methodology (Simonshon, 2015; 50 participants per site for fMRI and for MEG). Since we used a within-subject design, this sample size gives us >90% power to detect differences of medium effect size (Cohen's d=>0.5). For iEEG, data collection is variable, as it is based on patient availability.<br><br>iEEG (NYU, Harvard, WI): N= 34 (2 excluded)= 32<br>MEG (BU, PKU) :N= 102 (5 excluded)= 97<br>fMRI (DCCN, Yale): N=120 (12 excluded)= 108<br><br>Total collected; N = 256<br>Total datasets included in analyses; N=237<br>_________________<br><br>Pilot Study (reported in Supplementary Information): N= 39 |
|---|---|
| Data exclusions | Data from all modalities were checked at three levels by a Data Monitoring Team (DMT). The first level checks tested whether the datasets contained all expected files keeping their naming conventions, and that all personal information had been removed. The second level checks tested participant's performance with respect to behavior; participants were excluded if their hit rate was lower than 80% or false alarms (FAs) higher than 20% for MEG and fMRI, and for iEEG, a more relaxed criteria of 70% Hits and 30% FAs was used. The third level checks assessed the quality of the neural data. For iEEG, channel rejection was performed independently by both the DMT and iEEG teams, and compared to make sure there were no discrepancies. We then verified that the electrode reconstruction performed by the iEEG team matched the alignment of contacts in participants' MRI. Finally, we checked for massive disturbances in the spectra.<br><br>For M-EEG, the first stage of the third-level checks focused on system-related and external noise generators. It was tested using the signal spectra in the empty room recording, the resting state session, and the experiment itself for all sensors. Any sensor and/or specific frequency revealing extensive noise using visual inspection, was flagged to document potential problems. [Ultimately, this did not lead to any exclusions.] Next, all experimental data blocks were visually inspected for abnormalities in spectra (peaks not explainable by physiology), and in ICA components, and checked for extremely noisy (based on the score of differences between the original and Maxwell-filtered data> 7) and flat sensors. The latter step was performed in collaboration between the DMT and members of BU and PKU to check whether any potential changes in preprocessing for particular participants were needed. Finally, we tested if all experimental cells (i.e., task-relevant non-targets and task-irrelevant stimuli for each one of the four categories) had enough trials (N=30).<br><br>For fMRI, we combined visual inspection of structural and functional images with automatic criteria for motion-related artifacts. Third level checks of data quality were done using both MRIQC (Esteban et al., 2017) and fMRIprep (Esteban et al., 2020), separately for optimization and replication datasets. The output from MRIQC and fMRIprep was visually inspected. Datasets with clear artifacts and other indicators of low data quality (incorrect reconstructions, and substantial signal dropout or distortion) were marked by a trained observer, and if the detected problems were judged severe enough to warrant potential exclusion, data were additionally inspected together by the DMT and collaborators at DCCN and Yale. In practice, we rejected participants where a significant part of the cortex, roughly > 5%, was not covered by the brain mask (tissue was not segmented). Next, datasets were checked for extensive motion, using MRIQC image quality metrics. Specifically, the percentage of fMRI volumes that exceeded a threshold of 0.2mm framewise displacement (FD) and DVARS (Power et al., 2012) measure were calculated per run and averaged per session. Finally, each MRI session whose percentage framewise displacement or DVARS deviated by more than 2 standard deviations above the group mean were marked for rejection.<br>_________________<br><br>iEEG<br>Two (N=2) patients were excluded due to incomplete datasets. Three patients whose behavior fell short of the predefined behavioral criteria (i.e. hits< 70%, FA> 30%), were nonetheless included in the analysis: one kept the response button pressed for most of the time during experiment, the other's low performance was driven by one of the categories only (which the patient reported having difficulty to detect), and the third's performance was very close to the threshold (65%) and had very low FA rate (2%).<br><br>MEG<br>Five (N=5) participants were excluded from the MEG dataset: two due to failure to meet predefined behavioral criteria (i.e., hits< 80%, and/or FA > 20%), two due to excessive noise from sensors, and one due to incorrect sensor reconstruction<br><br>fMRI<br>Twelve (N=12) participants were excluded from the fMRI dataset: seven due to motion artifacts, two due to insufficient coverage, and two due to incomplete data<br><br>Eye movement analysis |

| | Eleven (N=ll) participants/patients were excluded from the eye movement analysis (3 iEEG patients due to no eye tracking data available; 8 for insufficient quality (iEEG, N=4; MEG, N=2; fMRI, N=2) |
|---|---|
| Replication | All findings from MEG and fMRI were replicated on an independent sample and are reported in the supplementary materials. Due to the limited number of iEEG datasets, replication was not conducted on those data.<br>An initial optimization phase was used on 1/3 of the MEG (N=32) and fMRI (N=35) data. Following optimization, pipelines were preregistered and applied tot he novel datasets containing twice as much data (MEG, N=65 and fMRI, N=73). |
| Randomization | We used a within-participant design that does not require randomization. |
| Blinding | We used a within-participant design that does not require blinding. |

# Reporting for specific materials, systems and methods

We require information from authors about some types of materials, experimental systems and methods used in many studies. Here, indicate whether each material, system or method listed is relevant to your study. If you are not sure if a list item applies to your research, read the appropriate section before selecting a response.

## Materials & experimental systems

| n/a | Involved in the study |
|---|---|
| ☒ | ☐ Antibodies |
| ☒ | ☐ Eukaryotic cell lines |
| ☒ | ☐ Palaeontology and archaeology |
| ☒ | ☐ Animals and other organisms |
| ☒ | ☐ Clinical data |
| ☒ | ☐ Dual use research of concern |
| ☒ | ☐ Plants |

## Methods

| n/a | Involved in the study |
|---|---|
| ☒ | ☐ ChIP-seq |
| ☒ | ☐ Flow cytometry |
| ☐ | ☒ MRI-based neuroimaging |

## Plants

| Seed stocks | N/A |
|---|---|
| Novel plant genotypes | N/A |
| Authentication | N/A |

## Magnetic resonance imaging

### Experimental design

| Design type | task, event related |
|---|---|
| Design specifications | fMRI Sites: DCCN & Yale<br>Stimuli were presented for one of three durations (0.5 s, 1.0 s or 1.5 s), followed by a blank period of a variable duration to complete an overall trial length fixed at 2.0 s. Random jitter was added at the end of each trial (mean inter-trial interval of 3 s, jittered 2.5-10 s, with truncated exponential distribution), with each trial lasting approximately 5.5 s. There were 8 runs containing 4 blocks each with 17-19 trials per block, 16 non-targets (4 per category) and 1-3 targets, for a total of 576 trials. Rest breaks between runs (at discretion) and blocks (12 seconds) were included.<br><br>*all other sites collected structural MRIs only; no design specifications to report |
| Behavioral performance measures | fMRI sites: DCCN & Yale<br>Log-linear corrected d'prime, false alarms (FA) and reaction times (RT) were computed per category and stimulus foration, separately (FAs were also calculated per task relevance, without duration), and per modality (iEEG, MEG, 'MRI). These measures were compared with Linear/Logistic mixed models, where appropriate. For the former, we ·eport ANOVA omnibus F tests, and for the latter, omnibus x' test from an analysis of deviance. We approximated fogrees of freedom using the Satterthwaite method. Pairwise t-tests following significant interactions were Bonferroni :orrected. To estimate Bayesian Information Criterion (BIC) differences between the original and null logistic models, we used the p-values and sample size (Wagenmakers: https://psyarxiv.com/egydq; p_to_bf package in R). |

*all other sites collected structural MRIs only; no behavioral performance measures to report

## Acquisition

**Imaging type(s)**

anatomical T1 scan (MRI) - NYU, Harvard, WI, BU, PKU, DCCN, Yale
functional MRI (fMRI) - DCCN and Yale

**Field strength**

3T

**Sequence & imaging parameters**

MEG - BU:
(Tl); 32-channel head coil (TR/TE= 2000/2.03; Tl= 880ms; Flip angle=S degrees; FOV=256 x 256 x 208; slices= 208; 1mm isotropic voxels)

MEG-PKU:
(Tl); 64-channel head coil (TR/TE= 2530/2.98ms; Tl = 1100 ms; 7° flip angle; FOV = 256x256x208 mm; 198 slices;l mm isotropic voxels, GRAPPA)

fMRI (DCCN & Yale):
Tl; 32-channel head coil, anatomical Tlw MPRAGE images (GRAPPA acceleration factor= 2, TR/TE= 2300/3.03 ms, 8° flip angle, 192 slices, 1 mm isotropic voxels)T2; whole-brain T2*-weighted multiband-4 sequence (TR/TE= 1500/39.6 ms, 75° flip angle, FOV = 210 mm, 68 slices, voxel size 2 mm isotropic, A/P phase encoding direction, BW = 2090 Hz/Px) EPI sequence; CMRR MB-4, TR/TE= 1500/39.6 ms, 68 slices, voxel size 2 mm isotropic, 75° flip angle, A/P phase encoding direction, FOV = 210 mm, BW = 2090 Hz/Px

**Area of acquisition**

whole-brain scan

**Diffusion MRI**  ☐ Used  ☒ Not used

## Preprocessing

**Preprocessing software**

MEG Sites (BU & PKU):
Freesurfer 6.0.1

fMRI Sites (DCCN & Yale):
Source DICOM data were converted to BIDS using BIDScoin v3.6.3.
(f)MRI data was preprocessed using fMRIPrep 20.2.3, based on Nipype 1.6.1. In addition, analysis specific preprocessing were performed using FSL 6.0.2 and custom Python scripts using the following packages: NumPy 1.19.2, Pandas 1.1.3, NiBabel 3.2.2, SciPy 1.8.0, Matplotlib 3.3.2 and Scikit-learn 0.23.2.
fMRIPrep anatomical data preprocessing
The Tl-weighted (Tlw) image was corrected for intensity non-uniformity (INU) with N4BiasFieldCorrection, distributed with ANTs 2.3.3 [RRID:SCR_004757], and used as Tlw-reference throughout the workflow. The Tlw-reference was then skull-stripped with a Nipype implementation of the antsBrainExtraction.sh workflow (from ANTs), using OASIS30ANTs as target template.

Brain tissue segmentation of cerebrospinal fluid (CSF), white-matter (WM) and gray-matter (GM) was performed on the brain-extracted Tlw using fast [FSL 5.0.9, RRID:SCR_002823].
Brain surfaces were reconstructed using recon-all [FreeSurfer 6.0.1, RRID:SCR_001847], and the brain mask estimated previously was refined with a custom variation of the method to reconcile ANTs-derived and FreeSurfer-derived segmentations of the cortical gray-matter of Mindboggle [RRID:SCR_002438].

Analysis-specific functional preprocessing
Additional, analysis-specific, fMRI data preprocessing was performed using FSL 6.0.2 (FMRIB Software Library; Oxford, UK; Smith et al., 2004], Statistical Parametric Mapping (SPM 12) software (Penny et al., 2007), and custom Python scripts. Functional data for univariate data analyses will be spatially smoothed (Gaussian kernel with full-width at half-maximum of 5 mm), grand mean scaled, and temporal high-pass filtered (128 s). No spatial smoothing was applied for multivariate analyses.

**Normalization**

BU & PKU:
All steps included in Freesurfer reconall

DCCN & Yale:
Volume-based spatial normalization to one standard space (MNl152NLin2009cAsym) was performed through nonlinear registration with antsRegistration (ANTs 2.3.3), using brain-extracted versions of both Tlw reference and the Tlw template.

**Normalization template**

BU & PKU:
Freesurfer "fsaverage"

DCCN & Yale:
ICBM 152 Nonlinear Asymmetrical template version 2009c (Fonov et al., (2009); RRID:SCR_008796; TemplateFlow ID: MNl152NLin2009cAsym).

**Noise and artifact removal**

BU & PKU:
-all steps included in Freesurer recon all

DCCN & Yale:
MRI data quality control was performed using MRIQC 0.16.1.
fMRIPrep functional data preprocessing
Head-motion parameters with respect to the BOLD reference (transformation matrices, and six corresponding rotation and translation parameters) were estimated before any spatiotemporal filtering using mcflirt [FSL 5.0.9].

A reference volume and its skull-stripped version were generated using a custom methodology of fMRIPrep. Several confounding time-series were calculated based on the preprocessed BOLD: framewise displacement (FD], DVARS and three region-wise global signals.
FD was computed using two formulations following Power (absolute sum of relative motions, Power et al., 2014) and Jenkinson (relative root mean square displacement between affines, Jenkinson et al., 2002).
FD and DVARS were calculated for each functional run, both using their implementations in Nipype (following the definitions by Power et al., 2014). The three global signals were extracted within the CSF, the WM, and the whole-brain masks. Additionally, a set of physiological regressors were extracted to allow for component-based noise correction (CompCor, Behzadi et al., 2007).
Principal components were estimated after high-pass filtering the preprocessed BOLD time-series (using a discrete cosine filter with 128s cut-off) for the two CompCor variants: temporal (tCompCor) and anatomical (aCompCor). tCompCor components were then calculated from the top 2% variable voxels within the brain mask. For aCompCor, three probabilistic masks (CSF, WM and combined CSF+WM) were generated in anatomical space.
The implementation differs from that of Behzadi et al. in that instead of eroding the masks by 2 pixels on BOLD space, the aCompCor masks are subtracted a mask of pixels that likely contain a volume fraction of GM. This mask is obtained by dilating a GM mask extracted from the FreeSurfer's aseg segmentation, and it ensures components are not extracted from voxels containing a minimal fraction of GM. Finally, these masks are resampled into BOLD space and binarized by thresholding at 0.99 (as in the original implementation).
Components are also calculated separately within the WM and CSF masks. For each CompCor decomposition, the k components with the largest singular values are retained, such that the retained components' time series are sufficient to explain 50 percent of variance across the nuisance mask (CSF, WM, combined, or temporal). The remaining components are dropped from consideration. The head-motion estimates calculated in the correction step were also placed within the corresponding confounds file. The confound time series derived from head motion estimates and global signals were expanded with the inclusion of temporal derivatives and quadratic terms for each (Satterthwaite et al., 2013).
Frames that exceeded a threshold of 0.5 mm FD or 1.5 standardised DVARS were annotated as motion outliers. All resamplings were performed with a single interpolation step by composing all the pertinent transformations (i.e. head-motion transform matrices, susceptibility distortion correction when available, and co-registrations to anatomical and output spaces). Gridded (volumetric) resamplings were performed using antsApplyTransforms (ANTs), configured with Lanczos interpolation to minimize the smoothing effects of other kernels (Lanczos, 1964). Non-gridded (surface) resamplings were performed using mri_vol2surf (FreeSurfer).

| Volume censoring | DCCN & Yale: The first three volumes of each run were discarded to allow for signal stabilization. |

## Statistical modeling & inference

**Model type and settings**

Mass univariate: Averaged across runs per participant using FSL's fixed effects analysis and subsequently averaged across participants using FSL's FLAME mixed effect analysis.

Univariate Connectivity Analyses: gPPI analyses were performed on the participant level and contrast maps were averaged across participants.

Multivariate decoding Analyses: Single trial estimates were obtained per each participant and decoding was performed on the participant level using these estimates as features. Decoding accuracies was averaged across participants.

**Effect(s) tested**

Conjunction analyses:
 Areas sensitive to task goal:
 targets> bsl & task relevant = bsl & task irrelevant = bsl

Areas sensitive to task-relevance:
    targets> bsl & task relevant " bsl & task irrelevant = bsl

Putative NCCs:
   (task relevant> bsl & task irrelevant> bsl) V (task relevant< bsl & task irrelevant< bsl)

Decoding analyses:
   Areas with decoding accuracy higher than the chance level

Connectivity analyses:
   Areas showing significant task-related connectivity

**Specify type of analysis:** ☐ Whole brain   ☐ ROI-based   ☒ Both

**Anatomical location(s)** | Freesurfer automatic parselation using the Destrieux 2010 and Wang 2015 atlases

**Statistic type for inference**

(See Eklund et al. 2016)

Univariate analyses: Gaussian random-field cluster thresholding was used to correct for multiple comparisons, using the default settings of FSL, with a cluster formation threshold of one sided $p < 0.001$ ($z \geq 3.1$,) and a cluster significance threshold of $p < 0.05$. Also JZS Bayes factor analyses.

Univariate task-related connectivity analyses: group level gPPI analyses were performed using cluster-based permutation testing ($p < 0.05$ ).

Multivariate analyses: Group level searchlight decoding analysis was performed using cluster-based permutation testing ($p < 0.05$) while group level ROI decoding was performed using one sample permutation test.

Correction | Multivariate analyses: We performed FDR correction ($p < 0.05$) across ROIs for ROI decoding.

## Models & analysis

| n/a | Involved in the study |
|---|---|
| ☐ | ☒ Functional and/or effective connectivity |
| ☒ | ☐ Graph analysis |
| ☐ | ☒ Multivariate modeling or predictive analysis |

Functional and/or effective connectivity | functional connectivity

Multivariate modeling and predictive analysis | Single trial estimates were obtained per participant and used as features for a support vector machines classifier. No feature selection was employed. Performance was evaluated using leave-one-run-one and through training on one condition and testing on the other condition.

