## [Peer Review File · Nature]

Adversarial testing of Global Neuronal Workspace and Integrated Information Theories of Consciousness

Corresponding Author: Professor Lucia Melloni

This file contains all reviewer reports in order by version, followed by all author rebuttals in order by version. Additionally, reviewer #4 version 1 comments is attached at the end of the file.

Version 0:

Reviewer comments:

Referee #1

(Remarks to the Author)

In this manuscript, the authors reported the first set of results of a highly publicized project (Cogitate) generously supported by the Templeton World Charity Foundation. The aim of the ongoing project is to empirically arbitrate between two theories of consciousness in the form of an adversarial collaboration. Overall, the project is strong in large part because of the scale, ample funding, and some aspects of the quality and quantity of the data. Unfortunately, there are several inherent flaws in the experimental design that limit the interpretations of these studies. Although some aspects of the results are informative, novel scientific content isn't yet a major strength of this project. However, the current experiments were skillfully executed, and the data will no doubt be very useful for future studies. Also, an upcoming second set of future studies, as anticipated in the discussion section, sound relatively promising. I'm sure the overall project will mature in time.

The biggest problem here concerns the very nature of this adversarial collaboration, which is also what has given it tremendous publicity and potential for huge impact. There have already been some general reservations expressed by others, regarding the tenuous connections between the hypotheses and the theories (1–3). Despite this, I was still read the manuscript with great interest. Unfortunately, something seems to have gone unexpectedly wrong in the ways the hypotheses were defined and applied in practice. Specifically, the selection of ROIs for IIT, which neglected the formalisms of the theory, excluded areas originally labeled as part of the so-called 'posterior hot zone' (e.g. precuneus, posterior cingulate (4)), and instead included important frontal areas as 'extended' ROIs, will puzzle many, and probably be considered false advertisement by others. Excluding parietal regions for GNWT likewise went against how the theory (GNWT) has been characterized in print. Although the study was overall pre-registered, these critical and somewhat surprising ROI definitions - without which the hypotheses were very much underspecified in practice - were only uploaded to OSF in Dec 2022, and made public as recently as late June 2023.

In other words, while the current studies admirably tested some 'pre-registered' hypotheses made by some theorists, these hypotheses were unfortunately developed and tested in somewhat idiosyncratic ways, such that they end up being basically irrelevant to theories of consciousness, or perhaps even consciousness in general. But the vast amount of data collected will no doubt still be tremendously useful for other researchers and for different scientific purposes. Below I discuss both some strengths and limitations of the manuscript in more detail.

Strengths

Overall, this project is highly commendable in a number of aspects.

1. iEEG coverage of PFC: The authors referred to the electrode coverage of the PFC as "exceptional", and indeed it was. It is often difficult to compare iEEG measures in PFC vs other brain regions, as the density and number of electrodes are often not matched. This makes the current dataset highly valuable.
2. Ample Data: Another remarkable feature of this project is the sheer quantity of data made possible by the generous funding support and outstanding organization effort. This allowed the studies to have statistical power far higher than usual. Also, multiple measurement methods were used, with studies conducted in different labs across the world.
3. Open Science Ethos: Overall, the experiments were well executed. Admirably, they mostly also followed some of the

latest standards of open science practices, including pre-registration, data sharing, etc. For a large-scale project of this kind, such coordination takes considerable effort, and should be applauded.

4. Clarity in Writing: Overall the writing is clear and easy to follow, especially in the sections describing the empirical results.

Limitations

Although there are some significant issues that are of some concern, they mostly stem from experimental design, hypothesis development, and conception. Overall, the studies seem well executed by the trainees involved.

1. The studies tested some hypotheses made by some theorists associated with those two theories. But unfortunately these hypotheses seem to be not so logically connected with, or constrained by, the theories. Specifically, for IIT, the selection of the ROIs were neither based on the mathematical formalisms of the theory (calculation of 'phi') nor previous empirical studies that helped to give rise of the notion of posterior 'hot zone' in the first instance, e.g. Siclari et al 2017 (4). Instead, curiously they look like a set of regions which anyone would have chosen in order to maximize decoding accuracy for classifying visual stimuli. Because IIT is not supposed to be specific only to vision, these ROIs do not appear to represent the theory in general. Notably, these ROIs also exclude the precuneus and posterior anterior cingulate which were supposed to be part of the posterior hot zone, based on previous publications from some of the authors themselves (4). From Extended Data Table 2, one also sees that large parts of the frontal cortex were actually included as extended ROI for IIT, meaning that if effects were found in these areas, they would by stipulation not count against IIT. This seems to go against the general premise of the project, of focusing on frontal vs posterior regions to distinguish between the theories. This important point should perhaps be made more clear up front in the main text; finding out what really happened from Extended Data Table 2 was quite a surprise for myself. For GNWT, not including parietal areas also went against how the theory has been characterized in print. I understand that the intention was to focus on the prefrontal cortex so as to allow the two theories to make different predictions. But logically this seems backwards: if the two theories are indeed empirically indistinguishable, perhaps we should just accept that fact. We can't instead contrive to modify a theory so substantially just for this ad hoc purpose, and still convincingly call it GNWT. Relatedly, these important ROI definitions also seem to be pre-registered relatively late in the project (Dec 2022, version 4 of their OSF upload, by which point data collection must have started if not completed?), and made public even later (late June 2023). The theoretical connection between IIT and functional neuroanatomy at a macro level is itself tenuous, as the authors acknowledge. In justifying the current ROIs, the authors cited in the discussion section a paper that appeared only recently in 2023 (5). Without the ROIs specified a priori, the hypotheses were very much underconstrained. This is especially concerning given how the IIT ROIs ended up looking so very different from what one may expect based on the literature. Perhaps this also compromises the 'pre-registered' status of the studies - an issue that may benefit from more discussion and clarification. In point 4 below I discuss a related problem with the predictions on dynamics. The predictions on connectivity depend on these same ROIs, and are therefore similarly tenuous. Overall, given these ROIs, it is difficult to make a convincing case that the project really tests the theories in a meaningful way.

2. The experiments are novel in that the scale and publicity have been unprecedented, even though this is not the first adversarial collaboration between theorists of consciousness (6). However, in terms of scientific content, novelty may be relatively limited. By now, there are already dozens of no-report studies on conscious perception, starting from the 1990s (7–13). Overall these studies converge on the same general conclusion (14), similar to the one presented here. Regarding temporal dynamics (prediction #2), several recent papers using intracranial EEG / ECoG directly addressed this exact question, using a similar approach, with arguably more sophisticated and thorough data analysis than what is presented here, at least in some cases (15–18); these studies should perhaps be cited and discussed. This is not to say that overall the project gives no new information at all. For example, the finding that adding PFC information to the analysis together with early and late visual activity did not help with decoding accuracy at all, even with iEEG, is interesting. However, as mentioned above the areas most important for PFC decoding have been excluded from this analysis, because they have been considered extended ROIs of IIT. I suggest that the authors clearly state in the main text that the analysis was performed after removing some frontal areas. Otherwise, statements such as "decoding of category and orientation was found to be the same, or to decrease, when including PFC" might be misleading. I also suggest that the authors try to include these areas in an independent analysis, as most people actually interested in the broader scientific debate would probably not see the logic in excluding those frontal areas.

3. The lack of non-conscious conditions unfortunately limits what we can say about consciousness with the present studies. This led to an uncontrolled stimulus confound, which could very well explain most of the presented results, especially in early visual areas. I have no doubt that the subjects consciously perceived the stimuli, but this doesn't mean that the conscious perception itself rather than the sheer presence of the stimuli was the explanation of the relevant brain activity. Just because conscious perception occurs doesn't mean that nonconscious perception doesn't also occur in parallel. Without a non-conscious condition, which has been standardly implemented in studies on the topic since the 1990s, the relationship between the current studies and consciousness per se is tenuous.

4. One interesting and novel prediction here concerns the activity in PFC associated with stimulus offset, supposedly 'made by GNWT'. While I appreciate that it was pre-registered, I'm not sure I agree with the logic of the prediction. I also could not find in print any reference supporting this prediction, or linking it to GNWT. When a random, i.e. not fully expected, stimulus arises, there is a considerable amount of information to be updated within the workspace, according to the theory. But upon stimulus offset, the next frame is invariably the same: blank. As some of the authors are no doubt aware, expectation can silence prediction errors. There is just overall less unpredictable information to be learned or to be updated over offset as compared to onset. As such, while the theorist representing GNWT here (Dehaene) made the prediction that there will be a

high level of activity at offset, I'm not sure GNWT or its broader community of supporters would have necessarily made that same prediction. In non-human primate electrophysiology there is known offset activity associated with stimulus offset, but that was observed in the visual cortex not PFC, and the magnitude is indeed considerably smaller than for stimulus onset (19).

5. The authors describe the chosen methods of combined fMRI, MEG, and iEEG, as capable of providing "exquisite temporal and spatial precision". I'm not sure this is right, especially regarding spatial resolution. This is a point that the authors seem to be well aware of, and was supposedly the reason the Cogitate project is also conducting invasive electrophysiological studies in non-human primates. Given the chosen emphasis on PFC, and the known inadequacies of current measurements relative to those that will be used in ongoing experiments within the same project, releasing the present results as anything other than preliminary seems somewhat premature.

6. The authors claim that "[t]he main discrepancy between the theories is [...] the necessity of PFC." However, according to the common definition of the neural correlates of consciousness (NCC), 'necessity' was never meant to be a requirement (14, 20); it is 'joint minimal sufficiency' that is the relevant logical concept. Further, if the intention is really to test for necessity as stated, the current methods may be rather suboptimal. Causal manipulation (e.g. ablation / inactivation, or even TMS) may be more appropriate and convincing.

7. The rationale of the pNCC analysis seems to rely on the assumption that a voxel/vertex is either contaminated by confounds e.g. reporting, or not. But of course, this is a false dichotomy, as a bit of brain tissue at this scale can very well serve multiple functions. In fact, this is true even for single neurons. There could be both linear and non-linearly interactive effects of both reporting and conscious perception. So I'm not so sure whether the conclusions here are valid.

8. One finding is that orientation information for stimuli, e.g. comparing faces viewed at different viewpoints, seems not to be decodable from PFC. However, one problem is that this information was never task relevant. So we do not know if such information would have been decodable, had it been made task relevant in some conditions. The interpretability of this finding could have been improved with a simple experimental design fix, which is to factorially manipulate task relevance for both category and orientation, in a simple 2x2 design.

9. Relatedly, the current manipulation of task irrelevance was also perhaps not the most effective. Subjects still had to attend to the irrelevant targets and made a decision to specifically not respond. A better manipulation may be to make the features or targets totally irrelevant to the task in some blocks (but not others), just as in how orientation was irrelevant throughout the current experiments.

Minor Issues

1. It would be useful to give more details regarding how the ROIs for the two theories were selected, including the rationale, empirical papers on which the considerations were based, and give specific references as needed. If this was all done based on intuitive guesses, please be transparent and say so. It may also help to provide some explanation as to why these important ROI definitions were only pre-registered late in the process (Dec 2022, in v4 of the OSF upload), and made public even later, and specifically whether it means that they were finalized only after data collection and analysis have started. For IIT, please also explain why no phi calculation seems to be involved here, and whether the original conception was to involve such calculation. Also, what exactly happened to the results of Siclari et al (2017) (4) which I thought were the (original?) empirical basis for the idea of "posterior hot zone" for IIT?

2. Some positive MEG results were found for the PFC but they were somewhat written off in the light of lack of both fMRI and iEEG activities identified for the same contrast. Overall I agree with the interpretation. But please clarify if such interpretation was based on pre-registered criteria. If not, please discuss whether this undermines the overall principle of anti-HARKing somewhat. Perhaps this is just the nature and limitations of this kind of exercise, but please be more transparent about this point, and discuss accordingly.

3. The authors wrote that "[a]ll research was conducted by theory-neutral teams to guard against confirmatory bias." But what is the evidence for this? Importantly, as I have pointed out above (Limitations point 1), in a sense, arguably the entire project is relatively theory-neutral. What is more important may just concern those specific ROIs, on which the specific hypotheses were based. Is there any record of whether the researchers a priori found some of these hypotheses to be more plausible than others? To the extent that there is likely some such bias, at least in some, can we really say that everyone is truly neutral just because they are not currently in the same labs as the theory PIs? In addition, it may also be helpful to clarify the theory-neutral status of the executive (non-theory) PIs.

4. The terms posterior "hot zone" and "posterior cortex" are often used. But these are not specific or meaningful anatomical labels suitable for an academic journal. As we see from Limitation #1, these unconventional labels really allow too much flexibility for the hypotheses to be meaningfully testable. Given the ROIs, I recommend just calling these early and late visual areas; otherwise would the authors have chosen these areas for another study testing IIT using other stimuli from a different modality? If the authors insist on using their own neologism, please give precise anatomical definitions for these labels.

5. The authors wrote that "[t]ogether, we identified divergent predictions of the theories and jointly developed an experimental design to test them". Please be more specific here, especially regarding who generated these predictions and who designed the studies (and who didn't). In the author contribution statement many authors seem to be involved in 'Conceptualization', but my understanding is that at least some of them played no role at all in experimental design.

Conceptualization is a relatively vague term. Please be more specific, as a clearer delineation would help with some authors, especially trainees, who might not have been given any chance to have any input to the process, to be freed from being held responsible for what some may see as inherent design flaws in the project. Some flaws may be inevitable given the challenges involved in organizing such a large scale project, but still, clarity can only help.

6. In different places the authors describe the two tested theories as “leading” and “two of the most well-established theories in the field”. I’m not sure about whether these somewhat subjective phrases are appropriate here. In this context, it may be helpful to also discuss an alternative interpretation as supported by some empirical studies (21, 22). Based on these studies there seems to be an argument to be made that one of the theories, IIT, was never really so well received within the scientific community, and that the current popularity may have something to do with hyperbolic promotion in the media. Therefore the status of being “leading” and “well-established” may be questionable.

7. One strength of these studies was the large number of subjects. Can we more explicitly discuss what was the statistical power achieved here. Specifically, can one not do power calculations based on the preliminary analysis from one-third of the data, and accordingly estimate the power one would get based on using the remaining two-third of the data? And what is the rationale for doing this 1:2 split? Why not any other ratio? Is there an empirical basis or reference for that?

8. In different places the authors refer the readers to please “see supplementary”. Please be more specific about what section or figure of the supplementary section they were referring to.

Hakwan Lau
Jul 23, 2023

References

1. S. Fleming, The state of consciousness science – and why the media got it wrong. *The Elusive Self* (2023), (available at <https://elusiveself.wordpress.com/2023/07/20/the-state-of-consciousness-science-and-why-the-media-got-it-wrong/>).
2. A. Seth, Finding the Neural Correlates of Consciousness Is Still a Good Bet. *Nautilus* (2023), (available at <https://nautil.us/finding-the-neural-correlates-to-consciousness-is-still-a-good-bet-352054/>).
3. S. Reardon, “Outlandish” competition seeks the brain’s source of consciousness. *Science* (2019), doi:10.1126/science.aaz8800.
4. F. Siclari, B. Baird, L. Perogamvros, G. Bernardi, J. J. LaRocque, B. Riedner, M. Boly, B. R. Postle, G. Tononi, The neural correlates of dreaming. *Nat. Neurosci.* 20, 872–878 (2017).
5. A. Watakabe, H. Skibbe, K. Nakae, H. Abe, N. Ichinohe, M. F. Rachmadi, J. Wang, M. Takaji, H. Mizukami, A. Woodward, R. Gong, J. Hata, D. C. Van Essen, H. Okano, S. Ishii, T. Yamamori, Local and long-distance organization of prefrontal cortex circuits in the marmoset brain. *Neuron* (2023), doi:10.1016/j.neuron.2023.04.028.
6. H. Zhou, M. Davidson, P. Kok, L. Y. McCurdy, F. P. de Lange, H. Lau, K. Sandberg, Spatiotemporal dynamics of brightness coding in human visual cortex revealed by the temporal context effect. *Neuroimage.* 205, 116277 (2020).
7. N. Block, What is wrong with the no-report paradigm and how to fix it. *Trends Cogn. Sci.* 23, 1003–1013 (2019).
8. J. K. Hesse, D. Y. Tsao, A new no-report paradigm reveals that face cells encode both consciously perceived and suppressed stimuli. *Elife.* 9 (2020), doi:10.7554/eLife.58360.
9. E. Hatamimajoumerd, N. A. Ratan Murty, M. Pitts, M. A. Cohen, Decoding perceptual awareness across the brain with a no-report fMRI masking paradigm. *Curr. Biol.* 32, 4139–4149.e4 (2022).
10. V. Kapoor, A. Dwarakanath, S. Safavi, J. Werner, M. Besserve, T. I. Panagiotaropoulos, N. K. Logothetis, Decoding internally generated transitions of conscious contents in the prefrontal cortex without subjective reports. *Nat. Commun.* 13, 1535 (2022).
11. P. U. Tse, S. Martinez-Conde, A. A. Schlegel, S. L. Macknik, Visibility, visual awareness, and visual masking of simple unattended targets are confined to areas in the occipital cortex beyond human V1/V2. *Proc. Natl. Acad. Sci. U. S. A.* 102, 17178–17183 (2005).
12. E. D. Lumer, G. Rees, Covariation of activity in visual and prefrontal cortex associated with subjective visual perception. *Proc. Natl. Acad. Sci. U. S. A.* 96, 1669–1673 (1999).
13. S. Kouider, S. Dehaene, A. Jobert, D. Le Bihan, Cerebral bases of subliminal and supraliminal priming during reading. *Cereb. Cortex.* 17, 2019–2029 (2007).
14. H. Lau, The Unfinished NCC Project (2022), doi:10.1093/oso/9780198856771.003.0003.
15. G. Vishne, E. M. Gerber, R. T. Knight, L. Y. Deouell, Distinct ventral stream and prefrontal cortex representational dynamics during sustained conscious visual perception. *Cell Rep.* 42, 112752 (2023).
16. R. Broday-Dvir, Y. Norman, M. Harel, A. D. Mehta, R. Malach, Perceptual stability reflected in neuronal pattern similarities in human visual cortex. *Cell Rep.* 42, 112614 (2023).
17. E. M. Gerber, T. Golan, R. T. Knight, L. Y. Deouell, Cortical representation of persistent visual stimuli. *Neuroimage.* 161, 67–79 (2017).
18. E. Podvalny, E. Yeagle, P. Mégevand, N. Sarid, M. Harel, G. Chechik, A. D. Mehta, R. Malach, Invariant Temporal Dynamics Underlie Perceptual Stability in Human Visual Cortex. *Curr. Biol.* 27, 155–165 (2017).
19. S. L. Macknik, M. S. Livingstone, Neuronal correlates of visibility and invisibility in the primate visual system. *Nat. Neurosci.* 1, 144–149 (1998).
20. M. Michel, H. Lau, On the dangers of conflating strong and weak versions of a theory of consciousness. *Philosophy and the Mind Sciences* (2020) (available at <https://www.philosophymindscience.org/index.php/phimisci/article/view/54>).
21. M. Michel, S. M. Fleming, H. Lau, A. L. F. Lee, S. Martinez-Conde, R. E. Passingham, M. A. K. Peters, D. Rahnev, C.

Sergent, K. Liu, An Informal Internet Survey on the Current State of Consciousness Science. *Front. Psychol.* 9, 2134 (2018).
22. A. W. K. Yeung, C. A. Cushing, A. L. F. Lee, A bibliometric evaluation of the impact of theories of consciousness in academia and on social media. *Conscious. Cogn.* 100, 103296 (2022).

Referee #2

(Remarks to the Author)

//Summary

A. Summary of the key results

The authors have demonstrated that it is possible to investigate competing theories of consciousness in a comprehensive and theory neutral way.

B. Originality and significance: if not novel, please include reference

The approach in this project is groundbreaking, and a major contribution to - and advancement of - the field.

C. Data & methodology: validity of approach, quality of data, quality of presentation

The data are impressive, and the approach is one-of-a-kind (but will hopefully become the standard in the future)

D. Appropriate use of statistics and treatment of uncertainties

I am not qualified to judge this.

E. Conclusions: robustness, validity, reliability

I had hoped for stronger conclusions, but understand why some of these may be postponed to future work. Do see my comments below though.

F. Suggested improvements: experiments, data for possible revision

I have no suggestions for improvements of the empirical work, but below are suggestions related to this manuscript specifically.

G. References: appropriate credit to previous work?

Did not find any critical omissions.

H. Clarity and context: lucidity of abstract/summary, appropriateness of abstract, introduction and conclusions

See below for comments.

//

Dear Authors,

Congratulations on this major accomplishment. I am honored and grateful to have been given the opportunity to review the results of something that I (and pretty much our entire field) have been eagerly following and anticipating over the last several years. You have done an excellent job, and your work is (to quote your manuscript) "paradigmatically scientific".

I have found no critical issues with this manuscript, but I do have a range of comments and recommendations. For this reason, I will recommend minor revisions, even though I do not think acceptance of this manuscript should be conditional on any one revision in relation to the comments below (let alone all of them). I do hope, however, that you will see the value in at least some of the comments and recommendations and make changes accordingly. But I will leave this at your discretion. Importantly, if the editors judge that any of my comments are of such character that acceptance should be conditional on them being addressed, I defer to their judgement on this.

Below are my comments in chronological order (in relation appearance in the manuscript) tagged with line numbers. The issues I see as most important have been given a headline surrounded by **.

Kind regards

Asger Kirkeby-Hinrup

//Detailed comments

Lines 91-99: This first paragraph is maybe a bit too succinct, leading to minor inaccuracies. For instance, it seems to suggest that the hard problem and explanatory gap 'caused' ("led to": line 93) theories of consciousness, when in fact many theories (including three considered in the adversarial collaboration project, i.e. HOT, GWT, and first order theories [which later became RPT]) were around before Chalmers' 1995 paper. While they generally do not predate Levine's 1983 paper, the formulation is still somewhat awkward. (I assume what the authors want to say is something along the lines of "the perplexing phenomenon, the issues with which have been described as the hard problem, or in terms of an explanatory gap, has led to..."). In any case, I'd recommend spending one or two lines more to spell out properly the background and connect the many claims better, which would make the introduction less abrupt and more readable.

Line 94: A minor point possibly worth noting that according to the Signorelli et al. paper referenced by the authors in line 94), some theories do not aim "to explain the subjective nature of consciousness" (line 92). In fact GNWT and IIT are on opposite

sides of the quantity-quality distinction proposed in the Signorelli paper (Signorelli et al., 2021, pp. 5-6. see also fig. 2 on p. 7). This, however, is not of any huge consequence here, since the crux of this manuscript does not hinge on this (and if Stan does not object to GNWT “trying to explain subjective nature of consciousness”, this point can be safely ignored).

Line 124-126: The formulations of the two positions in prediction #1 are not mutually exclusive. It is possible that conscious content can be “instantiated primarily in posterior brain areas” even if the PFC is necessary. Do the authors intend something stronger? (see below regarding formulation of prediction#1)

Line 251: This seems to be the best formulation of prediction#1 so far, but it is still a little ambiguous (in the sense noted above) and would perhaps be better characterized by saying “involvement of PFC” or “activation of PFC” rather than “necessity of PFC”. (see below regarding formulation of prediction#1)

****Figure 1****

Figure 1.a. Formulation of Prediction#1 is nontrivially different from the formulation in Line 124-126. The formulation here seems better (but above note regarding the formulation in line 124-126 applies here as well). (see below regarding formulation of prediction#1)

Figure 1.b seems out of place in the middle of a figure laying out empirical elements. My recommendation is to remove it, as it does not contribute significantly to advancing any point in the paper. Everything it says already is expressed in lines 166-173, and the illustration does not add anything on top of those lines. If the authors/editors decide to keep it, I recommend moving it into its own figure (to be placed around line 173), so it does not disrupt the “empirical flow” in figure 1.

Figure 1.h: Scales on top row are not identical (going to 0.14 and 0.12 respectively) yet with same color coding. This is potentially misleading (e.g. white color means different things in the pictures). Also, font formatting on top row middle picture seems off.

****Prediction#1****

Comparing the way this is presented in the paper with the pre-registered predictions (linked in line 261, the pdf file named “Preregistration_ver4.0-FINAL.pdf”) there seems to be some conflation going on. In the preregistered prediction (PRP), there are actually two predictions that seem to be collapsed into Prediction#1 in the manuscript. In PRP there is a prediction related to “Neural Activations related to consciousness”, which says something that sounds very much like the way Prediction#1 is presented (especially in the earlier sections I mentioned above), namely: “for GNW, activation of prefrontal areas is a prerequisite for conscious experience, while IIT, in contrast, submits that activity in posterior areas is sufficient for consciousness” (page 6 of Preregistration_ver4.0-FINAL.pdf). Immediately after in the PRP (also p. 6) there is a prediction related to decoding, which seems to be the prediction that is addressed in this manuscript. In the PRP that prediction is phrased “In short, GNW predicts that the content of consciousness should be decodable from prefrontal areas, whereas IIT predicts that they should be maximally decodable from posterior areas.”. This decoding prediction aligns much better with what would be the proper wording of prediction#1 in this manuscript. I would strongly encourage the authors to rephrase the presentations of Prediction#1 (in the introduction and figure 1, and maybe parts of the text under the headline prediction#1) to be cached in terms of the decoding prediction, rather than varieties of ‘instantiated primarily’ and ‘necessity’. This would: 1) make the prediction clearer, 2) prevent readers balking at its non-exclusivity (as I did above), and 3) align better with the data presented (e.g. especially the conclusion of prediction#1 in line 336-37).

Line 423: The combination of “despite” and “serendipitous” seems weird. Do you mean “Due to serendipitous electrode implantation”? or maybe something like “despite our lack of control over electrode implantation”? (The way I read it, the point is that you were “lucky” that the patients had electrodes that fitted well with your ROIs. On this reading “despite” doesn’t mesh well with “serendipitous”)

Line 503: “theories predictions” (missing possessive apostrophe).

Line 515: “theories predictions” (missing possessive apostrophe).

****Figure 3****

3.a and 3.c may need to be shown in bigger format (or use more contrasting colors). It is almost impossible to see the purple and pink electrodes on 3.a (IIT x Category/Face) and the Black electrode on 3.c (GNWT x phasic onset).

Furthermore, the legend on 3.a seems to show part of the legend for 3.c (last line saying GNWT in green), and 3.c does not have a legend.

3.e Line 552: shouldn’t it be “for category (left and middle left), identity (middle right) and orientation (right)” ?

3.f Line 559: shouldn’t it be “as in Figure 3e” ?

****Line 587-588: “the activity was not found in the gamma frequency predicted by IIT”. ****

I could not find any previous point in the manuscript where it was stated that IIT predicted specifically Gamma activity (maybe I missed it?). Gamma is mentioned as part of the GNWT prediction in line 569. Should this be reformulated to cover both the prediction of IIT and GNWT? I checked against the preregistered predictions (Preregistration_ver4.0-FINAL.pdf referred to above), and do not see any specification that it has to be gamma in the IIT prediction on synchronization (p.28 of PRP). It does mention gamma in the IIT prediction on “activity pattern analysis” immediately before (p. 27 on PRP), is this supposed to ‘carry over’ to the synchronization prediction? (I could not find any explicit statement on this in the PRP).(The connection between IIT and gamma is mentioned again in line 633)

Line 733: “Siloes” is a peculiar word choice here. (Does this refer to “silo theory”?), maybe use ‘echo chambers’ or something more recognizable to the average reader (I had to google silo theory..). Affixed to “theoretical siloes” there is a

reference to your Contrast paper, but “theoretical siloes” is not mentioned anywhere in that either(?).

Line 756-757: The claim that “orientation is a fundamental property of conscious experience” is maybe a bit strong. First of all, it is only a property of visual conscious experience, not conscious experience as such. Secondly, even within the visual domain there is a case to be made that only low level properties (colors, edges etc) are ‘fundamental’ in the sense that other properties are derived from these. (Also observe that there is a possible ambiguity here in the sense that when ‘orientation’ is used here [in my understanding] it refers to the direction a person is facing [c.f. figure 1.d], whereas there is another sense in which orientation is a fundamental [low level] property, namely when it refers to the angle that selective neurons in V1 are sensitive to, but this is not the meaning referred to in the manuscript [I think]).

Line 784: “Primal” is a weird word choice here. Do you still mean ‘fundamental’?

Line 812: From what I remember (its been a while since reading), Popper (referred to here) was concerned with falsifying theories through testing predictions, not falsifying predictions. This is a minor point though.

Line 923-926: Would this interpretation of the failure to see ignition at stimulus offset not entail more or persistent PFC activation as it seems to invoke cognitive (PFC reliant) processes (similar to Block’s bored monkey consideration)? Here is a (speculative) alternative interpretation that (I think) is a better explanation for GNWT for the lack of offset ignition (this is an elaboration on the considerations found in lines 931-935): If one assumes ignition serves to update on the arrival of new information, offset should not trigger ignition in cases where the disappearance of the stimulus results in less information than before (i.e. a grey screen contains less information than a grey screen with an image). Alternatively, there is another interpretation where the “grey color behind the stimulus image” was already inferred from (contained in) the stimulus image. If either of these is the case then there would be no need for updating (ignition) on the stimulus offset, because either there was ‘less’ information on offset, or the information was already contained in the previous broadcast.

Conclusion (Cogitate consortium)

For a project of this size, cost, duration, and ambition I think this conclusion is inadequate. You are not wrong that science is a collaborative (social) enterprise but leaving the reader with “make of this what you will” is a letdown. You have done excellent science (!) and the readers (and I guess maybe also your collaborators and the funders) deserve to know your conclusions. The conclusion does not have to appoint a winner (you could rename it to ‘concluding remarks’, so readers do not expect a “conclusion”), and I agree that the consortium should preserve neutrality, but still there are plenty of things you could bring up in this space. The following points are just a few suggestions.

1. Both the IIT and GNWT replies above indicate to limitations in the data (Line 836 and 907 respectively). What were these limitations, what do you think about them, could they have been avoided, will they be ameliorated in the next parts of the adversarial collaboration project?
2. Regarding the PFC, IIT and GNWT seem to draw opposing conclusions from the data (lines 841-42 and 895-96 respectively), what are the implications of this? Could this have been avoided?
3. Stan points to the future studies to help resolve outstanding questions here (line 947), maybe foreshadow these studies, discuss how they relate, and why he is optimistic.
4. What are the (long term) prospects of adversarial collaboration to resolve these debates? It is basic scientific practice to revise theories when they make failed predictions. However, when a theory is revised, we would need a whole new round of multi-year-multimillion dollar adversarial collaboration to test the new version. Is there a ‘halting problem’ in this regard? (c.f. “no one changes their mind”).
5. Maybe some general considerations and reflection on the project? What would you have done differently? What are the positive upshots? Has the current test led to new ideas for other tests? Have you had ideas on how to refine the paradigms used in this test that may avoid shortcomings?
6. You have set a magnificent and high bar for comparison of theories. However, the cost (in money, expertise, and time) is prohibitively high for almost everyone else in the field. Is there a way to emulate this approach on a smaller scale? Is there any way to make this approach “consumer grade”?

Line 1509: “missing data were excluded” sounds weird. Do you mean trials with missing data? Or do you mean cases where either gaze or pupil data were missing? (same in line 1518-19)

Line 1548-1552: Is there a reason the text here is grey rather than black? (maybe it’s a convert to pdf glitch?) (Same with lines 1634-1636 and lines 1654-1662, maybe I’ve been staring at my monitor for too long?)

Lines 1864-1871 (especially lines 1870-71) This is both very interesting and important and deserves to be in the manuscript (either in the pNCC section or the conclusion). Especially because of the potential implications outside of evaluating IIT and GNWT.

Extended figure 10: I think this deserves to be in the main manuscript.

Referee #3

(Remarks to the Author)

This first paper from the Cogitate consortium reports data from iEGG, fMRI and MEG experiments of visual object processing, collected across several different labs and involving a thorough pre-registered approach. A range of univariate

and multivariate analyses are applied to the imaging data to evaluate 3 predictions (about neural correlates, timing and connectivity) made by advocates of two theories of consciousness – global neuronal workspace theory (GNWT) and information integration theory (IIT). I admire the adversarial collaboration and pre-registration approach and applaud all the authors involved here for a considerable amount of high-quality experimental work. The iEEG data from >30 patients with widespread electrode coverage is particularly impressive and will provide a unique resource for the community.

That said, I have serious concerns about the scientific weight being placed on the current results for each of the theories, and for consciousness science more broadly. First, the results concern neural correlates of visual processing, not visual consciousness – as unlike in the ongoing Exp. 2 from the Consortium, there is no experimental operationalisation of conscious vs. unconscious processing here. The task is relatively standard, involving brief presentations of visual objects (faces, objects, letters, false fonts) from different viewpoints and for different stimulus durations, with an additional manipulation of task relevance. Neural decoding of many of these stimulus properties is maximal in posterior (visual) cortex – this is to be expected from mainstream vision science (eg the many papers from Kanwisher, Tsao, di Carlo, Rust and others). There is also no investigation of mechanism or computation (although future studies using these open data might aim for this). Instead, all the predictions pertain to neural correlates of visual processing over time and space, using different imaging techniques.

These design choices are justified by the consortium by appealing to the power of a simple suprathreshold task for falsifying tenets of each of the two theories. The scientific weight of the paper therefore turns on whether the neural measures provide strong tests of the theories of consciousness under consideration, IIT vs. GNWT. Here unfortunately it becomes clear as one reads the paper that this theoretical operationalisation is at best asymmetric. GNWT is under clearer test, as from its inception it has made anatomical commitments to the involvement of a frontoparietal network in conscious processing. But I struggled to understand the provenance of the IIT test (see specific comments below). As Dehaene points out in the discussion, the mathematical backbone of IIT was not invoked, and the results showing posterior cortical decoding of object features are “just what any physiologist familiar with the bottom-up response properties of those regions would predict, since visual neurons still respond selectively during inattention or general anesthesia” (p. 27). This is unfortunately not just an adversarial discussion point, but a serious indictment of the entire project and its relevance to either consciousness or IIT (and hence it’s somewhat surprising that the GNWT side signed off on these predictions in the first place).

Specific comments:

- I tried my best to track down the origin of the key IIT hypothesis (that is invoked by all the Predictions, but especially Prediction 1) that posterior cortex should be more associated with consciousness than other regions. In the current paper (p. 4 and p. 26), the reader is referred to Tononi et al. (2016) NRN (reference 5), where there is discussion of the potential physical substrates of consciousness (PSC) based on IIT. But the main claim in Tononi et al. (2016) was that cortex should have higher phi than the cerebellum – there was little mention of PFC vs. posterior cortex (or a putative hot zone). It was also proposed that large-scale computer simulations based on the known anatomy of cortical circuits would be needed to evaluate network predictions for phi. Such simulations were not conducted for the current study, and instead a hypothesis that posterior cortical regions should be the “main complex” is based on an assertion that they have “dense local connections arranged topographically into a hierarchical, divergent-convergent 3D lattice” (p. 26). I don’t see how one can establish this is the case for posterior but not prefrontal cortex without detailed knowledge and evaluation of the relevant micro-anatomy and connections involved. I then went back to the pre-registration document which states (p. 5) “current theoretical and neuroanatomical considerations suggest that a complex of maximum phi is likely to reside in the posterior cerebral cortex, in a temporo-parietal-occipital ‘hot zone’ (Koch, Massimini, Boly and Tononi, 2016b)”. But upon reading the cited Koch et al. (2016) NRN paper, IIT was barely mentioned, and the hot zone was instead motivated with reference to empirical data on visual awareness. In summary, then, the posterior cortex prediction is not derived from IIT, and seems chiefly motivated by empirical findings, which is circular. This is especially concerning given that standard visual neuroscience would predict robust decoding of visual stimuli in these regions, as noted above.

- There are similar albeit less prominent concerns about the GNWT predictions. Why should GNWT predict activity-silent maintenance, which underpins the onset/offset Prediction 2? The only reference for this given in both the paper and the pre-registration is the Stokes et al. TICS paper – but this paper does not mention GNWT or consciousness and is instead a model of working memory (which presumably could be conscious or unconscious).

- The “putative neural correlates of consciousness” analysis (Fig 5) is puzzling and out of place in the main text (it would be more appropriate in the supplement). The conjunction of contrasts relative to baseline implies the (already implicit) baseline is a special case of the contents of consciousness, and it’s not clear why this should be – as noted above, there is no manipulation of conscious vs. unconscious processing in this study. Instead, it’s a statistical map saying where in the brain is activated / deactivated by visual stimuli using fMRI, after masking out task-related activations. In this light, the widespread stimulus-triggered deactivation is also not surprising given that it looks like this is picking up the default-mode network (though this was difficult to assess from the axial slices).

- Was awareness of stimulus orientation / viewpoint tested in the surprise memory test? I couldn’t see these data reported in the supplement (p. 22-26), but one can imagine that this aspect of the stimulus might have a higher likelihood of being forgotten than other features. This is not to say that subjects were not conscious of orientation, but given orientation plays a strong role in testing Prediction 1, it seems important to establish behaviourally.

Referee #4

(Remarks to the Author)

Referee report on "An adversarial collaboration to critically evaluate theories of consciousness," Melloni et al.

Benjamin Kozuch
University of Alabama
Philosophy Dept.
bkozuch@ua.edu

The article that I reviewed (hereafter, "Article") empirically investigates the issue of what neuroscientific theory of consciousness is correct. Remarkably, the project is a collaboration between proponents of competing theories of consciousness, who are joined by theory-neutral researchers that take a leading role in the experimentation. The goal that the authors of this article had (hereafter, "Authors") was to evaluate two currently popular theories of consciousness, Global Neuronal Workspace Theory (GNW), and Integrative Information Theory (IIT). The work that the Authors carried out was paradigmatically scientific, in that they took a falsificationist approach to evaluating the theories, generating stringent predictions of each theory, then experimentally testing them. The researchers used the best of those methodological and technological tools that are available today, in abundance. The result is some of the more informative neuroscientific data concerning consciousness that we have to date. I recommend that this article be published in Nature.

At the same time, there are ways that the article could be improved and made even more informative. In the report that follows, I describe them.

1.

As stated above, the article focuses on GNW and IIT. However, if we were to produce a complete list of leading theories of consciousness, it would include not just GNW and IIT, but also local recurrency theory (LR) (Lamme 2010) and higher-order theory (HO) (Brown et al. 2019), perhaps along with others. It seems, furthermore, that the results of the experiments presented in the Article shed just as much light on these other two theories (i.e., LR and HO) as it does in the case of the two that the article focuses on. And so for the article to give no discussion of these theories seems to be a missed opportunity for advancing the field of consciousness studies. More importantly, it gives the reader the mistaken impression that the field has narrowed down the candidate theories of consciousness to just to just GNW and IIT, something that seems far from the truth (Michel et al. 2018). I will explain.

HO states that a mental state becomes conscious when it is represented by another mental state. According to how HO is usually construed (e.g., Rosenthal 2011; Brown 2015), the content that becomes conscious will be the content of the HO state (as opposed to the content of the lower-order state). Consider now that it is commonly thought that the kind of HO states that are essential for consciousness will be produced in the PFC (for review: Kozuch 2023). Given this, it seems that HO predicts, just like GNW, that whatever one consciously experiences should be decodable in the PFC. And so it seems that it is not just GNW, but also HO, that makes Prediction #1.

(Worth noting: In some ways, Prediction #1 seems to be a more stringent test of HO than it is of GNW, since HO might be (and often is) construed as requiring activity in just the PFC for consciousness (as long as certain background conditions obtain), whereas GNW takes the PFC to be just one of the members of the constellation of brain areas that GNW says gives rise to consciousness. Because of this, it might be easier for GNW to explain why their Prediction #1 is not met, since the GNW theorist might claim, e.g., that the failure to decode the stimuli orientation in the PFC occurs because such content is actually distributed across the frontoparietal areas, rather than being comprehensively represented in the PFC.)

A similar observation can be made in the case of IIT and LR. According to LR, visual consciousness arises from recurrent loops localized to the visual cortex, ones that are between lower- (V1/V2) and higher- (e.g., LO, IT, V4) level brain areas. These are of course brain areas that are comparable to those that figure into each of the IIT Predictions made in the Article. Given this, we can consider each of the IIT predictions to also be predictions of LR, in which case we can consider the experiments presented by the Authors to confirm and disconfirm LR in precisely the same way that it is taken to confirm and disconfirm IIT.

Notably, this seems to even be true of Prediction 3, which was in some ways much more specific than the first two predictions, since it identified not only those brain areas that should be representing the conscious content, but also the nature of the activity underlying this representation (i.e., synchronized activity between higher- and lower-level brain areas). This seems to fit very well with what we would expect LR to predict. Indeed, one might think that, in the case of experiments that the Researchers ran in this article, LR and IIT are actually empirically indistinguishable: They seem to both make the same predictions as far as what the results of the experiments should be.

The upshot of all this is that I think that the article could benefit greatly from (a) an acknowledgement that IIT and GNW are not the only leading theories of consciousness, and (b) a substantial discussion as to what the results presented in the Article mean for these other theories of consciousness. As well, I should mention again that the article, as it is, seems to implicitly misrepresent the field of consciousness studies, since these omissions might give the reader the impression that these are the only two theories of consciousness that anyone takes seriously anymore, something that is untrue.

2.

So far, I have pointed out how I think that the Article, as it currently is, misses out on opportunities to expand our neuroscientific knowledge of consciousness by showing how the results reported in the Article bear on HO and LR. The

same point can be made about another debate, one concerning a more general distinction in types of theory, this being between Local and Broad theories of consciousness (for distinction, see Michel & Doerig 2021). According to Local Theories (this includes LR and IIT), activity in just the visual cortex (given some background conditions) can be sufficient for visual consciousness; according to Broad Theories (this includes GNW and HO), areas outside of the visual cortex (which are usually considered to include the PFC) are necessary (perhaps in addition to areas in the visual cortex). The relevance of the results presented in the Article to this debate is obvious, given the strong similarities to be found between GNW and HO, and especially given the apparent empirical equivalence of IIT and LR (discussed in Point 1).

It would be good if the Article were to comment on what their results mean for the Local vs. Broad debate, not least of all because this debate is very consequential in regards to the issue of where future research efforts should be directed (i.e., whether it is more worthwhile to pursue the development of Local or Broad Theories of consciousness). It should be noted that one of main points of contention between Local and Broad Theories is what role the PFC plays in consciousness, this of course being one of the issues that the Article focuses on as well. Given this, the Article seems well-poised to comment on the Local vs. Broad debate.

3.

The Authors at one point write:

“When viewing the Mona Lisa (Figure 1c), one experiences it located in visual space, with a specific identity, a specific orientation, and the experience *continues as long as one looks at the painting* (143-45; italics mine).

This gives the reader the impression that all of the authors countenance the following plausible assumption about the nature of consciousness:

[CONTINUOUS] When looking at and focusing on a stimulus, one continues to visually experience it the entire time one is looking at it.

At the same time, one of the predictions that GNW makes in the article is that there will be “brief content-specific ignition in PFC ~0.3-0.5 s after stimulus onset and offset (when the workspace is updated), with content...being stored in a *non-conscious* silent state resembling activity-silent working memory in between” (207, italics mine).

This prediction suggests that the following thesis is true:

[BRIEF] When looking at and focusing on a stimulus, one is *not* conscious of it the entire time one is looking at it, but only when a representation of the stimulus is first introduced into the global workspace (i.e., just when one first starts looking at and focusing on it)

I became puzzled when I got to this part of the Article (around line 207), since what was said earlier in the Article made it seem as if all of the researchers agreed with CONTINUOUS, but now a prediction of GNW was being built around an idea that is in direct conflict with it (as it is based around the thesis of BRIEF).

Something should be added to the article (near line 143) to make clear whether the GNW theorists do or do not agree with taking CONTINUOUS to be one of the starting assumptions that the Authors collectively make.

4.

Continuing on the same topic: The thesis of BRIEF seems extremely controversial, because CONTINUOUS is strongly supported by introspection: As the Authors very plausibly state, when one visually experiences a painting, one has the impression that their experience of the painting “continues as long as one looks at the painting” (144-5). The thesis of BRIEF denies this.

Because of this, when GNW Prediction #2 is being offered, some justification for rejecting CONTINUOUS should be given. Later in the article (p. 27), Dehaene does indicate that he is skeptical about the thesis of CONTINUOUS, and gives justification for rejecting it (doing so by appealing to experimental phenomena such as inattention blindness). However, I do think that it would be better if this rejection were made more up-front in the article, so that the reader could have this in mind throughout the article’s presentation of the experiments, rather than being surprised by it at the end.

(I would think that, for the GNW Theorist, this would strategically make more sense, since if CONTINUOUS is rejected after the experimental results have already been unveiled, this rejection might be more easily mistaken as having been done just to save the theory of GNW.)

5.

It should be noted, however, that rejecting BRIEF seems fairly extreme, since it seems to be saying that, when viewing an object in front of our face for several seconds, we are only conscious of it for 50ms or so, and then the rest of the time...what?

Is it that we fail to have any kind of phenomenal experience in the part of the visual field where the object is represented? Or no visual experience at all, in any part of the visual field? If it is not either of these options, then *what is it* that the subject experiences in this interim?

In justifying the rejection of CONTINUOUS, Dehaene makes the familiar move of appealing to experimental phenomena

such as attentional blink and inattention blindness. But these experiments are typically appealed to in order to argue that our impression that visual experience is rich is mistaken, the alternative hypothesis being that we are only experiencing whatever objects we are attending to; however, what Dehaene is arguing goes beyond this, in that he argues that our impression that visual experience is continuous is wrong.

This difference helps to illuminate what I think is important here, which is that BRIEF seems to be not only a counterintuitive claim, but also a **novel** one, one that goes beyond the usual, already counterintuitive claims that we have heard many times, those about visual experience not containing as much content as it seems.

To me, this means that it would be best if the justification that Dehaene provides for rejecting CONTINUOUS receives supplementation. Some articles by Herzog, Drissi-Daoudi, and Doerig (et al.) might be helpful here, since their work gives some reason for thinking that our impression that visual experience is continuous is mistaken (a good place to start is Herzog et al. 2020).

6.
The stimuli were divided into three categories: target, task-relevant, and task-irrelevant. The motivation for having these three categories was to enable the researchers to subtract the neural activity created by task demands in order to zero in on just the neural activity that is actually constitutive of visual consciousness.

However, the subjects having to ignore the task-irrelevant stimuli still creates task demands. For example, they have to attend to it to determine that it is task irrelevant. This seems to indicate that we cannot take the neural activity that was recorded during presentation of task-irrelevant stimuli to reflect only those processes that are responsible for their consciousness of the stimulus. That the authors are aware of this is implied by the way that they couch some of their claims, e.g., they write, "We focused on the task irrelevant condition as it is **most** diagnostic for neural activity related to consciousness, **minimizing** the contribution of other, potentially confounding, cognitive processes" (408; italics mine). The Authors do not, however, comment further on the fact that there are task demands even when subjects are viewing the task irrelevant stimuli.

It would be good to hear what the Authors thought about how this affects the way that we interpret the results that they gathered. For example, could the ability to decode the category of the stimuli in the PFC be attributed to the task demands present when a subject must ignore a task irrelevant stimulus? Perhaps there are other observations that the Authors could make here.

7.
The Authors' goal to carry out experiments that looked for correlations between neural activity and **multiple** dimensions of conscious experience (e.g., category and orientation) at the same time is highly admirable, and the result of their focusing on this is that we get some highly informative data concerning how stimuli are consciously represented in the brain.

At the same time, it should be noted that their efforts are still very far off from measuring consciousness in a way that truly captures its apparent phenomenal richness: The typical visual experience does not consist merely of the representation of the category to which the object belongs, along with the orientation from which one views it, but so much more: One experiences the object's brightness and hue, its precise shape and location, and experiences the object from a highly specific viewpoint (something more fine-grained than a broad description such as "from the front" or "from the side"); one will also experience the hue and brightness of the object's background. (The list could continue here, but I'll stop so as to not court controversy...)

It should be noted, as well, that there is an active debate concerning whether object category is something that is even part of our conscious experience (qua **phenomenal** experience) (for review: Siegel & Byrne 2017). This means that finding a neural correlate of category representation might not even qualify as having found a neural correlate of phenomenal consciousness. As well, it is plausible that a representation of the category to which an object belongs is amodal (especially by the time it reaches the PFC), in which case its neural correlate would also not count as a neural correlate of **visual** consciousness. (On the other hand, something that might more clearly qualify as a neural correlate of visual phenomenal experience would be the neural correlates of one's experience of an object's color, or of its precise location in the visual field.)

This is not to say the Authors have been negligent here (indeed, they do a much better job of trying to experimentally capture the phenomenal richness of visual experience than has been done so far). Rather, I just think that this limitation of the Authors' studies should be acknowledged in the Article. As well, perhaps the Authors could make some suggestions as to how these limitations could be overcome in the future.

8.
Much to the credit of the project, the Authors take a falsificationist approach to evaluating GNW and IIT. For this reason, they intentionally chose to use fully visible and attended stimuli; the idea here was that, if the predictions of one of the theories could not be met—even under these circumstances—it would be particularly damaging to the theory.

However, in the falsificationist philosophy of science, results are more to the credit of a theory if they are based on predictions that not only turn to be correct, but were also **risky** predictions for the theory to make. But there appears to be a fairly pronounced asymmetry in the riskiness of the predictions that the two theories make in the Article, especially in the case of the first two predictions. As Dehaene points out in his analysis of the results, the predictions concerning posterior

areas are ones that “any physiologist familiar with the bottom-up response properties of those regions would predict” (911). This is in marked contrast to the predictions made by the GNW Theorists: The prediction that conscious content would be represented in the PFC, and that it would have that particular and idiosyncratic time course of activation, is far from what is routinely assumed to be the case by the average neuroscientist.

All this adds up to the idea that IIT being confirmed in the way that it is in the first two predictions should probably not be considered to be an especially strong confirmation of the theory. This should be made clear in the article.

It is good that Dehaene has a chance to point out something like this during his defense of GNW. This idea is important enough, however, that it should probably be given a more prominent place in the article. The reader should be made aware of this at the time that the predictions are being made, and I think that it would be fair if this asymmetry were acknowledged by all of the Authors, rather than being presented just as the view of one of GNW’s proponents.

The second round of experiments that Cogitate plans to perform, the ones in which conscious and unconscious conditions are to be contrasted, will probably hold much more risk for IIT (since these experiments might show stimuli to be represented in posterior brain areas without becoming conscious). The article should point out that the true test of IIT is yet to come.

9.
In the article, IIT is taken to be disconfirmed by the lack of sustained synchronization between face/object selective regions and V1/V2, this being one of the predictions that the IIT Theorists made. However, in the case of features such as object category or face identity, is it not possible that the relevant synchronization would be between face/object selective regions and *mid-level* areas such as V4, V5, and LO? Indeed, given that the properties that these mid-level regions represent (color, motion, and shape) are the properties that the brain would use to *identify* an object or face, perhaps hypothesizing the relevant type of synchronization to be with these mid-level brain areas, instead of lower-level brain areas, makes much more sense. This possibility should be considered in the Article.

10.
When discussing the failure to find the synchronization predicted by IIT (the one discussed in the last point), the IIT Theorists say (at 836) that the results “do not support prediction #3 concerning sustained synchrony, although there are potential explanations (see supplementary).”
It would be most helpful if the Authors pointed the reader to a much more specific place than this (e.g., a page or section number).

There are other places where the Article would benefit from similar cross-referencing. Given the large size of the document, I picture this as being very helpful to the reader as they try to navigate it.

11.
Authors write (91): “Philosophers and scientists have sought to explain the subjective nature of consciousness (e.g., the feeling of pain or of seeing a colorful rainbow) and how it relates to physical processes in the brain. This ‘explanatory gap’ or ‘hard problem’ has led to several competing theories of consciousness that have evolved in parallel.”

This thought is expressed unclearly. The explanatory gap/hard problem is explaining *how* and *why* subjectivity/consciousness arises from physical processes, not simply “how it relates to physical processes in the brain.” The latter phrase could be construed as being satisfied by merely finding correlations between mental and neural activity, rather than actually “bridging the gap” of the hard problem.

12.
Authors write (129): “GNW predicts brief content-specific ignition in PFC ~0.3-0.5 s after stimulus onset and offset (when the workspace is updated), with content...being stored in a *non-conscious silent state* resembling activity-silent working memory in between.”

To my knowledge, this is a novel prediction of GNW, one that did not appear until the article that was a precursor to the article that I am now reviewing (i.e., Melloni et al 2023). I believe that many readers will be surprised by it, especially given its counterintuitive nature (see Points 3 through 5 above).

More resources should be given to the reader to help them understand why this prediction is being made. A citation (preferably with a page number) to (Melloni et al 2023) should be provided with the sentence that I quoted just above (line 129), along with citations to some of the work that was cited when this prediction was being discussed in (Melloni et al 2023). There are sections in (Mashour et al 2020) that the reader might find helpful as well.

13.
I do not understand why, at line 423, the word “serendipitous” is being used. Is it that some kind of “happy accident” came from how the electrodes were placed? Or do the Authors just mean “stochastic” or “fortuitous”? Please clarify.

REFERENCES

Brown. (2015). The HOROR theory of phenomenal consciousness. *Philosophical Studies*, 172(7), 1783–1794.

- Brown ... LeDoux. (2019). Understanding the higher-order approach to consciousness. *Trends in Cognitive Sciences*.
- Herzog ... Doerig. (2020). All in good time: long-lasting postdictive effects reveal discrete perception. *Trends in Cognitive Sciences*, 24(10), 826–837.
- Kozuch. (2023). A Legion of Lesions: The Neuroscientific Rout of Higher-Order Thought Theory. *Erkenntnis*, 1–27.
- Lamme. (2006). Towards a true neural stance on consciousness. *Trends in Cognitive Sciences*, 10(11), 494–501.
- Melloni ... Lepauvre. (2023). An adversarial collaboration protocol for testing contrasting predictions of global neuronal workspace and integrated information theory. *Plos One*, 18(2), e0268577.
- Michel & Doerig. (2021). A new empirical challenge for local theories of consciousness. *Mind & Language*.
- Michel ... Liu. (2018). An informal internet survey on the current state of consciousness science. *Frontiers in Psychology*, 9, 2134.
- Rosenthal. (2011). Exaggerated reports: reply to Block. *Analysis*, 71(3), 431–437.
- Siegel & Byrne. (2017). Rich or thin?

Version 1:

Reviewer comments:

Referee #2

(Remarks to the Author)

The authors have adequately responded to my comments from round one of the review.

While I was positive then, the manuscript is significantly better now. Especially, the clarification of ROI's in response to reviewer 1 is an important addition. I still recommend publication of this manuscript.

Here are three minor notes:

1) I am still a little disappointed by the conclusion but accept the authors' arguments for it being the way it is, and look forward to reading the other paper they are preparing, and hope to see some of my comments addressed there (no pressure intended though).

2) in reply to my point 7 about prediction #1 not being mutually exclusive (rebuttal page 22) the authors reformulated the prediction to say

““Prediction #1: According to IIT, PFC is not necessary for consciousness and therefore decoding of conscious content should be maximal from posterior cortex. According to GNW, PFC is necessary for consciousness and therefore every content of consciousness should be decodable from the PFC.””

The conceptual pedantic in me still reacts to this formulation, albeit for different reasons than before.

Reason one: there is no entailment relation between 'necessity' and 'decodability', making the use of "therefore" unwarranted.

Reason two: the logical structure (suggested by the use of 'therefore') of the IIT part isn't right given that the premise says nothing about the conclusion (the premise is "X is not necessary", the conclusion is "Y").

Reason three: the mechanisms of 'extrinsic' theories (i.e. HOT and GWT, that require something other than the content itself to be involved in the coming about of conscious content) can be conceived of as 'content neutral' in the sense that whatever generates consciousness is ignorant of the content of the states it processes (Kirkeby-Hinrup, Asger. "Two caveats to the meta-problem challenge." *Journal of Consciousness Studies* 27.5-6 (2020): 74-81.) This would mean that 'content' would not be decodable from the mechanism (i.e. PFC on GNW).

Importantly, none of this is critical to the paper. I would recommend just ignoring reason 3. In reply to reasons 1 and 2, I recommend rewording this to exclude "therefore". Something along the lines of

““Prediction #1: According to IIT, PFC is not necessary for consciousness. Consequently, proponents of IIT predict decoding of conscious content should be maximal from posterior cortex. According to GNW, PFC is necessary for consciousness and consequently predict that every content of consciousness should be decodable from the PFC.””

3) The revision on page 19 (paragraph two) says: " Furthermore, within the framework of IIT, spiking activity is considered a constituent property of the physical substrate of consciousness according to IIT"

 Fix pleonasm of "within the framework of IIT" and "according to IIT" (i.e. delete one of them).

Referee #4

(Remarks to the Author)
Please see the attached.

(Remarks on code availability)
I have not reviewed the code because this is an area beyond my expertise.

Referee #5

(Remarks to the Author)
Explaining the first-person perspective (consciousness) from a third-person perspective is one of science's most formidable challenges. Over the past three decades, systems neuroscience witnessed an explosion of empirical insight into this research question. Yet, there is no widely accepted theoretical framework to provide a rigorous foundation. This paper is motivated by the fact that there are two leading, but competing, theories. Prior work was unable to conclusively decide between these rival theories. As a potential solution, the authors organized a large-scale collaborative research effort that included champions of either theory to collectively test previously published predictions. The outcome revealed that neither theory produced a full slate of successful predictions (though it favors one theory over the other). This work is original, significant, and remarkable in idea, scope, and outcome.

The experiments accurately test all three core predictions, which are logically deducible from the theories. It is noteworthy that the core predictions were devised by the primary authors of the respective theories, thus leaving little room for speculation. However, prior work already made clear that GNW posits involvement of the prefrontal cortex (PFC), whereas IIT predicts that PFC is not essential. Specifically, a series of peer-reviewed papers discussed the role of PFC from both the perspective of IIT and GNW, and identified this empirically testable question as key distinction (Boly M, et al. J Neurosci. 2017; Odegaard B, et al., J Neurosci. 2017). The experiments outlined the current paper explicitly test for that predicted difference on top of several more refined and quantified corollaries.

Importantly, the experimental design aimed to maximally avoid biases or other problematic forms of scientific conduct. That is, the research approach represents current best practices: (1) The chosen predictions are sufficiently distinct from each other and thus cannot both be true at the same time ("strong inference"). (2) The predictions were preregistered. That is, there is a written record of the predictions preceding the collection of data, thus preventing adaptation of theory to outcome. (3) The predictions included quantitative pass/fail criteria, thus preventing ambiguity in interpreting the outcome.

The experiments have been designed with appropriate care, and the analyses are reflecting the current state-of-the-art. The data are largely unambiguous, of high statistical power, and entirely novel. Taken together, the findings provide compelling evidence that both theories failed in the totality of their predictions. That is a significant achievement from a Popperian point of view, which urges scientists to find failures in theoretical predictions. From an alternative philosophical (Bayesian) point of view, the conclusion reaches even further in that IIT gains in likelihood over GNW. So, on either view, there is significant progress. The authors favor a third (Lakatosian) perspective, which is increasingly gaining popularity in neuroscience (Doerig et al., Nat. Rev. Neurosci. 2023). This epistemic stance argues that failing theoretical predictions of this kind spur scientific advance via further theoretical development. Importantly, all of these philosophical lenses converge on the notion of significant empirical progress in this paper. This is especially notable when considering the theoretical impasse that plagued the field (e.g. Seth & Payne, Nat. Rev. Neurosci. 2022). The authors thus arguably succeeded with the implementation of an "adversarial collaboration" approach to combat a theoretical stalemate in an important research field. Other fields likely will take note.

The extensive Supplementary Information provide an extremely useful addition. This material successfully addressed remaining questions that the main paper could not accommodate given the space constraints.

To conclude, both the approach and the conclusions of the paper are sound, highly original, and impactful. The findings are of immediate interest beyond neuroscience. This paper not only empirically solved a long-standing impasse in a research field of general concern by revealing flaws in leading theories. It also demonstrates innovative new ways of conducting research in our age of increased collaboration, open data, and transparent science.

minor issues:

As one of the previous reviewers pointed out, there are a few aspects to the final analyses that somewhat deviate from the preregistered details. The authors replied by providing reasonable rationale for these modifications. However, the final text does not indicate where, how, and why finalized predictions were modified. This should be clarified, especially since the paper dedicates an entire section to preregistration.

Several of the line plots seem to indicate variance using confidence intervals. This is not always indicated in the corresponding figure legends,

In the same vein, several of the bar plots feature error bars, but these are not defined as standard error of the mean (or alternative) in the figure legends.

(Remarks on code availability)

The associated code currently cannot be tested by the peer-reviewers. The linked GitHub repository seemingly lacks several directories required to recreate the figures.

Referee #6

(Remarks to the Author)

This manuscript shows how results on brain activation, decoding and connectivity in different experiments support, partially support or fail to support pre-specified hypotheses regarding conscious content, maintenance of conscious content, and interareal communication for these 2 theories of consciousness that have come together in this Adversarial Collaboration grant.

Conscious content: robust decoding for PFC across modalities but some differences between faces and orientation that are interpreted as partial support for GNWT; and for IIT clear support as decoding of conscious content (both category and orientation) was robust in posterior cortex (although this prediction was not deemed key for IIT).

Maintenance of conscious content: sustained activity in posterior cortex, including representation of category and identity across multiple stimuli partially support IIT, however, sustained responses were very rare in the intracranial data, and no sustained representation of orientation. GNWT was also partially supported by the patterns for later phase ignition after stimuli onset but not offset.

Interareal communication: While the authors express that there is no evidence in support of the prediction of the theories, in the exploratory analyses the amplitude-based metrics of connectivity (DFC and gPPI) show support for GNWT with PFC connectivity in timing and powerband; but not for IIT, with no sustained connectivity with V1/V2.

The results are novel in that it combines the same experimental design in direct cortical recordings in humans, with MEG and fMRI in separate cohorts. While the designs are not novel, they were, in principle, specifically agreed to test the three predictions that would differentiate between the theories.

The used of statistics and methods development are well explained and appropriate.

The key part of the paper and what I have been specifically asked to comment on is not a trivial endeavour as it pertains to whether core predictions of the two theories were tested and how strongly the data are able to confirm/refute either theory. I think that the paper tests the predictions as they were presented in the preregistrations and agreed by IIT and GNWT parties. I also think that these predictions are not core to the key aspects of consciousness for these theories but for different reasons.

For IIT this first paper of the Consortium seems more relevant as it is more directed to conscious "content" than "access". IIT theory gives primacy on content while GNWT is historically and conceptually closer to the conscious access claims. This difference, I interpret, make the contrastive approach proposed by the adversarial collaboration rather taxing. On one hand the adversaries agreed on a series of predictions to be tested, but on the other hand it either does not test core aspects of the theory on conceptual grounds (IIT), or it is not a strong way of testing the theory (GNWT).

IIT authors define some core aspect of IIT applied to neural system in humans (in the section Integrated Information Theory: Melanie Boly, Christof Koch, Giulio Tononi) and interpret the results in terms of content but they do not link the results to the core aspects of the theory. This is unfortunate as any result should be interpreted considering the theory, and not only be restricted to the confirmation of the predictions. Further, this paper does not "account for how experience feel" (last paragraph of that section) but uses neural data to decode the differences between different aspects of the stimuli. While I see that the predictions have been tested and evaluated, there is no effort in justifying how these results contributed to core aspect of IIT. The results are not linked to neural measures that might represent "the requirements for maximal integrated information", and they are also not testing the core aspect of "essential properties of experience"; making the predictions "not-core" to IIT.

On the other side of the adversarial, Dehaene, in the subsequent section, concludes that the design does not allow to differentiate between conscious and non-conscious process since these conditions are not present, and that creates a lot of problems interpreting that attentional lapses (in the form of blink, blindness, PRP or retro-cuing) could have decoupled consciousness. He also goes to say that these methods are purely correlational, and he looks forward to the 2nd and 3rd part of the COGITATE project to test how the theories explain consciousness and to define the necessary regions for conscious experience. So, also in his own words, I interpret, these predictions and their results in the current paper, do not test conscious access, hence they might not be testing conscious content adequately, and they do not test the necessary regions for conscious experience as they are purely correlational. Again, this paper does not test core aspect of GNWT.

The COGITATE endeavour is enormous amount of work and it has been carefully curated from the beginning in bringing together and trying to make very distinct theories to agree on specific predictions. The data is rich and will continue to nurture the cognitive neuroscience community for many years. What constitutes evidence in a normal research paper it is here elevated to a higher standard, this is a laudable effort and should receive the merit it deserves.

I am fully convinced of the results, I do not completely agree with the interpretation, as I do think that the predictions have been tested but they do not pertain to core aspects of the theories. I look forward to the next phase of experiments and results to integrate with these first pieces of evidence to see if and how these theories truly have explanatory power in our understanding of human (and maybe animal) consciousness.

Version 2:

Reviewer comments:

Referee #2

(Remarks to the Author)

The authors have adequately handled my previous suggested revisions, and I have no outstanding issues with this manuscript.

I again congratulate the authors on their excellent work, and thank them and Nature for being part of this very productive review process.

- Asger Kirkeby-Hinrup

Referee #4

(Remarks to the Author)

The paper is much improved. I am pleased to see this: Since it is such a strong piece of research, it deserves to have been sharpened up the way that it has been in these successive rounds of revision.

I have only one suggestion. It concerns what appears on p. 24 (~line 852) of the latest manuscript:

"Although this study was designed around IIT and GNWT, the results may have implications for other theories of consciousness. *For example, GNWT's prediction about PFC is shared by some (but not all) higher-order theories of consciousness that also give a central role to PFC.* As a result, the challenges to this prediction challenge not only GNWT but also those higher-order theories."

This is an awkward way to put it (the part between the asterisks), since it makes it sound as if there might be a significant number of HO theories that *don't* give the PFC a central role in producing consciousness. But nearly all do. (Rocco Gennaro is one exception that I can think of here, though his account seems inconsistent with our current neuroscientific knowledge, and hasn't been updated since 2012). At a minimum, this sentence should be cleaned up to remove this ambiguity, but this can happen in the course of addressing a second comment I had. I'll continue...

Now, while nearly all versions of HO theories give a central role to the PFC, not all of them take the PFC to actually supply the content of visual consciousness. I will explain: It is now common in NCC research to distinguish between two types of PFC theories (i.e., theories that hold that the PFC plays a central role in consciousness). This is between (1) those theories in which the PFC actually *supplies the content of visual experience, and (2) those in which the PFC merely enables the consciousness of content that is located in the back-of-the-brain visual areas. Probably the clearest explanation of the distinction is found in (Kozuch 2024, "An embarrassment of riches: The PFC isn't the content NCC"), where the distinction is made between "Upper-deck" and "Lower-deck" types of HO/PFC theories. The distinction is also found in (Kozuch 2021; Raccach, Block & Fox 2021; Michel 2022)), and implied in other places.

Anyway, it is just Upper-deck versions of HOT that are threatened by the results that we're considering, and this should be made clear. Doing so need not involve adding more than a handful of words to the manuscript. It could look like this:

"Although this study was designed around IIT and GNWT, the results may have implications for other theories of consciousness. *For example, GNWT's prediction about PFC is shared by those higher-order theories of consciousness that hypothesize the PFC to actually supply the content of visual consciousness (e.g., LeDoux & Brown 2017), rather than those that take it to merely enable the consciousness of content that is located in the back of the brain visual areas (e.g., Cleeremans et al. 2020; Fleming 2020).* As a result, the challenges to this prediction challenge not only GNWT but also those higher-order theories."

The references appearing just above can be found in the (Kozuch 2024) paper mentioned above (along with many others). So can more explanation of the distinction in types of PFC/HO theories. The Authors could probably pick different works to cite, but the ones that I put in the passage above are fairly clean ones.

Referee #5

(Remarks to the Author)

(Remarks on code availability)

I just did a spot test, evaluating whether I could get the EEG data to preprocess on my machine. It worked.

Referee #6

(Remarks to the Author)

Thanks for answering, I see we just disagree. However, the editorial decision is to go ahead with the paper as it is after the excellent responses to the other reviewers. I acknowledge that the non-core aspects are tested and that is somehow stated in the paper so I am satisfied.

Point-by-Point reply to the reviewers

We thank all four reviewers for their insightful feedback which was instrumental for improving the quality and focus of our manuscript. For the reviewers' convenience, modifications to the original manuscript are highlighted in red in the accompanying document. In what follows, we respond to each of the concerns expressed by the four reviewers. The reviewers' original comments are copied below (in black) followed by our responses (in blue italics).

Referee #1

In this manuscript, the authors reported the first set of results of a highly publicized project (Cogitate) generously supported by the Templeton World Charity Foundation. The aim of the ongoing project is to empirically arbitrate between two theories of consciousness in the form of an adversarial collaboration. Overall, the project is strong in large part because of the scale, ample funding, and some aspects of the quality and quantity of the data. Unfortunately, there are several inherent flaws in the experimental design that limit the interpretations of these studies. Although some aspects of the results are informative, novel scientific content isn't yet a major strength of this project. However, the current experiments were skillfully executed, and the data will no doubt be very useful for future studies. Also, an upcoming second set of future studies, as anticipated in the discussion section, sound relatively promising. I'm sure the overall project will mature in time.

The biggest problem here concerns the very nature of this adversarial collaboration, which is also what has given it tremendous publicity and potential for huge impact. There have already been some general reservations expressed by others, regarding the tenuous connections between the hypotheses and the theories (1–3). Despite this, I was still read the manuscript with great interest. Unfortunately, something seems to have gone unexpectedly wrong in the ways the hypotheses were defined and applied in practice. Specifically, the selection of ROIs for IIT, which neglected the formalisms of the theory, excluded areas originally labeled as part of the so-called 'posterior hot zone' (e.g. precuneus, posterior cingulate (4)), and instead included important frontal areas as 'extended' ROIs, will puzzle many, and probably be considered false advertisement by others. Excluding parietal regions for GNWT likewise went against how the theory (GNWT) has been characterized in print. Although the study was overall pre-registered, these critical and somewhat surprising ROI definitions - without which the hypotheses were very much underspecified in practice - were only uploaded to OSF in Dec 2022, and made public as recently as late June 2023.

In other words, while the current studies admirably tested some 'pre-registered' hypotheses made by some theorists, these hypotheses were unfortunately developed and tested in somewhat idiosyncratic ways, such that they end up being basically irrelevant to theories of consciousness, or perhaps even consciousness in general. But the vast amount of data collected will no doubt still be tremendously useful for other researchers and for different scientific purposes. Below I discuss both some strengths and limitations of the manuscript in more detail.

We thank the reviewer for his evaluation of our work. In the reviewer's preamble, two major points are raised: 1) reservations about the tenuous connection between the hypotheses and

the theories and 2) reservations about the predictions being tested in an idiosyncratic way that makes them irrelevant for the theories. We believe that these comments might stem from issues that were not clearly expressed in our original manuscript, e.g., concerning the aims of our adversarial collaboration, the timeline of events, and the specific ways in which theories can to be tested. We now clarify these points in our revised manuscript and explain these changes below in our specific replies.

Strengths

Overall, this project is highly commendable in a number of aspects.

1. iEEG coverage of PFC: The authors referred to the electrode coverage of the PFC as “exceptional”, and indeed it was. It is often difficult to compare iEEG measures in PFC vs other brain regions, as the density and number of electrodes are often not matched. This makes the current dataset highly valuable.
2. Ample Data: Another remarkable feature of this project is the sheer quantity of data made possible by the generous funding support and outstanding organization effort. This allowed the studies to have statistical power far higher than usual. Also, multiple measurement methods were used, with studies conducted in different labs across the world.
3. Open Science Ethos: Overall, the experiments were well executed. Admirably, they mostly also followed some of the latest standards of open science practices, including pre-registration, data sharing, etc. For a large-scale project of this kind, such coordination takes considerable effort, and should be applauded.
4. Clarity in Writing: Overall the writing is clear and easy to follow, especially in the sections describing the empirical results.

We thank the reviewer for acknowledging our strong commitment to Open Science Ethos, as well as the uniqueness of our approach with respect to data modalities and data standards. We indeed consider those to be key strengths of our work, alongside the results and their implications on the theories.

Limitations

Although there are some significant issues that are of some concern, they mostly stem from experimental design, hypothesis development, and conception. Overall, the studies seem well executed by the trainees involved.

1. The studies tested some hypotheses made by some theorists associated with those two theories. But unfortunately these hypotheses seem to be not so logically connected with, or constrained by, the theories. Specifically, for IIT, the selection of the ROIs were neither based on the mathematical formalisms of the theory (calculation of ‘phi’) nor previous empirical studies that helped to give rise of the notion of posterior ‘hot zone in the first instance, e.g. Siclari et al 2017 (4). Instead, curiously they look like a set of regions which anyone would have chosen in order to maximize decoding accuracy for classifying visual stimuli. Because IIT is not supposed to be specific only to vision, these ROIs do not appear to represent the theory in general. Notably, these ROIs also exclude the precuneus and posterior anterior cingulate which were supposed to be part of the posterior hot zone, based on previous

publications from some of the authors themselves (4). From Extended Data Table 2, one also sees that large parts of the frontal cortex were actually included as extended ROI for IIT, meaning that if effects were found in these areas, they would by stipulation not count against IIT. This seems to go against the general premise of the project, of focusing on frontal vs posterior regions to distinguish between the theories. This important point should perhaps be made more clear up front in the main text; finding out what really happened from Extended Data Table 2 was quite a surprise for myself. For GNWT, not including parietal areas also went against how the theory has been characterized in print. I understand that the intention was to focus on the prefrontal cortex so as to allow the two theories to make different predictions. But logically this seems backwards: if the two theories are indeed empirically indistinguishable, perhaps we should just accept that fact. We can't instead contrive to modify a theory so substantially just for this ad hoc purpose, and still convincingly call it GNWT. Relatedly, these important ROI definitions also seem to be pre-registered relatively late in the project (Dec 2022, version 4 of their OSF upload, by which point data collection must have started if not completed?), and made public even later (late June 2023). The theoretical connection between IIT and functional neuroanatomy at a macro level is itself tenuous, as the authors acknowledge. In justifying the current ROIs, the authors cited in the discussion section a paper that appeared only recently in 2023 (5). Without the ROIs specified a priori, the hypotheses were very much underconstrained. This is especially concerning given how the IIT ROIs ended up looking so very different from what one may expect based on the literature. Perhaps this also compromises the 'pre-registered' status of the studies - an issue that may benefit from more discussion and clarification. In point 4 below I discuss a related problem with the predictions on dynamics. The predictions on connectivity depend on these same ROIs, and are therefore similarly tenuous. Overall, given these ROIs, it is difficult to make a convincing case that the project really tests the theories in a meaningful way.

As this comment includes several different points, we address them separately below.

First, the reviewer is concerned that the hypotheses tested in this work do not genuinely represent the theories (referring to them as “some hypotheses made by some theorists associated with those two theories”). We respectfully disagree. These hypotheses were laid out in a joint effort between all participants in the adversarial collaborative process, including the main proponents of these theories (i.e., Tononi (IIT), Boly (IIT), Dehaene (GNWT), Panagiotaropoulos (GNWT), and others). These predictions have been further confirmed by Dehaene and Tononi as “suitable for testing the theory” in a formal letter sent to Cogitate’s center PIs, and in a series of pre-registrations (see all versions here: v1, 2019 <https://osf.io/wz8mj/> v2, 2020 <https://osf.io/f87xq/> v3, 2021 <https://osf.io/35ke9/> v4, 2022 <https://osf.io/92tbg/>), as well as in two separate publications by our consortium (Melloni et al., 2023; Melloni et al., 2021). Notably, the predictions – as well as the experimental design – were described in these publications (and in the original grant application to the Templeton Foundation) and therefore had been vetted and approved by independent reviewers (see open peer review in Melloni, et al 2023) before the results were revealed. A further testament to the centrality of these predictions is their featuring in another adversarial collaboration between IIT and GNWT (performed by a different consortium), conducting a very similar experiment and testing similar predictions in non-human primates and rodent models.

*The reviewer's comment also discloses a misconception about the focus of the adversarial collaboration: to begin with, our goal was to test **the neural implementations** of GNWT and IIT, not their conceptual or computational claims. For example, at the conceptual level, the global workspace theory (GWT; Dehaene, 2001; Mashour et al., 2020) posits that*

consciousness is associated with the broadcasting of information. We do not test this broadcast, but instead **focus on the associated neural prediction that prefrontal areas (as part of the proposed broadcasting network) should represent the content of perception** (a non-trivial prediction, which is only partially met by the data). Similarly, the Integrated Information Theory (IIT) posits that the substrate of any conscious experience corresponds to a maximum of integrated information, denoted as Phi (Albantakis et al., 2022; Tononi, 2012; Tononi et al., 2016). We do not test this proposal, or the axioms and translation to postulates on which it was based. Instead, we **do test the existence of content-specific and sustained synchronization between category-selective areas and V1/V2**. Such synchronization is inferred from IIT's hypotheses about the structure of consciousness: as conscious experience involves both experiencing the higher-level (e.g., the stimulus being a face) and the lower-level features of the stimulus (e.g. the contours of the face), there should be binding between these features, which should be manifested by synchrony. This is a non-trivial prediction of IIT that was not confirmed by the empirical results reported here.

Although the tested predictions focus on the neural implementations of the theories, they all **relate to key claims of each theory**. For GNWT, Predictions #1 -3 tested the expected neural ignition for which a code representing each conscious content should be detectable in PFC (and parietal areas; see below a justification for our decision to focus on the PFC only), as well as an expected outcome of the broadcasting and distribution of information: synchronization between PFC and category selective areas. For IIT, Predictions #1-3 tested the hypothesis that the involvement of the prefrontal cortex is not necessary for every content of experience, and that the activation and synchrony patterns in posterior cortex should support all aspects of experience, even when the perceived content is task-irrelevant. Both theories made non-trivial predictions (#2) about the duration of percepts: GNW predicted an onset and an offset response, while IIT predicted that both the content-specific activation and the synchronization should tracking the duration of the experienced content.

These predictions had the potential of posing substantial challenges to both theories, and as our data showed, some of these challenges indeed materialized. For GNWT, we found that some features of participants' conscious percepts were not decodable from PFC, and that the expected pattern of ignition did not appear at the offset of the stimulus. For IIT, the empirical data did not contain the expected pattern of synchrony, and our analyses did not reveal the predicted sustained representation. Thus, some components of the neural implementations of both theories were not borne out by the data. This, in turn, has implications for the theoretical components from which the neural implementations were derived.

Next, the reviewer raises several concerns regarding the definition of regions of interest (ROIs), that can be summarized into two points. One concern is that the selected ROIs do not represent how the theories have been characterized in the literature. The second concern is about the timeline of the ROI selection process. We respond to both concerns here, and have revised the manuscript to improve clarity regarding the ROI definitions used in each analysis (see supplementary section 11).

With respect to the first concern, we want to emphasize that the selected ROIs were aimed at **testing the predictions put forward by the theory proponents within the context of this particular study**, rather than representing the full set of brain regions relevant to each theory. This was done to make sure the results we obtain are meaningful given the experimental design. We accordingly selected only the ROIs relevant for the **visual** experiment performed here (and excluded the auditory cortex, the somatosensory cortex, etc.). This is also the

reason why the precuneus and posterior cingulate cortex were not included in the IIT ROIs (that are thought to play a role in self, episodic memory, spatial navigation, etc. Epstein et al., 2017; Jacobs et al., 2010; Kumral et al., 2021; Sestieri et al., 2017).

Similarly, for GNWT, we did not include parietal areas and focused on prefrontal ROIs, because the theory's prediction of "global broadcasting" involves these regions. Since according to the theory both parietal and prefrontal regions should be involved, and because prefrontal areas are where the two theories' predictions maximally diverged, we chose to focus only on the latter.

Crucially, the **IIT ROIs do not contain "large parts of the frontal cortex"** as mentioned by the reviewer. This confusion is likely to have arisen due to our use of an "IIT extended" ROI list in the reporting of results of the putative neural correlates of consciousness (pNCC) analysis (previous Extended Data Table 2 and Supplementary Tables 20 to 24). The pNCC analysis is considered exploratory and following the suggestions of Reviewer 3 comment #3, it was moved to the Supplemental section, and substantially deemphasized. **These "IIT extended" ROIs were not used to test any of the critical main predictions of the theories.**

Besides the pNCC analysis, the only other case in which the "IIT extended" ROIs were used was in the comparison of the decoding accuracy obtained using posterior ROIs (i.e., the "basic" IIT ROIs) vs. the decoding accuracy obtained using posterior + prefrontal ROIs (i.e., the basic IIT ROIs + the GNWT ROIs; prediction #1b; Extended Figure 5c. Note that this is not a crucial prediction by IIT). In this prediction, IIT explicitly addresses the prefrontal cortex, claiming that adding it to posterior areas should not increase decoding accuracy compared to decoding from posterior ROIs alone. For this reason, we asked the IIT theorists to specify which PFC regions should be included/excluded for this particular analysis; one sub-region of PFC (inferior frontal sulcus) was identified as having similar neuroanatomical characteristics as most posterior regions¹ and was thus not included in this analysis. Nevertheless, **to address the reviewer's concern of a possible bias due to the selection of ROIs, we have repeated this analysis with the ROIs suggested by the reviewer, which includes the inferior frontal sulcus as a GNWT ROI.** These new analyses produced virtually the same outcome (see Figure 1 and our reply to comment 2 below). These results have been added to the supplementary material section 5.1.3.

In addition, following the reviewer's comment, we realized our description of the ROIs included in each analysis was not clear enough in our original manuscript. In the revision, we now **present in each figure and for each analysis the set of ROIs used to test each of the theories.** This clarifies which ROIs were used in each analysis, while also demonstrating that the IIT defined ROIs did not include prefrontal regions.

We hope that our reply helps clarify why these particular ROIs were selected in the context of this adversarial collaboration, and why different ROI lists were used for different analyses.

¹ The fundamental criterion that IIT considers to be necessary for the neural substrate of consciousness (the "main complex") is having an appropriate anatomical organization. A 3D lattice of dense synaptic connections among topographically organized areas (ordered hierarchically through divergent-convergent connections) is a good substrate for consciousness because it ensures maximal irreducibility, whereas a modular or near-modular system is not (Albantakis et al., 2022; Tononi et al., 2016). The posterior cortex is a good candidate precisely because substantial portions of it are characterized by such a lattice-like synaptic organization. Conversely, there is little evidence that the PFC is organized in the same way. In fact, there is now positive evidence that primate PFC may be organized primarily in a modular manner (with superimposed diffuse connections, see; Watakabe et al, 2023). It is worth reiterating that while having the right anatomical organization is necessary for consciousness, it is not sufficient. Indeed, as predicted by IIT, when causal interactions in posterior cortex break down due to bistability in deep sleep or anesthesia, consciousness is lost (Tononi et al., 2016).

We emphasize that the reviewer's main concern about prefrontal ROIs being included in IIT's list is only relevant to 1 out of the 14 tests we performed to test the theories' predictions, as well as to an additional analysis that is now reported in the supplementary, and, importantly, the results don't change even when this subset of PFC ROIs is included.

To address the second concern regarding the timeline of ROI definitions:

When the Cogitate collaboration was launched (March 2018), the published literature did not contain rigorous definitions of ROIs for the neural implementation of either theory, not to mention a specific definition of ROIs that are tailored to the current experiment. Underspecified terms such as 'fronto-parietal network' for GNWT, and 'posterior hot zone' for IIT were commonly used. We stated in a previous publication (Yaron et al., 2022) that such imprecision was permissive of adaptive post-hoc interpretations of empirical findings, often self-serving one's preferred theory.

*Thus, early in the project (in 2021) we produced a list of ROIs, based on previous literature, using the Desikan/Killiany atlas (Desikan et al., 2006). Once we started working on the development of the analyses for the optimization phase, we realized that these ROIs were too coarse, and requested the adversaries to provide more fine-grained ROIs lists, which they provided in December 2021. These ROIs were then translated into the Destrieux atlas for improved precision (Destrieux et al., 2010). In June 2022, we asked the adversaries to provide a final approval of the ROI definitions. The GNWT proponents approved the listed ROIs. The IIT proponents added one ROI in V2 which they had inadvertently overlooked before. During this period, only preliminary results generated as part of the optimization and analysis development stage, were shared among the consortium members. **Only after the ROIs were fully defined, we proceeded to test the theories' predictions, using two-thirds of the total data that were held out for this purpose.** Importantly, as the final GNWT and IIT lists of ROIs provided by the adversaries are in line with the initial lists that we defined at the beginning of the project, we deem it very unlikely that the selection of ROIs had been biased in any way. This is however to be expected given the course ROIs used. For the reviewer and interested readers, in the spirit of Open Science, we now report this information in Supplementary section 11.*

2. The experiments are novel in that the scale and publicity have been unprecedented, even though this is not the first adversarial collaboration between theorists of consciousness (6). However, in terms of scientific content, novelty may be relatively limited. By now, there are already dozens of no-report studies on conscious perception, starting from the 1990s (7–13). Overall these studies converge on the same general conclusion (14), similar to the one presented here. Regarding temporal dynamics (prediction #2), several recent papers using intracranial EEG / ECoG directly addressed this exact question, using a similar approach, with arguably more sophisticated and thorough data analysis than what is presented here, at least in some cases (15–18); these studies should perhaps be cited and discussed. This is not to say that overall the project gives no new information at all. For example, the finding that adding PFC information to the analysis together with early and late visual activity did not help with decoding accuracy at all, even with iEEG, is interesting. However, as mentioned above the areas most important for PFC decoding have been excluded from this analysis, because they have been considered extended ROIs of IIT. I suggest that the authors clearly state in the main text that the analysis was performed after removing some frontal areas. Otherwise, statements such as “decoding of category and orientation was found to be the

same, or to decrease, when including PFC” might be misleading. I also suggest that the authors try to include these areas in an independent analysis, as most people actually interested in the broader scientific debate would probably not see the logic in excluding those frontal areas.

In this point, the reviewer raises four concerns regarding the novelty of this adversarial collaboration.

First, the reviewer states that the present project is not the first adversarial collaboration in consciousness science and points to one of his own projects (Zhou et al., 2020). We did not find any indications for the cited piece serving as a test for any theory of consciousness, or signs that this work adhered to an adversarial collaboration protocol (Mellers et al., 2001) (the acknowledged funding grant does refer to an adversarial collaboration, however this is not reflected in any way in the cited work). We apologize if this is an oversight on our side, and if so, we are more than willing to amend our manuscript to duly acknowledge and credit this prior endeavor.

Second, the reviewer states that “there are already dozens of no report studies on conscious perception”. We do not contest the existence of numerous report- and no-report studies on consciousness perception. In our manuscript, we acknowledge that our experimental design was inspired by two previous studies (Farooqui & Manly, 2018; Gerber et al., 2017). Yet notwithstanding the importance of these previous works, the current design, including a task manipulation, and a manipulation of several features of conscious content (category, identity, orientation, duration) has not been used before, and certainly not with three different neuroscientific techniques, geared at testing predefined predictions by competing theories.

An additional key novel aspect of our project was the process itself, whereby the theories had to (a) generate predictions, including some that were novel; (b) precisely specify existing claims and predictions, which had been vague in the past; (c) provide a list of specific ROIs that were previously underspecified in the literature; and (d) agree on a list of conditions for important challenges, which had not been done before in our field (see again Yaron et al., 2022). These novel aspects of our work, on their own, go far beyond any other empirical investigation of the theories that we are aware of.

In addition, many of our analyses and results are also novel. For example, this is (a) the first attempt to look for prefrontal offset signals that would mark the end of a conscious percept; (b) the first attempt to look for specific patterns of synchronization predicted by the theories; and (c) the first attempt to test theories of consciousness by decoding information about multiple task-irrelevant aspects of conscious perception (e.g., orientation, and also duration). The reviewer states that our conclusions are in line with “the same general conclusion” of previous no-report studies, and cites (Iemi et al., 2022, chapter 2), but while checking this reference, we failed to find conclusions that resemble the ones presented in our manuscript.

Third, the reviewer questions the novelty of the findings with respect to prediction #2, citing 4 papers (Broday-Dvir et al., 2023; Gerber et al., 2017; Podvalny et al., 2017; Vishne et al., 2023). Two of the cited papers (both from 2017) are indeed seminal papers that introduced the question of the mechanisms supporting the maintenance of conscious perception, and we indeed cited one of them, and another one that did the same (Gerber et al., 2017; Stigliani et al., 2017; we added the missing reference and thank the reviewer for mentioning it; p. 4, last

paragraph). Notably though, these studies were not performed in the context of testing theories of consciousness and in an adversarial collaboration manner.

The other two cited papers were published in 2023, one of them after our manuscript was submitted for publication (Broday-Dvir et al., 2023; Vishne et al., 2023). Both of these papers acknowledged our project (Melloni et al., 2021, connected to preregistration v3, 2021) as a source of inspiration for testing predictions of GNWT and IIT on preexisting data. The results reported in these two studies are impressively consistent with our own data, as we now mention in the manuscript (p. 14, fourth paragraph). Our consortium considers this an excellent example of how openly sharing new experimental ideas, and specific predictions of the theories, can accelerate progress in the science of consciousness, avoiding the pervasive post-hoc and confirmatory bias currently predominant in the field. We note that though impressive in their own right, those two studies could not have related their findings to our predictions, had this adversarial collaboration not existed and adhered to an open science ethos. This serves as a further testament of the impact and novelty of our collaboration,

Finally, the reviewer highlights the comparison of decoding with and without prefrontal areas as novel and interesting, but expresses concern because “the areas most important for PFC decoding have been excluded from this analysis”. As we explained above in our response to the first comment of this reviewer, the only area excluded from this analysis was the inferior frontal sulcus. Following the reviewer’s suggestion we repeated the analysis without excluding this area. None of the results changed (Figure 1 below and supplementary section 5.1.3). As we found in the original analysis, for both iEEG and MEG, **no increase in decoding accuracy was found (and in some cases it even decreased) when adding prefrontal areas.** For fMRI, which was not included in IIT’s original prediction due to its low temporal resolution, we did find an increase in decoding accuracy (specifically, a 2% increase, as opposed to the 1% increase we reported in the original manuscript). In our revision, these results are included in the supplementary, and referred to in the main manuscript (p. 10, fifth paragraph).

Figure 1 Results of the decoding analysis in which the decoders of the posterior ROIs were combined with those of a PFC ROIs, including inferior frontal sulcus. The ROI selection does not affect the results reported in the manuscript: a. Region of interest used in the decoding analysis including and excluding PFC areas. For iEEG (b) and MEG (c), no improvement was found, and in some cases, decoding accuracy decreased for the combined posterior and PFC ROI (blue) compared to posterior ROIs only (red). For fMRI (d), an advantage was found for the combined ROIs (posterior and frontal) compared to posterior ROIs only, reminiscent of the original result (see further discussion in the original manuscript). In the plot, the difference in accuracy between the combined ROIs and the posterior ones is plotted on the x-axis, with the number of subjects showing each value on the y-axis.

3. The lack of non-conscious conditions unfortunately limits what we can say about consciousness with the present studies. This led to an uncontrolled stimulus confound, which could very well explain most of the presented results, especially in early visual areas. I have no doubt that the subjects consciously perceived the stimuli, but this doesn't mean that the conscious perception itself rather than the sheer presence of the stimuli was the explanation of the relevant brain activity. Just because conscious perception occurs doesn't mean that nonconscious perception doesn't also occur in parallel. Without a non-conscious condition, which has been standardly implemented in studies on the topic since the 1990s, the relationship between the current studies and consciousness per se is tenuous.

*We agree with the reviewer that including an unconscious condition is critical when trying to isolate the neural correlates of consciousness (NCC). **Yet the goal of our adversarial project was to test divergent predictions by the two rivaling theories.** Theories of consciousness should not only explain what differentiates conscious from non-conscious processing, but should also account for the mechanisms underlying the different aspects of conscious perception. In our project, we asked where and when information about the content of visual perception is instantiated in the brain, how a percept is maintained over time, and what synchronization patterns are evident when a certain content is perceived. For all three questions, IIT and GNWT theorists provided clear predictions, and for both of them we identified challenges. We agree with the reviewer that a confirmatory result (whereby a theory “passes” a test) may be driven by unconscious processing of the stimulus (and this is acknowledged in the manuscript, in the Cogitate discussion section). However, disconfirmatory results – which pose the most important diagnostic challenges to the theories – are not. Moreover, these results are more insightful and interpretable when the stimuli are salient and undoubtedly consciously perceived. **If a prediction by the theory is not borne out by the data for clearly visible stimuli, the failure is more informative and the theory is significantly challenged.***

Notably, the reviewer agrees that, in the discussed study, there is “no doubt that the subjects consciously perceived the stimuli”. In fact, a large number of published studies of conscious perception, including from proponents of IIT and GNWT, have used suprathreshold stimuli, with no unconscious condition, to study consciousness (Bellet et al., 2022; Boly et al., 2015; Broday-Dvir et al., 2023; Cohen et al., 2020; Haynes, 2009; Vishne et al., 2023).

4. One interesting and novel prediction here concerns the activity in PFC associated with stimulus offset, supposedly ‘made by GNWT’. While I appreciate that it was pre-registered, I’m not sure I agree with the logic of the prediction. I also could not find in print any reference supporting this prediction, or linking it to GNWT. When a random, i.e. not fully expected, stimulus arises, there is a considerable amount of information to be updated within the workspace, according to the theory. But upon stimulus offset, the next frame is invariably the same: blank. As some of the authors are no doubt aware, expectation can silence prediction errors. There is just overall less unpredictable information to be learned or to be updated over offset as compared to onset. As such, while the theorist representing GNWT here (Dehaene) made the prediction that there will be a high level of activity at offset, I’m not sure GNWT or its broader community of supporters would have necessarily made that same prediction. In non-human primate electrophysiology there is known offset activity associated with stimulus offset, but that was observed in the visual cortex not PFC, and the magnitude is indeed considerably smaller than for stimulus onset (19).

The reviewer mentions he does not agree with the logic of the prediction made by GNWT theory lead. However, we do not consider this a relevant point against the manuscript, but rather another example of the new horizons and important discussions our work has evoked (see for example Reviewer 2's comments which suggest a third alternative). Additionally, the reviewer mentions that he could not find any reference supporting this prediction. Indeed, this prediction was novel and was a result of our collaboration's efforts to prompt these two theories out of their comfort zones, to explain basic aspects of conscious experience that had previously been neglected in the literature. It is however explained in length in our published (and peer-reviewed) registered protocol (Melloni et al., 2023).

Notably, asking theory leaders to make novel predictions about relevant test-cases is a way to make progress; pushing theories outside their comfort zone allows researchers to highlight the shortcomings of the theory, in case these predictions fail, or increase its explanatory power, in case they are confirmed. This is how normal science works. In an adversarial collaboration this process is amplified, as participants are willing to challenge their own hypotheses or viewpoints by interacting with those with opposing views. We consider this one of the greatest merits of our project: we did not only rely on existing predictions from the literature, but prompted the adversaries, thanks to the adversarial collaboration process, to generate new predictions (and make old predictions more specific), thereby contributing to the theoretical knowledge on consciousness in addition (and irrespective of) the obtained results. We now explicitly emphasize this in the revised manuscript (p. 5, second paragraph; p. 24, first paragraph).

GNWT predicted increases in amplitude at the beginning and at the end of an experience in PFC, as well as a reinstatement of information in the workspace at the offset of the experience bookmarking its end. This prediction was novel and at the time of the initial preregistration there was no systematic data to demonstrate this effect. Note that the lack of data per se is not a problem. This is what predictions are for: they should explain data from novel tests.

Based on the main published postulates of GNWT, it stands to reason that any update of the workspace should lead to an ignition. In our protocol, the main stimuli were always replaced by a blank screen, which is a new percept: instead of seeing a stimulus, one now sees a gray screen. Notably, the fact that we call it 'blank' doesn't mean that participants' consciousness is now void – they still perceive something, but this something is a gray screen rather than a gray screen with a stimulus in the middle. As such, this is a change in perception, which should lead to an update in conscious content. Thus, most scientists familiar with GNWT would predict an ignition of the workspace when a new stimulus enters into consciousness and will thus not find this prediction controversial (what could potentially be considered controversial is the prediction that the offset ignition should be content specific, but we did not find evidence for a non-content-specific ignition either, besides one electrode showing an offset response, albeit too early in time).

5. The authors describe the chosen methods of combined fMRI, MEG, and iEEG, as capable of providing “exquisite temporal and spatial precision”. I’m not sure this is right, especially regarding spatial resolution. This is a point that the authors seem to be well aware of, and was supposedly the reason the Cogitate project is also conducting invasive electrophysiological studies in non-human primates. Given the chosen emphasis on PFC, and the known inadequacies of current measurements relative to those that will be used in ongoing experiments within the same project, releasing the present results as anything other than preliminary seems somewhat premature.

We are perplexed by the reviewer’s remark here. By most standards in the field, any study including MEG, fMRI and iEEG under the same experimental protocol in a large sample of human subjects will be regarded as ‘providing exquisite temporal and spatial precision’. We didn’t include single unit recordings in human subjects to test the theories as it would be unfeasible since they almost invariably only sample the medial temporal lobe. As such we disagree with the reviewer’s statement that ‘releasing the present results as anything other than preliminary seems somewhat premature’. Under this standard, almost all studies conducted so far in cognitive neuroscience are premature.

Further, the invasive studies in non-human primates mentioned by the reviewer are not being conducted by the Cogitate Consortium. Those belong to a parallel adversarial collaboration led by Yuri Saalman, performing a comparable paradigm to our Experiment 1, using invasive electrophysiology and causal manipulations on non-human animals (primates and rodents). Though we very much look forward to seeing the results obtained by that consortium, we note that like most studies (including ours), that experiment also has limitations. Some limitations of this parallel adversarial collaboration include the problem of establishing a reliable measure of consciousness in non-human animals, the limited spatial coverage of the Neuropixel probes, and the potential confound of extensive training. We thus see these two adversarial collaborations as complementary, yet independent: we do not think that the results of either project are preliminary or premature.

Yet to accommodate the reviewer’s concern, we have added to the discussion section a further clarification of our claim with respect to the temporal and spatial resolution of our studies, and the relationship to the parallel adversarial collaboration (p. 26, third paragraph). For the reviewer’s convenience, we copied the relevant section below:

“Our study, while comprehensive, is not without its limitations. Firstly, [...]. Thirdly, although our study offers superior spatial and temporal resolution across the brain by integrating three distinct brain imaging techniques—fMRI, MEG, and iEEG—it stops short of incorporating single-unit recordings. Such recordings, typically reserved for epilepsy patients and limited to certain brain areas like the Medial Temporal Lobe, are impractical for directly testing our theories. Studies in other animal models, including Neuropixels and causal manipulations, are underway as part of another adversarial collaboration, and are expected to complement our findings. Despite the inherent challenges of using animal studies to probe consciousness (difficulty of measuring consciousness in non-human subjects, the limited spatial coverage of Neuropixels probes, and overtraining), we see these two adversarial collaborations as synergistic, providing a stronger test for the theories than either one alone”

6. The authors claim that “[t]he main discrepancy between the theories is [...] the necessity of PFC.” However, according to the common definition of the neural correlates of consciousness (NCC), ‘necessity’ was never meant to be a requirement (14, 20); it is ‘joint minimal sufficiency’ that is the relevant logical concept. Further, if the intention is really to test for necessity as stated, the current methods may be rather suboptimal. Causal manipulation (e.g. ablation / inactivation, or even TMS) may be more appropriate and convincing.

*We agree with the reviewer that the original definition of the NCC included “joint minimal sufficiency” rather than necessity. However, as we explained above, **our project did not aim***

*to isolate NCCs, but to test predictions of the theories. In that context, the PFC is necessary: GNWT predicts that PFC is part of the neural implementation of the global workspace, and that every conscious experience arises from updates to (or “ignitions” in) this hypothesized workspace. Therefore, according to the theory, PFC (as part of the larger fronto-parietal network) is **necessary** to all conscious experiences. Failure to detect effects in PFC in response to stimuli that were clearly consciously experienced – especially in such a large sample size and with three different modalities (fMRI, MEG, iEEG) – challenges this necessity claim and thereby the underlying theory. The logic the reviewer refers to is the inverse – i.e., if we find activation (or decoding, or synchrony, etc.) in PFC for stimuli that were consciously experienced, it is unjustified to conclude that PFC is necessary (because minimal sufficiency is the relevant logical concept in that case). We agree with this inverse logic, and accordingly do not make that claim.*

7. The rationale of the pNCC analysis seems to rely on the assumption that a voxel/vertex is either contaminated by confounds e.g. reporting, or not. But of course, this is a false dichotomy, as a bit of brain tissue at this scale can very well serve multiple functions. In fact, this is true even for single neurons. There could be both linear and non-linearly interactive effects of both reporting and conscious perception. So I’m not so sure whether the conclusions here are valid.

*We agree with the reviewer that the dichotomy he describes is indeed false, and apologize if the manuscript gave the impression that we hold otherwise. In fact, our analysis only excludes voxels if they respond to task goals or task relevance, **but do not respond to task-irrelevant stimuli**. Thus, if an area is involved both in perception and in report, as the reviewer rightfully suggests, it would not be excluded by our analysis. The conjunction that includes voxels into the NCC requires that a voxel should respond to both task-relevant and task-irrelevant stimuli (either similarly or with different magnitudes). Therefore, voxels whose activity is modulated by task factors can still be included, as long as they respond to task-irrelevant stimuli. However, irrespective of this point, following the suggestion made by Reviewer 3, we have decided to move this analysis, which was not central to begin with, to the supplementary materials (namely, Supplemental section 8). We have also revised the relevant section of the methods for clarity (p. 46). It now clearly states:*

“Thus, this analysis casts a wide net to identify areas that can potentially be the neural correlates of consciousness, while excluding areas that do not respond to task relevant/irrelevant stimuli (meaning that areas that respond both to the task and to the content of perception are still included)”.

8. One finding is that orientation information for stimuli, e.g. comparing faces viewed at different viewpoints, seems not to be decodable from PFC. However, one problem is that this information was never task relevant. So we do not know if such information would have been decodable, had it been made task relevant in some conditions. The interpretability of this finding could have been improved with a simple experimental design fix, which is to factorially manipulate task relevance for both category and orientation, in a simple 2x2 design.

Our experiment consisted of a 3x3x4x2 design. The factors were stimulus duration, stimulus orientation, stimulus category and, task relevance. We opted for an experimental design whereby one stimulus feature (i.e., category) can still be contaminated by a minimal degree

of task engagement, while another feature (i.e., orientation and also duration) is always task irrelevant. This sets the bar higher for the theories, as any effects found there cannot be explained by selective attention being directed at this feature due to – even minimal – task demands. We understand the motivation for including orientation, as a task relevant condition would have allowed us to demonstrate that it is decodable. However, this was not feasible; first, because it would have prolonged an already long experiment, and second, because to make it fully task irrelevant, as needed for testing the theories, the session would have had to be divided in two, whereby first orientation is task irrelevant and then task relevant, thereby introducing order effects. Most importantly, there is already previous evidence (albeit weak) showing that such decoding is possible (albeit weak), from Dehaene's lab, using orientation of Gabor patches (King et al., 2016). Thus, although we agree with the reviewer that a follow up study could provide further evidence for such decoding, the key point is that this does not invalidate the conclusions of our study. As the stimuli were highly visible (and to the extent that one does not deny that subjects consciously experienced the stimuli including their orientation – an assertion the reviewer seems to agree with), information about orientation should have been encoded in PFC and posterior areas, according to GNWT.

9. Relatedly, the current manipulation of task irrelevance was also perhaps not the most effective. Subjects still had to attend to the irrelevant targets and made a decision to specifically not respond. A better manipulation may be to make the features or targets totally irrelevant to the task in some blocks (but not others), just as in how orientation was irrelevant throughout the current experiments.

We fully agree with the reviewer's comment that the best manipulation is to make some stimuli totally irrelevant in some blocks but not others. Indeed, that was one of the key aspects of our design. To reiterate, in the current experiment, we intentionally grouped letters and false-fonts as relevant stimuli in some blocks, and faces and objects as relevant stimuli in other blocks, in order to make the irrelevant stimuli very easy to disregard as potential targets. Thus, as the reviewer suggests, our design had stimuli that were either task relevant or task irrelevant, in different blocks.

We agree however that this manipulation does not render the stimuli completely task irrelevant, and that it can still lead to some carry over effect turning the condition that was presumed to be irrelevant into more attention grabbing. To address these issues, we included in our design orientation and duration as features that remained consistently task irrelevant throughout the entire experiment (see also our reply to the reviewer's previous comment above).

Minor Issues

1. It would be useful to give more details regarding how the ROIs for the two theories were selected, including the rationale, empirical papers on which the considerations were based, and give specific references as needed. If this was all done based on intuitive guesses, please be transparent and say so. It may also help to provide some explanation as to why these important ROI definitions were only pre-registered late in the process (Dec 2022, in v4 of the OSF upload), and made public even later, and specifically whether it means that they were finalized only after data collection and analysis have started. For IIT, please also explain why no phi calculation seems to be involved here, and whether the original conception was to involve such calculation. Also, what exactly happened to the results of Siclari et al (2017) (4)

which I thought were the (original?) empirical basis for the idea of “posterior hot zone” for IIT?

This comment reiterates issues raised in Major Comment #1, which we addressed in full above. For this revision, we have included detailed information about the process of definition of the ROIs, including the literature used for that purpose, in supplementary section 11.

2. Some positive MEG results were found for the PFC but they were somewhat written off in the light of lack of both fMRI and iEEG activities identified for the same contrast. Overall I agree with the interpretation. But please clarify if such interpretation was based on pre-registered criteria. If not, please discuss whether this undermines the overall principle of anti-HARKing somewhat. Perhaps this is just the nature and limitations of this kind of exercise, but please be more transparent about this point, and discuss accordingly.

*In our pre-registered predictions, we stated that if PFC decoding for stimulus orientation was found for either iEEG or MEG, GNWT’s predictions would be supported. The reason for toning down the contribution of some positive MEG results was our uncertainty about their validity in their own right. To explain, the preregistered criteria assumed adequate sensitivity with respect to spatial resolution for the correct interpretation of the results. However, when conducting several control analyses to evaluate the point spread function of source modeled MEG data, we could not conclusively rule-out the alternative explanation of “leakage” from posterior to anterior cortex in the source-space MEG decoding analyses (see Extended Data Figure 5b). Thus, we could not be certain that the results indeed represent decoding coming from prefrontal areas. This doubt was further enhanced by the following factors: first, this result (i.e., decoding of orientation in PFC) was barely-above-chance. Second, it was found for a very brief period of time, which was earlier than predicted by GNWT. When weighing these factors with the inability to exclude the leakage confound, and with the null results in the other two techniques: namely, fMRI and iEEG, which have far superior spatial resolution compared with MEG, a lower weight was given to this MEG finding. **To be clear, if the MEG result had been reliable, the lack of results in the other modalities wouldn’t have made a difference, in line with the pre-registered criteria.***

Importantly, despite our low confidence in these results, they are clearly mentioned in the main text and reported in detail in the supplement, – to allow the readers to form their own opinion. We accordingly hold that we reported these results honestly and with appropriate nuance. Moreover, in the final figure summarizing the results (Figure 5), this result is highlighted in yellow (rather than red, marking a failure), and we write “no consistent decoding of orientation” as opposed to “no decoding”. And in the legend, we spell this out very clearly: “Yellow marks cases in which we considered that the results did not allow a confident interpretation... Namely, for GNWT’s prediction #1, we found cross-task generalization of decoding of faces vs. objects, in line with the prediction. However, the only evidence for orientation decoding was found in the MEG data, where we could not conclusively rule out leakage from posterior areas. Thus, as passing this prediction requires both decoding of category and orientation to be found, we cannot determine with high confidence if this prediction should be counted as a pass or a fail.” We accordingly believe that our approach should not be considered as HARKing, since we do not diverge from our preregistered criteria; had the MEG results been reliable, they would have fulfilled the preregistered criteria. Given the substantial doubts about their reliability, with which the reviewer seems to agree, we could not conclude that this was the case.

3. The authors wrote that “[a]ll research was conducted by theory-neutral teams to guard against confirmatory bias.” But what is the evidence for this? Importantly, as I have pointed out above (Limitations point 1), in a sense, arguably the entire project is relatively theory-neutral. What is more important may just concern those specific ROIs, on which the specific hypotheses were based. Is there any record of whether the researchers a priori found some of these hypotheses to be more plausible than others? To the extent that there is likely some such bias, at least in some, can we really say that everyone is truly neutral just because they are not currently in the same labs as the theory PIs? In addition, it may also be helpful to clarify the theory-neutral status of the executive (non-theory) PIs.

We are happy to provide additional information regarding the neutrality of the consortium members who conducted the study (and we note that the process of ROI selection has already been addressed above in full, so we will not reiterate it here). Neither the three center PIs nor the PIs of the labs collecting and analyzing the data are affiliated with any of the theories. We were also supported by eight external expert consultants who provided impartial advice on the respective analysis. Of course, all researchers (and generally, people) are biased in some way and this may affect the way they will conduct their studies; however, we believe that in a big collaborative study like ours, and with the open science practices we followed, we did our best to limit such biases. We are happy to consider any concrete practice that the reviewer can propose to further limit bias, and if appropriate, we will gladly incorporate it for the second experiment, whose results are currently being analyzed.

4. The terms posterior “hot zone” and “posterior cortex” are often used. But these are not specific or meaningful anatomical labels suitable for an academic journal. As we see from Limitation #1, these unconventional labels really allow too much flexibility for the hypotheses to be meaningfully testable. Given the ROIs, I recommend just calling these early and late visual areas; otherwise would the authors have chosen these areas for another study testing IIT using other stimuli from a different modality? If the authors insist on using their own neologism, please give precise anatomical definitions for these labels.

We agree with the Reviewer that the terms used for the ROIs were underspecified in the original version, and have accordingly added a specific description of the ROIs for each analysis (please see above our reply to Major point #2).

5. The authors wrote that “[t]ogether, we identified divergent predictions of the theories and jointly developed an experimental design to test them”. Please be more specific here, especially regarding who generated these predictions and who designed the studies (and who didn’t). In the author contribution statement many authors seem to be involved in ‘Conceptualization’, but my understanding is that at least some of them played no role at all in experimental design. Conceptualization is a relatively vague term. Please be more specific, as a clearer delineation would help with some authors, especially trainees, who might not have been given any chance to have any input to the process, to be freed from being held responsible for what some may see as inherent design flaws in the project. Some flaws may be inevitable given the challenges involved in organizing such a large scale project, but still, clarity can only help.

We thank the reviewer for making this suggestion. Among the many lessons learned by embarking on a project of this magnitude and complexity is the issue of credit assignment in team science. We adopted the strategy of the International Brain Lab (<https://www.internationalbrainlab.com/>) employed in their publication (Aguillon-Rodriguez et al., 2021) to visualize a contribution matrix. Here, we list the CRediT taxonomy roles (Brand et al., 2015), along the y-axis and the authors along the x-axis. Within the matrix, the respective level of contribution of each member is reported on a 3-point color scale (major, equal, minor). Our contribution matrix can be found in the last section of the supplementary materials (Supplementary section 12), and we provide it here for convenience (Figure 2). We believe that this already goes beyond what is typically done in author contributions, and gives a good enough account of the different roles taken by the Cogitate members in this work.

CRediT	Author																																										
	Cogitate Consortium	Oscar Ferrante	Urszula Gorska-Klimowska	Simon Hein	Rony Hirschhorn	Aya Khelaf	Alex Lepauvre	Ling Liu	David Richter	Yanni Vidal	Niccolò Bonacchi	Tanya Brown	Praveen Sripad	Marcelo Armendariz	Katarina Bendtz	Tara Ghafari	Dorothy Helenyi	Jay Jeschke	Csaba Kozma	David R. Mazumder	Stephanie Montenegro	Alla Seedorf	Abdelrahman Sharafeldin	Shujun Yang	Sylvain Baillet	David J. Chalmers	Radoslaw Marcin Cichy	Francis Fallon	Frans J. Panaigiotaropoulos	Hal Blumenfeld	Sasha Devore	Ole Jensen	Gabriel Kreiman	Flores P. de Lange	Huan Luo	Melanie Boly	Stanislas Dehaene	Christof Koch	Giulio Tononi	Michael Pitts	Liad Mudrik	Lucia Melloni	
Conceptualization																																											

0	Did not participate; null contribution
1	Provided support to a specific deliverable/task; contributed in a meaningful, yet minimal way
2	Contributed equally in relation to others who participated in a similar capacity
3	Major contributor to this effort/task; designated as the corresponding leader to specific deliverable/task

Figure 2 CrediT assignment for the Cogitate project, specifically for the conceptualization contribution.

- In different places the authors describe the two tested theories as “leading” and “two of the most well-established theories in the field”. I’m not sure about whether these somewhat subjective phrases are appropriate here. In this context, it may be helpful to also discuss an alternative interpretation as supported by some empirical studies (21, 22). Based on these studies there seems to be an argument to be made that one of the theories, IIT, was never really so well received within the scientific community, and that the current popularity may have something to do with hyperbolic promotion in the media. Therefore the status of being “leading” and “well-established” may be questionable.

This comment has been raised by this reviewer elsewhere, as part of more far-reaching claims about IIT. However, our description of IIT as one of the leading theories in the field is based on empirical data and not media propaganda. In a previous work, we found that more than 100 experiments have interpreted their findings in light of IIT (Yaron et al., 2022), and even more papers (224) did so for GNW. As we continue to expand the ConTraSt database, we now have 119 experiments for IIT, and 270 for GNW (see here). This suggests that both theories are indeed very prominent in consciousness research (compared to other theories, such as HOT; see Yaron et al., 2022). The papers mentioned by the reviewer as claiming otherwise, while interesting, are informal surveys of researchers and the general public’s opinions, or analyses performed on social media metrics. We believe that establishing the status of a theory as leading is more meaningful when relying on published empirical work. Importantly, these two theories are frequently described as leading (Mashour & Alkire, 2013; A. Seth & T. Bayne, 2022) or “major” (Lau & Rosenthal, 2011). Thus, we do not think that our interpretation deviates from accepted norms in the field. However, given the reviewer’s concern, we no longer use the word ‘leading’ to describe the theories.

7. One strength of these studies was the large number of subjects. Can we more explicitly discuss what was the statistical power achieved here. Specifically, can one not do power calculations based on the preliminary analysis from one-third of the data, and accordingly estimate the power one would get based on using the remaining two-third of the data? And what is the rationale for doing this 1:2 split? Why not any other ratio? Is there an empirical basis or reference for that?

The reviewer suggests calculating the achieved (i.e., post-hoc) power using the effect sizes observed in the first-third of the data and the sample size in the remaining two-thirds. However, such an approach has been criticized, and shown to be not particularly informative (see Lakens, 2022). Instead, to address this concern, we opted for conducting a sensitivity analysis. In this analysis, the sample size, the desired alpha level and the desired power are used to determine the smallest effect size that can be detected. For example, for a simple paired sample t-test (two-tailed), with a sample sizes of 65 and 73 subjects (second two-thirds of the data), the MEG and fMRI analyses have 90% power at an alpha level of 0.05% for effects of size $d = 0.40$ and $d = 0.38$ respectively. Taking the full dataset of included subjects (97 MEG and 108 fMRI) yields 90% power at an alpha level of 0.05% for effects of size $d = 0.33$ and $d = 0.31$ respectively. As iEEG data are not analyzed at the group level in a straightforward way (due to heterogeneity in electrode placement), group level sensitivity analyses are not informative. For some of the analyses performed, however, there are no exact analytical ways to calculate power or sensitivity, and it would be necessary to resort to surrogate methods which are computationally expensive and less meaningful. However, as this example demonstrates, the power of our study is high to detect medium-to-small effect sizes.

With respect to splitting the data, we agree that other data split ratios have their own merits. For example a 1:1 split would allow a >90% power test and a subsequent identically powered replication (using the same parameters as above). Alternatively, it could be possible to develop an analysis pipeline with a very small sample (e.g., 5 subjects) and use the majority of the data to perform a single high-powered test. We plan to discuss this and other decisions made in a forthcoming paper describing the process of this adversarial collaboration.

8. In different places the authors refer the readers to please “see supplementary”. Please be more specific about what section or figure of the supplementary section they were referring to.

We thank the reviewer for the suggestion. In the revised manuscript we have improved the cross-referencing within the main, extended, and supplementary sections of our paper.

Hakwan Lau

Jul 23, 2023

References

1. S. Fleming, The state of consciousness science – and why the media got it wrong. *The Elusive Self* (2023), (available at <https://elusiveself.wordpress.com/2023/07/20/the-state-of-consciousness-science-and-why-the-media-got-it-wrong/>).
2. A. Seth, Finding the Neural Correlates of Consciousness Is Still a Good Bet. *Nautilus* (2023), (available at <https://nautil.us/finding-the-neural-correlates-to-consciousness-is-still-a-good-bet-352054/>).
3. S. Reardon, “Outlandish” competition seeks the brain’s source of consciousness. *Science* (2019), doi:10.1126/science.aaz8800.
4. F. Siclari, B. Baird, L. Perogamvros, G. Bernardi, J. J. LaRocque, B. Riedner, M. Boly, B. R. Postle, G. Tononi, The neural correlates of dreaming. *Nat. Neurosci.* 20, 872–878 (2017).
5. A. Watakabe, H. Skibbe, K. Nakae, H. Abe, N. Ichinohe, M. F. Rachmadi, J. Wang, M. Takaji, H. Mizukami, A. Woodward, R. Gong, J. Hata, D. C. Van Essen, H. Okano, S. Ishii, T. Yamamori, Local and long-distance organization of prefrontal cortex circuits in the marmoset brain. *Neuron* (2023), doi:10.1016/j.neuron.2023.04.028.
6. H. Zhou, M. Davidson, P. Kok, L. Y. McCurdy, F. P. de Lange, H. Lau, K. Sandberg, Spatiotemporal dynamics of brightness coding in human visual cortex revealed by the temporal context effect. *Neuroimage.* 205, 116277 (2020).
7. N. Block, What is wrong with the no-report paradigm and how to fix it. *Trends Cogn. Sci.* 23, 1003–1013 (2019).
8. J. K. Hesse, D. Y. Tsao, A new no-report paradigm reveals that face cells encode both consciously perceived and suppressed stimuli. *Elife.* 9 (2020), doi:10.7554/eLife.58360.
9. E. Hatamimajoumerd, N. A. Ratan Murty, M. Pitts, M. A. Cohen, Decoding perceptual awareness across the brain with a no-report fMRI masking paradigm. *Curr. Biol.* 32, 4139–4149.e4 (2022).
10. V. Kapoor, A. Dwarakanath, S. Safavi, J. Werner, M. Besserve, T. I. Panagiotaropoulos, N. K. Logothetis, Decoding internally generated transitions of conscious contents in the prefrontal cortex without subjective reports. *Nat. Commun.* 13, 1535 (2022).

11. P. U. Tse, S. Martinez-Conde, A. A. Schlegel, S. L. Macknik, Visibility, visual awareness, and visual masking of simple unattended targets are confined to areas in the occipital cortex beyond human V1/V2. *Proc. Natl. Acad. Sci. U. S. A.* 102, 17178–17183 (2005).
12. E. D. Lumer, G. Rees, Covariation of activity in visual and prefrontal cortex associated with subjective visual perception. *Proc. Natl. Acad. Sci. U. S. A.* 96, 1669–1673 (1999).
13. S. Kouider, S. Dehaene, A. Jobert, D. Le Bihan, Cerebral bases of subliminal and supraliminal priming during reading. *Cereb. Cortex.* 17, 2019–2029 (2007).
14. H. Lau, The Unfinished NCC Project (2022), doi:10.1093/oso/9780198856771.003.0003.
15. G. Vishne, E. M. Gerber, R. T. Knight, L. Y. Deouell, Distinct ventral stream and prefrontal cortex representational dynamics during sustained conscious visual perception. *Cell Rep.* 42, 112752 (2023).
16. R. Broday-Dvir, Y. Norman, M. Harel, A. D. Mehta, R. Malach, Perceptual stability reflected in neuronal pattern similarities in human visual cortex. *Cell Rep.* 42, 112614 (2023).
17. E. M. Gerber, T. Golan, R. T. Knight, L. Y. Deouell, Cortical representation of persistent visual stimuli. *Neuroimage.* 161, 67–79 (2017).
18. E. Podvalny, E. Yeagle, P. Mégevand, N. Sarid, M. Harel, G. Chechik, A. D. Mehta, R. Malach, Invariant Temporal Dynamics Underlie Perceptual Stability in Human Visual Cortex. *Curr. Biol.* 27, 155–165 (2017).
19. S. L. Macknik, M. S. Livingstone, Neuronal correlates of visibility and invisibility in the primate visual system. *Nat. Neurosci.* 1, 144–149 (1998).
20. M. Michel, H. Lau, On the dangers of conflating strong and weak versions of a theory of consciousness. *Philosophy and the Mind Sciences* (2020) (available at <https://www.philosophymindscience.org/index.php/phimisci/article/view/54>).
21. M. Michel, S. M. Fleming, H. Lau, A. L. F. Lee, S. Martinez-Conde, R. E. Passingham, M. A. K. Peters, D. Rahnev, C. Sergent, K. Liu, An Informal Internet Survey on the Current State of Consciousness Science. *Front. Psychol.* 9, 2134 (2018).
22. A. W. K. Yeung, C. A. Cushing, A. L. F. Lee, A bibliometric evaluation of the impact of theories of consciousness in academia and on social media. *Conscious. Cogn.* 100, 103296 (2022).

Referee #2:

//Summary

A. Summary of the key results

The authors have demonstrated that it is possible to investigate competing theories of consciousness in a comprehensive and theory neutral way.

B. Originality and significance: if not novel, please include reference

The approach in this project is groundbreaking, and a major contribution to - and advancement of - the field.

C. Data & methodology: validity of approach, quality of data, quality of presentation

The data are impressive, and the approach is one-of-a-kind (but will hopefully become the standard in the future)

D. Appropriate use of statistics and treatment of uncertainties

I am not qualified to judge this.

E. Conclusions: robustness, validity, reliability

I had hoped for stronger conclusions, but understand why some of these may be postponed to future work. Do see my comments below though.

F. Suggested improvements: experiments, data for possible revision

I have no suggestions for improvements of the empirical work, but below are suggestions related to this manuscript specifically.

G. References: appropriate credit to previous work?

Did not find any critical omissions.

H. Clarity and context: lucidity of abstract/summary, appropriateness of abstract, introduction and conclusions

See below for comments.

//

We thank the reviewer for the positive evaluation of our manuscript and of our ability to test the theories. We further thank him for the many insightful and useful comments.

Dear Authors,

Congratulations on this major accomplishment. I am honored and grateful to have been given the opportunity to review the results of something that I (and pretty much our entire field) have been eagerly following and anticipating over the last several years. You have done an excellent job, and your work is (to quote your manuscript) “paradigmatically scientific”.

I have found no critical issues with this manuscript, but I do have a range of comments and recommendations. For this reason, I will recommend minor revisions, even though I do not think acceptance of this manuscript should be conditional on any one revision in relation to the comments below (let alone all of them). I do hope, however, that you will see the value in at least some of the comments and recommendations and make changes accordingly. But I will leave this at your discretion. Importantly, if the editors judge that any of my comments

are of such character that acceptance should be conditional on them being addressed, I defer to their judgement on this.

Below are my comments in chronological order (in relation appearance in the manuscript) tagged with line numbers. The issues I see as most important have been given a headline surrounded by **.

Kind regards
Asger Kirkeby-Hinrup

We are very grateful for the positive and enthusiastic evaluation of our work, and for recognizing our study as 'paradigmatically scientific'.

//Detailed comments

1. Lines 91-99: This first paragraph is maybe a bit too succinct, leading to minor inaccuracies. For instance, it seems to suggest that the hard problem and explanatory gap 'caused' ("led to": line 93) theories of consciousness, when in fact many theories (including three considered in the adversarial collaboration project, i.e. HOT, GWT, and first order theories [which later became RPT]) were around before Chalmers' 1995 paper. While they generally do not predate Levine's 1983 paper, the formulation is still somewhat awkward. (I assume what the authors want to say is something along the lines of "the perplexing phenomenon, the issues with which have been described as the hard problem, or in terms of an explanatory gap, has led to..."). In any case, I'd recommend spending one or two lines more to spell out properly the background and connect the many claims better, which would make the introduction less abrupt and more readable.

We thank the reviewer for this comment; indeed, the wording in the first paragraph was confusing. In the revision, we modified this part of the introduction (p. 4, first paragraph) and removed the reference to the explanatory gap.

2. Line 94: A minor point possibly worth noting that according to the Signorelli et al. paper referenced by the authors in line 94), some theories do not aim "to explain the subjective nature of consciousness" (line 92). In fact GNWT and IIT are on opposite sides of the quantity-quality distinction proposed in the Signorelli paper (Signorelli et al., 2021, pp. 5-6. see also fig. 2 on p. 7). This, however, is not of any huge consequence here, since the crux of this manuscript does not hinge on this (and if Stan does not object to GNWT "trying to explain subjective nature of consciousness", this point can be safely ignored).

Although GNWT and IIT do not have the same explanatory profile, as nicely explained in Signorelli et al. paper, we respectfully argue that GNWT does try to explain the subjective nature of experience. The proponents of the theory have explicitly stated this in print. For instance, in Dehaene and Naccache (2001) the authors wrote: "We postulate that this global availability of information through the workspace is what we subjectively experience as a conscious state." A more recent paper by a major GNW theorist is entirely focused on this same point (Naccache, 2018). Also, like all other co-authors, Dehaene read this manuscript and did not object to this sentence. Accordingly, we decided to keep it.

3. Line 124-126: The formulations of the two positions in prediction #1 are not mutually exclusive. It is possible that conscious content can be "instantiated primarily in posterior brain areas" even if the PFC is necessary. Do the authors intend something stronger? (see below regarding formulation of prediction#1)

Indeed, the reviewer is correct in identifying that the two positions in prediction #1 are not mutually exclusive. The major point of contention is the necessity of the prefrontal cortex. We have reworded the description to make this point clearer (p. 9, first paragraph). For the reviewer's convenience, we copied the corresponding paragraph below:

“According to IIT, PFC is not necessary for visual consciousness and therefore decoding of conscious content should be maximal from posterior cortex. According to GNWT, PFC is necessary for consciousness and therefore every content of consciousness should be decodable from the PFC”

4. Line 251: This seems to be the best formulation of prediction#1 so far, but it is still a little ambiguous (in the sense noted above) and would perhaps be better characterized by saying “involvement of PFC” or “activation of PFC” rather than “necessity of PFC”. (see below regarding formulation of prediction#1)

We appreciate the reviewer's critical assessment of the terms used to describe this prediction, yet we still believe that “necessity” is an important term to convey GNWT's predictions. We have reformulated the sentences as shown above (p. 9, first paragraph).

5. **Figure 1**

Figure 1.a. Formulation of Prediction#1 is nontrivially different from the formulation in Line 124-126. The formulation here seems better (but above note regarding the formulation in line 124-126 applies here as well). (see below regarding formulation of prediction#1)

Figure 1.b seems out of place in the middle of a figure laying out empirical elements. My recommendation is to remove it, as it does not contribute significantly to advancing any point in the paper. Everything it says already is expressed in lines 166-173, and the illustration does not add anything on top of those lines. If the authors/editors decide to keep it, I recommend moving it into its own figure (to be placed around line 173), so it does not disrupt the “empirical flow” in figure 1.

We thank the reviewer for these comments, we have revised the wording of prediction 1 (as stated in the previous response; p. 9, first paragraph) and also removed figure 1b.

6. Figure 1.h: Scales on top row are not identical (going to 0.14 and 0.12 respectively) yet with same color coding. This is potentially misleading (e.g. white color means different things in the pictures). Also, font formatting on top row middle picture seems off.

We again thank the reviewer for these suggestions, which have been fixed in the revision.

7. **Prediction#1**

Comparing the way this is presented in the paper with the pre-registered predictions (linked in line 261, the pdf file named “Preregistration_ver4.0-FINAL.pdf”) there seems to be some conflation going on. In the preregistered prediction (PRP), there are actually two predictions that seem to be collapsed into Prediction#1 in the manuscript. In PRP there is a prediction related to “Neural Activations related to consciousness”, which says something that sounds very much like the way Prediction#1 is presented (especially in the earlier sections I mentioned above), namely: “for GNW, activation of prefrontal areas is a prerequisite for conscious experience, while IIT, in contrast, submits that activity in posterior areas is sufficient for consciousness” (page 6 of Preregistration_ver4.0-FINAL.pdf). Immediately

after in the PRP (also p. 6) there is a prediction related to decoding, which seems to be the prediction that is addressed in this manuscript. In the PRP that prediction is phrased “In short, GNW predicts that the content of consciousness should be decodable from prefrontal areas, whereas IIT predicts that they should be maximally decodable from posterior areas.”. This decoding prediction aligns much better with what would be the proper wording of prediction#1 in this manuscript. I would strongly encourage the authors to rephrase the presentations of Prediction#1 (in the introduction and figure 1, and maybe parts of the text under the headline prediction#1) to be cached in terms of the decoding prediction, rather than varieties of ‘instantiated primarily’ and ‘necessity’. This would: 1) make the prediction clearer, 2) prevent readers balking at its non-exclusivity (as I did above), and 3) align better with the data presented (e.g. especially the conclusion of prediction#1 in line 336-37).

We appreciate the reviewer's suggestion and have amended the definition of Prediction#1 as follows (p. 9, first paragraph):

“Prediction #1: According to IIT, PFC is not necessary for consciousness and therefore decoding of conscious content should be maximal from posterior cortex. According to GNW, PFC is necessary for consciousness and therefore every content of consciousness should be decodable from the PFC”.

8. Line 423: The combination of “despite” and “serendipitous” seems weird. Do you mean “Due to serendipitous electrode implantation”? or maybe something like “despite our lack of control over electrode implantation”? (The way I read it, the point is that you were “lucky” that the patients had electrodes that fitted well with your ROIs. On this reading “despite” doesn’t mesh well with “serendipitous”)

We thank the reviewer for noticing this point; this sentence was rephrased in the revised manuscript as follows (p. 13, fifth paragraph):

“Although we lacked control over the placement of electrodes, the sampling density of electrodes in both the posterior cortex and the prefrontal cortex (PFC) was consistently high and evenly distributed across ROIs pertinent to the theories.”

9. Line 503: “theories predictions” (missing possessive apostrophe).
10. Line 515: “theories predictions” (missing possessive apostrophe).

We thank the reviewer for pointing out the typos at line 503 and 515. Both have been fixed in the revision (p. 15, fifth and sixth paragraphs).

11. ****Figure 3****
 - 3.a and 3.c may need to be shown in bigger format (or use more contrasting colors). It is almost impossible to see the purple and pink electrodes on 3.a (IIT x Category/Face) and the Black electrode on 3.c (GNWT x phasic onset). Furthermore, the legend on 3.a seems to show part of the legend for 3.c (last line saying GNWT in green), and 3.c does not have a legend.
 - 3.e Line 552: shouldn’t it be “for category (left and middle left), identity (middle right) and orientation (right)” ?
 - 3.f Line 559: shouldn’t it be “as in Figure 3e” ?

We thank the reviewer for the excellent suggestions and corrections; and apologize for having missed that in the original submission. The manuscript and figures were amended accordingly.

12. ****Line 587-588: “the activity was not found in the gamma frequency predicted by IIT”. ****I could not find any previous point in the manuscript where it was stated that IIT predicted specifically Gamma activity (maybe I missed it?). Gamma is mentioned as part of the GNWT prediction in line 569. Should this be reformulated to cover both the prediction of IIT and GNWT? I checked against the preregistered predictions (Preregistration_ver4.0-FINAL.pdf referred to above), and do not see any specification that it has to be gamma in the IIT prediction on synchronization (p.28 of PRP). It does mention gamma in the IIT prediction on “activity pattern analysis” immediately before (p. 27 on PRP), is this supposed to ‘carry over’ to the synchronization prediction? (I could not find any explicit statement on this in the PRP).(The connection between IIT and gamma is mentioned again in line 633)

We thank the reviewer for the thoughtful and attentive reading of the manuscript. We realize the rationale for the frequency selection was not clearly stated in the preregistration. As spiking activity is considered a constituent property of the physical substrate of consciousness by IIT (Tononi et al, 2016), in discussions with IIT proponents we decided to focus on gamma activity, as it is known to closely reflect neuronal spiking (Cardin et al., 2009). As suggested by the reviewer, we reformulated the paragraph covering the prediction of IIT and GNWT with respect to the gamma frequency (p. 18, first paragraph), and explained its rationale (p. 18, third paragraph). We also studied alpha activity (8-13 Hz), especially in MEG, to complement the gamma analysis. To explain, alpha activity is known to be anticorrelated with neuronal spiking, thereby serving as a good proxy for it (Haegens et al., 2011; Jemi et al., 2022). We emphasize that the activation analysis was also performed on ERP/ERF (0-30 Hz), and the synchronization analysis on the entire available spectrum, in light of debate regarding the significance of gamma activity as a marker of consciousness (Koch et al., 2016). These results are reported in the main text (e.g., p. 19, third paragraph; p. 20, first paragraph) and in the additional analyses reported in the supplementary materials (supplemental section 7.1) and their findings consistently align with those observed in the gamma range.

13. Line 733: “Siloes” is a peculiar word choice here. (Does this refer to “silo theory”?), maybe use ‘echo chambers’ or something more recognizable to the average reader (I had to google silo theory..). Affixed to “theoretical siloes” there is a reference to your Contrast paper, but “theoretical siloes” is not mentioned anywhere in that either(?).

We thank the reviewer for this comment and have rephrased it to: “echo chambers” (p. 24, first paragraph).

14. Line 756-757: The claim that “orientation is a fundamental property of conscious experience” is maybe a bit strong. First of all, it is only a property of visual conscious experience, not conscious experience as such. Secondly, even within the visual domain there is a case to be made that only low level properties (colors, edges etc) are ‘fundamental’ in the sense that other properties are derived from these. (Also observe that there is a possible ambiguity here in the sense that when ‘orientation’ is used here [in my understanding] it refers to the direction a person is facing [c.f. figure 1.d], whereas there is another sense in which orientation is a fundamental [low level] property, namely when it refers to the angle that selective neurons in V1 are sensitive to, but this is not the meaning referred to in the manuscript [I think]).

We agree that the original sentence was too strong, and have accordingly changed it (p. 24, fourth paragraph). The original sentence referred to orientation being a part of conscious experience, and thus, according to some mid-level accounts (Prinz, 2017) is fundamental to conscious experience. We have rephrased it to:

“More broadly, although IIT passed the predefined criteria for the duration prediction (#2), there was no evidence for a sustained representation of orientation, despite being a property of the consciously perceived stimuli”.

15. Line 784: “Primal” is a weird word choice here. Do you still mean ‘fundamental’?

We agree and have changed the sentence (p. 25, second paragraph) to:

“Another key challenge for GNWT pertains to representing the contents of experience: though we found representation of category in PFC irrespective of the task, hereby demonstrating the sensitivity of our methods, no representation of identity was found, and representation of orientation was only evident in MEG (without being able to exclude source leakage effects), although these dimensions were clearly a part of subjects’ conscious experience of the stimuli.”

16. Line 812: From what I remember (its been a while since reading), Popper (referred to here) was concerned with falsifying theories through testing predictions, not falsifying predictions. This is a minor point though.

We thank the reviewer for pointing out this confusion. We have reworded the sentence (p. 25, last paragraph) to:

“Within the framework of this adversarial collaboration, our aim is to challenge and potentially falsify (Lakatos, 1976; Popper, 1935) IIT and GNW , by examining where their predictions differ...”

17. Line 923-926: Would this interpretation of the failure to see ignition at stimulus offset not entail more or persistent PFC activation as it seems to invoke cognitive (PFC reliant) processes (similar to Block’s bored monkey consideration)? Here is a (speculative) alternative interpretation that (I think) is a better explanation for GNWT for the lack of offset ignition (this is an elaboration on the considerations found in lines 931-935): If one assumes ignition serves to update on the arrival of new information, offset should not trigger ignition in cases where the disappearance of the stimulus results in less information than before (i.e. a grey screen contains less information than a grey screen with an image). Alternatively, there is another interpretation where the “grey color behind the stimulus image” was already inferred from (contained in) the stimulus image. If either of these is the case then there would be no need for updating (ignition) on the stimulus offset, because either there was ‘less’ information on offset, or the information was already contained in the previous broadcast.

The reviewer raises an interesting interpretation, and we agree that it could potentially explain the results. Reviewer 1 (point 4) proposed an alternative explanation of the absence of a prefrontal offset response, in terms of predictions silencing the response to the (expected) blank screen. Both possibilities could be explored in future studies (unfortunately, due to

space limitations, we cannot discuss them in the paper itself). In the context of our adversarial collaboration, we consider this finding an opportunity for the theory to refine its account for the maintenance of a conscious percept over time, and hope that our results will continue to inspire researchers to explore alternative explanations for the current findings.

18. ****Conclusion (Cogitate consortium)****

For a project of this size, cost, duration, and ambition I think this conclusion is inadequate. You are not wrong that science is a collaborative (social) enterprise but leaving the reader with “make of this what you will” is a letdown. You have done excellent science (!) and the readers (and I guess maybe also your collaborators and the funders) deserve to know your conclusions. The conclusion does not have to appoint a winner (you could rename it to ‘concluding remarks’, so readers do not expect a “conclusion”), and I agree that the consortium should preserve neutrality, but still there are plenty of things you could bring up in this space. The following points are just a few suggestions.

A) Both the IIT and GNWT replies above indicate to limitations in the data (Line 836 and 907 respectively). What were these limitations, what do you think about them, could they have been avoided, will they be ameliorated in the next parts of the adversarial collaboration project?

b) Regarding the PFC, IIT and GNWT seem to draw opposing conclusions from the data (lines 841-42 and 895-96 respectively), what are the implications of this? Could this have been avoided?

c) Stan points to the future studies to help resolve outstanding questions here (line 947), maybe foreshadow these studies, discuss how they relate, and why he is optimistic.

d) What are the (long term) prospects of adversarial collaboration to resolve these debates? It is basic scientific practice to revise theories when they make failed predictions. However, when a theory is revised, we would need a whole new round of multi-year-multimillion dollar adversarial collaboration to test the new version. Is there a ‘halting problem’ in this regard? (c.f. “no one changes their mind”).

e) Maybe some general considerations and reflection on the project? What would you have done differently? What are the positive upshots? Has the current test led to new ideas for other tests? Have you had ideas on how to refine the paradigms used in this test that may avoid shortcomings?

f) You have set a magnificent and high bar for comparison of theories. However, the cost (in money, expertise, and time) is prohibitively high for almost everyone else in the field. Is there a way to emulate this approach on a smaller scale? Is there any way to make this approach “consumer grade”?

We thank the reviewer for these valuable suggestions. We agree that more discussion would be beneficial, and we plan to provide such elaborated discussion in an upcoming publication that focuses on the process of adversarial collaborations, using our project as a case study.

Generally speaking, although we agree that more discussion would have been helpful, we could not cover all relevant points given the word limitation. Moreover, we adhered to the guidelines for adversarial collaborations put forth by Daniel Kahneman and Barbara Mellers (see Table 1 below) when structuring our discussion. We see the current publication as a prompt for further discussions in the community, and for readers to become a part of the scientific process reaching their own conclusions in light of the presented evidence. We understand that this might appear as an unorthodox decision, but we hope the compromise of providing a discussion in a follow up publication is well received. In the revised

manuscript, we added a reference to the process of adversarial collaboration to bring more context to our rationale (p. 24, first paragraph, reference #17).

Table 1. *Suggestions for adversarial collaboration*

1. When tempted to write a critique or to run an experimental refutation of a recent publication, consider the possibility of proposing joint research under an agreed protocol. We call the scholars engaged in such an effort participants. If theoretical differences are deep or if there are large differences in experimental routines between the laboratories, consider the possibility of asking a trusted colleague to coordinate the effort, referee disagreements, and collect the data. We call that person an arbiter.
2. Agree on the details of an initial study, designed to subject the opposing claims to an informative empirical test. The participants should seek to identify results that would change their mind, at least to some extent, and should explicitly anticipate their interpretations of outcomes that would be inconsistent with their theoretical expectations. These predictions should be recorded by the arbiter to prevent future disagreements about remembered interpretations.
3. If there are disagreements about unpublished data, a replication that is agreed to by both participants should be included in the initial study.
4. Accept in advance that the initial study will be inconclusive. Allow each side to propose an additional experiment to exploit the fount of hindsight wisdom that commonly becomes available when disliked results are obtained. Additional studies should be planned jointly, with the arbiter resolving disagreements as they occur.
5. Agree in advance to produce an article with all participants as authors. The arbiter can take responsibility for several parts of the article: an introduction to the debate, the report of experimental results, and a statement of agreed-upon conclusions. If significant disagreements remain, the participants should write individual discussions. The length of these discussions should be determined in advance and monitored by the arbiter. An author who has more to say than the arbiter allows should indicate this fact in a footnote and provide readers with a way to obtain the added material.
6. The data should be under the control of the arbiter, who should be free to publish with only one of the original participants if the other refuses to cooperate. Naturally, the circumstances of such an event should be part of the report.
7. All experimentation and writing should be done quickly, within deadlines agreed to in advance. Delay is likely to breed discord.
8. The arbiter should have the casting vote in selecting a venue for publication, and editors should be informed that requests for major revisions are likely to create impossible problems for the participants in the exercise.

Table 1, taken from (Mellers et al., 2001).

19. Line 1509: “missing data were excluded” sounds weird. Do you mean trials with missing data? Or do you mean cases where either gaze or pupil data were missing? (same in line 1518-19)

The sentence was indeed unclear, and we thank the reviewer for pointing this out. We now rephrased it as “trials with missing data were excluded” (p. 38).

20. Line 1548-1552: Is there a reason the text here is grey rather than black? (maybe it’s a convert to pdf glitch?) (Same with lines 1634-1636 and lines 1654-1662, maybe I’ve been staring at my monitor for too long?)

We apologize, but we couldn’t locate those differences. It may have been a pdf conversion issue.

21. Lines 1864-1871 (especially lines 1870-71) This is both very interesting and important and deserves to be in the manuscript (either in the pNCC section or the conclusion). Especially because of the potential implications outside of evaluating IIT and GNWT.

We are glad to learn that the reviewer considers these results important, an opinion we also share. However, following the comments of Reviewer 3 (and, to some extent, Reviewer 1), we decided to remove the pNCC analysis from the main paper, as to begin with it was not defined as crucial for testing the theories’ predictions. It is now fully described in supplemental section 8, where we also shortly discuss its implications (supplemental section 8.1.4).

22. Extended figure 10: I think this deserves to be in the main manuscript.

We thank the reviewer for this suggestion, which we adopted. The figure has been incorporated into the main paper as Figure 5 (which was possible since the pNCC figure was moved to the supplementary material). It’s important to highlight that this figure has

undergone some adjustments to now differentiate between the core and peripheral challenges faced by the theories. This includes an illustration of the importance of the predictions and measures for testing the theories, as well as the relative weighting of these predictions.

Referee #3

This first paper from the Cogitate consortium reports data from iEEG, fMRI and MEG experiments of visual object processing, collected across several different labs and involving a thorough pre-registered approach. A range of univariate and multivariate analyses are applied to the imaging data to evaluate 3 predictions (about neural correlates, timing and connectivity) made by advocates of two theories of consciousness – global neuronal workspace theory (GNWT) and information integration theory (IIT). I admire the adversarial collaboration and pre-registration approach and applaud all the authors involved here for a considerable amount of high-quality experimental work. The iEEG data from >30 patients with widespread electrode coverage is particularly impressive and will provide a unique resource for the community.

We thank the reviewer for the positive feedback and for recognizing the strengths and uniqueness of our work.

That said, I have serious concerns about the scientific weight being placed on the current results for each of the theories, and for consciousness science more broadly. First, the results concern neural correlates of visual processing, not visual consciousness – as unlike in the ongoing Exp. 2 from the Consortium, there is no experimental operationalisation of conscious vs. unconscious processing here. The task is relatively standard, involving brief presentations of visual objects (faces, objects, letters, false fonts) from different viewpoints and for different stimulus durations, with an additional manipulation of task relevance. Neural decoding of many of these stimulus properties is maximal in posterior (visual) cortex – this is to be expected from mainstream vision science (eg the many papers from Kanwisher, Tsao, di Carlo, Rust and others). There is also no investigation of mechanism or computation (although future studies using these open data might aim for this). Instead, all the predictions pertain to neural correlates of visual processing over time and space, using different imaging techniques.

The reviewer makes several points here, which we will address one by one.

*First, the reviewer is correct that the current experiment does not include an experimental operationalization of conscious vs. unconscious processing. As we explained in our reply to comment 3 by Reviewer 1, this contrast would be needed if our goal was to identify the neural correlates of consciousness. This was not the case here; instead, we aimed at **testing the neural predictions of two theories of consciousness**. All predictions pertain to what should happen in the brain, according to the theories, when participants are undeniably conscious of the stimuli. In that respect, we argue that it's precisely the simplicity of the paradigm that confers its strength, as it allows us to create **critical tests for which a failure cannot be easily dismissed**. To put it differently, while confirming the theories predictions can be contaminated by unconscious processing, failing to confirm predictions for a clearly visible stimulus is highly meaningful when challenging the theories, which is exactly what we have been aiming for. Thus, and importantly, the lack of unconscious conditions cannot explain the most critical results obtained here – the challenges to the theories, which is the main trust of this study.*

Second, the reviewer suggests that the task is relatively standard. As we explain in our reply to comment #2 by Reviewer 1, although our task was indeed inspired by a few studies that

used a similar design (which we acknowledge in the manuscript), none of these studies included all the different manipulations that our design had (category, orientation, identity, duration and task), none of them were run using three different techniques, and none of them was designed to test these theoretical predictions. We are in fact quite proud that we were able to adopt an already established, simple task, to achieve all these goals in one experiment, and consider that a merit of our design.

*Finally, the reviewer notes that finding maximal decoding in posterior areas is to be expected from mainstream vision science. We agree – which is one of the reasons why **this prediction is not one of the key tests for IIT**. The weakness of prediction 1a is precisely why we introduced the stronger prediction 1b (decodability from posterior areas should not be enhanced by adding certain prefrontal areas, which was also not defined as a key test), and – most importantly – why the pre-registration marks predictions 2 and 3 as IIT’s most critical predictions. We realize this was not clear enough in the original manuscript, although it is clearly stated in the pre-registration. We have accordingly revised the manuscript to state this explicitly (p., 9, first paragraph; and for GNW, p. 14, last paragraph), and also modified figure 5 (previously Extended Figure 10) to make this distinction clearer. The two other IIT predictions are much riskier; and in fact, both provided key challenges for IIT. To recap our findings, only a small fraction of our electrodes showed sustained activity (25 electrodes out of 657), demonstrating that this is in fact not a very prevalent finding (which we have now emphasized in the revised version), and there was no sustained representation of orientation, nor sustained synchronization. This shows that the predictions were not trivial after all, and had the potential to challenge IIT, as indeed happened.*

These design choices are justified by the consortium by appealing to the power of a simple suprathreshold task for falsifying tenets of each of the two theories. The scientific weight of the paper therefore turns on whether the neural measures provide strong tests of the theories of consciousness under consideration, IIT vs. GNWT. Here unfortunately it becomes clear as one reads the paper that this theoretical operationalisation is at best asymmetric. GNWT is under clearer test, as from its inception it has made anatomical commitments to the involvement of a frontoparietal network in conscious processing. But I struggled to understand the provenance of the IIT test (see specific comments below). As Dehaene points out in the discussion, the mathematical backbone of IIT was not invoked, and the results showing posterior cortical decoding of object features are “just what any physiologist familiar with the bottom-up response properties of those regions would predict, since visual neurons still respond selectively during inattention or general anesthesia” (p. 27). This is unfortunately not just an adversarial discussion point, but a serious indictment of the entire project and its relevance to either consciousness or IIT (and hence it’s somewhat surprising that the GNWT side signed off on these predictions in the first place).

We thank the reviewer for this important remark as it gives us the opportunity to provide further context and nuance to what is being tested in this adversarial collaboration and in what form.

*We beg to differ on the claim that the operationalization of the test is asymmetrical. First, as explained above in our reply to comment #1 by Reviewer 1, the objective of the Cogitate consortium is to test hypotheses concerning the respective **neurobiological implementations of two theories** of consciousness rather than their conceptual or computational claims. For example, at the conceptual level, the global workspace theory (Baars, 1988) posits that consciousness is associated with the broadcasting of information. This broadcast is held to*

trigger global availability, integration and distribution of information across a series of otherwise isolated modules. Dehaene and colleagues have developed a neural version of GWT, namely the global neuronal workspace theory (GNWT), and it is this theory that we aimed to test (Dehaene, 2001; Dehaene & Changeux, 2011; Dehaene et al., 1998; Mashour et al.). According to GNWT, the workspace is implemented by neurons with long-range axons connecting fronto-parietal and higher-order sensory areas. Therefore, despite not testing broadcasting itself, we tested GNWT's central claim concerning the necessity of prefrontal cortex (PFC) in the hypothesized broadcasting of information via decoding of visual content and synchronization between PFC and high-level visual areas in the occipital-temporal cortex. We focused specifically on PFC because it is in that region that the predictions from the two tested theories maximally diverged.

Conceptually, the Integrated Information Theory (IIT) posits that the substrate of any conscious experience corresponds to a maximum of integrated information, denoted as Φ (Haun & Tononi, 2019; Tononi, 2012; Tononi et al., 2016). Computer simulations have shown that connected graphs in the form of 'pyramid-of-grids' specify an exceedingly large number of distinctions and relations, ensuring a maximum of integrated information (or Φ), whereas a (near)-modular system can be broken down to its components, leading to lower integrated information. In a neural version of IIT, Tononi and colleagues rely on the anatomy of the temporo-parietal-occipital cortex: their claim is based on previous neurophysiological, anatomical and functional imaging studies of mammalian neocortex, showing that these areas comprise 3D lattices of dense synaptic connections among topographically organized areas (Haber et al., 2022; Lund et al., 1993; Oh et al.; Watakabe et al., 2023). Based on these findings, IIT claims that these regions, by and large, should be a prime candidate for the substrate of consciousness. To test the neural predictions of IIT about the temporo-parietal-occipital cortex, our study focused on activation and synchrony patterns that are held to underlie such a causal structure in a content-specific manner.

As we explained in the above reply, we agree with the reviewer that prediction 1a (Extended table 1) poses a stronger test for GNW than for IIT, and this was the reason for it not being defined as a critical prediction for IIT. **We stress that prediction #2 and #3 by IIT are far from being trivial or easy to confirm.** We point the reviewer to our response to point #8 by Reviewer 4, where we comment further on why the patterns of responses predicted by IIT are not trivial, nor do they comply with what any physiologist would expect.

In fact, as we wrote above, IIT's prediction 2 (at least in the case of orientation) and prediction 3 were both falsified, substantially challenging IIT as a result. **The fact that these predictions were falsified suggests that the predictions were not overly weak.** Thus, IIT provided at least two non-trivial predictions, much like GNW.

Finally, and more generally, adversarial collaborations are not "science as usual". They are complex, but with due care in the process of formulating predictions about the same experiments from advocates of theories, theories can be brought out of their comfort zones and challenged. This is the process this consortium has set out to achieve, and the result is that theories have been challenged.

Specific comments:

1. I tried my best to track down the origin of the key IIT hypothesis (that is invoked by all the Predictions, but especially Prediction 1) that posterior cortex should be more associated with consciousness than other regions. In the current paper (p. 4 and p. 26), the reader is referred to Tononi et al. (2016) NRN (reference 5), where there is discussion of the potential physical substrates of consciousness (PSC) based on IIT. But the main claim in Tononi et al. (2016) was that cortex should have higher Φ than the cerebellum – there was little mention of PFC vs. posterior cortex (or a putative hot zone). It was also proposed that large-scale computer simulations based on the known anatomy of cortical circuits would be needed to evaluate network predictions for Φ . Such simulations were not conducted for the current study, and instead a hypothesis that posterior cortical regions should be the “main complex” is based on an assertion that they have “dense local connections arranged topographically into a hierarchical, divergent-convergent 3D lattice” (p. 26). I don’t see how one can establish this is the case for posterior but not prefrontal cortex without detailed knowledge and evaluation of the relevant micro-anatomy and connections involved. I then went back to the pre-registration document which states (p. 5) “current theoretical and neuroanatomical considerations suggest that a complex of maximum Φ is likely to reside in the posterior cerebral cortex, in a temporo-parietal-occipital ‘hot zone’ (Koch, Massimini, Boly and Tononi, 2016b)”. But upon reading the cited Koch et al. (2016) NRN paper, IIT was barely mentioned, and the hot zone was instead motivated with reference to empirical data on visual awareness. In summary, then, the posterior cortex prediction is not derived from IIT, and seems chiefly motivated by empirical findings, which is circular. This is especially concerning given that standard visual neuroscience would predict robust decoding of visual stimuli in these regions, as noted above.

We first want to reiterate that this prediction was not a key prediction by IIT, as was clearly stated in the pre-registration. We regret not making that clearer in the original manuscript, and for not clarifying the rationale behind it.

The reviewer also asks about the support of this prediction in previous literature. Below we provide some quotes in that spirit. For example, the relevant section in Tononi et al. (2016):

“In a healthy, awake participant, the set of neural elements specifying the conceptual structure with the highest Φ_{max} is assumed, based on current evidence, to be a complex of neuronal groups distributed over the posterior cortex and portions of the anterior cortex⁵” (p.456, Figure 3b caption; the reference cited at the end of this sentence is: (Tononi et al., 2016).

“In general, the coexistence of functional specialization and integration in the cerebral cortex is ideally suited to integrating information... Specifically, the grid-like horizontal connectivity among neurons in topographically organized areas in the posterior cortex, augmented by converging–diverging vertical connectivity linking neurons along sensory hierarchies, should yield high values of Φ_{max} ” (same paper, p.458, first full paragraph).

We agree with the reviewer that the Koch et al. (2016) paper focused on empirical evidence, but it also contained several meaningful references to IIT and GNWT. For example:

“It is generally recognized that consciousness also requires an integrated neural substrate” (p.315, third full paragraph; the reference cited is (Tononi, 2012), a paper on IIT).

“The anatomical basis of the full NCC and content-specific NCC do not comprise the wide fronto-parietal network emphasized in past studies, but are primarily localized to a more restricted temporo-parietal-occipital hot zone with additional contributions from some anterior regions.” (same paper, p.315, penultimate paragraph).

“Experiencing specific contents associated with activity in the posterior hot zone does not require the amplification of fronto-parietal network activity¹⁴².” (same paper, p.315, last paragraph; the reference cited is (Dehaene & Changeux, 2011), a paper on GNWT).

2. There are similar albeit less prominent concerns about the GNWT predictions. Why should GNWT predict activity-silent maintenance, which underpins the onset/offset Prediction 2? The only reference for this given in both the paper and the pre-registration is the Stokes et al. TICS paper – but this paper does not mention GNWT or consciousness and is instead a model of working memory (which presumably could be conscious or unconscious).

We thank the reviewer for highlighting this aspect. We understand that more details with respect to the rationale of the predictions is desired, but space limitations prevent us from covering them in further detail in the manuscript. We have however published a registered protocol (Melloni et al., 2023) covering in more detail the rationale for this prediction and the other predictions. We have added the reference to this publication in the revised version (p. 13, first paragraph; p. 25, third paragraph). In a nutshell, for GNWT, the activity-silent maintenance is the neuronal mechanism by which the information that falls outside the focus of attention is maintained (see Mashour et al., 2020, Neuron, for a review on GNWT). Although this prediction is indeed new, it is rooted in previous publications by the proponents of GNWT. Specifically, in the above-mentioned review by Mashour et al. (2020), GNWT proposes that the content of “working memory is conscious only when it is coded by global, highly distributed persistent neural firing, as occurs during both initial encoding, during the later refresh stage, and when the memory item influences other mental processing steps”, and “conscious ignition is a first step leading to the entry of information into working memory”. This was the basis for the novel claim by GNWT.

3. The “putative neural correlates of consciousness” analysis (Fig 5) is puzzling and out of place in the main text (it would be more appropriate in the supplement). The conjunction of contrasts relative to baseline implies the (already implicit) baseline is a special case of the contents of consciousness, and it’s not clear why this should be – as noted above, there is no manipulation of conscious vs. unconscious processing in this study. Instead, it’s a statistical map saying where in the brain is activated / deactivated by visual stimuli using fMRI, after masking out task-related activations. In this light, the widespread stimulus-triggered deactivation is also not surprising given that it looks like this is picking up the default-mode network (though this was difficult to assess from the axial slices).

We agree with the reviewer’s interpretation of this analysis, which is why we referred to it as “putative NCCs” (as opposed to simply NCCs) and did not feature it as part of the main predictions in which the theories were tested. We presented it for completion as while it does not test the theories per se, we still deemed it informative. Given the reviewer’s comment, which also resonates with one from Reviewer 1, we decided to deemphasize it even further and move it to the supplementary materials (supplementary section 8). We further explained why it is less informative for testing the theories, as opposed to the tested predictions #1-#3

(p. 23, first paragraph). We think that this puts the putative NCC analysis in the right context, and thank the reviewer for highlighting its limited nature.

4. Was awareness of stimulus orientation / viewpoint tested in the surprise memory test? I couldn't see these data reported in the supplement (p. 22-26), but one can imagine that this aspect of the stimulus might have a higher likelihood of being forgotten than other features. This is not to say that subjects were not conscious of orientation, but given orientation plays a strong role in testing Prediction 1, it seems important to establish behaviourally.

The reviewer is correct that the surprise memory test did not contain an explicit test for orientation. Instead, the memory test was geared to evaluate whether the stimulus as a whole, including its identity and orientation, was explicitly remembered. This is because the surprise memory task aimed at excluding the more fundamental objection that stimuli in the task-irrelevant condition were not consciously perceived. Specifically, in the exposure phase, participants saw 80 unique stimuli in various combinations of identity/ orientation/ duration/ task demands (counterbalanced across subjects). In the memory task phase, 40 out of the 80 stimuli were presented alongside 20 new fillers. The 40 old stimuli were presented in the same orientation as in the exposure phase. We thus reason that if we establish that the stimuli were consciously perceived (and therefore remembered), their orientation should have been perceived alongside – (and the reviewer seems to agree that this rationale is plausible).

We agree with the reviewer that a specific memory test for orientation would have been informative. The interpretation of such a test is however not equivocal: A positive finding confirming memory of orientation would suggest that subjects were indeed aware of the orientation. In contrast, a negative result would likely be inconclusive, as it could either reflect a memory lapse even when orientation was consciously perceived (i.e., attribute amnesia, which was claimed to occur even for attended and consciously perceived attributes; Chen & Wyble, 2015; Wang et al., 2021), or a true failure of perceiving the stimulus. Since we have already established conscious perception of the stimuli, we believe conducting such an experiment is unnecessary.

Referee #4

Referee report on “An adversarial collaboration to critically evaluate theories of consciousness,” Melloni et al.

Benjamin Kozuch
University of Alabama
Philosophy Dept.
bkozuch@ua.edu

The article that I reviewed (hereafter, “Article”) empirically investigates the issue of what neuroscientific theory of consciousness is correct. Remarkably, the project is a collaboration between proponents of competing theories of consciousness, who are joined by theory-neutral researchers that take a leading role in the experimentation. The goal that the authors of this article had (hereafter, “Authors”) was to evaluate two currently popular theories of consciousness, Global Neuronal Workspace Theory (GNW), and Integrative Information Theory (IIT). The work that the Authors carried out was paradigmatically scientific, in that they took a falsificationist approach to evaluating the theories, generating stringent predictions of each theory, then experimentally testing them. The researchers used the best of those methodological and technological tools that are available today, in abundance. The result is some of the more informative neuroscientific data concerning consciousness that we have to date. I recommend that this article be published in Nature.

We thank the reviewer for his positive appreciation of our work. These remarks resonate with the strong support of Reviewer 2 of the predictions made by the theories, of our experimental approach, and the methodologies used yielding informative results which they considered robust and scientifically paradigmatic.

At the same time, there are ways that the article could be improved and made even more informative. In the report that follows, I describe them.

1. As stated above, the article focuses on GNW and IIT. However, if we were to produce a complete list of leading theories of consciousness, it would include not just GNW and IIT, but also local recurrency theory (LR) (Lamme 2010) and higher-order theory (HO) (Brown et al. 2019), perhaps along with others. It seems, furthermore, that the results of the experiments presented in the Article shed just as much light on these other two theories (i.e., LR and HO) as it does in the case of the two that the article focuses on. And so for the article to give no discussion of these theories seems to be a missed opportunity for advancing the field of consciousness studies. More importantly, it gives the reader the mistaken impression that the field has narrowed down the candidate theories of consciousness to just to just GNW and IIT, something that seems far from the truth (Michel et al. 2018). I will explain.

We thank the reviewer for these insightful remarks. We very much agree that our results bear implications on many theories of consciousness (and are accordingly highly meaningful even outside the scope of this adversarial collaboration). In the original manuscript, we did not include a discussion of other theories due to a lack of space, but more importantly because we followed the adversarial collaboration protocol previously defined in Mellers et al 2002, including limiting the number of words for each section of the discussion a priori i.e., cogitate, IIT and GNWT discussion section. We understand this may have appeared as a

neglectful omission. Yet, it is hard to satisfy all constraints, including the strict word limits of journals and the adversarial collaboration protocols. With those constraints in mind, we have revised the manuscript to make clear that GNWT and IIT are just two theories among many others, while we also justify the selection of those theories (p. 4, second paragraph).

For convenience the corresponding passage is included here:

“... We focus on GNWT and IIT, among many other theories of consciousness (A. K. Seth & T. Bayne, 2022; Signorelli et al., 2021), as they feature prominently in the field of consciousness science, and have received substantial empirical support as shown in a recent systematic review of the literature (Yaron et al., 2021)”

After deliberation, however, we decided to avoid linking our findings to other theories, although we see great merit in doing so and completely agree that this should be done (in a separate paper). The main reason is, again, the word limit and the already dense nature of the manuscript. In addition, the HOT and RP theories have many variants, and only some of those are challenged by our data. There are currently at least two other adversarial collaborations (one led by Biyu He and another by Steve Fleming and Axel Cleeremans) that directly test predictions from those theories. We consider it more prudent to let those projects report their findings and relate to ours. Moreover, we are now preparing an additional paper reporting on the mechanics of the adversarial collaboration, where we will touch on the issue of how to relate findings of an adversarial collaboration to other theories, while also adhering to adversarial protocols i.e., defining predictions and designing experiments geared to challenge those predictions. We understand that this might seem disappointing, yet we hope the reviewer understands the constraints we are under.

As we anticipate that others will probably share the reviewer’s opinion, and rightfully so, we changed the concluding paragraph to explicitly acknowledge that expectation. Although we cannot meet it here, as we explain above, we hope to be able to do so in future papers about this work (e.g., after Experiment 2 is also complete, we plan to write a final paper integrating the results of the two experiments, which could be a good venue for this more elaborate discussion concerning how our results relate to other theories besides GNWT and IIT).

For the reviewer’s convenience, below is the revised excerpt from the discussion section (p. 30, last paragraph):

“At this point, the reader might expect the consortium to draw a final conclusion, or to relate the findings to other theories of consciousness, which without doubt put pressure on other theories i.e., some variants of higher-order (Kozuch, 2023) and recurrent processing (Lamme, 2015; Malach, 2021)”.

HO states that a mental state becomes conscious when it is represented by another mental state. According to how HO is usually construed (e.g., Rosenthal 2011; Brown 2015), the content that becomes conscious will be the content of the HO state (as opposed to the content of the lower-order state). Consider now that it is commonly thought that the kind of HO states that are essential for consciousness will be produced in the PFC (for review: o Kozuch 2023). Given this, it seems that HO predicts, just like GNW, that whatever one consciously experiences should be decodable in the PFC. And so it seems that it is not just GNW, but also HO, that makes Prediction #1.

(Worth noting: In some ways, Prediction #1 seems to be a more stringent test of HO than it is of GNW, since HO might be (and often is) construed as requiring activity in just the PFC for consciousness (as long as certain background conditions obtain), whereas GNW takes the PFC to be just one of the members of the constellation of brain areas that GNW says gives rise to consciousness. Because of this, it might be easier for GNW to explain why their Prediction #1 is not met, since the GNW theorist might claim, e.g., that the failure to decode the stimuli orientation in the PFC occurs because such content is actually distributed across the frontoparietal areas, rather than being comprehensively represented in the PFC.)

This is an excellent observation, with which we very much agree. The reason why we have abstained in the original and the revised manuscript from relating our findings to other HO theories is because there are several HOT variants, and only some of those make explicit predictions about a higher-order representation in PFC, while others invoke other mechanisms such as pointers (e.g., Brown, 2015). To avoid mischaracterizing any of those theories, we preferred not to relate our findings to them, especially given the two concurrent adversarial collaborations explicitly testing these theories. However, it is also our hope that when those adversarial collaborations release their preregistrations, we and others in the community will be able to explicitly relate our findings, or even test those preregistered predictions on our openly available dataset.

A similar observation can be made in the case of IIT and LR. According to LR, visual consciousness arises from recurrent loops localized to the visual cortex, ones that are between lower- (V1/V2) and higher- (e.g., LO, IT, V4) level brain areas. These are of course brain areas that are comparable to those that figure into each of the IIT Predictions made in the Article. Given this, we can consider each of the IIT predictions to also be predictions of LR, in which case we can consider the experiments presented by the Authors to confirm and disconfirm LR in precisely the same way that it is taken to confirm and disconfirm IIT.

Again this is an excellent and completely fair point, and we very much agree with the reviewer. Yet under the same rationale as the one applied on HOT, here too we intentionally refrained from referring to Recurrent Processing (RP), both due to space limitations and due to the several variants of RP theory. Some theories, like those suggested by Lamme (Lamme, 2015), emphasize extrinsic connections between early and mid-level visual areas, whereas others, such as the one proposed by Malach (Malach, 2021), argue that local, intrinsic connectivity (horizontal connections) is essential for consciousness. An ongoing adversarial collaboration is currently comparing first-order and higher-order theories and will likely be in a better position to provide evidence challenging these different variants. Therefore, while we concur with the reviewer's perspective, we have chosen to strictly adhere to the theoretical predictions given to us by proponents of GNWT and IIT, without extending our discussion to other theories in the current paper to avoid mischaracterizing those theories, to stay within page limits, to avoid over-complexity, etc. However, this suggestion will be highly valuable for a follow-up study, and we do hope that the field will consider our results under a wider perspective, also taking into account their meaning to other theories.

Notably, this seems to even be true of Prediction 3, which was in some ways much more specific than the first two predictions, since it identified not only those brain areas that should be representing the conscious content, but also the nature of the activity underlying this representation (i.e., synchronized activity between higher- and lower-level brain areas). This seems to fit very well with what we would expect LR to predict. Indeed, one might think that, in the case of experiments that the Researchers ran in this article, LR and IIT are actually

empirically indistinguishable: They seem to both make the same predictions as far as what the results of the experiments should be.

Again, an excellent remark, and a case in point for how complex it is to relate our findings to other theories–

When preparing the revision, we reached out to Victor Lamme to confirm whether RP indeed also predicts sustained synchronization between higher and lower visual areas, measured as phase-locked synchrony. We learned that RPT expects sustained interaction between visual areas as long as the stimulus is consciously visible. However, this sustained interaction is not necessarily expressed as sustained synchrony. This is very different from IIT, for which synchrony is necessary. According to the theory, any content of experience (in our experiment the high-level invariant content face, and low-level features such as the particular contour of the face) should correspond to substructures. These are further bound by relations – overlaps between causes and effects - which in the brain should typically be accompanied by firing synchrony. Other measures that can evaluate connectivity, like for instance dynamical connectivity or psychophysiological interactions are not good proxy measures for IIT, as those are amplitude-based measures.

This discrepancy underscores how theoretical predictions are intertwined with the preferred metrics that may (or may not) directly reflect the underlying mechanisms. Given these complexities, we refrain from linking our findings to RPT in this revision, opting instead to explore this connection in a subsequent piece. Additionally, we are aware that others, including Robert Chris-Ciure and Georg Northoff, are composing a comprehensive comparison of empirical measures within consciousness theories. This work promises to further elucidate how mechanistic accounts relate to specific measures and in turn how those fare between theories. This might help us and others, in the future, to better connect our findings to other theories than the ones tested.

The upshot of all this is that I think that the article could benefit greatly from (a) an acknowledgement that IIT and GNW are not the only leading theories of consciousness, and (b) a substantial discussion as to what the results presented in the Article mean for these other theories of consciousness. As well, I should mention again that the article, as it is, seems to implicitly misrepresent the field of consciousness studies, since these omissions might give the reader the impression that these are the only two theories of consciousness that anyone takes seriously anymore, something that is untrue.

These are both points well taken. Certainly, it was not our intention to portray the field as consisting of just these two theories of consciousness. As stated above, we have modified the introduction to avoid giving that impression (p. 4, second paragraph), as well as in the final conclusion (p. 30, last paragraph), to mention that our results might also relate to other theories to alleviate the impression that we are misrepresenting the field of consciousness studies. We hope that this, and the future publication we are planning where this discussion will take place, mitigate the reviewer's concerns.

2. So far, I have pointed out how I think that the Article, as it currently is, misses out on opportunities to expand our neuroscientific knowledge of consciousness by showing how the results reported in the Article bear on HO and LR. The same point can be made about another

debate, one concerning a more general distinction in types of theory, this being between Local and Broad theories of consciousness (for distinction, see Michel & Doerig 2021). According to Local Theories (this includes LR and IIT), activity in just the visual cortex (given some background conditions) can be sufficient for visual consciousness; according to Broad Theories (this includes GNW and HO), areas outside of the visual cortex (which are usually considered to include the PFC) are necessary (perhaps in addition to areas in the visual cortex). The relevance of the results presented in the Article to this debate is obvious, given the strong similarities to be found between GNW and HO, and especially given the apparent empirical equivalence of IIT and LR (discussed in Point 1).

It would be good if the Article were to comment on what their results mean for the Local vs. Broad debate, not least of all because this debate is very consequential in regards to the issue of where future research efforts should be directed (i.e., whether it is more worthwhile to pursue the development of Local or Broad Theories of consciousness). It should be noted that one of main points of contention between Local and Broad Theories is what role the PFC plays in consciousness, this of course being one of the issues that the Article focuses on as well. Given this, the Article seems well-poised to comment on the Local vs. Broad debate.

We very much agree with the reviewer that our results are not just informative with respect to the two theories directly under scrutiny but also illuminate other discussions in the field. However, as we explained above, we are afraid this is not possible in the limits of this manuscript. However, as we explained, we plan to explore this subject in depth in a forthcoming publication, and look forward to developing these interesting theoretical points.

3. The Authors at one point write:

“When viewing the Mona Lisa (Figure 1c), one experiences it located in visual space, with a specific identity, a specific orientation, and the experience *continues as long as one looks at the painting* (143-45; italics mine).

This gives the reader the impression that all of the authors countenance the following plausible assumption about the nature of consciousness:

[CONTINUOUS] When looking at and focusing on a stimulus, one continues to visually experience it the entire time one is looking at it.

At the same time, one of the predictions that GNW makes in the article is that there will be “brief content-specific ignition in PFC ~0.3-0.5 s after stimulus onset and offset (when the workspace is updated), with content...being stored in a *non-conscious* silent state resembling activity-silent working memory in between” (207, italics mine).

This prediction suggests that the following thesis is true:

[BRIEF] When looking at and focusing on a stimulus, one is *not* conscious of it the entire time one is looking at it, but only when a representation of the stimulus is first introduced into the global workspace (i.e., just when one first starts looking at and focusing on it)

I became puzzled when I got to this part of the Article (around line 207), since what was said earlier in the Article made it seem as if all of the researchers agreed with CONTINUOUS, but now a prediction of GNW was being built around an idea that is in direct conflict with it (as it is based around the thesis of BRIEF).

Something should be added to the article (near line 143) to make clear whether the GNW theorists do or do not agree with taking CONTINUOUS to be one of the starting assumptions that the Authors collectively make.

This point is addressed below.

4. Continuing on the same topic: The thesis of BRIEF seems extremely controversial, because CONTINUOUS is strongly supported by introspection: As the Authors very plausibly state, when one visually experiences a painting, one has the impression that their experience of the painting “continues as long as one looks at the painting” (144-5). The thesis of BRIEF denies this.

Because of this, when GNW Prediction #2 is being offered, some justification for rejecting CONTINUOUS should be given. Later in the article (p. 27), Dehaene does indicate that he is skeptical about the thesis of CONTINUOUS, and gives justification for rejecting it (doing so by appealing to experimental phenomena such as inattention blindness). However, I do think that it would be better if this rejection were made more up-front in the article, so that the reader could have this in mind throughout the article’s presentation of the experiments, rather than being surprised by it at the end.

(I would think that, for the GNW Theorist, this would strategically make more sense, since if CONTINUOUS is rejected after the experimental results have already been unveiled, this rejection might be more easily mistaken as having been done just to save the theory of GNW.)

This point is also addressed below, as it continues the same topic.

5. It should be noted, however, that rejecting BRIEF seems fairly extreme, since it seems to be saying that, when viewing an object in front of our face for several seconds, we are only conscious of it for 50ms or so, and then the rest of the time...what?

Is it that we fail to have any kind of phenomenal experience in the part of the visual field where the object is represented? Or no visual experience at all, in any part of the visual field? If it is not either of these options, then *what is it* that the subject experiences in this interim? In justifying the rejection of CONTINUOUS, Dehaene makes the familiar move of appealing to experimental phenomena such as attentional blink and inattention blindness. But these experiments are typically appealed to in order to argue that our impression that visual experience is rich is mistaken, the alternative hypothesis being that we are only experiencing whatever objects we are attending to; however, what Dehaene is arguing goes beyond this, in that he argues that our impression that visual experience is continuous is wrong.

This difference helps to illuminate what I think is important here, which is that BRIEF seems to be not only a counterintuitive claim, but also a *novel* one, one that goes beyond the usual, already counterintuitive claims that we have heard many times, those about visual experience not containing as much content as it seems.

To me, this means that it would be best if the justification that Dehaene provides for rejecting CONTINUOUS receives supplementation. Some articles by Herzog, Drissi-Daoudi, and Doerig (et al.) might be helpful here, since their work gives some reason for thinking that our impression that visual experience is continuous is mistaken (a good place to start is Herzog et al. 2020).

Here we address all three points raised above. We thank the reviewer for making them. We believe that our phrasing might have evoked some confusion, which we have now amended, and we further clarify this point below.

According to GNWT, once a certain content has been globally broadcast, it is consciously perceived, and it continues to be consciously perceived until a new broadcast occurs. In between, the information can still be stored in a silent state (i.e., it is not ‘lost’) in the

workspace, persisting as long as there is no new broadcast. This silent state also characterizes ‘unconscious working memory’, which is why it was mentioned in the text, but we now understand that this was confusing. Importantly though, in between these two broadcasts, perception persists and stays continuous. Thus, while the underlying brain response (the workspace update) is temporally discrete (i.e., an onset and an offset response), the conscious experience can be temporally continuous (lasting from one workspace update to the next).

Therefore, GNWT’s view differs from Herzog’s view, with the latter suggesting that conscious experience is itself discrete (i.e., temporally sparse in addition to spatially sparse, and the apparent continuousness of experience is illusory) while non-conscious processing can be continuous. Thus, although we understand why the reviewer found this relevant, we believe that GNWT makes a different prediction here. We have changed the wording in the manuscript (p. 13, first paragraph), to avoid this confusion. For convenience the revised paragraph is provided below:

“According to IIT, the state of the network that specifies the content of consciousness in posterior cortex is actively maintained for the duration of the conscious experience (manipulated here via different stimulus durations). In contrast, GNWT predicts brief content-specific ignition in PFC ~0.3-0.5s after stimulus onset, when the workspace is updated. Then, activity decays to baseline, with information being maintained in a latent state, until another ignition marks the offset of the current percept and the onset of a new percept (in our paradigm, the fixation screen following stimulus offset). Thus, while the underlying brain response (the workspace update) is temporally discrete (i.e., an onset and an offset response), the conscious experience can be temporally continuous (lasting from one workspace update to the next).”

Furthermore, in his discussion, Dehaene puts forward the suggestion that the lack of ignition at the offset might indicate a decoupling between the temporal dynamics of stimuli presentation and subjective experience. According to this account, it is possible that participants failed to experience the offset of the visual stimulus due to its low task relevance and instead engaged in mind-wandering. Crucially, this does not necessarily entail that conscious experience is discrete, but rather that when the sensory stimulation is irrelevant, its conscious experience might last shorter than its presentation time, as other content (e.g., mind-wandering) captures one’s conscious experience. This will lead to a difficulty to find an offset response, due to jitter in time (since the percept will end at different times in each trial).

6. The stimuli were divided into three categories: target, task-relevant, and task-irrelevant. The motivation for having these three categories was to enable the researchers to subtract the neural activity created by task demands in order to zero in on just the neural activity that is actually constitutive of visual consciousness.

However, the subjects having to ignore the task-irrelevant stimuli still creates task demands. For example, they have to attend to it to determine that it is task irrelevant. This seems to indicate that we cannot take the neural activity that was recorded during presentation of task-irrelevant stimuli to reflect only those processes that are responsible for their consciousness of the stimulus. That the authors are aware of this is implied by the way that they couch some of their claims, e.g., they write, “We focused on the task irrelevant condition as it is **most** diagnostic for neural activity related to consciousness, **minimizing** the contribution of other, potentially confounding, cognitive processes” (408; italics mine). The Authors do not,

however, comment further on the fact that there are task demands even when subjects are viewing the task irrelevant stimuli.

We thank the reviewer for making this point, which resonates with Comment 9 by Reviewer 1 (please see our reply there). As we wrote there, our experiment consisted of a 3x3x4x2 design. The factors were task relevance, stimulus duration, stimulus category, and stimulus orientation. We opted for an experimental design whereby one stimulus feature (i.e., category) can still be contaminated by a minimal degree of task engagement (even when task irrelevant), while another feature (i.e., orientation) is always completely task irrelevant (this also applied to duration). Notably, this was done precisely due to the concern raised by the reviewer, according to which making a feature task relevant on some trials and not in others, can turn a condition that was presumed 'irrelevant' into more attention grabbing. The inclusion of these completely irrelevant features, then, sets the bar higher for the theories, as any effects found there cannot be explained by selective attention being directed at this feature due to – even minimal – task demands. Yet to clarify that the task irrelevant condition is not free of task demands, in line with the reviewer's comment, we have updated the discussion section in the revised manuscript to make this limitation clearer (p. 25, third paragraph). Below is the relevant addition:

“Our study, while comprehensive, is not without its limitations. Firstly, despite our best efforts to minimize the contribution of task relevance by making some stimulus features relevant on some trials and irrelevant in others, we cannot rule out some residual task engagement with respect to category. However, this potential bias is addressed by our deliberate choice to make features like orientation and duration always irrelevant to the task. This approach strengthens the test for the theories we are examining, as any detected effects on these features cannot be attributed to selective attention driven by task requirements, no matter how minimal.”

7. The Authors' goal to carry out experiments that looked for correlations between neural activity and *multiple* dimensions of conscious experience (e.g., category and orientation) at the same time is highly admirable, and the result of their focusing on this is that we get some highly informative data concerning how stimuli are consciously represented in the brain.

At the same time, it should be noted that their efforts are still very far off from measuring consciousness in a way that truly captures its apparent phenomenal richness: The typical visual experience does not consist merely of the representation of the category to which the object belongs, along with the orientation from which one views it, but so much more: One experiences the object's brightness and hue, its precise shape and location, and experiences the object from a highly specific viewpoint (something more fine-grained than a broad description such as “from the front” or “from the side”); one will also experience the hue and brightness of the object's background. (The list could continue here, but I'll stop so as to not court controversy...)

It should be noted, as well, that there is an active debate concerning whether object category is something that is even part of our conscious experience (qua *phenomenal* experience) (for review: Siegel & Byrne 2017). This means that finding a neural correlate of category representation might not even qualify as having found a neural correlate of phenomenal consciousness. As well, it is plausible that a representation of the category to which an object

belongs is amodal (especially by the time it reaches the PFC), in which case its neural correlate would also not count as a neural correlate of *visual* consciousness. (On the other hand, something that might more clearly qualify as a neural correlate of visual phenomenal experience would be the neural correlates of one's experience of an object's color, or of its precise location in the visual field.)

This is not to say the Authors have been negligent here (indeed, they do a much better job of trying to experimentally capture the phenomenal richness of visual experience than has been done so far). Rather, I just think that this limitation of the Authors' studies should be acknowledged in the Article. As well, perhaps the Authors could make some suggestions as to how these limitations could be overcome in the future.

We agree with the reviewer regarding the importance of studying phenomenal richness of a visual experience and designing paradigms capable of capturing such richness. Following this comment, we now acknowledge that our attempt to do so is only partial, and hope this will be further pursued in future studies in the field (though, due to space limitations, we sufficed with saying that it should be done in future studies, without being able to suggest how this should be done). We accordingly added this limitation to the discussion section (p. 25, third paragraph), which now reads:

"...Secondly, although we made our best efforts to capture the richness of experience by investigating multiple dimensions of conscious experience (i.e., category, orientation, identity and duration), we acknowledge that our efforts are still far from measuring consciousness in a way that truly captures its apparent phenomenal richness. Future studies will be needed to address that further."

8. Much to the credit of the project, the Authors take a falsificationist approach to evaluating GNW and IIT. For this reason, they intentionally chose to use fully visible and attended stimuli; the idea here was that, if the predictions of one of the theories could not be met—even under these circumstances—it would be particularly damaging to the theory.

However, in the falsificationist philosophy of science, results are more to the credit of a theory if they are based on predictions that not only turn to be correct, but were also *risky* predictions for the theory to make. But there appears to be a fairly pronounced asymmetry in the riskiness of the predictions that the two theories make in the Article, especially in the case of the first two predictions. As Dehaene points out in his analysis of the results, the predictions concerning posterior areas are ones that "any physiologist familiar with the bottom-up response properties of those regions would predict" (911). This is in marked contrast to the predictions made by the GNW Theorists: The prediction that conscious content would be represented in the PFC, and that it would have that particular and idiosyncratic time course of activation, is far from what is routinely assumed to be the case by the average neuroscientist.

All this adds up to the idea that IIT being confirmed in the way that it is in the first two predictions should probably not be considered to be an especially strong confirmation of the theory. This should be made clear in the article.

It is good that Dehaene has a chance to point out something like this during his defense of GNW. This idea is important enough, however, that it should probably be given a more prominent place in the article. The reader should be made aware of this at the time that the

predictions are being made, and I think that it would be fair if this asymmetry were acknowledged by all of the Authors, rather than being presented just as the view of one of GNW's proponents.

The second round of experiments that Cogitate plans to perform, the ones in which conscious and unconscious conditions are to be contrasted, will probably hold much more risk for IIT (since these experiments might show stimuli to be represented in posterior brain areas without becoming conscious). The article should point out that the true test of IIT is yet to come.

*Please see response to reviewer #3 who raised a similar point. To reiterate, when it comes to testing theories at the **neural** level, we believe that both theories were being strongly tested, and that both provided risky predictions. For both theories there were plausible alternative outcomes, as well as prior evidence indicating that the predictions might not be borne out by the data. In fact, predictions #2 and #3 by IIT are far from being trivial. First, based on previous electrophysiological findings, prediction 2 (posterior activation associated with conscious perception will be sustained) was unlikely due to known, and very strong, stimulus-specific adaptation effects which clamp responses in the posterior cortex (e.g., Muller et al., 1999), which made the sustained responses predicted by IIT uncommon (something IIT proponents touch upon in their respective discussion section). A much more prevalent and common pattern of responses in early visual cortex is a transient onset and offset response (or a single transient response followed by stimulus specific adaptation) (McLelland et al., 2010). The results indeed indicate that sustained responses are considerably less common in posterior cortex, with only 25 out of 657 electrodes (approximately 3.8% of the total dataset) exhibiting such responses, in contrast to transient responses. If this has been the typical, already known, pattern of response in posterior areas, we would have expected to find it in a much more robust and spread manner, which was not the case. We now emphasize this better in the manuscript (p. 15, seventh paragraph):*

“Considering the primary preregistered tests, their respective weight and interpretations for both theories (Extended Table 1), for prediction #2, results were in line with IIT’s prediction as activation and representation of conscious content was sustained in posterior cortex, including representation of category and identity across multiple stimuli. Yet, sustained responses were rather rare in posterior cortex (found only in 3.8% of the electrodes in the iEEG data). Also, there was no sustained representation of orientation”.

IIT also predicted that content-specific information should be sustained, another non-trivial prediction considering adaptation effects and also the effects of eye movements on areas with small receptive fields (Kravitz et al., 2013).

Second, as we explained in our replies above, not all of IIT’s predictions are shared by other theories. For example, Prediction 3 (sustained connectivity between high-level and low-level visual areas) is especially distinctive to IIT and is not shared by RPT. In fact, IIT's prediction 2 (at least in the case of orientation) and prediction 3 were both falsified, substantially challenging IIT as a result. The fact that these predictions were falsified suggests that they were not overly weak or devoid of risks. In retrospect, we realize that our failure to specify that the first prediction by IIT is not a key test for the theory, also because to it being less risky, was a mistake. It probably primed readers to think that all of IIT’s predictions were not risky enough, which was not the case. For the revision, we now make explicit that prediction #1a, #2a and #3 are critical for GNWT, while prediction #2a, #2b and #3 are critical for IIT.

Also, as we stated in our reply to Reviewer 3's similar point, we think that one of the merits of our work lies in exposing more specific predictions by the theories, allowing the field to evaluate their informativeness and riskiness. We now explicitly state that in the manuscript (p. 5, second paragraph; p. 24, first paragraph). Thus, even if the predictions are not symmetrical, this is yet another measure by which the theories can be evaluated following this work.

Finally, with respect to the discussion section of both GNWT and IIT, as a consortium we made a point not to censor, or significantly edit any of the discussion sections. This does not imply that we concur with the suggested interpretations nor that we believe the exposition of the literature is balanced. Instead, we present the readership with the data and uncensored reactions from the theory proponents, with the intention that readers themselves can interpret the evidence and can evaluate the responses, including any omissions and selective exposition of the data and background literature. In the revised version we make this more explicit (p. 30, last paragraph). The relevant section, now reads:

“At this point, the reader might expect the consortium to draw a final conclusion, or to relate the findings to other theories of consciousness, which without doubt put pressure on other theories i.e., some variants of higher-order (Kozuch, 2023) and recurrent processing (Lamme, 2015; Malach, 2021). Instead, we invite the reader to form their own conclusions considering the relative evidence we presented for each of the preregistered predictions, the scope of the evidence with > 250 subjects using the most sophisticated techniques available to human neuroscience, the hindsight bias, and the challenges in changing people's minds including those of the theory proponents but also the reader's own mind. Science is a social enterprise - evidence is interpreted based on prior beliefs and expectations. As such, the reader is as much a part of this social enterprise as any of the authors from this consortium who have made their best attempt to present the evidence, and the adversaries' reactions, as unprocessed and uncensored as possible, even when disagreeing (Clark et al., 2023). Science needs openness to collectively converge to true explanations of complex phenomena in nature such as consciousness.”

9. In the article, IIT is taken to be disconfirmed by the lack of sustained synchronization between face/object selective regions and V1/V2, this being one of the predictions that the IIT Theorists made. However, in the case of features such as object category or face identity, is it not possible that the relevant synchronization would be between face/object selective regions and *mid-level* areas such as V4, V5, and LO? Indeed, given that the properties that these mid-level regions represent (color, motion, and shape) are the properties that the brain would use to *identify* an object or face, perhaps hypothesizing the relevant type of synchronization to be with these mid-level brain areas, instead of lower-level brain areas, makes much more sense. This possibility should be considered in the Article.

We thank the reviewer for raising this point. Prediction#3 was hashed out with IIT proponents who posited that for IIT, any content of experience should correspond to substructures. In our study, this is applied on the high-level invariant face content, and the low-level features such as the particular contour of the face (thought to be represented in early visual areas i.e., V1/V2). As such, to our understanding of IIT, connectivity to activated areas within low-level visual cortices (i.e., V1/V2) is necessitated (and predicted) by the theory, setting it apart from predictions made by alternative theories. We agree that connectivity to mid-level regions

(e.g., V4) might be easier to obtain, but as this is a lower bar to obtain, the results will be less informative in testing IIT's prediction regarding interareal synchronization (and might raise more concerns about asymmetrical predictions, as mentioned above). As such, we prefer to keep the connectivity analysis to V1/V2, as it is both more informative for testing IIT and also unique with respect to other RP theories.

10. When discussing the failure to find the synchronization predicted by IIT (the one discussed in the last point), the IIT Theorists say (at 836) that the results “do not support prediction #3 concerning sustained synchrony, although there are potential explanations (see supplementary).”

It would be most helpful if the Authors pointed the reader to a much more specific place than this (e.g., a page or section number).

There are other places where the Article would benefit from similar cross-referencing. Given the large size of the document, I picture this as being very helpful to the reader as they try to navigate it.

We agree with the reviewer, and this comment has also been made by Reviewer 1. We have reformatted the supplementary to contain different sections which are cross-referenced more explicitly in the manuscript. For the IIT discussion, the cross-referencing reads:

“They do not support prediction #3 concerning sustained synchrony, although there are potential explanations (see supplementary section 9).”

11. Authors write (91): “Philosophers and scientists have sought to explain the subjective nature of consciousness (e.g., the feeling of pain or of seeing a colorful rainbow) and how it relates to physical processes in the brain. This ‘explanatory gap’ or ‘hard problem’ has led to several competing theories of consciousness that have evolved in parallel.”

This thought is expressed unclearly. The explanatory gap/hard problem is explaining *how* and *why* subjectivity/consciousness arises from physical processes, not simply “how it relates to physical processes in the brain.” The latter phrase could be construed as being satisfied by merely finding correlations between mental and neural activity, rather than actually “bridging the gap” of the hard problem.

We agree with the reviewer that this statement was not clear, a remark that also resonates with a comment made by reviewer #2. We have accordingly revised this paragraph of the introduction (p. 4, first paragraph) to:

“Philosophers and scientists have sought to explain the subjective nature of consciousness (e.g., the feeling of pain or of seeing a colorful rainbow) and how it relates to physical processes in the brain (Chalmers, 2000; Crick & Koch). This ongoing endeavor has led to a number of competing theories of consciousness that have evolved in parallel (A. K. Seth & T. Bayne, 2022; Signorelli et al., 2021; Yaron et al., 2021). Yet, those theories offer incompatible accounts of the neural basis of consciousness (A. K. Seth & T. Bayne, 2022; Signorelli et al., 2021)”.

12. Authors write (129): “GNW predicts brief content-specific ignition in PFC ~0.3-0.5 s after stimulus onset and offset (when the workspace is updated), with content...being stored in a *non-conscious silent state* resembling activity-silent working memory in between.”

To my knowledge, this is a novel prediction of GNW, one that did not appear until the article that was a precursor to the article that I am now reviewing (i.e., Melloni et al 2023). I believe that many readers will be surprised by it, especially given its counterintuitive nature (see Points 3 through 5 above).

This remark was also made by Reviewer 1 comment# 4. We refer the reviewer to the response there, where we also provided the background for that prediction and explained its rationale. Notably, in the manuscript we now also refer to the 2023 paper where the rationale is described in greater detail (which was not possible here due to space limitations).

13. I do not understand why, at line 423, the word “serendipitous” is being used. Is it that some kind of “happy accident” came from how the electrodes were placed? Or do the Authors just mean “stochastic” or “fortuitous”? Please clarify.

This comment was also made by Reviewer 2 (comment# 8); we agree and have changed the wording (p. 13, fifth paragraph) to:

“Although we lacked control over the placement of electrodes, the sampling density of electrodes in both the posterior cortex and the prefrontal cortex (PFC) was consistently high and evenly distributed across ROIs pertinent to the theories”.

References

- Aguillon-Rodriguez, V., Angelaki, D., Bayer, H., Bonacchi, N., Carandini, M., Cazettes, F., Chapuis, G., Churchland, A. K., Dan, Y., Dewitt, E., Faulkner, M., Forrest, H., Haetzel, L., Häusser, M., Hofer, S. B., Hu, F., Khanal, A., Krasniak, C., Laranjeira, I., Mainen, Z. F., Meijer, G., Miska, N. J., Mrsic-Flogel, T. D., Murakami, M., Noel, J.-P., Pan-Vazquez, A., Rossant, C., Sanders, J., Socha, K., Terry, R., Urai, A. E., Vergara, H., Wells, M., Wilson, C. J., Witten, I. B., Wool, L. E., & Zador, A. M. (2021). Standardized and reproducible measurement of decision-making in mice. *Elife*, *10*. <https://doi.org/10.7554/eLife.63711>
- Albantakis, L., Barbosa, L., Findlay, G., Grasso, M., Haun, A., Marshall, W., Mayner, W., Zaeemzadeh, A., Boly, M., Juel, B., Sasai, S., Fujii, K., David, I., Hendren, J., Lang, J., & Tononi, G. (2022). *Integrated information theory (IIT) 4.0: Formulating the properties of phenomenal existence in physical terms*. <https://doi.org/10.48550/arXiv.2212.14787>
- Baars, B. J. (1988). *A Cognitive Theory of Consciousness*. Cambridge University Press. <https://doi.org/10.1017/cbo9780511816789.009>
- Bellet, J., Gay, M., Dwarakanath, A., Jarraya, B., van Kerkoerle, T., Dehaene, S., & Panagiotaropoulos, T. I. (2022). Decoding rapidly presented visual stimuli from prefrontal ensembles without report nor post-perceptual processing. *Neuroscience of Consciousness*, *2022*(1). <https://doi.org/10.1093/nc/niac005>
- Boly, M., Sasai, S., Gosseries, O., Oizumi, M., Casali, A. G., Massimini, M., & Tononi, G. (2015). Stimulus set meaningfulness and neurophysiological differentiation: a functional magnetic resonance imaging study. *PLoS ONE*, *10*(5), e0125337.
- Brand, A., Allen, L., Altman, M., Hlava, M., & Scott, J. (2015). Beyond authorship: attribution, contribution, collaboration, and credit. *Learned Publishing*, *28*(2), 151-155. <https://doi.org/10.1087/20150211>
- Brodsky-Dvir, R., Norman, Y., Harel, M., Mehta, A. D., & Malach, R. (2023). Perceptual stability reflected in neuronal pattern similarities in human visual cortex. *Cell Reports*, *42*(6). <https://doi.org/10.1016/j.celrep.2023.112614>
- Brown, R. (2015). The HOROR theory of phenomenal consciousness. *Philosophical Studies*, *172*, 1783-1794.
- Cardin, J. A., Carlén, M., Meletis, K., Knoblich, U., Zhang, F., Deisseroth, K., Tsai, L. H., & Moore, C. I. (2009). Driving fast-spiking cells induces gamma rhythm and controls sensory responses. *Nature*, *459*(7247), 663-667.
- Chalmers, D. (2000). What Is a Neural Correlate of Consciousness? In *Neural Correlates of Consciousness*. The MIT Press. <https://doi.org/10.7551/mitpress/4928.003.0004>

- Chen, H., & Wyble, B. (2015). Amnesia for object attributes: Failure to report attended information that had just reached conscious awareness. *Psychological science*, 26(2), 203-210.
- Cohen, M. A., Ortego, K., Kyroudis, A., & Pitts, M. (2020). Distinguishing the neural correlates of perceptual awareness and postperceptual processing. *Journal of Neuroscience*, 40(25), 4925-4935.
- Crick, F., & Koch, C. (1990). Towards a neurobiological theory of consciousness. *Seminars in the Neurosciences*,
- Dehaene, S. (2001). Towards a cognitive neuroscience of consciousness: basic evidence and a workspace framework. *Cognition*, 79(1-2), 1-37. [https://doi.org/10.1016/s0010-0277\(00\)00123-2](https://doi.org/10.1016/s0010-0277(00)00123-2)
- Dehaene, S., & Changeux, J.-P. (2011). Experimental and Theoretical Approaches to Conscious Processing. *Neuron*, 70(2), 200-227. <https://doi.org/10.1016/j.neuron.2011.03.018>
- Dehaene, S., Kerszberg, M., & Changeux, J.-P. (1998). A neuronal model of a global workspace in effortful cognitive tasks. *Proceedings of the National Academy of Sciences*, 95(24), 14529-14534. <https://doi.org/10.1073/pnas.95.24.14529>
- Desikan, R. S., Ségonne, F., Fischl, B., Quinn, B. T., Dickerson, B. C., Blacker, D., Buckner, R. L., Dale, A. M., Maguire, R. P., & Hyman, B. T. (2006). An automated labeling system for subdividing the human cerebral cortex on MRI scans into gyral based regions of interest. *Neuroimage*, 31(3), 968-980.
- Destrieux, C., Fischl, B., Dale, A., & Halgren, E. (2010, 2010/10//). Automatic parcellation of human cortical gyri and sulci using standard anatomical nomenclature. *NeuroImage*, 53(1), 1-15. <https://doi.org/10.1016/j.neuroimage.2010.06.010>
- Epstein, R. A., Patai, E. Z., Julian, J. B., & Spiers, H. J. (2017). The cognitive map in humans: spatial navigation and beyond. *Nature neuroscience*, 20(11), 1504-1513.
- Farooqui, A. A., & Manly, T. (2018). When attended and conscious perception deactivates fronto-parietal regions. *Cortex*, 107, 166-179. <https://doi.org/10.1016/j.cortex.2017.09.004>
- Gerber, E. M., Golan, T., Knight, R. T., & Deouell, L. Y. (2017). Cortical representation of persistent visual stimuli. *NeuroImage*, 161, 67-79. <https://doi.org/10.1016/j.neuroimage.2017.08.028>
- Haber, S. N., Liu, H., Seidlitz, J., & Bullmore, E. (2022, Jan). Prefrontal connectomics: from anatomy to human imaging. *Neuropsychopharmacology*, 47(1), 20-40. <https://doi.org/10.1038/s41386-021-01156-6>

- Haegens, S., Nácher, V., Luna, R., Romo, R., & Jensen, O. (2011). α -Oscillations in the monkey sensorimotor network influence discrimination performance by rhythmical inhibition of neuronal spiking. *Proceedings of the National Academy of Sciences*, *108*(48), 19377-19382. <https://doi.org/10.1073/pnas.1117190108>
- Haun, A., & Tononi, G. (2019). Why Does Space Feel the Way it Does? Towards a Principled Account of Spatial Experience. *Entropy*, *21*(12). <https://doi.org/10.3390/e21121160>
- Haynes, J.-D. (2009). Decoding visual consciousness from human brain signals. *Trends in Cognitive Sciences*, *13*(5), 194-202. <https://doi.org/10.1016/j.tics.2009.02.004>
- Iemi, L., Gwilliams, L., Samaha, J., Auksztulewicz, R., Cycowicz, Y. M., King, J.-R., Nikulin, V. V., Thesen, T., Doyle, W., Devinsky, O., Schroeder, C. E., Melloni, L., & Haegens, S. (2022). Ongoing neural oscillations influence behavior and sensory representations by suppressing neuronal excitability. *NeuroImage*, *247*. <https://doi.org/10.1016/j.neuroimage.2021.118746>
- Jacobs, J., Korolev, I. O., Caplan, J. B., Ekstrom, A. D., Litt, B., Baltuch, G., Fried, I., Schulze-Bonhage, A., Madsen, J. R., & Kahana, M. J. (2010). Right-lateralized brain oscillations in human spatial navigation. *Journal of cognitive neuroscience*, *22*(5), 824-836.
- King, J.-R., Pescetelli, N., & Dehaene, S. (2016). Brain Mechanisms Underlying the Brief Maintenance of Seen and Unseen Sensory Information. *Neuron*, *92*(5), 1122-1134. <https://doi.org/10.1016/j.neuron.2016.10.051>
- Koch, C., Massimini, M., Boly, M., & Tononi, G. (2016). Neural correlates of consciousness: progress and problems. *Nature Reviews Neuroscience*, *17*(5), 307-321. <https://doi.org/10.1038/nrn.2016.22>
- Kozuch, B. (2023). A Legion of Lesions: The Neuroscientific Rout of Higher-Order Thought Theory. *Erkenntnis*. <https://doi.org/10.1007/s10670-023-00669-4>
- Kravitz, D. J., Saleem, K. S., Baker, C. I., Ungerleider, L. G., & Mishkin, M. (2013). The ventral visual pathway: an expanded neural framework for the processing of object quality. *Trends in cognitive sciences*, *17*(1), 26-49.
- Kumral, E., Bayam, F. E., & Özdemir, H., N. (2021). Cognitive and behavioral disorders in patients with precuneal infarcts. *European Neurology*, *84*(3), 157-167.
- Lakatos, I. (1976). Falsification and the Methodology of Scientific Research Programmes. In S. G. Harding (Ed.), *Can Theories be Refuted? Essays on the Duhem-Quine Thesis* (pp. 205-259). Springer Netherlands. https://doi.org/10.1007/978-94-010-1863-0_14

- Lakens, D. (2022). Sample size justification. *Collabra: Psychology*, 8(1), 33267.
- Lamme, V. (2015). The Crack of Dawn: Perceptual Functions and Neural Mechanisms that Mark the Transition from Unconscious Processing to Conscious Vision. In T. K. Metzinger & J. M. Windt (Eds.), *Open MIND*. MIND Group. <https://doi.org/10.15502/9783958570092>
- Lau, H., & Rosenthal, D. (2011). Empirical support for higher-order theories of conscious awareness. *Trends in cognitive sciences*, 15(8), 365-373.
- Lund, J. S., Yoshioka, T., & Levitt, J. B. (1993, Mar-Apr). Comparison of intrinsic connectivity in different areas of macaque monkey cerebral cortex. *Cerebral Cortex*, 3(2), 148-162. <https://doi.org/10.1093/cercor/3.2.148>
- Malach, R. (2021). Local neuronal relational structures underlying the contents of human conscious experience. *Neuroscience of Consciousness*, 2021(2). <https://doi.org/10.1093/nc/niab028>
- Mashour, G. A., & Alkire, M. T. (2013). Consciousness, anesthesia, and the thalamocortical system. *The Journal of the American Society of Anesthesiologists*, 118(1), 13-15.
- Mashour, G. A., Roelfsema, P., Changeux, J.-P., & Dehaene, S. (2020). Conscious Processing and the Global Neuronal Workspace Hypothesis. *Neuron*, 105(5), 776-798. <https://doi.org/10.1016/j.neuron.2020.01.026>
- McLelland, D., Baker, P. M., Ahmed, B., & Bair, W. (2010). Neuronal responses during and after the presentation of static visual stimuli in macaque primary visual cortex. *Journal of Neuroscience*, 30(38), 12619-12631.
- Mellers, B., Hertwig, R., & Kahneman, D. (2001, Jul). Do frequency representations eliminate conjunction effects? An exercise in adversarial collaboration. *Psychol Sci*, 12(4), 269-275. <https://doi.org/10.1111/1467-9280.00350>
- Melloni, L., Mudrik, L., Pitts, M., Bendtz, K., Ferrante, O., Gorska, U., Hirschhorn, R., Khalaf, A., Kozma, C., Lepauvre, A., Liu, L., Mazumder, D., Richter, D., Zhou, H., Blumenfeld, H., Boly, M., Chalmers, D. J., Devore, S., Fallon, F., de Lange, F. P., Jensen, O., Kreiman, G., Luo, H., Panagiotaropoulos, T. I., Dehaene, S., Koch, C., & Tononi, G. (2023). An adversarial collaboration protocol for testing contrasting predictions of global neuronal workspace and integrated information theory. *PLoS One*, 18(2). <https://doi.org/10.1371/journal.pone.0268577>
- Melloni, L., Mudrik, L., Pitts, M., & Koch, C. (2021, May 28). Making the hard problem of consciousness easier. *Science*, 372(6545), 911-912. <https://doi.org/10.1126/science.abj3259>
- Muller, J. R., Metha, A. B., Krauskopf, J., & Lennie, P. (1999). Rapid adaptation in visual cortex to the structure of images. *Science*, 285(5432), 1405-1408.

- Naccache, L. (2018). Why and how access consciousness can account for phenomenal consciousness. *Philosophical Transactions of the Royal Society B: Biological Sciences*, 373(1755), 20170357.
- Oh, S. W., Harris, J. A., Ng, L., Winslow, B., Cain, N., Mihalas, S., Wang, Q., Lau, C., Kuan, L., Henry, A. M., Mortrud, M. T., Ouellette, B., Nguyen, T. N., Sorensen, S. A., Slaughterbeck, C. R., Wakeman, W., Li, Y., Feng, D., Ho, A., Nicholas, E., Hirokawa, K. E., Bohn, P., Joines, K. M., Peng, H., Hawrylycz, M. J., Phillips, J. W., Hohmann, J. G., Wahnoutka, P., Gerfen, C. R., Koch, C., Bernard, A., Dang, C., Jones, A. R., & Zeng, H. (2014, Apr 10). A mesoscale connectome of the mouse brain. *Nature*, 508(7495), 207-214. <https://doi.org/10.1038/nature13186>
- Podvalny, E., Yeagle, E., Mégevand, P., Sarid, N., Harel, M., Chechik, G., Mehta, A. D., & Malach, R. (2017). Invariant Temporal Dynamics Underlie Perceptual Stability in Human Visual Cortex. *Current Biology*, 27(2), 155-165. <https://doi.org/10.1016/j.cub.2016.11.024>
- Popper, K. (1935). *The Logic of Scientific Discovery*. Routledge. <https://doi.org/10.4324/9780203994627>
- Prinz, J. (2017). The intermediate level theory of consciousness. *The Blackwell companion to consciousness*, 257-271.
- Sestieri, C., Shulman, G. L., & Corbetta, M. (2017). The contribution of the human posterior parietal cortex to episodic memory. *Nature Reviews Neuroscience*, 18(3), 183-192.
- Seth, A., & Bayne, T. (2022). Theories of consciousness. *Nature Reviews Neuroscience*, 1-14.
- Seth, A. K., & Bayne, T. (2022, 2022/05/03). Theories of consciousness. *Nature Reviews Neuroscience*. <https://doi.org/10.1038/s41583-022-00587-4>
- Signorelli, C. M., Szczotka, J., & Prentner, R. (2021). Explanatory profiles of models of consciousness - towards a systematic classification. *Neuroscience of Consciousness*, 2021(2). <https://doi.org/10.1093/nc/niab021>
- Stigliani, A., Jeska, B., & Grill-Spector, K. (2017). Encoding model of temporal processing in human visual cortex. *Proceedings of the National Academy of Sciences*, 114(51). <https://doi.org/10.1073/pnas.1704877114>
- Tononi, G. (2012, Jun-Sep). Integrated information theory of consciousness: an updated account. *Arch Ital Biol*, 150(2-3), 56-90. <https://doi.org/10.4449/aib.v149i5.1388>
- Tononi, G., Boly, M., Massimini, M., & Koch, C. (2016). Integrated information theory: from consciousness to its physical substrate. *Nature Reviews Neuroscience*, 17(7), 450-461. <https://doi.org/10.1038/nrn.2016.44>

- Vishne, G., Gerber, E. M., Knight, R. T., & Deouell, L. Y. (2023). Distinct ventral stream and prefrontal cortex representational dynamics during sustained conscious visual perception. *Cell Reports*, 42(7). <https://doi.org/10.1016/j.celrep.2023.112752>
- Wang, R., Fu, Y., Chen, L., Chen, Y., Zhou, J., & Chen, H. (2021). Consciousness can overflow report: Novel evidence from attribute amnesia of a single stimulus. *Consciousness and cognition*, 87, 103052.
- Watakabe, A., Skibbe, H., Nakae, K., Abe, H., Ichinohe, N., Rachmadi, M. F., Wang, J., Takaji, M., Mizukami, H., Woodward, A., Gong, R., Hata, J., Van Essen, D. C., Okano, H., Ishii, S., & Yamamori, T. (2023). Local and long-distance organization of prefrontal cortex circuits in the marmoset brain. *Neuron*. <https://doi.org/10.1016/j.neuron.2023.04.028>
- Yaron, I., Melloni, L., Pitts, M., & Mudrik, L. (2021). The Consciousness Theories Studies (ConTraSt) database: analyzing and comparing empirical studies of consciousness theories. *bioRxiv*, 2021.2006.2010.447863. <https://doi.org/10.1101/2021.06.10.447863>
- Yaron, I., Melloni, L., Pitts, M., & Mudrik, L. (2022). The ConTraSt database for analysing and comparing empirical studies of consciousness theories. *Nature Human Behaviour*, 6(4), 593-604. <https://doi.org/https://doi.org/10.1038/s41562-021-01284-5>
- Zhou, H., Davidson, M., Kok, P., McCurdy, L. Y., de Lange, F. P., Lau, H., & Sandberg, K. (2020). Spatiotemporal dynamics of brightness coding in human visual cortex revealed by the temporal context effect. *NeuroImage*, 205. <https://doi.org/10.1016/j.neuroimage.2019.116277>

Response to the reviewers

We want to thank all reviewers for their positive appraisal of our work, their insightful comments, and above all, for diligently helping us improve our work by reviewing the manuscript and the extensive supplementary material. We are deeply indebted to you for your time, effort, and thoughtful comments. To ease readability, we provide our answers to the reviewers' comments in italics and blue. Changes to the manuscript follow the same convention.

Referee #2 (Remarks to the Author):

The authors have adequately responded to my comments from round one of the review.

While I was positive then, the manuscript is significantly better now. Especially, the clarification of ROI's in response to reviewer 1 is an important addition. I still recommend publication of this manuscript.

We thank the reviewer for the continuous support during the evaluation of our work and for the excellent feedback.

Here are three minor notes:

1) I am still a little disappointed by the conclusion but accept the authors' arguments for it being the way it is, and look forward to reading the other paper they are preparing, and hope to see some of my comments addressed there (no pressure intended though).

At the request of multiple reviewers and editors, we have expanded the conclusions (within the space constraints), indicating that our results have broader implications for other theories beyond just GWT and IIT. The current paragraph reads as follows:

“Although this study was designed around IIT and GNWT, the results may have implications for other theories of consciousness. For example, GNWT's prediction #1 about PFC is shared by some (but not all) higher-order theories of consciousness that also give a central role for PFC⁵⁵. As a result, the challenges to this prediction challenge not only GNWT but also those higher-order theories. Predictions #2 and #3 about timing and connectivity are more distinctive to GNWT but could also be shared by other theories in principle. Likewise, IIT's prediction #1 about posterior cortex is also shared by many other theories (e.g., recurrent processing theory), and its prediction #2 about timing may be shared by some posterior theories of consciousness, such as the local recurrency theory⁵⁶. Its challenged prediction #3 about interareal connectivity is perhaps more distinctive to IIT (e.g., it is not shared by synchrony theory, so the challenge here is more specific as well.

All this highlights that our adversarial collaboration is designed more to challenge theories than to confirm them. Both theories have some predictions confirmed, but these predictions are also consistent with other theories, so the successful predictions cannot serve as evidence for IIT or GNWT specifically. However, the disconfirmed predictions are certainly challenges to both theories (and to others, as discussed above). These challenges can be met by altering the theories or their proposed biological implementation, but such alteration typically comes at some cost to the theoretical framework, because the relevant features of the theory or the implementation were motivated by the framework. In this respect, our adversarial collaboration approach subscribes to the approach advocated by Lakatos⁴⁸, a sophisticated version of Popper's falsificationism⁵⁴, whereby scientific

knowledge advances through a process of conjectures and refutations. When a theory makes an unsuccessful prediction, the challenged theory can survive by refining its details. But if unsuccessful predictions continue, the theory can be deemed a degenerate rather than a progressive research program⁴⁹. This process is expected to be continued by the results of our second experiment (reported in a future manuscript), alongside those of a follow-up adversarial collaboration using a comparable experimental design in animal models (i.e., mice and non-human primates). With time, we hope that substantial evidence will be gathered, allowing the scientific community to form an informed judgment about both theories and possibly others (through the open data). This might be important, as some have proposed a theory-inspired approach to inferring consciousness in non-responsive populations such as unresponsive patients, infants, non-human animals and artificial systems⁵⁷⁻⁵⁹. “

2) in reply to my point 7 about prediction #1 not being mutually exclusive (rebuttal page 22) the authors reformulated the prediction to say

““Prediction #1: According to IIT, PFC is not necessary for consciousness and therefore decoding of conscious content should be maximal from posterior cortex. According to GNW, PFC is necessary for consciousness and therefore every content of consciousness should be decodable from the PFC”.”

The conceptual pedantic in me still reacts to this formulation, albeit for different reasons than before.

Reason one: there is no entailment relation between ‘necessity’ and ‘decodability’, making the use of “therefore” unwarranted.

Reason two: the logical structure (suggested by the use of ‘therefore’) of the IIT part isn’t right given that the premise says nothing about the conclusion (the premise is “X is not necessary”, the conclusion is “Y”).

Reason three: the mechanisms of ‘extrinsic’ theories (i.e. HOT and GWT, that require something other than the content itself to be involved in the coming about of conscious content) can be conceived of as ‘content neutral’ in the sense that whatever generates consciousness is ignorant of the content of the states it processes (Kirkeby-Hinrup, Asger. "Two caveats to the meta-problem challenge." *Journal of Consciousness Studies* 27.5-6 (2020): 74-81.) This would mean that ‘content’ would not be decodable from the mechanism (i.e. PFC on GNW).

Importantly, none of this is critical to the paper. I would recommend just ignoring reason 3. In reply to reasons 1 and 2, I recommend rewording this to exclude ‘therefore’. Something along the lines of

““Prediction #1: According to IIT, PFC is not necessary for consciousness. Consequently, proponents of IIT predict decoding of conscious content should be maximal from posterior cortex. According to GNW, PFC is necessary for consciousness and consequently predict that every content of consciousness should be decodable from the PFC”.”

We have adopted the proposed formulation for prediction #1. In the revised manuscript, it reads:

“Prediction #1: According to IIT, PFC is not necessary for consciousness. Consequently, proponents of IIT predict that decoding of conscious content should be maximal from the

posterior cortex. According to GNWT, PFC is necessary for consciousness and consequently predicts that every content of consciousness should be decodable from the PFC.”

3) The revision on page 19 (paragraph two) says: ” Furthermore, within the framework of IIT, spiking activity is considered a constituent property of the physical substrate of consciousness according to IIT”

 Fix pleonasm of “within the framework of IIT” and “according to IIT” (i.e. delete one of them).

We thank the reviewer for noticing this duplicity. We have updated the formulation to: “Furthermore, within the framework of IIT, spiking activity is considered a constituent property of the physical substrate of consciousness.”

Referee #4 (Remarks to the Author):

I’ll begin by saying that I am highly appreciative of all of the work that the Authors have put into addressing the comments that I made in the last report, and am deeply honored that they would take the time to do so. I especially appreciate how, in their replies to my comments, they’ve provided very patient explanations of what the reasoning is behind some of the ways in which they’ve chosen to present their work. I also want to recognize the numerous ways in which they’ve very effectively clarified how some of their ideas are presented in the manuscript. Because of the Authors’ explanations, additions, and clarifications, most of my worries have been very convincingly dispelled. What you’ll find below are just those handful of items that I still have reservations about.

We thank the reviewer for the many insightful comments provided during the first and second revisions. This has been a very fruitful exchange, through which we have gained a deeper understanding of the nuances of other theories of consciousness and better grasped the rationale for relating our results to these theories i.e., the significance for attributing consciousness to other animals and infants. For ease, we grouped together the reviewer’s comments based on topics, and have replied to each topic separately.

A. GNW and IIT portrayal, and reference to other theories

A few comments about this addition to the manuscript (referring to the justification of focusing on GNW and IIT):

1. Small point: In the way that it is phrased, there’s multiple ambiguities in the sentence quoted from the manuscript: What does “they” refer to? Are we focusing on just GNWT and IIT, or focusing on them along with many other theories? Saying that you “focus on GNWT and IIT among many other theories” (italics mine) could be interpreted either way. I know that, in the manuscript, it soon becomes clear to the reader that the article focuses on just GNWT and IIT, but this momentary ambiguity is unnecessary.

This is an excellent point. To avoid ambiguities we have updated the sentence to: “We focus on GNWT and IIT, two theories of consciousness out of several others widely discussed e.g., Recurrent processing theory and Higher-order theories^{1,2}, since these theories feature

prominently in the field of consciousness science as shown in a recent systematic review of the literature⁴.”

2. This addition fails to convey the fact that the other theories that I thought that should be discussed in the manuscript (LR and HO) are roughly on equal footing (or close enough) with IIT and GNWT: They are on equal footing both in popularity among consciousness researchers, and in the amount of supporting evidence that each of these four theories have. This sentence doesn't convey this at all, in part because it avoids mentioning the theories by name. Note that I'm not saying that LR and HO need to be mentioned at this point specifically, but rather that this sentence isn't dispelling the worry that I had (which it was intended to do, according to the Authors).

And elsewhere: My concern is not that readers think that IIT and GNWT are the only theories of consciousness, but rather that the reader thinks that these two theories are much better supported, and taken much more seriously in the field of consciousness studies, than all of the other theories (including LR and HO).

And again: I thank the Authors for conscientiously making the changes and additions that they have made in response to my comments. At the same time, and for reasons stated above and in the last report that I created, I feel that the way that the issue is presented still runs the risk appearing to misrepresent the standing of GNWT and IIT relative to the two other major theories of consciousness (LR and GNWT). In my view, this should still be corrected.

We have corrected the introductory paragraph to reduce ambiguities and avoid giving the impression that GNW and IIT are the most endorsed theories. We have also mentioned explicitly other theories. The current sentence reads:

“We focus on GNWT and IIT, two theories of consciousness out of several others widely discussed e.g., Recurrent processing theory and Higher-order theories^{1,2}, since these theories feature prominently in the field of consciousness science as shown in a recent systematic review of the literature⁴.”

We have also added a qualifier to the following sentence

“With this goal as a starting point, we make a concerted effort to test two theories of consciousness, among several widely discussed ones¹, through a large-scale, open science, adversarial collaboration^{12,13,22-24} aimed at accelerating progress in consciousness research by building upon constructive disagreement.”

We also reread the entire manuscript and made sure we don't give the false impression that these two theories are much better supported than others.

B. Corrections of formal referencing mistakes:

1. While I of course like to be cited, especially in an article that presents such impressive research, I am not a higher-order theorist (rather I more often serve as a critic), so I am

puzzled by my being cited here.

2. With all due respect, I believe that this misconstrues Brown's position, which is in fact committed to the content of HO states being represented in the PFC. This is something that he has confirmed to me in conversation. Fortunately, the necessary conceptual work has already been done for dividing HO theories into those that are strongly committed to the content of consciousness being represented in (and therefore decodable from) the PFC: There is now a distinction commonly made in NCC research (by myself, Block, Lau, Brown, Michel, and probably others) between Upper- and Lower-Deck PFC Theory (though not always using these terms), one which maps neatly onto forms of HO Theory that do and do not hold the contents of consciousness to be represented. The clearest explications of this distinction are available in (Kozuch 2021, in press) and (Michel 2022). One can find an up-to-date list of the forms of HO/PFC Theory that are and are not committed to the decodability of content from the PFC in (Kozuch, in press (p. 5)).

We have corrected the referencing mistakes from the previous version. Due to a reference manager issue, we incorrectly referred to the reviewer's paper and cited Brown in the wrong section. We concur with the reviewer that Brown has been explicit in his commitment to decoding in the PFC. When referring to the pointer mechanism, we intended to cite Lau, not Brown. The citations have been corrected accordingly.

C. Relation of results to other theories of consciousness

Moving on now to the other issue here, that of what kind of discussion these other two theories should be given in the manuscript: While the Authors have convinced me that an extended discussion of these issues might be out of place, I don't see how the paper is better for not having at least a cursory discussion of what these results do and do not say about the other two major theories of consciousness.

The results that the researchers present in this article are some of the most informative that we've had in the last couple decades, and therefore not using them to shed at least a small amount of light on each of our four preeminent theories of consciousness seems to be a lost opportunity.

I would make the same observations here as I made just above, in relation to the issue of what bearing these results have on LR and HO. I think, in fact, that this is a more important issue to discuss, for reasons mentioned in my original report. I note, too, something that I neglected to mention up until this point, which is that part of the reason for which this issue is so important is because of the implications it has for animal and infant consciousness (given that each of these types of being has an absent or underdeveloped PFC) (See, for example, the discussion in Kozuch 2023b).

And elsewhere: I appreciate greatly the care with which the Authors wish to proceed here in regard to the conclusions that they want to draw. At the same time, one might reasonably be under the impression that LR is committed to the same predictions here that IIT is; if this turns out to be not the case, this is interesting, and probably worth discussing in the manuscript.

We have been convinced by the reviewer that omitting the impact of our results on other theories of consciousness, particularly the challenges they impose, might send an undesirable signal. This omission could give the impression that only GNWT and IIT are challenged, whereas our results actually affect a broader range of theories. We have accordingly added a paragraph to address this, highlighting the challenges for HOT regarding the lack of decoding in the PFC and the challenges for Recurrent Processing Theories concerning the lack of connectivity between early and higher-level visual areas. In doing so, we differentiated the predictions of these theories from those of GNW and IIT, as suggested by the reviewer. In Figure 5, we have also added a reference to prompt the reader to consult the discussion section for information on how our results impact other theories. Finally, we noted how this could affect inferences about the distribution of consciousness in non-human cases. See answers to reviewer #2, point 1.

D. Richness of consciousness

With respect to our previous revision: I believe this is a good addition (though perhaps the Authors are being too modest when they say that their “efforts are still far from” adequately measuring consciousness). My only suggestion here is that it would be better if the reader did not have to guess what this “apparent phenomenal richness” consists of. Instead, the Authors could specifically mention some of what constitutes this richness (e.g., an object’s brightness and hue, its precise shape and location, the highly specific viewpoint from which an object is perceived, etc.).

Following the suggestion of the reviewer, we have updated the manuscript to better specify what is meant by “apparent phenomenal richness.” The revised paragraph now reads:

“Second, although we made our best efforts to capture the richness of experience by investigating multiple dimensions of conscious experience (i.e., category, orientation, identity and duration), we acknowledge that our efforts are still far from measuring consciousness in a way that truly captures its apparent phenomenal richness (e.g., an object’s brightness and hue, its precise shape and location, the highly specific viewpoint from which an object is perceived, etc.).”

Referee #5 (Remarks to the Author):

Explaining the first-person perspective (consciousness) from a third-person perspective is one of science's most formidable challenges. Over the past three decades, systems neuroscience witnessed an explosion of empirical insight into this research question. Yet, there is no widely accepted theoretical framework to provide a rigorous foundation. This paper is motivated by the fact that there are two leading, but competing, theories. Prior work was unable to conclusively decide between these rival theories. As a potential solution, the authors organized a large-scale collaborative research effort that included champions of either theory to collectively test previously published predictions. The outcome revealed that neither theory produced a full slate of successful predictions (though it favors one theory over the other). This work is original, significant, and remarkable in idea, scope, and outcome.

The experiments accurately test all three core predictions, which are logically deducible from the theories. It is noteworthy that the core predictions were devised by the primary authors of the respective theories, thus leaving little room for speculation. However, prior work already

made clear that GNW posits involvement of the prefrontal cortex (PFC), whereas IIT predicts that PFC is not essential. Specifically, a series of peer-reviewed papers discussed the role of PFC from both the perspective of IIT and GNW, and identified this empirically testable question as key distinction (Boly M, et al. J Neurosci. 2017; Odegaard B, et al., J Neurosci. 2017). The experiments outlined the current paper explicitly test for that predicted difference on top of several more refined and quantified corollaries.

Importantly, the experimental design aimed to maximally avoid biases or other problematic forms of scientific conduct. That is, the research approach represents current best practices: (1) The chosen predictions are sufficiently distinct from each other and thus cannot both be true at the same time ("strong inference"). (2) The predictions were preregistered. That is, there is a written record of the predictions preceding the collection of data, thus preventing adaptation of theory to outcome. (3) The predictions included quantitative pass/fail criteria, thus preventing ambiguity in interpreting the outcome.

The experiments have been designed with appropriate care, and the analyses are reflecting the current state-of-the-art. The data are largely unambiguous, of high statistical power, and entirely novel. Taken together, the findings provide compelling evidence that both theories failed in the totality of their predictions. That is a significant achievement from a Popperian point of view, which urges scientists to find failures in theoretical predictions. From an alternative philosophical (Bayesian) point of view, the conclusion reaches even further in that IIT gains in likelihood over GNW. So, on either view, there is significant progress. The authors favor a third (Lakatosian) perspective, which is increasingly gaining popularity in neuroscience (Doerig et al., Nat. Rev. Neurosci. 2023). This epistemic stance argues that failing theoretical predictions of this kind spur scientific advance via further theoretical development. Importantly, all of these philosophical lenses converge on the notion of significant empirical progress in this paper. This is especially notable when considering the theoretical impasse that plagued the field (e.g. Seth & Payne, Nat. Rev. Neurosci. 2022). The authors thus arguably succeeded with the implementation of an "adversarial collaboration" approach to combat a theoretical stalemate in an important research field. Other fields likely will take note.

The extensive Supplementary Information provide an extremely useful addition. This material successfully addressed remaining questions that the main paper could not accommodate given the space constraints.

To conclude, both the approach and the conclusions of the paper are sound, highly original, and impactful. The findings are of immediate interest beyond neuroscience. This paper not only empirically solved a long-standing impasse in a research field of general concern by revealing flaws in leading theories. It also demonstrates innovative new ways of conducting research in our age of increased collaboration, open data, and transparent science.

We are deeply thankful to the reviewer for the positive evaluation of our study and for highlighting the potential impact of our research on the field of consciousness and beyond.

minor issues:

As one of the previous reviewers pointed out, there are a few aspects to the final analyses that somewhat deviate from the preregistered details. The authors replied by providing reasonable rationale for these modifications. However, the final text does not indicate where, how, and

why finalized predictions were modified. This should be clarified, especially since the paper dedicates an entire section to preregistration.

We thank the reviewer for pointing this out. We very much agree with the reviewer that we could have done a better job detailing the deviations from the preregistration and the motivations behind them. Prompted by the reviewer's comment, and aiming to make these changes more explicit and transparent to the reader, we have added a section to the supplementary material. In that section, we detailed every deviation from the preregistration, providing the corresponding justification. Those deviations are also referenced in the main text, alongside notes on additional control and exploratory analyses. We hope that by providing them as a consolidated report in the SI (alongside in-text referrals), it will be easier for the readers to identify and judge the importance of these deviations.

During the revision, and in further scrutinizing the preregistration, we also noticed an inconsistency in the way in which the RSA was computed for the MEG data i.e., orientation was defined as a two-way class as opposed to a three-way class as done for all other analysis. We have redone the analysis, correcting this omission. The results do not change. We have updated the supplementary material with the corrected analysis (see Supplementary Figure 45).

Several of the line plots seem to indicate variance using confidence intervals. This is not always indicated in the corresponding figure legends,

In the same vein, several of the bar plots feature error bars, but these are not defined as standard error of the mean (or alternative) in the figure legends.

We sincerely apologize for those oversights. In the revised manuscript, we have corrected them by indicating in every figure the type of error and variance.

Referee #5 (Remarks on code availability):

The associated code currently cannot be tested by the peer-reviewers. The linked GitHub repository seemingly lacks several directories required to recreate the figures.

We thank the reviewer for pointing out this problem, and apologize for the inconvenience. We have found out that the name of the file containing the list of participants used to run the demo was placed in a wrong directory. This has been solved. We also found that for the MEG and iEEG modalities, the links to download the demo data were missing. This has also been corrected. Please note that our code has several 3rd party dependencies that the user will need to install if not already available. Also please note that the code is tailored to be used in a Linux HPC, and the use in other platforms will require the code to be adapted.

Referee #6 (Remarks to the Author):

This manuscript shows how results on brain activation, decoding and connectivity in different experiments support, partially support or fail to support pre-specified hypotheses regarding conscious content, maintenance of conscious content, and interareal communication for these 2 theories of consciousness that have come together in this Adversarial Collaboration grant.

Conscious content: robust decoding for PFC across modalities but some differences between faces and orientation that are interpreted as partial support for GNWT; and for IIT clear support as decoding of conscious content (both category and orientation) was robust in posterior cortex (although this prediction was not deemed key for IIT).

Maintenance of conscious content: sustained activity in posterior cortex, including representation of category and identity across multiple stimuli partially support IIT, however, sustained responses were very rare in the intracranial data, and no sustained representation of orientation. GNWT was also partially supported by the patterns for later phase ignition after stimuli onset but not offset.

Interareal communication: While the authors express that there is no evidence in support of the prediction of the theories, in the exploratory analyses the amplitude-based metrics of connectivity (DFC and gPPI) show support for GNWT with PFC connectivity in timing and powerband; but not for IIT, with no sustained connectivity with V1/V2.

The results are novel in that it combines the same experimental design in direct cortical recordings in humans, with MEG and fMRI in separate cohorts. While the designs are not novel, they were, in principle, specifically agreed to test the three predictions that would differentiate between the theories.

The use of statistics and methods development are well explained and appropriate.

We thank the reviewer for highlighting the main results and acknowledging the appropriateness of our analysis methods.

The key part of the paper and what I have been specifically asked to comment on is not a trivial endeavour as it pertains to whether core predictions of the two theories were tested and how strongly the data are able to confirm/refute either theory. I think that the paper tests the predictions as they were presented in the preregistrations and agreed by IIT and GNWT parties. I also think that these predictions are not core to the key aspects of consciousness for these theories but for different reasons.

For IIT this first paper of the Consortium seems more relevant as it is more directed to conscious “content” than “access”. IIT theory gives primacy on content while GNWT is historically and conceptually closer to the conscious access claims. This difference, I interpret, makes the contrastive approach proposed by the adversarial collaboration rather taxing. On one hand the adversaries agreed on a series of predictions to be tested, but on the other hand it either does not test core aspects of the theory on conceptual grounds (IIT), or it is not a strong way of testing the theory (GNWT).

IIT authors define some core aspect of IIT applied to neural system in humans (in the section Integrated Information Theory: Melanie Boly, Christof Koch, Giulio Tononi) and interpret the results in terms of content but they do not link the results to the core aspects of the theory. This is unfortunate as any result should be interpreted considering the theory, and not only be restricted to the confirmation of the predictions. Further, this paper does not “account for how experience feels” (last paragraph of that section) but uses neural data to decode the differences between different aspects of the stimuli. While I see that the predictions have been tested and evaluated, there is no effort in justifying how these results contributed to the core aspect of IIT. The results are not linked to neural measures that might represent “the requirements for maximal integrated information”, and they are also not testing the core aspect of “essential

properties of experience”; making the predictions “not-core” to IIT.

On the other side of the adversarial, Dehaene, in the subsequent section, concludes that the design does not allow to differentiate between conscious and non-conscious process since these conditions are not present, and that creates a lot of problems interpreting that attentional lapses (in the form of blink, blindness, PRP or retro-cuing) could have decoupled consciousness. He also goes to say that these methods are purely correlational, and he looks forward to the 2nd and 3rd part of the COGITATE project to test how the theories explain consciousness and to define the necessary regions for conscious experience. So, also in his own words, I interpret, these predictions and their results in the current paper, do not test conscious access, hence they might not be testing conscious content adequately, and they do not test the necessary regions for conscious experience as they are purely correlational. Again, this paper does not test core aspect of GNWT.

The COGITATE endeavour is enormous amount of work and it has been carefully curated from the beginning in bringing together and trying to make very distinct theories to agree on specific predictions. The data is rich and will continue to nurture the cognitive neuroscience community for many years. What constitutes evidence in a normal research paper it is here elevated to a higher standard, this is a laudable effort and should receive the merit in deserves.

I am fully convinced of the results, I do not completely agree with the interpretation, as I do think that the predictions have been tested but they do not pertain to core aspects of the theories. I look forward to the next phase of experiments and results to integrate with these first pieces of evidence to see if and how these theories truly have explanatory power in our understanding of human (and maybe animal) consciousness.

We thank the reviewer for this detailed evaluation of our work. We are very grateful for the acknowledgment of the merits and uniqueness of our research. We also appreciate the feedback regarding the experiment not testing the core predictions of the theories.

Prompted by the reviewer’s comment and the editor’s suggestions, we have revised the manuscript to clarify the aims of this adversarial collaboration, which are to test contradictory predictions among theories as well as their neurobiological implementation. A key aspect for doing so is identifying points of disagreement. We acknowledge that certain core claims of the theories, such as the computationalist (functionalist) stance of GNWT and the mathematical core of IIT, are not being evaluated in this collaboration. Instead, we have focused on their neurobiological implementation, as this can be empirically evaluated and progress can be made.

While this approach leaves more degrees of freedom for the theories, allowing them to respond to challenging data by retaining their mathematical/computational cores and modifying their proposed biological implementations, these adjustments should be justified based on theoretical grounds. This flexibility is a natural feature of a project designed to test theoretical proposals about the neural correlates of consciousness. We hold that this endeavor is nonetheless meaningful, even if the core computational and mathematical predictions are not tested. It seems R5 and the editor share our views, and we have made this point very explicit in the manuscript. We hope the reviewer finds this acknowledgment satisfactory and agrees that the merits of this work and its contribution to the scientific community and our understanding of consciousness outweigh this issue.

If we interpreted the reviewer correctly, it is stated that GNWT and IIT do not share the same explanatory target, with GNWT historically focusing on conscious access and IIT on conscious content. Despite GNWT's empirical emphasis on conscious access, the theory aims to account for two different aspects: conscious processing (which processes require or do not require consciousness) and conscious phenomenology (what a participant experiences at a given moment). Historically, most studies testing GNWT have focused on conscious processing. For the purpose of this adversarial collaboration, we have focused in this experiment on conscious phenomenology to evaluate the theories on comparable grounds (while conscious processing as opposed to unconscious processing, is tested in Experiment 2, whose results are now being analyzed).

GNWT postulates that every conscious content should be represented in the prefrontal cortex (PFC). Assuming a falsificationist approach, and following the central tenet of GNWT that all content should be represented in the PFC, we derived prediction #1. For the theory to fail, it is sufficient not to observe decoding for one stimulus feature when the subjects are conscious of the stimulus. Thus, having an unconscious condition is not a prerequisite to falsify the theory. While the reasons for the challenges to both GNWT and IIT might be multiple, such as coverage, spatial resolution, or other factors, it is worth noting that most of the research testing these theories has used the methods employed in this adversarial collaboration, which were agreed upon ex-ante by the adversaries as adequate to test the theories.

From our perspective, the results we present do pose a challenge to both theories, and we await the scientific process to run its course and for further evidence to accumulate to evaluate the significance of the challenges we currently pose to the theories.

I'll begin by saying that I am *highly* appreciative of all of the work that the Authors have put into addressing the comments that I made in the last report, and am deeply honored that they would take the time to do so. I especially appreciate how, in their replies to my comments, they've provided very patient explanations of what the reasoning is behind some of the ways in which they've chosen to present their work. I also want to recognize the numerous ways in which they've very effectively clarified how some of their ideas are presented in the manuscript.

Because of the Authors' explanations, additions, and clarifications, most of my worries have been very convincingly dispelled. What you'll find below are just those handful of items that I still have reservations about.

So as to make this report easy to understand, I've often included excerpts from the original report, along with their replies, and then finally my replies to their replies.

The conventions are as follows:

- (1) my comments from the last report are in black font
- (2) the Authors' replies to the last report are in blue font
- (3) my replies to their replies are in red font

The report starts with a quoted passage from the original report, to be followed by the Authors' reply:

1. As stated above, the article focuses on GNW and IIT. However, if we were to produce a complete list of leading theories of consciousness, it would include not just GNW and IIT, but also local recurrency theory (LR) (Lamme 2010) and higher-order theory (HO) (Brown et al. 2019), perhaps along with others. It seems, furthermore, that the results of the experiments presented in the Article shed just as much light on these other two theories (i.e., LR and HO) as it does in the case of the two that the article focuses on. And so for the article to give no discussion of these theories seems to be a missed opportunity for advancing the field of consciousness studies. More importantly, it gives the reader the mistaken impression that the field has narrowed down the candidate theories of consciousness to just to just GNW and IIT, something that seems far from the truth (Michel et al. 2018). I will explain.

We thank the reviewer for these insightful remarks. We very much agree that our results bear implications on many theories of consciousness (and are accordingly highly meaningful even outside the scope of this adversarial collaboration). In the original manuscript, we did not include a discussion of other theories due to a lack of space, but more importantly because we followed the adversarial collaboration protocol previously defined in Mellers et al 2002, including limiting the number of words for each section of the discussion a priori i.e., cogitate, IIT and GNWT discussion section. We understand this may have appeared as a neglectful omission. Yet, it is hard to satisfy all constrains, including the strict word limits of journals and the adversarial collaboration protocols. With those constrains in mind, we have revised the

manuscript to make clear that GNWT and IIT are just two theories among many others, while we also justify the selection of those theories (p. 4, second paragraph).

For convenience the corresponding passage is included here:

“... We focus on GNWT and IIT, among many other theories of consciousness (A. K. Seth & T. Bayne, 2022; Signorelli et al., 2021), as they feature prominently in the field of consciousness science, and have received substantial empirical support as shown in a recent systematic review of the literature (Yaron et al., 2021)”

A few comments about this addition to the manuscript:

1. Small point: In the way that it is phrased, there's multiple ambiguities in the sentence quoted from the manuscript: What does “they” refer to? Are we focusing on just GNWT and IIT, or focusing on them along with many other theories? Saying that you “focus on GNWT and IIT *among* many other theories” (italics mine) could be interpreted either way. I know that, in the manuscript, it soon becomes clear to the reader that the article focuses on just GNWT and IIT, but this momentary ambiguity is unnecessary.

2. This addition fails to convey the fact that the other theories that I thought that should be discussed in the manuscript (LR and HO) are roughly on *equal footing* (or close enough) with IIT and GNWT: They are on equal footing both in popularity among consciousness researchers, and in the amount of supporting evidence that each of these four theories have. This sentence doesn't convey this at all, in part because it avoids mentioning the theories by name.

Note that I'm not saying that LR and HO need to be mentioned at this point specifically, but rather that this sentence isn't dispelling the worry that I had (which it was intended to do, according to the Authors).

After deliberation, however, we decided to avoid linking our findings to other theories, although we see great merit in doing so and completely agree that this should be done (in a separate paper). The main reason is, again, the word limit and the already dense nature of the manuscript.

To my knowledge, there are no word limits at this stage of the review process, so this shouldn't be a concern of the authors. I think a paragraph or two discussing how these results bear on these other major theories of consciousness would be sufficient. More on this below.

In addition, the HOT and RP theories have many variants, and only some of those are challenged by our data

I'll discuss this more below, where the authors mention this again.

There are currently at least two other adversarial collaborations (one led by Biyu He and another by Steve Fleming and Axel Cleeremans) that directly test predictions from those theories.

We consider it more prudent to let those projects report their findings and relate to ours. Moreover, we are now preparing an additional paper reporting on the mechanics of the adversarial collaboration, where we will touch on the issue of how to relate findings of an adversarial collaboration to other theories, while also adhering to adversarial protocols i.e., defining predictions and designing experiments geared to challenge those predictions. We understand that this might seem disappointing, yet we hope the reviewer understands the constraints we are under.

As we anticipate that others will probably share the reviewer's opinion, and rightfully so, we changed the concluding paragraph to explicitly acknowledge that expectation. Although we cannot meet it here, as we explain above, we hope to be able to do so in future papers about this work (e.g., after Experiment 2 is also complete, we plan to write a final paper integrating the results of the two experiments, which could be a good venue for this more elaborate discussion concerning how our results relate to other theories besides GNWT and IIT).

For the reviewer's convenience, below is the revised excerpt from the discussion section (p. 30, last paragraph):

"At this point, the reader might expect the consortium to draw a final conclusion, or to relate the findings to other theories of consciousness, which without doubt put pressure on other theories i.e., some variants of higher-order (Kozuch, 2023) and recurrent processing (Lamme, 2015; Malach, 2021)".

(While I of course like to be cited, especially in an article that presents such impressive research, I am not a higher-order theorist (rather I more often serve as a critic), so I am puzzled by my being cited here.)

HO states that a mental state becomes conscious when it is represented by another mental state. According to how HO is usually construed (e.g., Rosenthal 2011; Brown 2015), the content that becomes conscious will be the content of the HO state (as opposed to the content of the lower-order state). Consider now that it is commonly thought that the kind of HO states that are essential for consciousness will be produced in the PFC (for review: see Kozuch 2023a). Given this, it seems that HO predicts, just like GNW, that whatever one consciously experiences should be decodable in the PFC. And so it seems that it is not just GNW, but also HO, that makes Prediction #1.

(Worth noting: In some ways, Prediction #1 seems to be a more stringent test of HO than it is of GNW, since HO might be (and often is) construed as requiring activity in just the PFC for consciousness (as long as certain background conditions obtain), whereas GNW takes the PFC to be just one of the members of the constellation of brain areas that GNW says gives rise to consciousness. Because of this, it might be easier for GNW to explain why their Prediction #1 is not met, since the GNW theorist might claim, e.g., that the failure to decode the stimuli orientation in the PFC occurs because such content is actually distributed across the frontoparietal areas, rather than being comprehensively represented in the PFC.)

This is an excellent observation, with which we very much agree. The reason why we have abstained in the original and the revised manuscript from relating our findings to other HO theories is because there are several HOT variants, and only some of those make explicit predictions about a higher-order representation in PFC, while others invoke other mechanisms such as pointers (e.g., Brown, 2015).

(With all due respect, I believe that this misconstrues Brown's position, which is in fact committed to the content of HO states being represented in the PFC. This is something that he has confirmed to me in conversation.)

Fortunately, the necessary conceptual work has already been done for dividing HO theories into those that are strongly committed to the content of consciousness being represented in (and therefore decodable from) the PFC: There is now a distinction commonly made in NCC research (by myself, Block, Lau, Brown, Michel, and probably others) between Upper- and Lower-Deck PFC Theory (though not always using these terms), one which maps neatly onto forms of HO Theory that do and do not hold the contents of consciousness to be represented.

The clearest explications of this distinction are available in (Kozuch 2021, in press) and (Michel 2022). One can find an up-to-date list of the forms of HO/PFC Theory that are and are not committed to the decodability of content from the PFC in (Kozuch, in press (p. 5)).

To avoid mischaracterizing any of those theories, we preferred not to relate our findings to them, especially given the two concurrent adversarial collaborations explicitly testing these theories. However, it is also our hope that when those adversarial collaborations release their preregistrations, we and others in the community will be able to explicitly relate our findings, or even test those preregistered predictions on our openly available dataset.

A similar observation can be made in the case of IIT and LR. According to LR, visual consciousness arises from recurrent loops localized to the visual cortex, ones that are between lower- (V1/V2) and higher- (e.g., LO, IT, V4) level brain areas. These are of course brain areas that are comparable to those that figure into each of the IIT Predictions made in the Article. Given this, we can consider each of the IIT predictions to also be predictions of LR, in which case we can consider the experiments presented by the Authors to confirm and disconfirm LR in precisely the same way that it is taken to confirm and disconfirm IIT.

Again this is an excellent and completely fair point, and we very much agree with the reviewer. Yet under the same rationale as the one applied on HOT, here too we intentionally refrained from referring to Recurrent Processing (RP), both due to space limitations and due to the several variants of RP theory. Some theories, like those suggested by Lamme (Lamme, 2015), emphasize extrinsic connections between early and mid-level visual areas, whereas others, such as the one proposed by Malach (Malach, 2021), argue that local, intrinsic connectivity (horizontal connections) is essential for consciousness. An ongoing adversarial collaboration is currently comparing first-order and higher-order theories and will likely be in a better position to provide

evidence challenging these different variants. Therefore, while we concur with the reviewer's perspective, we have chosen to strictly adhere to the theoretical predictions given to us by proponents of GNWT and IIT, without extending our discussion to other theories in the current paper to avoid mischaracterizing those theories, to stay within page limits, to avoid over-complexity, etc. However, this suggestion will be highly valuable for a follow-up study, and we do hope that the field will consider our results under a wider perspective, also taking into account their meaning to other theories.

Notably, this seems to even be true of Prediction 3, which was in some ways much more specific than the first two predictions, since it identified not only those brain areas that should be representing the conscious content, but also the nature of the activity underlying this representation (i.e., synchronized activity between higher- and lower-level brain areas). This seems to fit very well with what we would expect LR to predict. Indeed, one might think that, in the case of experiments that the Researchers ran in this article, LR and IIT are actually empirically indistinguishable: They seem to both make the same predictions as far as what the results of the experiments should be.

Again, an excellent remark, and a case in point for how complex it is to relate our findings to other theories—

When preparing the revision, we reached out to Victor Lamme to confirm whether RP indeed also predicts sustained synchronization between higher and lower visual areas, measured as phase-locked synchrony. We learned that RPT expects sustained interaction between visual areas as long as the stimulus is consciously visible. However, this sustained interaction is not necessarily expressed as sustained synchrony. This is very different from IIT, for which synchrony is necessary. According to the theory, any content of experience (in our experiment the high-level invariant content face, and low-level features such as the particular contour of the face) should correspond to substructures. These are further bound by relations – overlaps between causes a–d effects - which in the brain should typically be accompanied by firing synchrony. Other measures that can evaluate connectivity, like for instance dynamical connectivity or psychophysiological interactions are not good proxy measures for IIT, as those are amplitude-based measures.

I appreciate greatly the care with which the Authors wish to proceed here in regard to the conclusions that they want to draw. At the same time, one might reasonably be under the impression that LR is committed to the same predictions here that IIT is; if this turns out to be not the case, this is interesting, and probably worth discussing in the manuscript.

This discrepancy underscores how theoretical predictions are intertwined with the preferred metrics that may (or may not) directly reflect the underlying mechanisms. Given these complexities, we refrain from linking our findings to RPT in this revision, opting instead to explore this connection in a subsequent piece. Additionally, we are aware that others, including Robert Chris-Ciure and Georg Northoff, are composing a comprehensive comparison of empirical measures within consciousness theories. This work promises to further elucidate how

mechanistic accounts relate to specific measures and in turn how those fare between theories. This might help us and others, in the future, to better connect our findings to other theories than the ones tested.

The upshot of all this is that I think that the article could benefit greatly from (a) an acknowledgement that IIT and GNW are not the only leading theories of consciousness, and (b) a substantial discussion as to what the results presented in the Article mean for these other theories of consciousness. As well, I should mention again that the article, as it is, seems to implicitly misrepresent the field of consciousness studies, since these omissions might give the reader the impression that these are the only two theories of consciousness that anyone takes seriously anymore, something that is untrue.

These are both points well taken. Certainly, it was not our intention to portray the field as consisting of just these two theories of consciousness.

My concern is not that readers think that IIT and GNWT are the *only* theories of consciousness, but rather that the reader thinks that these two theories are much better supported, and taken much more seriously in the field of consciousness studies, than all of the other theories (including LR and HO).

As stated above, we have modified the introduction to avoid giving that impression (p. 4, second paragraph), as well as in the final conclusion (p. 30, last paragraph), to mention that our results might also relate to other theories to alleviate the impression that we are misrepresenting the field of consciousness studies. We hope that this, and the future publication we are planning where this discussion will take place, mitigate the reviewer's concerns.

I thank the Authors for conscientiously making the changes and additions that they have made in response to my comments. At the same time, and for reasons stated above and in the last report that I created, I feel that the way that the issue is presented *still* runs the risk appearing to misrepresent the standing of GNWT and IIT relative to the two other major theories of consciousness (LR and GNWT). In my view, this should still be corrected.

Moving on now to the other issue here, that of what kind of discussion these other two theories should be given in the manuscript: While the Authors have convinced me that an extended discussion of these issues might be out of place, I don't see how the paper is *better* for not having at least a cursory discussion of what these results do and do not say about the other two major theories of consciousness.

The results that the researchers present in this article are some of the most informative that we've had in the last couple decades, and therefore not using them to shed at least a small amount of light on each of our four preeminent theories of consciousness seems to be a lost opportunity.

2. So far, I have pointed out how I think that the Article, as it currently is, misses out on opportunities to expand our neuroscientific knowledge of consciousness by showing how the results reported in the Article bear on HO and LR. The same point can be made about another debate, one concerning a more general distinction in types of theory, this being between Local and Broad theories of consciousness (for distinction, see Michel & Doerig 2021). According to Local Theories (this includes LR and IIT), activity in just the visual cortex (given some background conditions) can be sufficient for visual consciousness; according to Broad Theories (this includes GNW and HO), areas outside of the visual cortex (which are usually considered to include the PFC) are necessary (perhaps in addition to areas in the visual cortex). The relevance of the results presented in the Article to this debate is obvious, given the strong similarities to be found between GNW and HO, and especially given the apparent empirical equivalence of IIT and LR (discussed in Point 1).

It would be good if the Article were to comment on what their results mean for the Local vs. Broad debate, not least of all because this debate is very consequential in regard to the issue of where future research efforts should be directed (i.e., whether it is more worthwhile to pursue the development of Local or Broad Theories of consciousness). It should be noted that one of main points of contention between Local and Broad Theories is what role the PFC plays in consciousness, this of course being one of the issues that the Article focuses on as well. Given this, the Article seems well-poised to comment on the Local vs. Broad debate.

We very much agree with the reviewer that our results are not just informative with respect to the two theories directly under scrutiny but also illuminate other discussions in the field. However, as we explained above, we are afraid this is not possible in the limits of this manuscript. However, as we explained, we plan to explore this subject in depth in a forthcoming publication, and look forward to developing these interesting theoretical points.

I would make the same observations here as I made just above, in relation to the issue of what bearing these results have on LR and HO. I think, in fact, that this is a more important issue to discuss, for reasons mentioned in my original report.

I note, too, something that I neglected to mention up until this point, which is that part of the reason for which this issue is so important is because of the implications it has for animal and infant consciousness (given that each of these types of being has an absent or underdeveloped PFC) (See, for example, the discussion in Kozuch 2023b).

8. The Authors' goal to carry out experiments that looked for correlations between neural activity and *multiple* dimensions of conscious experience (e.g., category and orientation) at the same time is highly admirable, and the result of their focusing on this is that we get some highly informative data concerning how stimuli are consciously represented in the brain.

At the same time, it should be noted that their efforts are still very far off from measuring consciousness in a way that truly captures its apparent phenomenal richness: The typical visual

experience does not consist merely of the representation of the category to which the object belongs, along with the orientation from which one views it, but so much more: One experiences the object's brightness and hue, its precise shape and location, and experiences the object from a highly specific viewpoint (something more fine-grained than a broad description such as "from the front" or "from the side"); one will also experience the hue and brightness of the object's background. (The list could continue here, but I'll stop so as to not court controversy...)

...

This is not to say the Authors have been negligent here (indeed, they do a much better job of trying to experimentally capture the phenomenal richness of visual experience than has been done so far). Rather, I just think that this limitation of the Authors' studies should be acknowledged in the Article. As well, perhaps the Authors could make some suggestions as to how these limitations could be overcome in the future.

We agree with the reviewer regarding the importance of studying phenomenal richness of a visual experience and designing paradigms capable of capturing such richness. Following this comment, we now acknowledge that our attempt to do so is only partial, and hope this will be further pursued in future studies in the field (though, due to space limitations, we sufficed with saying that it should be done in future studies, without being able to suggest how this should be done). We accordingly added this limitation to the discussion section (p. 25, third paragraph), which now reads:

"...Secondly, although we made our best efforts to capture the richness of experience by investigating multiple dimensions of conscious experience (i.e., category, orientation, identity and duration), we acknowledge that our efforts are still far from measuring consciousness in a way that truly captures its apparent phenomenal richness. Future studies will be needed to address that further."

I believe this is a good addition (though perhaps the Authors are being *too* modest when they say that their "efforts are still far from" adequately measuring consciousness).

My only suggestion here is that it would be better if the reader did not have to guess what this "apparent phenomenal richness" consists of. Instead, the Authors could specifically mention some of what constitutes this richness (e.g., an object's brightness and hue, its precise shape and location, the highly specific viewpoint from which an object is perceived, etc.).

This concludes my report. I again give many thanks to the Authors for paying such close attention and to my comments and for writing such careful replies to them. I am honored to have been asked to act as a reviewer on such a fine piece of research.

Signed,
Benjamin Kozuch
University of Alabama
Philosophy Department
bkozuch@ua.edu

REFERENCES

- Brown. (2015). The HOROR theory of phenomenal consciousness. *Philosophical Studies*, 172(7), 1783–1794.
- Brown ... LeDoux. (2019). Understanding the higher-order approach to consciousness. *Trends in Cognitive Sciences*.
- Herzog ... Doerig. (2020). All in good time: long-lasting postdictive effects reveal discrete perception. *Trends in Cognitive Sciences*, 24(10), 826–837.
- Kozuch. (2023a). A Legion of Lesions: The Neuroscientific Rout of Higher-Order Thought Theory. *Erkenntnis*, 1–27.
- Kozuch. (2023b). A new low: Reassessing (and revising) the local recurrency theory of consciousness. *British Journal for the Philosophy of Science*, online only(<https://doi.org/10.1086/727890>). <https://doi.org/10.1086/727890>
- Kozuch. (in press) An embarrassment of riches: The PFC's not the content NCC. available at <https://www.dropbox.com/scl/fi/aa9e7qva3kdw1wsagp3rr/kozuch-24-richnesses.pdf?rlkey=rzo7u0itmre53q6oeh5palvyk&dl=0>
- Lamme. (2006). Towards a true neural stance on consciousness. *Trends in Cognitive Sciences*, 10(11), 494–501.
- Melloni ... Lepauvre. (2023). An adversarial collaboration protocol for testing contrasting predictions of global neuronal workspace and integrated information theory. *Plos One*, 18(2), e0268577.
- Michel & Doerig. (2021). A new empirical challenge for local theories of consciousness. *Mind & Language*.
- Michel ... Liu. (2018). An informal internet survey on the current state of consciousness science. *Frontiers in Psychology*, 9, 2134.
- Rosenthal. (2011). Exaggerated reports: reply to Block. *Analysis*, 71(3), 431–437.
- Siegel & Byrne. (2017). Rich or thin?